pedology/environmental science

agricultural land conversion, farmland preservation, land take, soil ecosystem services, soil governance, soil security

**Author for correspondence:**
Lewis Peake
e-mail: l.peake@uea.ac.uk

# Saving the ground beneath our feet: Establishing priorities and criteria for governing soil use and protection

## Lewis Peake[1,2] and Cairo Robb[3]

[1]School of Environmental Science, and [2]Tyndall Centre for Climate Change Research, University of East Anglia, Norwich, UK
[3]Legal Research Fellow, Centre for International Sustainable Development Law

(iD) LP, 0000-0001-8835-7909

The continual loss and impairment of soil ecosystem services (SES) across the globe calls for a fundamental reconsideration of soil governance mechanisms. This critical synthesis charts the history and evolution of national and international soil law and seeks to unravel certain challenges that have contributed to this failure in governance. It describes and categorizes law and policy responses to different soil threats, and identifies a worrying widespread absence of legislation for oversight and protection of agricultural soils from urbanization, as well as a lack of clear legal mechanisms to determine national priorities for soil protection. A reduction in the world's prime farmland threatens SES, including food security, carbon storage and biodiversity. Falling between the stalls of agricultural and environmental law, the fate of farmland is often left to planners who do not see themselves as responsible for soils. Consequently, legal instruments with the greatest power to affect soil, sometimes irreversibly, are often framed and worded with little or no reference to the soil. Nevertheless, emerging conceptual frameworks might offer positive outcomes. The authors advocate robust holistic policies of soil governance and land use planning that place SES and natural capital at the heart of decision making.

# 1. Introduction

Soil is a vital multifunctional resource that could be regarded as the metaphorical as well as the literal foundation of human civilization. Soil ecosystem services (SES) [1], and the broader

concept of soil-derived nature's contributions to people (NCP) [2], is a relatively new academic subdiscipline, but human awareness of the life-supporting role of soil is as old as civilization, if not older. Clearly, the ground beneath our feet provides so much more than the ground beneath our feet, and transcends title deeds, boundary fences and even national borders. Arguably, we are more dependent on soil than ever, as our dominance as a species has resulted in greater demands and stressors on our environment. As the world's population has soared and almost every parcel of its terrestrial surface has been assigned to the various national states, and then further sub-divided and transformed by governments, private corporations or individuals, the need to safeguard this increasingly degraded resource is more urgent than ever.

In addition to the familiar and very direct role that soil has in providing our food, fibre and timber, we often forget its many other interconnected functions, for example:

— Physical: providing building or landscaping material, and a stable substrate for infrastructure;
— Chemical: regulating the supply of plant nutrients;
— Biological: constituting biodiverse ecosystems and communities of useful organisms, converting toxic sewage and decomposing matter into plant nutrients and providing a genetic reservoir;
— Hydrological: facilitating water absorption, storage and filtration and attenuating flooding;
— Cultural: preserving cultural heritage and archaeological information, and providing recreation;
— Climatic: storing carbon (C), regulating greenhouse gas (GHG) fluxes and heat buffering.

The last of these would have been almost inconceivable to our forebears, only assuming critical importance approximately since the beginning of the new millennium.

However, in certain circumstances soil, like any other natural resource, can deliver negative impacts, such as emitting GHGs, harbouring pests and diseases, or contributing to the accumulation of sediment, dust storms and landslides. These events are reminders that good governance of soil is also about minimizing ecosystem 'disservices' as well as safeguarding essential ecosystem services. We live in a time of anthropogenic environmental crisis, unprecedented climate change and species extinction. A report recently released by WWF states that: 'Since 1970, our Ecological Footprint has exceeded the Earth's rate of regeneration,' and currently exceeds the world's capacity to support humanity by 56% [3]. Soil is an integral component within this system, both as an enabler and a victim of exponential human expansion.

This interdisciplinary critical synthesis reviews the role of soil governance in helping to achieve the objectives of the UN Sustainable Development Goals (SDGs), to mitigate and adapt to climate change and contribute to a more secure future for humanity. We trace the historical path of the status and governance of soil; categorize anthropogenic threats to soil, and legal mechanisms that can be applied to address them; and discuss the semantics surrounding legal aspects of soil and land, aiming to disambiguate and illuminate the terminology. The current state of the art of soil governance is explored in depth and a key finding is a failure to prevent or barely acknowledge the worldwide loss of prime farmland and the implications of this for society. The review concludes by looking to emerging approaches such as the soil security conceptual framework, proposed as a way of translating critical soil knowledge into sustainable development policy, and to holistic land evaluation methodologies that incorporate SES into land use planning.

## 2. Methods

An open-ended literature review was conducted, using combinations of the following search terms: 'agriculture', 'biodiversity', 'conservation', 'contamination', 'conversion', 'degradation', 'ecosystem services', 'erosion', 'farmland', 'governance', 'land', 'land take', 'land use planning', 'law', 'legislation', 'non-agricultural', 'policy', 'preservation', 'prime', 'protection', 'SDG', 'urbanization', 'urban sprawl', 'soil', 'soil sealing', etc. By selective analysis we synthesized key points, such as our own categorizations, thus adding new content. By presenting in this way, the intention was twofold: to provide the reader with some clear signposts within a large and complex domain; and to provide ourselves with a framework within which the context of any given issue could be better understood and deconstructed.

To complement data and cases studies from the literature, a comprehensive search was conducted to abstract data from FAOLEX, the UN Food and Agriculture Organization (FAO) online database of policies and laws [4]. FAOLEX documents legal instruments in force in nearly 200 countries. This

database is organized by country, into the following groups, of which those marked with an asterisk were reviewed by the lead author (where they existed for the country in question): *Policies*\*; *Legislation: Agricultural and rural development*\*, *Climate change*, *Cultivated plants*, *Disaster risk management*, *Environment*\*, *Fisheries*, *Food and nutrition*, *Forestry*\*, *Land and soil*\*, *Livestock*, *Sea*, *Water*, *Wild species and ecosystems*\*; *International Agreements*\*. *Land and soil* was the most relevant, but it was necessary to search further, because in many cases this group is primarily devoted to the administration of land ownership and land reform. The groups are not mutually exclusive, so some legislation appears more than once.

The name of the FAOLEX entry alone was sometimes broadly sufficient to explain its meaning, e.g. 'Soil conservation law', but in many cases, it was necessary to select the embedded hyperlink to access its summary (usually in English). Ambiguous or simplified wording often made it necessary to select a further link to the full text, either within FAOLEX or on the websites of national governments. These documents were often in the local language, sometimes in the local script, and where necessary Google Translate was used. In some cases, individuals professionally familiar with the legislation were contacted for clarification and asked whether the laws in question were implemented and effectively governed. Approximately 1% of countries had either created no such instruments or lacked any accessible data in FAOLEX. This information was compiled from the summer of 2020 until the summer of 2021. A spreadsheet was created with one row per country and columns indicating categories of legal instrument, and from this, it was possible to calculate percentages of countries with such measures in force. The FAOLEX groups are powerful retrieval mechanisms, but unrelated to the categories we derived. While our research was being conducted FAO was in the process of creating and populating a similar but soil-specific repository, SoiLEX, including data from FAOLEX[1]. In terms of structural organization and ease of use, SoiLEX is superior to FAOLEX but limited to highly soil-specific categories of legislation, whereas FAOLEX is wider in scope and, crucially for our purposes, contains laws relating to the protection or preservation of categories of land such as prime farmland and natural ecosystems.

Some caveats are necessary. Such a synthesis can only ever be approximate and is undoubtedly incomplete. Several polities listed in FAOLEX as countries actually comprise a number of countries (e.g. the UK), or separate provinces or states (e.g. the US), with differing laws. Many instruments, especially policies, are generalized, e.g.: 'The land is the main national wealth, which is under the special protection of the state … ensuring rational use and protection of land' [5], iterated many times, but with little or no further qualification. A law stating that land will be classified as urban or rural could imply protected status or simply a degree of spatial planning; the latter was assumed unless explicitly otherwise. Furthermore, this is primarily a review of officially documented policies and laws, not a comprehensive assessment of enforcement or effective governance. While an overview of international soil law and policy context is provided, the synthesis does not deal in detail with states' external relations or human and peoples' rights relating to land and soil, nor does it address global or national soil information systems or soil education and literacy measures, all of which are also important aspects of comprehensive soil governance.

# 3. Soil as a valued resource and legal entity

## 3.1. The evolution of soil law

The ancient Egyptians called their nation state *Kemet* which means black land or dark earth, in contrast to the barren *Deshret*, or red land, stretching out on either side. The black silty clay of the Nile valley was not simply a metaphor for their country; in their minds, it *was* their country. Human dependence on productive soil and the impact of its degradation on civilization throughout history is well attested [6–10]. Mesopotamia experienced at least two catastrophic regime changes due, respectively, to soil salinity and soil erosion [11,12]. The Greek and Roman writers make multiple references both to the benefits of soil quality and the threat of soil degradation, and by the 1500s in Europe soil was regarded as the key factor of an economy [11]. History is full of examples of land-related wars and conquests, often correlated with soil quality [13–15] and repression and killing of those seeking to protect land and soil quality continues today [16,17].

---

[1]SoiLEX is described as a global database on national legislation on soil protection, conservation and restoration to facilitate access to information on the existing legal instruments in force and bridge the gap between the various soil stakeholders.: www.fao.org/soils-portal/soilex/en/.

The earliest known attempts to put an economic value on productive land were Chinese soil classification and maps created 2000 years BCE for taxation purposes [18]. Other ancient peoples also independently derived similar systems, for example in Mesoamerica [19]. However, while there are many historical examples of land ownership rights and transactions [20], it is difficult to pin down when laws or policies were first introduced to protect soil or land as a vulnerable or scarce resource in its own right, that is, part of the natural capital for the common good. The *Statutes on Agriculture*, compiled during the Qin Dynasty of the late fourth century BCE is China's, and possibly the world's, earliest known set of soil-related laws [21]. The *Statutes* were followed a century later, during the Han Dynasty (206 BCE–8 CE), by the *Book of Fan Shengzhi*, China's oldest surviving agronomic treatise which provides extraordinarily detailed instructions on most aspects of soil management and may well have been regarded by farmers as *de facto* law [22].

Perhaps the oldest surviving documented legislation explicitly addressing soil is that within the *Digesta* of the Eastern Roman emperor Justinian, published in the sixth century, but including laws dating from centuries earlier. While there was no overt implication that soil *per se* was a scarce resource in the Roman Empire, these edicts leave the reader in no doubt that soil was of great importance to everyone involved in its use, from the question of who benefited from its bounty to who was responsible for its maintenance [23].

In the modern era, the first specific instances of soil legislation were usually initiated in response to the threat of severe soil erosion, typically the result of inappropriate land management combined with extreme climatic conditions. The most iconic example is the US Dust Bowl of the 1930s, leading to the first major piece of soil-related legislation in the US, the 1935 Soil Conservation Act, but even before then soil erosion was perceived as an ever-present threat to American farmers and food production. By the 1870s there was widespread awareness of the problem of erosion [24]. The emerging environmental movement in Europe and America also had a growing influence on policy [25,26], and forest protection laws incorporating the principle of soil conservation appeared, for example in the US from 1873 [26] and New Zealand in 1874 [27]. In 1894, the US banned grazing in federal reserves to prevent land degradation [26] and in 1901 the Indian state of Punjab, under British administration, passed what might be the first soil and water conservation law, the *Punjab Land Preservation Act*, still in force today [28]. In 1907, Iceland passed the first national parliamentary soil conservation act [29].

A significant result of the Dust Bowl and President Roosevelt's generous response to it was the huge contribution made by the United States Department of Agriculture (USDA) and American scientists to the worldwide study and management of soils. Despite some criticism [30–33], the far-reaching impact of this intense period of activity, barely perceived outside the sphere of soil science, is perhaps unparalleled in human history. Building on a legacy of international scholarship, especially in Germany and Russia [34], in the space of a decade at least three trailblazing conceptual tools were created: the Soil Taxonomy, the land capability classification (LCC) methodology and the universal soil loss equation (USLE). These would radically transform the international science, management and legislation of soils.

From the 1930s other nations rapidly passed their own soil and land legislation. Food security was a major driver, but so was the loss of rural landscapes and wilderness. Other kinds of environmental protection were also implemented throughout the twentieth century, such as the creation of green belts around cities, even as early as 1901 [35]. In the 1940s, however, World War II impacted food production and distribution, especially in Europe, fostering more self-reliance. In the postwar world agricultural productivity took on a new significance and the new technique of land evaluation was enthusiastically applied [36,37].

In the UK, even before the end of the war, the Scott Report on rural land use recommended the protection of: 'good agricultural land', but also signalled a new approach to planning that incorporated landscape amenity [38], and has been cited as prescient in its ethos of sustainable development [39]. The report fed into the 1944 White Paper, *The Control of Land Use* [40] which paved the way for one of the earliest and most radical planning policies of its kind anywhere, enshrined in the UK 1947 Town and Country Planning Act which nationalized land use regulations and has been emulated worldwide [41]. While the driving force of the USDA LCC had been farm management and soil conservation, the British system was primarily a tool of the new planning system, being put to work by soil surveyors in Britain not to find new land to cultivate, but to identify and protect existing prime farmland. The latest incarnation of this system in England and Wales is the 1988 agricultural land classification (ALC) system for identifying the 'best and most versatile' (BMV) land [42]. In 1984, the USDA Natural Resources Conservation Service (NRCS) developed its own system oriented towards planning legislation, land evaluation and site assessment (LESA) [43].

The postwar era is notable for the creation of global and regional institutions such as the FAO, the International Union for Conservation of Nature (IUCN) and the European Union (EU), all of which have been instrumental in relation to land and soil governance. Shortly after its creation in 1945, the FAO embarked on the world's first formal international attempt to address soil conservation as a global issue, via a report [44] and a conference in 1948 [45]. Underpinning such initiatives was a growing awareness of the economic impact of soil degradation, to society at large as well as individual land users, with far-sighted hints at the valuation of ecosystem services and implications for government policy [46]. Similar analyses followed in the ensuing decades [47,48], and soil law continued to evolve, with the US taking the lead [49].

In 1949, the botanist Aubreville coined the term desertification to describe the transition of agricultural land in arid or semi-arid areas to an uncultivable state lacking ecological viability due to a combination of climatic and human factors [50]. By 1958 the Chinese government acknowledged the threat of desertification to the wellbeing of nearly 200 million people, and has initiated afforestation programmes since 1978 [51]. Despite controversy over its meaning, the concept of desertification became and continues to be a significant driver of sustainable land management (SLM) initiatives designed to tackle land degradation [52,53].

## 3.2. The conflicted role of agriculture

In a sign of things to come, the UK postwar agricultural policy came in for subsequent criticism for virtually exempting farmers from planning regulations, the assumption being that the role of farmers in protecting the rural landscape made controls on agricultural land use unnecessary [39,54]. After WWII, geopolitics stabilized and agricultural productivity soared, due to a range of technological innovations [31], which from the 1960s reached lower-income economies as the Green Revolution [55]. This period represents a radical reframing of society's attitudes towards the environment and agriculture [56]. In some parts of the world food security was no longer regarded as a serious threat, whereas agriculture itself was increasingly perceived as the primary threat to the environment, and hence a mixed blessing. Both the expansion and intensification of agriculture led to unprecedented wildlife habitat loss, encompassing deforestation, wetland drainage, conversion of grassland to arable and hedgerow clearance. Added to this were many other environmental impacts associated with industrial agriculture, including pollution, soil degradation and fuel inefficiency [57–59].

The publication of *Silent Spring* by Carson [60] led to a major international turning point in changing attitudes and policies in response to the excessive use of pesticides and herbicides. In Europe, in the 1970s and 1980s the Common Agricultural Policy (CAP) came to epitomize the ironic twin dilemma of taxpayers subsidizing overproduction at the expense of environmental destruction [40,54,56]. With the public emergence of climate change science since the late 1980s [61], and growing evidence of the significant climate-forcing influence of GHG emissions from farming [62], alongside the ongoing loss of biodiversity [63], agriculture finds itself more implicated than ever as a cause of the environmental crisis as well as its victim. The negative image of modern agriculture as an industry that degrades soil and threatens ecosystems risks devaluing the public perception of agricultural land, as if the sealing of such land would not result in a loss of environmental benefits. Meanwhile, soil science, straddling geology and biology, became virtually a subdiscipline of agriculture at least halfway into the twentieth century [11]. Hence soil, by association and also because largely hidden, rarely evokes the kind of intrinsic concern that attaches to wildlife or landscape.

## 3.3. The relationship between soil and land

Soil and land are inextricably linked in the context of governance and, indeed, many languages conflate the two in common parlance. Weigelt *et al.* [64] stress a distinction between soil and land, but also identify a gap in the literature to address the critical importance of governing them together to achieve sustainability. An absence of land rights also often encourages poor soil management and land degradation [65–68]. From a legal perspective, the concept of land is usually operative in matters of ownership and boundaries, and also in terms of spatial and territorial planning. Soil tends to enter the legal fray when it is transformed, harmed or threatened by a specific activity, whether by the owner of the land containing the soil, or by another party. This distinction signals two very different aspects of soil law. The former perspective generally encompasses property rights, whereby the injured party is primarily the landowner. The latter perspective offers protection to soil as a legal entity in its own right, even from its 'owner', for the common good.

Where the blurring of land and soil becomes problematic is in instances of proposed changes that are framed only in relation to land despite having significant impact on the soil within, or adjacent to, that land. This opacity can apply to legislation specific to agriculture, the environment or planning, and as a result important soil implications of land use conversion may fall into the gaps between all three. The creators of agricultural laws may feel that their remit regarding soil protection extends only to the direct impact of farming on soil, while the creators of environmental laws may feel that their remit for soil protection and conservation extends only to an unspecified component within a larger ecosystem or protected landscape. This then leaves rural land use conversion in the hands of the creators and implementers of planning laws, who may feel that soil *per se* is outside their jurisdiction, despite the soil-related implications of the land use changes that their legislation may encompass. There is in effect a blind spot whereby the legal instruments with the greatest power to affect soil, sometimes irreversibly, are often framed and worded with little or no reference to the soil. Raising this in a meeting with government soil scientists caused heads to nod in agreement and perusing comparative laws reveals that this is a recurring theme throughout the world [4].

## 3.4. Anthropogenic threats to soil and land

### 3.4.1. Types of threat

Humans have learnt to transform soil to their advantage, and the overall impact of human society on soil, both intended and unintended, has been profound. Short-term gains have often been at the expense of long-term harms. The picture is further complicated by the fact that human impact exacerbates or unbalances natural processes such as soil erosion, salinization or flooding. Soil or land is typically vulnerable to three types of anthropogenic threat, set out in table 1, and described further below.

### 3.4.2. Soil degradation

In this context, degradation refers to any harm done to the soil which is potentially ameliorated by remedial measures and includes acute problems such as contamination, salinization, acidification, sodification and compaction or other forms of structural collapse[2]. The term soil degradation has also been widely used to describe the gradual and chronic loss of productivity, typically the result of over-exploitation and inadequate management, invariably bound up with erosion, organic C depletion, reduction in soil biodiversity and low nutrient status. While acute soil degradation is often the result of breaches of regulations, chronic soil degradation tends to be associated with either intensive agriculture or rural poverty, and generally requires supportive rather than punitive legislation. These problems can be complex and linked to natural causes or intrinsic to certain soil types. In the worst cases, large tracts of land have been abandoned or declared unfit for food production with severe economic impacts. Gisladottir & Stocking [52] articulate the interlinkages of land degradation with other environmental problems such as climate change and biodiversity loss.

In the first study of its kind, UNEP commissioned the International Soil Reference and Information Centre (ISRIC) to conduct the Global Assessment of Soil Degradation (GLASOD). GLASOD concluded that from World War II to 1990 15% of all land worldwide (almost 2 billion hectares) was degraded, equating to 23% of inhabited land (14% seriously). Soil erosion was by far the most widespread form of soil degradation, affecting 83% of degraded land. Approximately 20 years later ISRIC scientists reviewed GLASOD and used satellite imagery to measure an overlapping and subsequently ongoing period (1981–2003) [70,71]. The authors detected a further 24% of the total land area that was degrading *mostly in addition to* GLASOD's 15% already degraded, i.e. a cumulative process, implying that at least a third of the world's land appeared to be degraded to some extent by 2003.[3] Additional key findings were that land degradation was not primarily a problem of drylands, as had long been proposed, but was a worldwide phenomenon.

In a 2015 follow-up study, the global estimate of degrading land was revised down slightly to 22%, with 14% of land showing some improvement [72]. Also in 2015 FAO/ITPS[4] claimed that 33% of the land

---

[2]The term 'soil degradation' can also, more generally, be understood to encompass the other two threats listed here, including soil sealing.

[3]The authors made no attempt to merge the data because of the contrasting methods of the separate studies and the fact that the GLASOD data were unverifiable, but one of the authors opines that: 'It is not unreasonable to judge that all land now under anything less than natural climax vegetation is degraded in terms of biodiversity and stored carbon and nitrogen, perhaps 66% of the world's land surface and rising'. (D. L. Dent 2020, personal communication).

[4]The Intergovernmental Technical Panel on Soils.

**Table 1.** Types of anthropogenic threat to soil or land.

1. *soil degradation* (i.e. harm to soil, other than soil sealing, which is covered in 2 and 3 below)

2. *conversion of natural ecosystems* or other semi-natural and uncultivated land to another form of land use, such as agriculture, forestry, urbanisation or industry, including mining or energy infrastructure (conversion is also known as 'land take')

3. *conversion of farmland* to urban or industrial use[10] [69] (agricultural land take)

was degraded, and could reach 90% by 2050 [73]. A 2016 study produced a global figure of 30% degraded land [74]. A UNCCD analysis in 2017 revised the total to 23% [75], but states that: '…over the last two decades, approximately 20 per cent of the Earth's vegetated surface shows persistent declining trends in productivity'. [76]. In 2018 IPBES[5] concluded that 75% of the Earth's land surface is either transformed or degraded to some extent by human activity, impairing 3.2 billion livelihoods and costing 10% of the annual global gross product, whereas the benefits of restoration are typically 10 times higher than the costs [77].

Most countries now have some form of regulation addressing soil degradation, typically embedded within their agricultural legislation, and in a number of these countries, the law is strictly enforced. In many cases, soil policies go beyond prohibitive and punitive laws dealing with acute soil degradation, and incorporate support and incentives to foster improved long-term soil health and protection [78]. Compared to land use conversion, soil degradation, even though it may be the result of deliberate law-breaking, is generally recognized as unintended and unwelcome, so there are fewer obvious social pressures to oppose such laws. The global trend has been greater governance, but the cost of remediation can be prohibitive, whether for those held responsible or those who bear the cost, e.g. taxpayers, so circumvention is a constant risk [79]. China stands out as the country probably most afflicted by soil degradation, of every kind, but by contamination in particular, on a scale that appears irreparable and unaffordable, requiring more than the world's entire wealth to remediate, according to *The Economist* [80]. Nevertheless, no country seems more acutely aware of this or determined to address it than China itself [81,82].

### 3.4.3. Conversion of natural ecosystems

Natural ecosystems or semi-natural landscapes, that is land that is uncultivated and largely unmanaged [83], are ubiquitously valued for a variety of reasons: their importance for Indigenous peoples and other local communities; their aesthetic characteristics and recreational potential; their educational and scientific worth; and, with increasing urgency, their vital ecosystem service benefits. Aside from largely uninhabitable areas, this category of land includes forest, wetland, native grassland (e.g. prairie or steppe) and heath-/moor-/peatland. Globally the primary threat to such land is from conversion to agriculture, responsible for 80% of deforestation according to WWF [3], but the full picture is confused by claims of both overestimation and underestimation [84].

The dynamics of forest change are also complicated by temporary deforestation and forest gain, e.g. via afforestation, reforestation or regrowth on abandoned farmland. A recent study [85] using satellite imagery estimated that between 2001 and 2015, only 27% of global tree cover loss was permanent land use change for commodity production, i.e. large-scale agriculture, mining and energy infrastructure. Urbanization accounted for less than 1%. Impermanent changes included forestry (26%), shifting agriculture (24%) and wildfire (23%). Regional contrasts were stark with permanent deforestation accounting for most of the tree loss that occurred in Latin America and Southeast Asia, mainly driven, respectively, by ranching and oil palm plantations [85]. Indonesia and Malaysia stand out as areas of increased deforestation. The rate of loss had declined markedly in Brazil, but since the change of regime in Brazil in 2018 forest destruction is reported to have increased again dramatically [86]. Other regions have relatively low rates of permanent deforestation, with temporary disturbance associated with managed forestry and, in North America, Russia and Australia, wildfire. A few countries have achieved a net forest gain.

[5]The Intergovernmental Science-Policy Platform on Biodiversity and Ecosystem Services.

Natural ecosystems tend to be better protected in the higher income countries, but historically this was largely because they lack the natural resources that facilitate intensive land use and often present severe obstacles to development. Very little 'productive' wilderness remains, and some parts of the tropics have reached or are close to this point [87]. In sub-Saharan Africa, even where natural forest exists near settlements, it is difficult and expensive for farmers to encroach on such land, which also often has poor soil [88]. The alarm has also been raised concerning the impact of rising and uncontrolled urbanization on biodiversity in Africa [89]. In 2009 Rockström *et al.* [90] suggested we are approaching the limits of planetary land use conversion. This makes the preservation of natural ecosystems worldwide all the more urgent, especially given their far-reaching environmental benefits, such as C storage, biodiversity and soil and water conservation. A combination of agricultural support and incentives, and rational territorial planning can raise farm incomes, improve food security and reduce loss and expansion of farmland [91–95].

Nevertheless, conversion of natural lands to agriculture is rarely as drastic or permanent as urbanization, and many countries are increasingly rewarding landowners, land managers or farmers, for example via payments for ecosystem services (PES), to implement more environmentally benign land use and practices [96]. Putting aside the question of how effectively each government responds to the loss of natural ecosystems, the issue is relatively unambiguous and virtually every country on Earth, theoretically at least, protects such land with environmental laws and policies. Furthermore, soil protection *per se* is rarely the driving force behind such legislation, which tends to be framed in terms of ecology, biodiversity or watershed protection, or occasionally preservation of landscape or culture.

### 3.4.4. Conversion of farmland to urban or industrial use

The vast majority of land consumed by urban and industrial expansion throughout the world is agricultural land and this type of threat, which encompasses much soil sealing, is arguably the least reversible and the most profound of these three types of anthropogenic threats. Furthermore, for sound socioeconomic reasons, urban settlement has historically developed near, and often surrounded by, prime farmland. Hence it is frequently the best quality land that is most vulnerable to conversion [97–100]. Peri-urban farmland is particularly attractive to developers because unless it is subject to strict green belt or zoning laws, it lacks the environmental protections of natural ecosystems, is usually cheaper to develop than brownfield sites [101] and tends to be conveniently situated in terms of existing facilities and infrastructure. In the UK there has been a gradual weakening of the protection afforded to BMV land, despite an official policy to the contrary [102,103]. Furthermore, a study of 25 EU countries found that the land most at risk was that slightly further out from city boundaries, which typically has more fertile soil than that closer to the city [104], while, interestingly, the influence of CAP subsidies reduced the rate of land take in general.

Although agricultural land take is rarely covered by either environmental or agricultural legislation, an exception to this is where farmland is protected for its above-ground biodiversity value, e.g. high natural value (HNV) [104], but this is not the same thing as protecting land for its intrinsic SES value. The last resort for farmland preservation, if it occurs at all, is usually some form of spatial or territorial planning, but the primary motive for such planning regulation is typically urban expansion driven by population and commercial pressure, rather than rural protection [105,106]. There is a distinct lack of robust legal instruments to prohibit or restrict agricultural land take and the approach taken varies considerably between and within countries, producing policies that are frequently complex and sometimes ambiguous or conflicting [40,106–113]. Added to this is the prevalence of 'informal' governance of planning regulations in many low- and middle-income countries [111,114,115].

The mere potential for urban development, even without planning permission, typically increases the value of farmland by, for example, fivefold in South Korea [116], sixfold in Morocco [117], nearly 10-fold in New Zealand [107] and even by orders of magnitude more than this: 100-fold in Ghana over 10 years [118] and similar multipliers in Britain and Japan [119]. This creates enormous commercial pressure for landowners to sell, especially in lower-income countries where agricultural livelihoods may be precarious and lack government support [120], but do not necessarily leave poorer farmers with any long-term benefits. With the exception of a few who may be able to exploit new urban markets, most farmers will use this temporary windfall to pay off debts and become landless farmers or seek other occupations [121]. The prospects of re-investing in cheaper land to cultivate are greatly diminished by the fragmentation of holdings and the creeping outward shift of the agricultural zone onto lower quality land [121].

It is worth addressing arguments advocating, or not opposed to, farmland conversion. Visser takes this logic to a purely economic extreme of treating natural resources as tradeable commodities or assets and even suggests that land markets would benefit from greater scarcity of farmland [122]. Satterthwaite *et al.* [123] argue that the world has abundant arable land and a global economy from which urban populations can import food without depending on their own agricultural hinterland. This suggestion presupposes reliable food supplies and reliable trading partners, yet geopolitical instability and climate change increasingly threaten global food security [124], not to mention the additional risks associated with a global pandemic.

In defence of conversion one might argue that much farmland, often government-owned, has been abandoned or under-used for various reasons, such as de-population of rural areas or, as in the case of the former Soviet Union, incomplete land reform [125], and that this land could be recultivated. There may be limited capacity for rehabilitated farmland to compensate for conversion elsewhere [126], but this raises other issues. One reason for abandoning farmland is the difficulty in extracting an adequate livelihood because the land is degraded [127] or marginal [125]. Such land will be intrinsically less productive so, even if farmers have the means and incentive to restore it, a greater area might be needed to achieve requisite returns. Where productive land has been abandoned for socioeconomic reasons, such as a lack of infrastructure in remote areas which could be remedied, recultivation is feasible [128,129]. However, abandoned farmland presents another global narrative, for it is swathes of such land, in Russia, US, China, Australia, Latin America and elsewhere, that have inadvertently facilitated a global pattern of reforestation and rewilding [126] that goes some way towards buffering the effects of deforestation elsewhere [126,130].

Some advocates of farmland conversion in high-income countries like the UK employ primarily socioeconomic arguments, for example with respect to a scarcity of land for housing in the London Metropolitan Green Belt (MGB), arguing that the MGB policy is rigid and anachronistic in protecting all farmland regardless of its environmental value, while constraining suburban gardens which can harbour more biodiversity [54,131–133]. However, residential land undergoes much sealing and topsoil removal, while the biodiversity of its gardens, though potentially of great value, is entirely arbitrary. Farmland, on the other hand, remains largely unsealed and, given appropriate policy incentives, such as PES, can become more biodiverse and provide greater ecosystem services, as has been a gradual trend in the UK since the late 1980s [102], and looks to continue with the proposed new environmental land management (ELM) schemes [134]. Where such authors have a much stronger case is in criticizing the binary distinction between green belt land, where all farmland is protected regardless of its agricultural quality, and non-green belt rural land where prime farmland should be protected but is often developed [132].

The MGB, one of the oldest green belt zones in the world which constrains one of Europe's largest cities, is continually under strain from pressure for housing and transit development. In spite of strict regulations, planning permission is devolved and inconsistent, and much development leapfrogs onto agricultural land beyond. Similar but younger zones have yet to experience such pressures, but in the case of the Ontario Greenbelt, for example, this is partly due to an approach which is arguably more integrated and enlightened, incorporating green infrastructure, principles of environmental, economic and social sustainability, greater public participation and underpinned by a legal framework, centrally owned and controlled by the provincial government, yet both robust and flexible [35,131].[6] Farmland preservation in Ontario is not simply a matter of locally *ad hoc* prohibition in the teeth of fierce opposition from planners and developers, but is integrated into a holistic policy that includes agricultural support, local food marketing, employment opportunities, recreation, tourism and ecosystem services in a regional context [35,131,135].[6]

The view that urbanization represents economic progress [123] has led many administrators and politicians, especially in lower-income countries, to embrace urban expansion policies with enthusiasm [136]. However, at the local level, there are serious concerns, even within government [108], that the wider implications of farmland loss are increasing food prices and imports, escalating rural poverty, land degradation and conflict [108,120,121,137–139], as well as the less appreciated impact of soil sealing [100,108,140], including poor sanitation, flooding and pollution, which disproportionately affect rural communities [141,142]. Much of the land consumed by urban sprawl is common land on which many rural communities depend [108]. Developers and officials or agents may benefit from these transactions [143], but the cumulative negative impacts are felt by the whole community. Those highlighting these issues include local academics [120,139,144], journalists and blighted farmers taking

---

[6]A. Shortly (Greenbelt Foundation, Toronto) 2020, personal communication.

their grievances to the courts, if they can [143,145]. Social inequalities drive many to facilitate the very land use changes from which they gain the least, sometimes converting natural ecosystems to replace what they have lost. There is evidence that appropriate government support reduces farmland sale for non-agricultural development, and enhances food security and biodiversity [104,105,146] so any attempt to prohibit or restrict agricultural land conversion, must be accompanied by economic support and viable alternatives. No amount of legislation or governance will succeed without this.

While all three types of anthropogenic threats to land and soil listed in table 1 above endanger ecosystem services, from a law and policy point of view, compared to soil degradation and natural ecosystem protection, agricultural land take is often 'out of sight and out of mind'.

### 3.4.5. Cross-cutting issues

A few academics argue that land conversion or degradation may be justified in some cases, on the basis that the economic benefits may outweigh the environmental costs [147], but despite the best efforts of environmental economists, such costs remain intangible [148] and often profound. Furthermore, such benefits are not necessarily sustainable and are ultimately dependent on ecosystem services that economists have traditionally treated as 'free' and, more to the point, inexhaustible [90]. This market-driven worldview has been strongly criticized for at least 50 years [149,150] and arguably much longer [151], underscoring what is becoming known as the Anthropocene crisis [152]. Economic benefits also accrue unevenly and not necessarily in the national interest. The escalating profits from development can foster an 'unholy alliance' between the public and private sector that rewards elaborate circumvention of the law [105,153] or outright infringements [66,75–77,112,144]. However, it is also important to appreciate that what may appear to outside observers to be infringements may, in some cases, simply be the result of longstanding traditions of informal governance or customary tenure [143,154], or simply a lack of institutional capacity [123].

Mining, energy infrastructure and other forms of industrial land use represent a very small proportion of the land loss or degradation overall [155], although this is increasing [76] and cannot go unmentioned because of their extreme impact in some locations. In western Ghana, for example, gold and diamond mining has had devastating effects on rural communities and natural ecosystems [156,157]. This activity includes both corporate mechanized extraction and illegal and artisanal hand-digging, a form of low-input mining which also provides construction material. These can involve all three types of anthropogenic threats to soil listed above, going far beyond the loss of land, and including local water contamination and depletion, illegal logging, extensive soil and subsoil removal and sometimes violent land disputes. The highly lucrative and largely unregulated context in which these enterprises operate tolerates and even encourages infringements of the law, from which corporations operating legal concessions are not exempt [157].

# 4. The jurisprudence[7] of soil and land

## 4.1. The implementation of soil legislation

There are essentially two components to legally protecting or preserving soil resources: (i) applying a method of evaluating and prioritizing areas or bodies of soil and (ii) implementing the requisite (and effective) legal instruments and governance structures. Both of these components exist throughout the world, but only within a fragmented and, in some cases, theoretical patchwork of initiatives. Before dealing effectively with the issues of which forms of governance structures or legal mechanisms might be most applicable to soils, one must consider what criteria to apply. This is essential whether focusing primarily on specified spatial areas of land or on the functional aspects of soils in relation to their current status or use. However, preceding even all of this is a minefield of ambiguous terminology to navigate.

## 4.2. Semantics and legal terminology

When legal documents and policies refer to the way in which soil is affected by those managing the land it occupies, and hence the possible harms it may encounter, the word 'protection' is the term most widely

---

[7]Jurisprudence here refers to both legal theory and legal systems.

used. Many if not most countries have some form of explicit or implicit soil protection policy in place, but this is more often than not subsumed within other legislative and policy domains, for example, the environment, agriculture or spatial planning. Soil legislation is commonly sub-divided further into categories of protection, reflecting the severity of particular problems within a given country or the gravity attached to the problem by its authorities. For whatever reason, certain soil laws are almost always framed in negative harmful terms and others in positive or remedial terms. Hence there are soil pollution or contamination laws, but hardly ever soil erosion laws. Instead, there are soil conservation laws or occasionally, in a similar vein, soil improvement laws.

Harder to pin down is the terminology applied to the conversion (effectively the loss) of agricultural or other undeveloped rural land to non-agricultural use which typically incorporates substantial additions of infrastructure (e.g. housing). In this domain several terms are used, some in the negative or destructive sense, and others in the positive or protective sense. It can be argued that the term land conversion, though technically describing the act of change or loss, is neutral, because unlike the unintended consequences of the harms done to soil, this refers to a deliberate act with an intended outcome and various beneficiaries.

The many approximate synonyms for rural land conversion with a more negative connotation include urbanization, urban spread/sprawl/expansion/encroachment, land take/consumption/loss, long-term land cover change and (in France) artificialization [158]. Other phrases commonly used in connection with rural land conversion are land competition and fragmentation, of farmland in particular, because this is one of the ways in which urbanization becomes self-perpetuating [159]. Soil sealing, the permanent covering of soil, for example by concrete or tarmac, is a phrase widely used in conjunction with land take, because they typically occur together, but the meanings are distinct. While land take refers to a change of use which, by definition, usually entails some loss of land resource from its former use, sealing constitutes a much more permanent and intrinsic alteration of the land surface, typically with more far-reaching consequences for soil functions, especially drainage and potentially severe effects on biodiversity [100,160,161]. Nevertheless, land take almost always involves some degree of sealing, and where it does, that constitutes a double impact—the spatial loss of land resource and the additional degradation of the ecosystem services that that resource provides in the round. However, sealing also occurs independently of more general land take, even in highly protected areas, for instance in the form of roads [160]. There are a few examples of unsealing, as a form of compensation for sealing elsewhere, but it is notable that these did not equate to total restoration [162].

On the positive land-saving side of the same lexical coin, the term predominantly used in North America, and in many countries, is land preservation [163,164], but the terms land conservation or land protection are used almost interchangeably with it in the US [165,166] and throughout the world. In most cases, these phrases are applied to specific bounded areas, at various scales, which could mean a protected zone or alternatively a cluster of holdings, a single estate or even one field. The expressions 'no net land take' or 'zero land take', which have appeared in EU documents in recent years [101,167], are similar in principle, but refer to overall quotas or targets, with the additional challenges relating to relevant spatial scale and overall quantification. In contrast 'land sparing' is used in juxtaposition to 'land sharing', referring to the concept of safeguarding biodiversity, either by sparing natural lands from agricultural use (land sparing) or by incentivizing farmers to support more biodiversity (land sharing) [168].

Globally words or phrases used to mean approximately the same thing as land preservation can sometimes have other connotations. Land conservation has historically been most strongly associated with nature conservation, in the sense of the land that is not recognized as developed or cultivated and has some form of protected status. Such land is often threatened by agricultural incursion at least as much as by urbanization. Land conservation is not necessarily synonymous with 'soil conservation', a term used professionally and more widely to refer specifically to the prevention of soil erosion with respect to land management practices.

Land protection is often synonymous with land preservation, but in some examples of legislation it is intended to mean soil protection, that is *in situ* safeguarding. Soil protection is sometimes intended to mean or encompass land preservation. These semantic issues may sometimes result from translation because the full texts of many laws appear only in their original language and script [4]. Occasionally countries have all-embracing 'soil protection' policies which include the concept of spatial land preservation in context [169,170] or conversely all-embracing 'land preservation (or protection)' policies which include the concept of soil protection [5,170]. In some cases, this blurring seems to be deliberate, such as when land preservation is promoted as a means to protect SES [5,160].

Pragmatically soil scientists may need to accommodate themselves to the language of policy and spatial land use planning in order to engage with the process.

Finally, it is important to note that while many legal instruments have been designed specifically to protect only prime (i.e. highly productive) farmland, this is not always the main driver of land preservation or protection, which may prioritize other factors such as landscape, heritage or SES. In this article, this distinction is highlighted by using the term prime farmland preservation (PFP) where appropriate.

## 4.3. Putting a value on soil

At the root of our current environmental crisis is that humanity has traditionally treated abundant natural resources as free goods and services. Economists and accountants are no exception: cost-benefit analyses include tradeable assets and products, but routinely exclude the ever-present prerequisites of life. Furthermore, traditional cost-benefit analysis (CBA) is short-termism by definition because it usually does not even consider impacts outside a 25-year window [171]. When a tonne of soil or water is traded, the allocated price is normally derived from the aggregate cost of acquisition, processing and delivery, with no attempt to assess the intrinsic value of the substance. This applies not only to the soil as an intrinsic good, but also to some extent to its economic potential, i.e. the land that contains it will have a market value and that value will be related to the land's productivity, yet unquantified and uncosted SES have far-reaching economic implications. Even in a society without monetary currency it would be relatively straightforward to calculate the economic value of a given volume of soil, in conjunction with the land it occupies, purely in terms of its capacity for life-sustaining primary production; ask any farmer. Although we now have modern techniques to assess the productive capacity of land resources, a conceptual process of 'following the money' has been at the heart of most farm or settlement emplacement decisions since the Neolithic.

For at least 80 years soil survey and land evaluation have been the traditional tools at our disposal for assessing the productive and economic potential of an area of land, alongside any limitations or hazards it may present. A range of soil properties is recorded alongside other local environmental data and, where appropriate, socioeconomic data. This data are combined to grade the capability or suitability of each parcel or zone of land according to specified use criteria. The modern terminology of multicriteria decision analysis (MCDA) is more a change of style than substance, to reflect the much greater reliance on information technology, especially GIS [172]. In the above approaches 'criteria' is the operative word. Where these methodologies differ from analogous *ad hoc* historical techniques, is in their capacity to distil decades of practical and scientific observation to apply much greater predictive accuracy and precision to the land use decision-making process [173].

Land evaluation is not the same thing as land valuation, though they have always been interrelated and both are relevant to spatial planning. Land evaluation methodologies were never intended to calculate the true, i.e. total and holistic, economic value of an area of land or a volume of soil (which are themselves two very different things), but rather the capability or suitability of distinct parcels of land for specified forms of primary production, such as crops, pasture or forestry. The purpose is for land use decisions, albeit often with profound economic implications. The purpose of land valuation, however, is generally to set prices for landowners and taxes for governing authorities, and has always primarily been based on location, which admittedly is historically bound up with soil quality and land use, but with a myriad of other factors too.

In contrast, the soil has always provided a variety of extrinsic benefits beyond primary production which were recognized and appreciated long before we referred to them as ecosystem services. To what extent our ancestors valued (or devalued) specific areas of soil or land for reasons other than primary production, or obvious physical location, is not always tangible, though it is clear that many indigenous communities have developed deep understanding of, and spiritual connection to, the land and soil on which they depend [83]. A wider all-embracing appreciation of soil and a few attempts to evaluate it in that context date from the 1960s [174]. This concept has gathered momentum and, more recently above all, with respect to C.

While it may seem too neat to earmark the last decade of the previous millennium as a critical turning point in time, it seems inescapable that this period represented an important paradigm shift in our collective understanding of the role of soils. The 1990s shines out as a lightbulb moment when soil science and climate science came together. Although the role of C in the soil has been studied for at least 150 years and its place in the C cycle appreciated for much of that time, it was only in the 1990s that a flurry of papers explicitly linked soil C fluxes to GHG concentrations in the atmosphere and

climate change [175–177] which, after several decades of growing evidence, had started attracting worldwide media attention since 1988 [178].

At the onset of the new millennium soil scientists have more than ever before started to deconstruct and articulate the critical importance of SES [179,180]. Experts on soil law have responded accordingly, emphasizing that the value and status of soil far exceeds its role in agriculture [181]. Taking this a step further, several groups of researchers, building on earlier attempts to evaluate ecosystem services [182] and characterizing soil as a form of natural capital, have developed tentative frameworks and quantitative methods to enumerate and evaluate SES in the fullest sense [183–187], sometimes using case studies to apply monetary values [188]. To what extent this is entirely feasible or even desirable, is debatable [174], especially with regard to qualitative or ethical issues, but such approaches may help policymakers and planners apply more meaningful criteria and priorities to the governance of soil and land protection.

## 4.4. Categories of soil governance

Juerges *et al.* [65] summarize the types of instruments applied to soil governance (i.e. regulatory, economic and so on) and the levels at which these operate, from global to local. In this section, we present a very different cross-cutting breakdown, based on issues and intended outcomes. One can identify three broad categories of governance approaches applied to preserve or protect soil or land, usually at a national or subnational level, with various implementation mechanisms, set out in table 2 and elaborated further below:

The first category comprises regulations and laws to *prohibit* (or guidance to discourage) *certain actions* by landowners that may degrade land, e.g. pollution, sealing, construction, stubble burning, tree felling, or cultivation methods leading to soil erosion, acidification, salinization, compaction and so on; whereby best practice may be either:

(a) Enforced by penalties for infringement or
(b) Facilitated by financial support, e.g. farming subsidies, PES or C credits

The second category comprises *restrictions* (or guidance) *on development involving conversion of use*, either to protect terrestrial natural resources (for the benefit of any or all of primary production, watershed management, biodiversity, landscape or cultural or historical value) or simply to retain a certain quota of agricultural or forested land within a state or designated region; whereby land conservation or preservation may be:

(a) Enforced by laws based on zoning, e.g. National Parks, nature reserves, green belt, urban growth areas, sites of special scientific interest (SSSI) and heritage landscapes such as the UK Areas of Outstanding Natural Beauty (AONB) or UNESCO World Heritage Sites;
(b) Enforced by more general laws (or encouraged by guidance) not specific to the protected areas described in 2(a), based on *ad hoc* site evaluation, such as an environmental impact assessment (EIA), and thereby affording protection to sites of ecological or landscape value, or prime farmland (e.g. based on ALC or LESA criteria);[8]
(c) Enforced by the public acquisition of land, such as land trusts;
(d) Facilitated by financial support, e.g. using LESA to obtain purchase of development rights (PDR) conservation easements (US).

The third category encompasses *generic incentives* to preserve land or to enhance its ecosystem services value (or disincentives to develop, such as the removal of subsidies), in the form of:

(a) Taking land out of agricultural production, purely for the purpose of conservation, e.g. EU set-aside, US land retirement, rewilding;
(b) Converting an area of intensive agricultural production (arable cropping) to more extensive agricultural production (such as pasture, silvopasture or agroforestry) or forestry;
(c) Declining to develop or intensify the use of uncultivated land that could otherwise be legally developed or intensified, in order to maintain ecological or ecosystem services value.

---

[8]A legally binding example is the Indian Prohibition on Conversion of Agricultural Land for Non-Agricultural Use (no. 16 of 2010).

**Table 2.** Categories of governance approaches and mechanisms to preserve or protect soil or land.

| | category | description |
|---|---|---|
| | 1 | *regulations to prohibit (or guidance to discourage) certain actions* |
| | (a) | —enforced by penalties |
| | (b) | —facilitated by financial support |
| | 2 | *restrictions on development involving change of use* |
| | (a) | —enforced by zoning laws |
| | (b) | —enforced by laws (or encouraged by guidance) based on *ad hoc* site evaluation |
| | (c) | —enforced by public acquisition of land |
| | (d) | —facilitated by financial support |
| | 3 | *generic incentives to preserve land or to enhance its ecosystem services value* |
| | (a) | —taking land out of agricultural production |
| | (b) | —converting intensively farmed land via intensification |
| | (c) | —declining to develop land or intensify its use |

## 4.5. Overlapping categories

The above categories are presented as a useful framework for the analysis of soil governance. There are of course many instances of overlap. For example, regimes that are not targeted specifically at soils can still fall within the categories above. There are also numerous cases where soil governance instruments span a number of the mechanisms listed above. The examples below illustrate these points.

Environmental impact assessment (EIA) of projects, strategic environmental assessment (SEA) of plans and programmes, and related types of assessment, may apply to some of these categories, especially categories 2(a) or 2(b), and can be helpful in drawing attention to threats to natural resources, but there can be limitations in their application to soils. In the US, for example, EIA is applied primarily for federally funded projects (T. Daniels (Professor of City and Regional Planning, Weitzman School of Design, University of Pennsylvania) 2020, personal communication) which is partly why the NRCS developed the LESA system for smaller projects targeted at farmland. In the UK EIA is more widely applicable, but generally only where part of the area intended for development is uncultivated or only partially cultivated and hence might not be invoked where only arable land is at risk [189]. EIA regulations vary slightly in Scotland, however, where the site being assessed may consist exclusively of farmland where it exceeds 200 ha. The Scottish EPA also provides specific guidance on the application of SEA to soils [190].

The broad North American planning term land preservation covers many of these categories with the main focus on zoning [163,191]. Every state and city has its own subset or version of land preservation laws and regulations which are varied and complex. One example is a 'conservation easement' whereby a landowner is bound by a covenant set by the government or some other organization and in return receives an incentive, such as a tax rebate; this could fall under categories 1(b), 2(d) or 3(a–c).

Apple Valley City, Minnesota has adopted a natural resources management plan (NRPM) which essentially constitutes a 2(b) type mechanism via a permitting system, but incorporates elements of 1(a) because it could apply to a single aspect of a development. It differs from most local planning laws and regulations by applying the broad principle of environmental impact to every development [192].

There is an interesting example of environmental scientists trying to set a legal precedent as expert witnesses in a 2(b) type scenario in New Zealand in 2011, using soil natural capital and ecosystem services arguments to prevent urban development on horticultural land. The lawyer representing the developer argued that the only measured ecosystem service of this soil was food production. The judge, while declining to engage in the natural capital debate, nevertheless upheld the local authority decision not to allow development in favour of the 'holistic' argument to protect natural resources which, from the point of view of the scientists, came to the same thing [193].

# 5. The modern governance of soil resources worldwide

## 5.1. A global environmental awakening

While the piecemeal governance of soil use is almost as old as human history, formal supranational global oversight of soil resources is only approximately 50 years old. In terms of the natural environment, the early 1970s stand out as a period when national policies, international initiatives and academic analyses of global problems coalesced in an unprecedented step change in attitudes; 1970 saw the conception of Earth Day, initially in the US but later worldwide, and consequently the creation of the US Environmental Protection Agency, an institution which also became replicated worldwide. In 1972, the Club of Rome think tank, published the highly influential *Limits to Growth*, with its prediction of societal collapse in the twenty-first century, and frequent references to soil degradation [194]. This period also saw the emergence of multidisciplinary stakeholder groups joining forces for sustainable development [195].

## 5.2. Emergence of soil in modern international law and policy

The 1968 *African Convention on the Conservation of Nature and Natural Resources* (the 'Algiers Convention') was a continent-wide treaty devoted to natural resource protection that included a reference to the soil. Although it lacked the resources and institutional framework to be particularly effective, the Algiers Convention was nevertheless regarded as a milestone in international environmental law [196]. At around the same time FAO was publishing several of its soils bulletins every year, including two landmark studies in 1971, on land degradation [197] and on legislative principles of soil conservation [198]. The latter was a relatively concise set of guidelines that any nation could use as a template for creating or improving its own legal framework for governing soil. The 1972 UN Conference on the Human Environment (UNCHE) in Stockholm was effectively the first major international conference on environmental protection and sustainable development, and has been referred to by Boer *et al*. as marking the: '…first phase of international soil protection law' [199,200].

In 1972, the Council of Europe created and adopted the *European Soil Charter,* a concise but relatively holistic set of soil protection aspirations [201] which was non-binding, but regarded nevertheless as the first international legal instrument dedicated specifically to protecting soils [199,202]. In 1977, the UN held its first Conference on Desertification (UNCOD), and produced its *Plan of Action to Combat Desertification* (PACD) [203]. In 1981, FAO adopted the *World Soil Charter* [204] and in 1982 UNEP developed a global soils policy [205] and the IUCN/UN *World Charter for Nature* explicitly referenced soil [206].

## 5.3. Soil as a feature of common concern

In 1987, the UN commissioned the Brundtland report, *Our Common Future*, which reinvigorated the debate, again stressing the severity of soil degradation, and annexed a summary of proposed legal principles for environmental protection and sustainable development [207]. In order to address these issues financially, in 1991 UN agencies and the World Bank created the Global Environment Facility (GEF), as a prerequisite for the 1992 UN Conference on Environment and Development (UNCED) or Rio Earth Summit [208], a milestone event. From this emerged the UN Commission on Sustainable Development (UNCSD) and three legally binding international treaties: the 1992 *UN Convention on Biological Diversity* (CBD) [209], the 1992 *UN Framework Convention on Climate Change* (UNFCCC) [210] and the 1994 *UN Convention to Combat Desertification* (UNCCD) [211], which came into force in 1993, 1994 and 1996, respectively (the Rio Conventions). There was appreciation of the interrelationships linking all three conventions to land degradation [212], but funding for the latter was mainly tied to the UNCCD which was always the weakest and poorest of the three due to donor scepticism [52]. The UNCCD has been called the first and only 'legally binding global agreement directly dealing with the promotion of bio-productive land' [199], although it contains little in the way of substantive obligations, and formally applies only to 'arid, semi-arid and dry sub-humid areas' [213]. Nevertheless, UNCCD is gaining momentum as a focal point for efforts to address the global land degradation neutrality (LDN) target [203,214].

The interconnected nature of different environmental problems, as well as their linkages to socioeconomics and SLM, was taken a step further at the 2012 UN Conference on Sustainable

Development (Rio + 20). The UNCCD brought the concept of zero net land degradation (ZNLD) for adoption at Rio + 20, reflected in the outcome document, *The Future We Want* [203], and this later became embedded into the 2015 SDGs as the global LDN target in SDG 15.3 [203,215]. Land degradation was finally being acknowledged as a global problem that was intrinsically linked with climate change, biodiversity loss and poverty. The concept of ZNLD/LDN was that every effort should be made either to prevent further degradation or, where this is not possible, to rehabilitate equivalent areas of degraded land elsewhere [215]. The importance of land and soils is gaining more prominence in the context of the CBD [209], the UNFCCC [216] and related *Paris Agreement* [217]. The parties to the CBD have addressed soil biodiversity via an international initiative [218] and a global report [219]. It remains to be seen how soils will be reflected in the Global Biodiversity Framework to be adopted by the CBD COP 15, currently re-scheduled to take place in China (in phases spanning 2021 and 2022) especially given China's rapid advance in the field of soil science.[9]

At its first Plenary Assembly, at the FAO Headquarters in Rome in June 2013, the Global Soil Partnership (GSP) created the Intergovernmental Technical Panel on Soils (ITPS), made up of 27 soil experts representing all the regions of the world. [73]. The GSP also proposed 2015 as the International Year of Soils (IYS) and the annual observance of a World Soil Day (on 5 December), both of which were adopted at the 68th UN General Assembly later that year [220]. Under the auspices of the GSP the *Revised World Soil Charter* [221] was adopted by FAO members in 2015, and the *Voluntary Guidelines for Sustainable Soil Management* (VGSSM) [222] were endorsed by the FAO Council in 2016.[10] The work of the ITPS, UNCCD Science-Policy Interface (SPI), IPBES, the IPCC, and in particular the publication of the 2015 *Status of the World's Soil Resources* [73], the 2017 *Scientific Conceptual Framework for Land Degradation Neutrality* [223], the 2018 *Global Land Degradation and Restoration Assessment* [77] and the 2019 *Report on Climate Change and Land* [51], by each, respectively, stand out as key influences in the soil governance narrative. The significance of the science-policy interface is increasingly important in determining the scope and extent of legal obligations [224].

In 2015, to coincide with UNFCCC COP21, the French government issued a bold entreaty to the global community to raise average soil organic C (SOC) levels by 0.4% (the '4 per mille Soils for Food Security and Climate' initiative), to offset annual global C emissions into the atmosphere, as well as improving food security [225,226]. The choice of SOC as the target was fundamental and twofold, because it represents both a means of accumulating and sequestering atmospheric C, and the most widely accepted measure of soil health or productivity. The 4 per 1000 Initiative, to which many countries have signed up, has succeeded in highlighting the critical role of soil and agriculture in climate change mitigation and adaptation. There has also been criticism [227], especially of the feasibility of the initiative and underpinning data. While the authors have defended the science, with caveats, they also stress that 4 per 1000 was never intended to be a precisely calculated solution, but a positive, politically driven and symbolic aspirational target [228]. Other soil scientists agree with them, judging that the initiative would be a technically feasible, 'no-regret' and indispensable climate action [229].

## 5.4. Regional and sectoral developments

At the regional level the Algiers Convention was revised in 2003 to become the *Revised African Convention on the Conservation of Nature and Natural Resources* (Maputo Convention), but only entered in force in 2016 [230]. In 1998, the Alpine Soil Protocol was the first legally binding treaty expressly devoted to the soil, entering in force in 2006 [231]. For both the Maputo Convention and the Alpine Soil Protocol, implementation has not yet matched aspiration, but both serve as useful focal points for awareness raising regarding soil conservation, management and best practice. The ASEAN Agreement on the Conservation of Nature and Natural Resources, signed in 1985, included an article specifically on soil, but it has never come into force [199].

The Council of Europe revised the *European Soil Charter* in 2003 to become the *Revised European Charter for the Protection and Sustainable Management of Soil* [232]. In contrast, in the European Union, agreement on a soil instrument resembles a triumph of hope over experience. In 2006, the European Commission presented its *Thematic Strategy on Soil Protection* and presented a *Proposal for a Soil Framework Directive* to place healthy soil on a par with clean water and air [233]. The proposal was

---

[9]According to the SJR International Science Ranking website China ranks second only to the US in soil science (https://www. scimagojr.com; accessed 28 April, 2021).

[10]http://www.fao.org/3/bl813e/bl813e.pdf.

eventually withdrawn in 2014, because five-member states were not willing to agree to strengthened EU-wide legislation addressing soil sealing and liability for contaminated land [234]. A new EU Soil Strategy is planned for adoption in 2021 as a key component of a European Green Deal (EGD), aimed at making Europe the first climate-neutral continent by 2050 [235]. The European Parliament has called for a binding legislative framework for soils. [236]

## 5.5. Sustainable development goals (SDGs)

All countries have signed up to the seventeen SDGs [237]. Land and soil have profound relevance for most, if not all, SDGs, though only a relatively small number of SDG targets make specific reference to them [237,238]. In particular *SDG2: Zero hunger*, aims in target 2.4 to ensure sustainable food production systems and resilient agricultural practices that increase productivity and production, help maintain ecosystems, strengthen capacity for adaptation to climate change, extreme weather, drought, flooding and other disasters and progressively improve land and soil quality; and *SDG 15: Life on land*, aims to protect, restore and promote sustainable use of terrestrial ecosystems, including in target 15.3 combatting desertification, restoring degraded land and soil, and striving to achieve a land degradation neutral (LDN) world. SDG targets 3.9, 6.4, 6.5, 12.4 and 14.1 also have particularly obvious direct relevance to soils. [238], and land and soil are fundamentally related to many other SDG targets and indicators, including those for SDG 13 on urgent climate action. While the SDGs themselves are not-legally binding, they do in some respects echo or amplify existing and emerging binding international commitments.

## 5.6. National soil legislation

Last year FAO and UNEP jointly published a key document, which is candid in its acknowledgement that implementation of international commitments can be difficult to enforce and monitor, especially when they are purely aspirational rather than legally binding [214]. The report provides many examples of innovative and progressive legislation in relation to land and soil, though questions still remain in relation to enforcement [68]. On a global scale the most urgent action needed may relate to category 2(a) in table 2—restrictions or guidance on development involving conversion of use, to protect terrestrial natural resources and enforced by laws based on zoning, e.g. National Parks, nature reserves, etc. to address issues such as deforestation in Amazonia or Southeast Asia. The instruments to regulate this type of anthropogenic threat are relatively straightforward and consistently defined across the world, but the obstacles are political and socioeconomic. This is also true of other existing laws that directly or indirectly protect soil, the success of which require that entrenched power imbalances are challenged [64].

Lack of enforcement and governance on the ground remain serious obstacles throughout much of the world. See, for example, the studies referenced at [66,89,92,112,114,120,144,153,239,240]. Those ten studies cited, merely a subset of many more, draw on a range of data for their evidence, in addition to literature reviews, including land use and land evaluation records, census and demographic statistics, case studies, planning and legal decisions, stakeholder interviews and, perhaps most telling of all, spatial remote sensing of land use change, using GIS software tools such as CORINE. Nevertheless, the process of enacting soil and land legislation is ongoing and ubiquitous, and reflects each country's circumstances and priorities. China, for example, with possibly the greatest absolute area of contaminated soil on Earth [241–243], as well as a long history of soil erosion [8], has separate soil laws to address both of these problems.

Table 3 presents a condensed numerical summary of derived categories of soil-related legal instruments, presented as approximate percentages of countries that have created or adopted such instruments (as detailed in *Methods* above). It is evident that soil and land protection laws are conspicuous by their relative absence worldwide. Environmental legislation is not specifically recorded here but, for comparison, it has been almost universally adopted. It is also conspicuous from the data below that for many countries the word *soil* would be entirely absent from their legal portfolio were it not for such environmental legislation.

## 5.7. Prime farmland preservation (PFP)

The data presented in table 3, along with our literature review, leads the authors to conclude that one of the most serious failings with respect to the ambiguity or absence of legal instruments relates to what we

**Table 3.** Proportions of countries with soil-related legislation.

| (a) soil-specific legal instruments | countries (%) |
| --- | --- |
| explicit soil policy | 7 |
| soil conservation/erosion law(s) [explicit/explicit + implicit] | 34/79 |
| soil contamination/pollution law(s) [explicit/explicit + implicit] | 27/65 |
| soil sealing law(s) [explicit/explicit + implicit] | 3/27 |
| generic or other soil protection law(s) [explicit/explicit + implicit] | 28/71 |
| soil protection monitoring and/or targets [explicit/explicit + implicit] | 21/47 |
| reference to soil embedded within environmental legislation | 88 |
| reference to soil embedded within agricultural or land rights legislation | 72 |
| reference to soil embedded within spatial/regional planning legislation | 50 |
| **(b) legal instruments to preserve agricultural land** | |
| policies designating zoning, including rural and/or agricultural land | 35 |
| policy advocating land take avoidance | 24 |
| land take targets | 9 |
| legal framework to facilitate farmland preservation schemes | 19 |
| PFP guidance based on soil or land classification | 27 |
| law prohibiting or strongly restricting loss of all prime farmland | 21 |
| land degradation neutrality (LDN) policy (commitments) | 66 |
| **(c) soil-related agro-environmental policies** | |
| policies promoting organic or regenerative agriculture, agroecology or PES | 40 |
| policies referring specifically to the critical ecosystem services of soils | 15 |

have termed category 2(b) in table 2, that is land take outside protected areas. At most risk and of most concern is the conversion of high quality or prime farmland, which is usually afforded far less protection than natural ecosystems, or none at all, yet can sometimes provide equivalent, or even greater, ecosystem services. This is an important and often misunderstood point that cannot be overestimated. It is easy to fall into the trap of assuming that agricultural land cannot provide environmental benefits comparable to those of natural ecosystems, especially when based on current land management practices, but this certainly does not reflect the intrinsic value of such land nor necessarily even its existential value [244].

Land which is densely covered in vegetation, with undisturbed topsoil, will tend to store more water and C, and foster greater biodiversity, but the principle reason land becomes prime farmland is its highly valued ecosystem attributes: low altitude and gentle gradients; deep soil with favourable texture and structure that retains water, nutrients and C; benign biochemistry; and a favourable moisture regime that is not susceptible to extremes of drought or waterlogging. These are the desirable attributes that facilitate not just food production, but terrestrial life in general, which ecologists assess in aggregate as NPP. As crude as it might be in some respects, especially in relation to cultural or scientific value, NPP is widely regarded as one of the best proxy measures of total ecosystem service contribution [245]. In contrast to this, many of the varied soil landscapes that currently support natural ecosystems can be biologically marginal, exhibiting low NPP. Even putting aside the utilitarian criterion of agricultural potential, these peripheral zones can also be limited with respect to a range of critical ecosystem services, including in terms of the biodiversity they support.

Prime farmland is not usually an obvious candidate for voluntary conversion to natural regeneration or rewilding, but high-grade arable land, even in its cultivated state, can provide abundant benefits in terms of flood control, water storage and filtration, wildlife habitat, landscape value and recreation. However, and critically, such benefits are greatly affected by land management. The negative impacts of modern intensive farming, on the soil in particular, such as compaction, erosion, pollution and reduced SOC, are precisely the factors that undermine the ability of the soil to provide ecosystem services [246,247].

One very welcome trend in recent decades has been to roll back many of these harmful methods. A range of techniques that are loosely grouped under the term of conservation, or regenerative, agriculture,

agroecology or even 'carbon farming' are increasingly being promoted and applied in many countries [78,248–250]. Included within this approach are minimum or zero tillage, permanent soil cover, crop residue retention, cover cropping, intercropping and enhanced rotations including, for example, deep rooting 'tillage crops', leguminous leys and rotational livestock grazing, agroforestry and other forms of mixed cropping, perennial crops, integrated pest management (IPM) and biological control, and many other ways of conserving and enhancing soil quality and, by default, ameliorating environmental impacts. Hence, our advocacy of PFP is proposed in tandem with the adoption of such methods.

## 5.8. Emerging holistic conceptual frameworks

Amundson [251] sums up soil governance in the US as a complex patchwork of transient interventions, as opposed to long-term solutions. Other authors describe comparable situations elsewhere, for example, in the EU [252,253], in South America [254], in Australia [255], in Russia [256] and in other examples already cited in this article. Fromherz summarizes many of the existing soil governance initiatives, as well as the failures, and makes an impassioned appeal for a dedicated, legally binding international soil governance instrument on the basis that, '…individual states lack both the power and the incentives to make these changes.' [257]. This has been echoed by others [181,200,236,258]. Fromherz also highlights a key point alluded to already in this article, the low profile (literal as well as metaphorical, one could say) of soil, which is often conspicuous in its absence from risk assessments of other natural resources [259,260]. Gonzalez Lago *et al.* refer to a global soils policy vacuum and call for an urgent transdisciplinary framework approach to 're-politicize' soil [261], while a number of authors and institutions highlight the degree to which soil protection is inescapably enmeshed with ethics [262].

Participatory modelling and conceptual frameworks have been applied to complex cross-cutting problems such as addressing the SDGs, and these approaches continue to evolve [263]. The latest stages in the process of soil governance so far, in the last decade, are encouraging to some extent, at least with regard to what soil scientists are bringing to the table. An overarching conceptual framework that has been proposed by some of the world's leading soil scientists, broadly as a memorandum of understanding, is 'soil security' [264]. This concept has five dimensions: capability, condition, capital, connectivity and codification, which encompasses the translation of soil knowledge into policy and legislation [265], e.g. aimed at achieving the SDGs [266], although an attempt to establish soil security as one of the SDGs was unsuccessful [267]. The soil security initiative is still at a high level, with the emphasis on policy rather than active governance, but progress is being made and reported on in some areas [268].

Also emerging are more formalized frameworks that place the concept of SES further than ever in the context of socioeconomic decisions, policy-making and governance. One such methodology developed and tested in Switzerland is SQUID (Soil QUality InDicator), an index for mapping soil quality with respect to SES, to guide spatial development [187]. Another example from New Zealand is the Land Resource Circle (LRC) framework, which goes further than any other approach known to the authors in extending land evaluation to incorporate SES. The framework identifies in-depth environmental and socioeconomic implications of land-use decisions via a scoring system which avoids reducing all outcomes to simplistic monetary values [269]. A recent LRC paper includes a very detailed hypothetical example which indicates how quantified soil variables can be combined and converted into societal costs and benefits. By contrast, the resilience–effectiveness–efficiency–legitimacy (REEL) framework from Germany provides a purely qualitative means of comparing different approaches to soil governance according to the four criteria (dimensions) embedded within its title [270]. The REEL framework also highlights interconnectedness and attempts to address situations where policy targets mismatch the causes of problems, either spatially or in terms of scale or even time. This is almost a meta-framework, with the emphasis on socioeconomics and due diligence.

As with other forms of natural resource governance, politics and socioeconomics are often obstacles to effective soil governance, as are weak governance structures, but a critical distinction in relation to PFP is that there is much scope for creating more effective legal instruments to tackle this problem than currently exist in most countries, and also for improving clarity and consistency to bring PFP further in line with ecosystem and soil protection.

# 6. Conclusion

An appreciation of the tangible benefits of soil is woven into the fabric of history and reflected by the value and protections afforded to productive soil by society over millennia. Such protections include legislation,

but *ad hoc*, inadequately enforced and rarely if ever proportionate to the total SES. Two challenges identified as hindering progress are terminological obstacles associated with soil and land, and the absence of a soil-centric policy framework. Moreover, soil exists as a component of land that is subject to both extra demands, e.g. of food production, and competing demands, such as urbanization. Agriculture has responded to these demands by expansion and intensification, each approach posing threats to soil, the former often encroaching on natural ecosystems and the latter fostering soil degradation. These threats have spawned national and international legislation. However, the problem of land take, in particular the conversion of prime agricultural land to non-agricultural use, including largely irreversible soil sealing, is arguably at least as serious a threat as intensification because of the disproportionate loss of SES this represents, yet tends to be relatively inconspicuous, and subject to ambiguous legislation and governance.

Threats to natural ecosystems are explicitly addressed by environmental laws, and to soil degradation by agricultural or soil-specific laws. A few of these laws have been in place for centuries and more have appeared in recent decades as global environmental awareness has grown; but neither environmental nor agricultural legislation normally encompasses the loss of prime farmland, the fate of which then tends to fall within the remit of spatial planning. Planners' procedures must navigate many competing demands and interest groups, political as well as economic, and rarely prioritize soil or its intrinsic value to future generations. As a result, legal instruments that are frequently designed with little, if any, reference to soil, often have the greatest power to transform soil and land use. This situation is further compounded by the fact that intense commercial pressure along with conventional economics, which has traditionally ignored the benefits of long-term ecosystem services, promotes land use options that are more profitable than farming.

In a time of so much environmental concern, a further paradoxical factor that has emerged in the last half-century to weaken the case for farmland preservation (as opposed to soil protection *per se*), is the conflicted nature of agriculture: on the one hand the producer of our sustenance and custodian of rural landscapes, while on the other, a significant cause of environmental harm and climate forcing. Lacking both the visual impact and the iconic status of natural landscapes, soil in an agricultural context does not constitute an obvious rallying point in the public consciousness and so the constant attrition of prime farmland occurs in something of a legal and ecological grey area, attracting far less attention than other environmental issues.

However, the case for protecting soil as a critical part of our natural capital is separately gaining ground. A key paradigm shift occurred in the 1990s when it became more widely appreciated that soil C is inextricably linked with atmospheric C. This development also approximately coincided with a broader gradual acceptance that land degradation was interrelated with climate change and biodiversity loss. This view of soil as a central component in limiting global warming, adapting to climate change and addressing the ecological crisis, is being framed unequivocally in the wider context of sustainability and the linking of science to policy. Emerging multidisciplinary frameworks, such as the concept of soil security and methodologies for holistically evaluating SES, contribute to this process.

Politicians and activists are calling for 'green new deals' to emulate Roosevelt's New Deal following the Great Depression, but with an emphasis on climate justice and ecosystem restoration. This encompasses sustainable land use, by default, and important components of that must be soil protection and land preservation, consistent with climate justice, including the attendant social and economic incentives necessary to achieve local and global climate goals, biodiversity objectives and food security. Profound reform can reach a tipping point when society perceives a binary moral choice that serves the common good. Two successful examples of global co-operation of this kind were the signing of the *Montreal Protocol* in 1987 to protect the ozone layer, and persuading the global community to reduce GHG emissions in the *Paris Agreement*, though hard-won and far from over. Ginzky [271] cites other examples and, despite poor progress to date, suggests that a binding international treaty on soils is both necessary and achievable.

Clearly, a lack of effective soil governance is often more significant a problem than an absence of legal instruments. The lesson here, for saving our soil, is to strive for a clear message, backed up by consistent policies and laws, and sound criteria to decide how to prioritize soil types or areas of land. This alone cannot solve the vast problem, but it will make the process more streamlined and more transparent, and facilitate governance for those who genuinely want to govern, and make infringements that bit more difficult to effect and pass unnoticed. The urgent need is to couple a compelling case with an achievable solution.

Data accessibility. The only original data presented are the categorization percentages presented in table 3.
Authors' contributions. L.R.P. conceived the idea, conducted much of the research and wrote the bulk of the text. C.R. made considerable contributions to the text, especially in relation to soil law, and performed much editing.

(https://research-portal.uea.ac.uk/en/publications/erosion-crop-yields-and-time-a-reassessment-of-quantitative-relat)

34. Konyushkov D. 2014 *Russian: Evolution and Examples. Reference Module in Earth Systems and Environmental Sciences*. Netherlands: Elsevier. (https://doi.org/10.1016/B978-0-12-409548-9.09291-5)

35. Carter-Whitney M, Esakin TC, Canadian Institute for Environmental Law and, P. 2010 *Ontario's greenbelt in an international context*. Toronto, Ont: Canadian Institute for Environmental Law and Policy. Report No.: 0981210341 9780981210346. (https://www.deslibris.ca/ID/222794)

36. Simonson RW. 1989 *Historical Highlights of Soil Survey and Soil Classification with Emphasis on the United States, 1899-1970*. Wageningen, the Netherlands: International Soil Reference and Information Centre (ISRIC). (https://edepot.wur.nl/488417)

37. Young A. 1980 *Tropical soils and soil survey*. Cambridge: Cambridge University Press.

38. Robinson A. 1943 The Scott and Uthwatt reports on land utilisation. *Econ. J.* **53**, 28–38. (doi:10.2307/2226286)

39. Sheail J. 1997 Scott revisited: post-war agriculture, planning and the British countryside. *J. Rural Stud.* **13**, 387–398. (doi:10.1016/S0743-0167(97)00028-4)

40. CPRE (Campaign to Protectf Rural England). 2017 *Landlines: why we need a strategic approach to land. Pamphlet*. London: CPRE. (https://www.cpre.org.uk/resources/landlines-why-we-need-a-strategic-approach-to-land/)

41. Monk S, Whitehead C, Burgess G, Tang C. 2013 *International review of land supply and planning systems*. York, UK: Joseph Rowntree Foundation. (https://www.jrf.org.uk/report/international-review-land-supply-and-planning-systems)

42. Natural England. 2012 Agricultural Land Classification: protecting the best and most versatile agricultural land. Technical Information Note TIN049. (http://publications.naturalengland.org.uk/publication/35012)

43. Daniels T. 1990 Using LESA in a purchase of development rights program. *Journal of Soil and Water Conservation*. **45**, 617–621. (https://www.jswconline.org/content/jswc/45/6/617.full.pdf)

44. Lee ATM. 1948 Soil conservation—an international study, food and agriculture organization of the United Nations, Washington, D. C., 1948. *Am. J. Agric. Econ.* **30**, 784–786. (doi:10.2307/1232795)

45. United Nations. 1949 *UN Yearbook 1947-48*. Washington DC: United Nations.

46. Bunce AC. 1942 *The economics of soil conservation*. Ames, IA: Iowa State college Press.

47. Ciriacy-Wantrup SV. 1952 *Resource Conservation: Economics and Policies*. University of California, Division of Agricultural Sciences, Agricultural Experiment Station. (doi:10.1017/S0003055400274273)

48. Sauer EL. Economics of Soil Conservation, Reclamation and Rehabilitation. *2nd West Indies Agricultural Economics Conference. St. Augustine, Trinidad*; *1967*. (doi:10.22004/ag.econ.263877)

49. Morgan RJ. 1966 *Governing Soil Conservation: Thirty Years of the New Decentralization*. Baltimore: Johns Hopkins Press for Resources for the Future. (doi:10.4324/9781315063980)

50. Aubreville A. 1949 Climats, Forêts, et Désertification de l'Afrique Tropicale Paris: Société d'Editions Géographiques, Maritimes et Tropicales.

51. IPCC. 2019 *Climate Change and Land: an IPCC special report on climate change, desertification, land degradation, sustainable land management, food security, and greenhouse gas fluxes in terrestrial ecosystems* (eds PR Shukla, J Skea, E Calvo Buendia, V Masson-Delmotte, HO Pörtner, DC Roberts, P Zhai, R Slade, S Connors, R van Diemen, M Ferrat, E Haughey, S Luz, S Neogi, M Pathak, J Petzold, J Portugal Pereira, P Vyas, E Huntley, K Kissick, M Belkacemi, J Malley). Geneva: IPCC Secretariat. (https://www.ipcc.ch/srccl/chapter/summary-for-policymakers/)

52. Gisladottir G, Stocking M. 2005 Land degradation control and its global environmental benefits. *Land Degrad. Dev.* **16**, 99–112. (doi:10.1002/ldr.687)

53. Thomas DSG, Middleton NJ. 1994 *Desertification: exploding the myth*. London, UK: Wiley.

54. Davidson J, Wibberley GP. 1977 Planning and the rural environment.. Frankfurt: Pergamon Press.

55. Pingali PL. 2012 Green Revolution: impacts, limits, and the path ahead. *Proc. Natl Acad. Sci. USA* **109**, 12 302–12 308. (doi:10.1073/pnas.0912953109)

56. Mannion AM. 1991 *Global environmental change: a natural and cultural environmental history*. Harlow, UK: Longman Scientific & Technical.

57. Agricultural Advisory Council. 1970 *Modern farming and the soil: report of the agricultural advisory council on soil structure and soil fertility*. London, UK: HM Stationery Office.

58. Biniek JP. 1979 *Agricultural and environmental relationships issues and priorities: report*. Washington DC: Library of Congress, US Govt. Print. Off. (http://hdl.handle.net/2027/mdp.39015081271325)

59. USDA. 1973 Monoculture in Agriculture: Extent, Causes, and Problems-report of the Task Force on Spatial Heterogeneity in Agricultural Landscapes and Enterprises. Washington DC: U.S. Govt. Print. Off., 64 pp.

60. Carson R. 1962 *Silent spring*, x, 368 p. Boston, Cambridge, MA: Houghton Mifflin; Riverside Press.

61. Schneider SH. 1989 The greenhouse effect: science and policy. *Science* **243**, 771–781.

62. Leahy S, Clark H, Reisinger A. 2020 Challenges and prospects for agricultural greenhouse gas mitigation pathways consistent with the Paris agreement. *Front. Sustain. Food Syst.* **4**, 69. (doi:10.3389/fsufs.2020.00069)

63. Shivanna KR. 2020 The sixth mass extinction crisis and its impact on biodiversity and human welfare. *Resonance* **25**, 93–109. (doi:10.1007/s12045-019-0924-z)

64. Weigelt J, Müller A, Janetschek H, Töpfer K. 2015 Land and soil governance towards a transformational post-2015 Development Agenda: an overview. *Curr. Opin. Environ. Sustain.* **15**, 57–65. (doi:10.1016/j.cosust.2015.08.005)

65. Juerges N, Hansjürgens B. 2016 Soil governance in the transition towards a sustainable bioeconomy – a review. *J. Clean. Prod.* **170**, 1628–1639. (doi:10.1016/j.jclepro.2016.10.143)

66. Guereña A. 2016 *Unearthed: Land, power and inequality in Latin America*. Oxford: Oxfam International. (https://oxfamilibrary.openrepository.com/bitstream/handle/10546/620158/bp-land-power-inequality-latin-america-301116-summ-en.pdf?sequence=13&isAllowed=y)

67. van Schaik L, Dinnissen R. 2014 *Terra Incognita: Land degradation as underestimated threat amplifier. Clingendael report for the Netherlands Environmental Assessment Agency (PBL)*. The Hague: Clingendael. (https://edepot.wur.nl/481424)

68. Baba SH, Wani MH, Zargar BA, Bhat IF. 2020 Determinants of land degradation in Jammu and Kashmir: implications for land governance. *Agricultural Economics Research Review*. **32**, 303645. (https://EconPapers.repec.org/RePEc:ags:aerrae:303645)

69. Sloan T *et al.* 2018 Peatland afforestation in the UK and consequences for carbon storage. *Mires and Peat*. **23**. (doi:10.19189/MaP.2017.OMB.315)

70. Bai ZG, Dent DL, Olsson L, Schaepman ME. 2008 Proxy global assessment of land degradation. *Soil Use Manag.* **24**, 223–234. (doi:10.1111/j.1475-2743.2008.00169.x)

71. Sonneveld BGJS, Dent DL. 2009 How good is GLASOD? *J. Environ. Manage* **90**, 274–283. (doi:10.1016/j.jenvman.2007.09.008)

72. Bai ZG, Dent DL, Olsson L, Tengberg A, Tucker C, Yengoh G. 2015 A longer, closer, look at land degradation. *Agriculture for Development*. **24**, 3–9. (https://edepot.wur.nl/353408)

73. FAO, ITPS. 2015 *The Status of the World'sSoil Resources (SWSR) – Main Report*. Rome: Food and Agriculture Organization of the United Nations and Intergovernmental Technical Panel on Soils. (https://www.fao.org/documents/card/en/c/c6814873-efc3-41db-b7d3-2081a10ede50/)

74. Nkonya E, Mirzabaev A, von Braun J. 2016 Economics of Land Degradation and Improvement – A Global Assessment for Sustainable Development. *Springer Nature*. (doi:10.1007/978-3-319-19168-3)

75. Esch S, Brink B, Stehfest E, Bakkenes M, Sewell A, Bouwman A, Meijer J, Westhoek H, van den Berg M. 2017 *Exploring future changes in land use and land condition and the impacts on food, water, climate change and biodiversity: Scenarios for the UNCCD Global Land Outlook*. The Hague: PBL Netherlands Environmental Assessment Agency. (https://www.pbl.nl/en/publications/exploring-future-changes-in-land-use)

76. United Nations Convention to Combat Desertification. 2017 *The Global Land Outlook, first edition*. Bonn, Germany (https://knowledge.unccd.int/glo/GLO_first_edition)

77. IPBES. 2018 *The IPBES assessment report on land degradation and restoration*. (eds L Montanarella, R Scholes, A Brainich), Bonn, Germany: Secretariat of the Intergovernmental Science-Policy Platform on Biodiversity and

Ecosystem Services. (https://digitallibrary.un.org/record/3794559?ln=en)

78. Zimmerer KS. 2011 "Conservation booms" with agricultural growth? Sustainability and Shifting Environmental Governance in Latin America, 1985-2008 (Mexico, Costa Rica, Brazil, Peru, Bolivia). *Latin American Research Review*. 82–114. (https://www.jstor.org/stable/41261393)

79. Stubenrauch J, Garske B, Ekardt F. 2018 Sustainable land use, soil protection and phosphorus management from a cross-national perspective. *Sustainability* **10**, 1988. (doi:10.3390/su10061988)

80. The Economist. 2017 Briefing,The bad earth: The most neglected threat to public health in China is toxic soil; and fixing it will be hard and costly. 8th June 2017. (https://www.economist.com/briefing/2017/06/08/the-most-neglected-threat-to-public-health-in-china-is-toxic-soil)

81. Liqiang H. 2019 *New law on soil pollution will pinpoint responsibility*. China Daily. (http://www.chinadaily.com.cn/a/201901/02/WS5c2c0b6fa310d91214051f8b.html)

82. Li T, Liu Y, Lin S, Liu Y, Xie Y. 2019 Soil pollution management in China: a brief introduction. *Sustainability* **11**, 556. (doi:10.3390/su11030556)

83. Hendlin YH. 2014 From Terra Nullius to Terra Communis reconsidering wild land in an era of conservation and indigenous rights. *Environ. Philos.* **11**, 141–174. (doi:10.5281/zenodo.260245)

84. Pearce F. Conflicting Data: How Fast Is the World Losing its Forests? 2018;[cited 2020; Available from: https://e360.yale.edu/features/conflicting-data-how-fast-is-the-worlds-losing-its-forests]

85. Curtis PG, Slay CM, Harris NL, Tyukavina A, Hansen MC. 2018 Classifying drivers of global forest loss. *Science* **361**, 1108–1111. (doi:10.1126/science.aau3445)

86. Escobar H. 2020 Deforestation in the Brazilian Amazon is still rising sharply. *Science* **369**, 613. (doi:10.1126/science.369.6504.613)

87. Young A. 1999 Is there really spare land? A critique of estimates of available cultivable land in developing countries. *Environ. Dev. Sustain.* **1**, 3–18. (doi:10.1023/A:1010055012699)

88. Chamberlin J, Jayne TS, Headey D. 2014 Scarcity amidst abundance? Reassessing the potential for cropland expansion in Africa. *Food Policy* **48**, 51–65. (doi:10.1016/j.foodpol.2014.05.002)

89. Güneralp B, Lwasa S, Masundire H, Parnell S, Seto KC. 2017 Urbanization in Africa: challenges and opportunities for conservation. *Environ. Res. Lett.* **13**, 015002. (doi:10.1088/1748-9326/aa94fe)

90. Rockström J *et al.* 2009 A safe operating space for humanity. *Nature*. **461**, 472–475. (doi:10.1038/461472a)

91. Byerlee D, Stevenson J, Villoria N. 2014 Does intensification slow crop land expansion or encourage deforestation? *Glob. Food Secur.* **3**, 92–98. (doi:10.1016/j.gfs.2014.04.001)

92. Abu Hatab A, Cavinato MER, Lindemer A, Lagerkvist C-J. 2019 Urban sprawl, food security and agricultural systems in developing countries: a systematic review of the literature.

*Cities* **94**, 129–142. (doi:10.1016/j.cities.2019.06.001)

93. Metternicht G. 2018 *Land Use and Spatial Planning: Enabling Sustainable Management of Land Resources*. Springer International Publishing. (doi:10.1007/978-3-319-71861-3)

94. Seitzinger SP *et al.* 2012 Planetary Stewardship in an Urbanizing World: Beyond City Limits. *AMBIO*. **41**, 787–794. (doi:10.1007/s13280-012-0353-7)

95. Smith P, Gregory PJ, Vuuren D, Obersteiner M, Havlík P, Rounsevell M, Woods J, Stehfest E, Bellarby J. 2010 Competition for land. *Phil. Trans. R. Soc. B* **365**, 2941–2957. (doi:10.1098/rstb.2010.0127)

96. FAO. 2011 *Payments for ecosystem services and food security*. Rome: Food and Agriculture Organization of the United Nations (FAO). (https://www.fao.org/3/i2100e/i2100e00.htm)

97. van Vliet J, Eitelberg DA, Verburg PH. 2017 A global analysis of land take in cropland areas and production displacement from urbanization. *Glob. Environ. Change* **43**, 107–115. (doi:10.1016/j.gloenvcha.2017.02.001)

98. Bren d'Amour C, Reitsma F, Baiocchi G, Barthel S, Güneralp B, Erb KH, Haberl H, Creutzig F, Seto KC. 2017 Future urban land expansion and implications for global croplands. *Proc. Natl Acad. Sci. USA* **114**, 8939–8944. (doi:10.1073/pnas.1606036114)

99. Chen G *et al.* 2020 Global projections of future urban land expansion under shared socioeconomic pathways. *Nature Communications*. **11**. (doi:10.1038/s41467-020-14386-x)

100. Gardi C. 2017 *Urban expansion, land cover and soil ecosystem services*. London: Taylor & Francis. (doi:10.4324/9781315715674)

101. Pedroli B, Meiner A. 2017 *Landscapes in transition: An account of 25 years of land cover change in Europe*. Copenhagen: European Environment Agency. (doi:10.2800/81075)

102. Green Balance. 2000 Valuing the land: planning for the best and most versatile agricultural land. London: Council for the Preservation of Rural England (CPRE).

103. Thurston N, Kenyon D, Starkings D, Taylor K. 2011 Review of the weight that should be given to the protection of best and most versatile (BMV) land, Defra Soil Research Programme. UK: Defra. Report No.: Technical Report SP1501/TR. (http://randd.defra.gov.uk)

104. Ustaoglu E, Williams B. 2017 Determinants of urban expansion and agricultural land conversion in 25 EU countries. *Environ. Manage.* **60**, 717–746. (doi:10.1007/s00267-017-0908-2)

105. OECD. 2009 *Farmland Conversion – The Spatial Implications of Agricultural and Land-use Policies*. Paris: OECD. (doi:10.1787/ae50672e-en)

106. Perrin C, Clément C, Melot R, Nougarèdes B. 2020 Preserving farmland on the urban fringe: a literature review on land policies in developed countries. *Land* **9**, 223. (doi:10.3390/land9070223)

107. Silva C. 2019 Auckland's urban sprawl, policy ambiguities and the peri-urbanisation to Pukekohe. *Urban Sci.* **3**, 1. (doi:10.3390/urbansci3010001)

108. Government of India. 2009 *Report of the Committee on State Agrarian Relations and the Unfinished Task in Land Reforms*. New Delhi: Department of Land Resources, Ministry of Rural Development. (https://dolr.gov.in/documents/report-of-committee-on-state-agrarian-relations)

109. OECD. 2010 *Regional Development Policies in OECD Countries*. Paris: OECD. (doi:https://doi.org/10.1787/9789264087255-en)

110. Nishi M. 2019 *Multi-Level Governance of Agricultural Land in Japan: Farmers' Perspectives and Responses to Farmland Banking*. New York: Columbia. (doi:10.7916/D8-HTAA-VV53)

111. Oberndorf RB. 2012 Legal review of recently enacted farmland law and vacant, fallow and virgin lands management law: improving the legal & policy frameworks relating to land management in Myanmar. *Food Security Working Group and Land Core Group*. 1–41. (https://www.forest-trends.org/publications/legal-review-of-recently-enacted-farmland-law-and-vacant-fallow-and-virgin-lands-management-law/)

112. Fernández-Maldonado AM. 2019 Unboxing the black box of Peruvian planning. *Plan. Pract. Res.* **34**, 368–386. (doi:10.1080/02697459.2019.1618596)

113. Slätmo E. 2017 Preservation of agricultural land as an issue of societal importance. Rural landscapes: society, environment. *History* **4**, 1–12. (doi:10.16993/rl.39)

114. Jain M. 2018 Contemporary urbanization as unregulated growth in India: the story of census towns. *Cities* **73**, 117–127. (doi:10.1016/j.cities.2017.10.017)

115. Sili M, Soumoulou L. 2011 *The issue of land in Argentina: Conflicts and dynamics of use, holdings and concentration*. Rome: International Fund for Agricultural Development (IFAD). (https://farmlandgrab.org/19130)

116. Cho J. 2005 Urban planning and urban sprawl in Korea. *Urban Policy Res.* **23**, 203–218. (doi:10.1080/08111470500143304)

117. Debolini M, Valette E, François M, Chéry J-P. 2015 Mapping land use competition in the rural–urban fringe and future perspectives on land policies: a case study of Meknès (Morocco). *Land Use Policy* **47**, 373–381. (doi:10.1016/j.landusepol.2015.01.035)

118. Naab FZ, Dinye R, Kasanga RK. 2013 Urbanisation and its impact on agricultural lands in growing cities in developing countries: a case study of Tamale in Ghana. *European scientific journal*. **2**, 256–287. (https://www.semanticscholar.org/paper/Urbanisation-and-its-impact-on-agricultural-lands-a-Naab-Dinye/dbc6e81df2e898925202c0d812c54de01a6a079d)

119. Mori H. 1998 Land conversion at the urban fringe: a comparative study of Japan, Britain and the Netherlands. *Urban Stud.* **35**, 1541–1558. (doi:10.1080/0042098984277)

120. Ladu JLC, Athiba AL, Ondogo EC. 2019 An assessment of the impact of urbanization on agricultural land use in Juba City, Central Equatoria State, Republic of South Sudan. *J. Appl. Agric. Econ. Policy Anal.* **2**, 22–30. (doi:10.12691/jaaepa-2-1-4)

121. Ullah S, Khan MA, Rahman A, Mahmood S. 2019 Evaluation of urban encroachment on

farmland: a threat to urban agriculture in Peshawar City District, Pakistan. *Erdkunde* **73**, 127–142. (doi:10.3112/erdkunde.2019.02.04)

122. Visser O. 2017 Running out of farmland? Investment discourses, unstable land values and the sluggishness of asset making. *Agric. Hum. Values* **34**, 185–198. (doi:10.1007/s10460-015-9679-7)

123. Satterthwaite D, McGranahan G, Tacoli C. 2010 Urbanization and its implications for food and farming. *Phil. Trans. R. Soc. B* **365**, 2809–2820. (doi:10.1098/rstb.2010.0136)

124. Cottrell RS *et al.* 2019 Food production shocks across land and sea. *Nature Sustainability* **2**, 130–137. (doi:10.1038/s41893-018-0210-1)

125. Prishchepov AV, Müller D, Dubinin M, Baumann M, Radeloff VC. 2013 Determinants of agricultural land abandonment in post-Soviet European Russia. *Land Use Policy* **30**, 873–884. (doi:10.1016/j.landusepol.2012.06.011)

126. Lambin EF, Meyfroidt P. 2011 Global land use change, economic globalization, and the looming land scarcity. *Proc. Natl Acad. Sci. USA* **108**, 3465–3472. (doi:10.1073/pnas.1100480108)

127. Kaz'min MA. 2016 Transformation of agricultural land use in Russian regions in the course of modern socioeconomic reforms. *Reg. Res. Russ.* **6**, 87–94. (doi:10.1134/S20799705 16010056)

128. Liefert WM, Liefert O, Vocke G, Allen EW. Former Soviet Union Region To Play Larger Role in Meeting World Wheat Needs. *Amber Waves: The Economics of Food, Farming, Natural Resources, and Rural America*. US: United States Department of Agriculture, Economic Research Service; 2010:12–19. (doi:10.22004/ag.econ. 121958)

129. Meyfroidt P, Schierhorn F, Prishchepov AV, Müller D, Kuemmerle T. 2016 Drivers, constraints and trade-offs associated with recultivating abandoned cropland in Russia, Ukraine and Kazakhstan. *Glob. Environ. Change* **37**, 1–15. (doi:10.1016/j.gloenvcha. 2016.01.003)

130. Schierhorn F, Müller D, Beringer T, Prishchepov AV, Kuemmerle T, Balmann A. 2013 Post-Soviet cropland abandonment and carbon sequestration in European Russia, Ukraine, and Belarus. *Global Biogeochem. Cycles* **27**, 1175–1185. (doi:10.1002/2013gb004654)

131. Amati M, Taylor L. 2010 From green belts to green infrastructure. *Plan. Pract. Res.* **25**, 143–155. (doi:10.1080/02697451003740122)

132. Papworth T. 2015 *The green noose*. London: Adam Smith Institute. (https://www. adamsmith.org/research/the-green-noose)

133. Cheshire P. Turning houses into gold: the failure of British planning. CentrePiece - The Magazine for Economic Performance. 2014 7th May 2014. (https://EconPapers.repec.org/RePEc:cep: cepcnp:421)

134. Defra. Farming for the future: Policy and progress update. 2020;[cited; Available from: https://assets.publishing.service.gov.uk/ government/uploads/system/uploads/ attachment_data/file/868041/future-farming-policy-update1.pdf.

135. Loghrin H. Forthcoming *Global greenbelts updates since 2009 (2019)*. Toronto: Greenbelt Foundation.

136. Suu NV. 2009 Agricultural land conversion and its effects on farmers in contemporary Vietnam. *Focaal* **2009**, 106. (doi:10.3167/fcl.2009. 540109)

137. Paudel B, Pandit J, Reed B. Fragmentation and conversion of agriculture land in Nepal and Land Use Policy 2012. MPRA Paper No. 58880.; 2013, posted 26 Sep 2014. (https://mpra.ub. uni-muenchen.de/58880/)

138. Chen A, He H, Wang J, Li M, Guan Q, Hao J. 2019 A study on the arable land demand for food security in China. *Sustainability* **11**, 4769. (doi:10.3390/su11174769)

139. Govindaprasad PK, Manikandan K. Farm Land Conversion and Food Security: Empirical Evidences from Three Villages of Tamil Nadu. *Indian Journal of Agricultural Economics*; 2016:493–503. (doi:10.22004/ag.econ.302235)

140. Islam GMT, Islam A, Shopan AA, Rahman MM, Lázár AN, Mukhopadhyay A. 2015 Implications of agricultural land use change to ecosystem services in the Ganges delta. *J. Environ. Manage* **161**, 443–452. (doi:10.1016/j.jenvman.2014.11.018)

141. Robson JS, Ayad HM, Wasfi RA, El-Geneidy AM. 2012 Spatial disintegration and arable land security in Egypt: a study of small- and moderate-sized urban areas. *Habitat Int.* **36**, 253–260. (doi:10.1016/j.habitatint.2011.10.001)

142. Dai W, Dai W. 2019 Effects of urban expansion on environment by morphological study. *IOP Conf. Ser.: Earth Environ. Sci.* **227**, 052004. (doi:10.1088/1755-1315/227/5/052004)

143. Mouël C, de Lattre-Gasquet M, Mora O. 2018 *Land Use and Food Security in 2050: a Narrow Road. Agrimonde-Terra*. France: Éditions Quæ. (doi:10.35690/978-2-7592-2880-5)

144. Bezbradica L, Pantić M, Gajić A. 2019 The land use and soil protection: planning and legal regulations in Serbia. *Zemljiste i biljka* **68**, 51–71. (doi:10.5937/ZemBilj1902051B)

145. Owusu Ansah B, Chigbu UE. 2020 The nexus between peri-urban transformation and customary land rights disputes: effects on peri-urban development in Trede, Ghana. *Land* **9**, 187. (doi:10.3390/land9060187)

146. Jiang L, Zhang Y. 2016 Modeling urban expansion and agricultural land conversion in Henan Province, China: an integration of land use and socioeconomic data. *Sustainability* **8**, 920. (doi:10.3390/su8090920)

147. Li S. 2018 Change detection: how has urban expansion in Buenos Aires metropolitan region affected croplands. *Int. J. Digit. Earth* **11**, 195–211. (doi:10.1080/17538947.2017. 1311954)

148. Carreño L, Frank FC, Viglizzo EF. 2012 Tradeoffs between economic and ecosystem services in Argentina during 50 years of land-use change. *Agric. Ecosyst. Environ.* **154**, 68–77. (doi:10. 1016/j.agee.2011.05.019)

149. Meadows D, Randers J. 2004 *Limits to Growth: The 30-Year Update*. White River Junction, VT, US: Chelsea Green Publishing. (doi:10.1108/ 14636680510611831)

150. Schumacher EF. 1973 *Small is beautiful; economics as if people mattered*. New York, NY: Harper & Row.

151. Kuper S. 2014 Thoreau, leopold, & carson: challenging capitalist conceptions of the natural

environment. *Consilience: The Journal of Sustainable Development* **December 2015**. (doi:10.7916/consilience.v0i13.3927)

152. Kennel CF. 2021 The gathering anthropocene crisis. *Anthr. Rev.* **8**, 83–95. (doi:10.1177/ 2053019620957355)

153. Źróbek-Różańska A, Zielińska-Szczepkowska J. 2019 National land use policy against the misuse of the agricultural land—causes and effects. Evidence from Poland. *Sustainability* **11**, 6403. (doi:10.3390/su11226403)

154. McPike JL. Creating Space for the Formal Amongst the Informal: An Examination of Urban Housing Policies, State Power, and Multi-Scalar Politics in Indian Cities. RC21 International Conference on "The Ideal City: between myth and reality. Representations, policies, contradictions and challenges for tomorrow's urban life". Urbino (Italy); 2015. (http://www. rc21.org/en/wp-content/uploads/2014/12/G2.2-McPike.pdf)

155. Bartz D. 2015 *Soil atlas: Facts and figures about earth, land and fields*. Berlin & Potsdam, Germany: Heinrich Böll Foundation & Institute for Advanced Sustainability Studies. (https:// www.boell.de/sites/default/files/soilatlas2015_ ii.pdf)

156. Appiah DO, Asante F, Nketiah B. 2019 Perspectives on agricultural land use conversion and food security in rural Ghana. *Sci* **1**, 14. (doi:10.3390/sci1010014.v1)

157. Schueler V, Kuemmerle T, Schröder H. 2011 Impacts of surface gold mining on land use systems in Western Ghana. *Ambio* **40**, 528–539. (doi:10.1007/s13280-011-0141-9)

158. Franco J, Borras Jr SM. 2013 *Land concentration, land grabbing and people's struggles in Europe*. Amsterdam: Transnational Institute (TNI). (https://www.tni.org/en/publication/land-concentration-land-grabbing-and-peoples-struggles-in-europe-0)

159. Khalid H, Yusuf MD. Resource management: Fragmentation of land ownership and its impact on sustainability of agriculture. *11th International Annual Symposium on Sustainability Science and Management (UMTAS 2012)*. Kuala Trengganu, Malaysia; 2012. (http://irep.iium.edu.my/27046/)

160. Prokop G, Jobstmann H, Schönbauer A. 2011 Report on Best Practices for Limiting Soil Sealing and Mitigating Its Effects. European Commission. (doi:10.2779/15146)

161. Scalenghe R, Marsan FA. 2009 The anthropogenic sealing of soils in urban areas. *Landsc. Urban Plan.* **90**, 1–10. (doi:10.1016/j. landurbplan.2008.10.011)

162. Tobias S, Conen F, Duss A, Wenzel LM, Buser C, Alewell C. 2018 Soil sealing and unsealing: state of the art and examples. *Land Degrad. Dev.* **29**, 2015–2024. (doi:10.1002/ldr.2919)

163. Daniels T, Lapping M. 2005 Land preservation: an essential ingredient in smart growth. *J. Plan. Lit.* **19**, 316–329. (doi:10.1177/088541220 4271379)

164. Desrousseaux M, Schmitt B, Billet P, Béchet B, Le Bissonnais Y, Ruas A. 2019 Artificialised Land and Land Take: What Policies Will Limit Its Expansion and/or Reduce Its Impacts? In *International Year book of Soil Law and Policy*

2018 (eds H Ginzky, E Dooley, IL Heuser, E Kasimbazi, T Markus, T Qin), pp. 149–165. Cham, Switzerland: Springer. (doi:10.1007/978-3-030-00758-4_7)

165. Hellerstein DR. 2002 *Farmland protection : the role of public preferences for rural amenities. Agricultural Economic Report No. 815*. Washington, D.C: U.S. Dept. of Agriculture, Economic Research Service. (https://www.ers.usda.gov/webdocs/publications/41479/17324_aer815_1_.pdf?v=41061)

166. Gosnell H, Kline JD, Chrostek G, Duncan J. 2011 Is Oregon's land use planning program conserving forest and farm land? A review of the evidence. *Land Use Policy* **28**, 185–192. (doi:10.1016/j.landusepol.2010.05.012)

167. Ludlow D, Falconi M, Carmichael L, Croft N, Di Leginio M, Fumanti F, Sheppard A, Smith N. 2013 Land Planning and Soil Evaluation Instruments in EEA Member and Cooperating Countries (with inputs from Eionet NRC Land Use and Spatial Planning). Final Report for EEA from ETC/SIA (EEA project managers: G. Louwagie and G. Dige). (http://www.eea.europa.eu/themes/landuse/document-library)

168. Grass I et al. 2019 Land-sharing/-sparing connectivity landscapes for ecosystem services and biodiversity conservation. *People and Nature*. **1**, 262–272. (doi:10.1002/pan3.21)

169. Huang Q, Lu J, Li M, Chen Z, Li F. 2015 Developing planning measures to preserve farmland: a case study from China. *Sustainability* **7**, 13 011–13 028. (doi:10.3390/su71013011)

170. Law for the preservation of the agricultural lands. India; 2003.

171. Sartori D, Catalano G, Genco M, Pancotti C, Sirtori E, Vignetti S, Bo C. 2014 *Guide to Cost-benefit Analysis of Investment Projects. Economic appraisal tool for Cohesion Policy 2014-2020*. Luxembourg: European Union. (doi:10.2776/97516)

172. Malczewski J. 2004 GIS-based land-use suitability analysis: a critical overview. *Prog. Plan.* **62**, 3–65. (doi:10.1016/j.progress.2003.09.002)

173. Dent D, Young A. 1981 *Soil Survey and Land Evaluation*. London: George Allen & Unwin.

174. Baveye PC, Baveye J, Gowdy J. 2016 Soil 'Ecosystem' services and natural capital: critical appraisal of research on uncertain ground. *Front. Environ. Sci.* **4**, 41. (doi:10.3389/fenvs.2016.00041)

175. Bouwman AF, Sombroek WG. 1990 *Inputs to climatic change by soils and agriculture related activities: Present status and possible future trends. In Soils on a Warmer Earth*, pp. 15–30. Amsterdam: Elsevier. (http://dx.doi.org/10.1016/S0166-2481(08)70478-1)

176. Bouwman AF. 1990 Land use related sources of greenhouse gases: present emissions and possible future trends. *Land Use Policy* **7**, 154–164. (doi:10.1016/0264-8377(90)90006-K)

177. Batjes NH, Bridges EM. 1992 *A Review of Soil Factors and Processes that Control Fluxes of Heat, Moisture and Greenhouse Gases*. Wageningen, The Netherlands: ISRIC. (https://edepot.wur.nl/494028)

178. Weart SR. 2008 *The Discovery of Global Warming: Revised and Expanded Edition*. Harvard University Press. (doi:10.4159/9780674417557)

179. Wall DH. 2004 *Sustaining biodiversity and ecosystem services in soils and sediments*. Washington, DC: Island Press.

180. Karlen DL, Mausbach MJ, Doran JW, Cline RG, Harris RF, Schuman GE. 1997 Soil quality: a concept, definition, and framework for evaluation (A Guest Editorial). *Soil Sci. Soc. Am. J.* **61**, 4–10. (doi:10.2136/sssaj1997.03615995006100010001x)

181. Hannam I, Boer B. 2002 *Legal and institutional frameworks for sustainable soils: a preliminary report*. Gland, Switzerland & Cambridge, UK: International Union for Conservation of Nature and Natural Resources (IUCN). Report No.: IUCN environmental policy and law paper no. 45. (http://www.iucn.org/themes/law/pdfdocuments/EPLP45EN.pdf)

182. Costanza R et al. 1997 The value of the world's ecosystem services and natural capital. *Nature*. **387**, 253–260. (doi:10.1038/387253a0)

183. Breure AM, De Deyn GB, Dominati E, Eglin T, Hedlund K, Van Orshoven J, Posthuma L. 2012 Ecosystem services: a useful concept for soil policy making!. *Curr. Opin. Environ. Sustain.* **4**, 578–585. (doi:10.1016/j.cosust.2012.10.010)

184. Lescourret F et al. 2015 A social–ecological approach to managing multiple agro-ecosystem services. *Current Opinion in Environmental Sustainability* **14**, 68–75. (doi:10.1016/j.cosust.2015.04.001)

185. Robinson DA et al. 2013 Natural capital and ecosystem services, developing an appropriate soils framework as a basis for valuation. *Soil Biology and Biochemistry*. **57**, 1023–1033. (doi:10.1016/j.soilbio.2012.09.008)

186. Ellili-Bargaoui Y, Walter C, Lemercier B, Michot D. 2021 Assessment of six soil ecosystem services by coupling simulation modelling and field measurement of soil properties. *Ecol. Indic.* **121**, 107211. (doi:10.1016/j.ecolind.2020.107211)

187. Drobnik T, Greiner L, Keller A, Grêt-Regamey A. 2018 Soil quality indicators – from soil functions to ecosystem services. *Ecol. Indic.* **94**, 151–169. (doi:10.1016/j.ecolind.2018.06.052)

188. Dominati E, Mackay A, Green S, Patterson M. 2014 A soil change-based methodology for the quantification and valuation of ecosystem services from agro-ecosystems: a case study of pastoral agriculture in New Zealand. *Ecol. Econ.* **100**, 119–129. (doi:10.1016/j.ecolecon.2014.02.008)

189. Natural England. EIA (Agriculture) regulations: apply to make changes to rural land, first published 16 September 2014.[cited October 2020]; Available from: https://www.gov.uk/guidance/eia-agriculture-regulations-apply-to-make-changes-to-rural-land.

190. Scottish Government. Guidance on consideration of soil in Strategic Environmental Assessment, v5, 5 April 2019.[cited October 2020]; Available from: https://www.sepa.org.uk/media/162986/lups-sea-gu2-consideration-of-soil-in-sea.pdf.

191. Caldwell WJ, Hilts S, Wilton B. 2017 *Farmland Preservation: Land for Future Generations*.

Winnipeg, Canada: University of Manitoba Press. (doi:10.1080/01944363.2017.1362304)

192. Nolon J. 2003 Land Preservation. In: *Environmental Law Practice Guide: State and Federal Law* (eds MB Gerrard), New York: Matthew Bender. (http://digitalcommons.pace.edu/lawfaculty/604)

193. Robinson DA, Jackson BM, Clothier BE, Dominati EJ, Marchant SC, Cooper DM, Bristow KL. 2013 Advances in soil ecosystem services: concepts, models, and applications for earth system life support. *Vadose Zone J.* **12**, vzj2013.2001.0027. (doi:10.2136/vzj2013.01.0027)

194. Meadows DH. 1972 *The Limits to growth; a report for the Club of Rome's project on the predicament of mankind*. New York: Universe Books. (doi:10.1177/000276427201500672)

195. Norström AV et al. 2020 Principles for knowledge co-production in sustainability research. *Nature Sustainability*. **3**, 182–190. (doi:10.1038/s41893-019-0448-2)

196. Erinosho BT. 2013 The revised African convention on the conservation of nature and natural resources: prospects for a comprehensive treaty for the management of Africa's natural resources. *Afr. J. Int. Comp. Law* **21**, 378–397. (doi:10.3366/ajicl.2013.0069)

197. FAO. 1971 *Land degradation (Soils Bulletin 13)*. Rome: Food and Agriculture Organization of the United Nations (FAO). (https://www.fao.org/3/c1243e/c1243e.pdf)

198. FAO. 1971 *Legislative principles of soil conservation (Soils bulletin 15)*. Rome: Food and Agriculture Organization of the United Nations (FAO). (https://www.fao.org/publications/card/en/c/ef46ba3d-4063-54d8-8181-29645edd8673/)

199. Boer B, Ginzky H, Heuser I. 2017 International Soil Protection Law: History, Concepts and Latest Developments. In *International Yearbook of Soil Law and Policy 2016* (eds ILH Harald Ginzky, Tianbao Qin, Oliver C. Ruppel, Patrick Wegerdt), pp. 49–72. Cham, Switzerland: Springer. (doi:10.1007/978-3-319-42508-5_7)

200. Bodle R, Stockhaus H, Wolff F, Scherf C-S, Oberthür S. 2019 *Improving international soil governance - Analysis and recommendations*. Berlin: Ecologic Institute and Öko-Institute. (https://researchportal.vub.be/en/publications/improving-international-soil-governance-analysis-and-recommendati)

201. Council of Europe. 1972 *European Soil Charter*. Strasbourg: Council of Europe. (https://rm.coe.int/090000168067e296)

202. Tóth Z. 2018 International dimensions of EU soil policy – the main binding and non-binding legal instruments. *Hung. J. Leg. Stud. Acta Jurid. Hung.* **59**, 290. (doi:10.1556/2052.2018.59.3.4)

203. Byron-Cox R. 2020 From Desertification to Land Degradation Neutrality: The UNCCD and the Development of Legal Instruments for Protection of Soils. In *Legal Instruments for Sustainable Soil Management in Africa* (eds H Yahyah, H Ginzky, E Kasimbazi, R Kibugi, OC Ruppel), pp. 1–13. Cham, Switzerland: Springer. (doi:10.1007/978-3-030-36004-7_1)

204. FAO. 1981 *World Soil Charter*. Rome: Food and Agriculture Organization of the United Nations

(FAO). (https://www.fao.org/3/p8700e/p8700e.pdf)

205. UNEP. 1982 *World Soils Policy*. Nairobi, Kenya: United Nations Environment Programme (UNEP). (https://edepot.wur.nl/481087)

206. Wood Harold, W. 1985 The United Nations World Charter for Nature: The Developing Nations' Initiative to Establish Protections for the Environment. *Ecology Law Quarterly*. **12**, 977. (http://dx.doi.org/10.15779/Z38783Q)

207. World Commission on Environment and Development. 1987 *Our common future (The Brundtland Report)*. Oxford; New York: Oxford University Press.

208. World Bank. 2002 *The first decade of the GEF: second overall performance study*. Washington, DC: World Bank.

209. Wolff F, Kaphengst T. 2017 The UN Convention on Biological Diversity and Soils: Status and Future Options. In *International Yearbook of Soil Law and Policy 2016* (eds H Ginzky, IL Heuser, T Qin, OC Ruppel, P Wegerdt), pp. 129–148. Cham, Switzerland: Springer. (doi:10.1007/978-3-319-42508-5_11)

210. United Nations Framework Convention on Climate Change. 1992 (https://unfccc.int/resource/docs/convkp/conveng.pdf)

211. United Nations. Convention to Combat Desertification in those Countries Experiencing Serious Drought and/or Desertification, Particularly in Africa. International Legal Materials. 2017/02/27 ed. Cambridge: Cambridge University Press; 1994:1328-1382. (doi:10.1017/S0020782900026711)

212. Global Environment Facility (GEF). 1999 *Clarifying linkages between land degradation and the GEF focal areas: an action plan For enhancing GEF support: an action plan for enhancing GEF support, GEF/C.14/4, November 17, 1999*. Washington, DC: World Bank Group.

213. Hannam I, Boer B. 2019 Land degradation and international environmental law. In *Response to Land Degradation*. (eds IDHE Michael Bridges, L Roel Oldeman, Frits W.T Penning de Vries, Sara J Scherr, Samran Sombatpanit, Robin N Leslie, Tanadol Compo, Apuntree Prueksapong), pp. 429–440 (doi:10.1201/9780429187957-43)

214. FAO and UNEP. 2020 *Legislative approaches to sustainable agriculture and natural resources governance. FAO Legislative Study No. 114*. Rome: Food and Agriculture Organization of the United Nations (FAO). (https://doi.org/10.4060/ca8728en)

215. Chasek P, Safriel U, Shikongo S, Fuhrman VF. 2015 Operationalizing Zero Net Land Degradation: the next stage in international efforts to combat desertification? *J. Arid Environ.* **112**, 5–13. (doi:10.1016/j.jaridenv.2014.05.020)

216. UNFCCC. Land Use, Land-Use Change and Forestry (LULUCF). 2020;[cited 2020 October 2020]; Available from: https://unfccc.int/topics/land-use/workstreams/land-use--land-use-change-and-forestry-lulucf

217. Fee E. 2019 Implementing the Paris Climate Agreement: Risks and Opportunities for Sustainable Land Use. In *International Yearbook of Soil Law and Policy 2018* (eds H Ginzky, E

Dooley, IL Heuser, E Kasimbazi, T Markus, T Qin), pp. 249–270. Cham, Switzerland: Springer. (doi:10.1007/978-3-030-00758-4_12)

218. Convention on Biological Diversity (CBD). 2020 *Review of the International Initiative for the Conservation and Sustainable Use of Soil Biodiversity and Updated Plan of Action, Twenty-fourth meeting, Montreal, Canada, 17-22 August 2020: UNEP*. (https://www.cbd.int/doc/c/f25f/ac08/fac2443375cabc303ef45c22/sbstta-24-07-en.pdf)

219. FAO, ITPS, GSBI, SCBD, EC. 2020 *State of knowledge of soil biodiversity - Status, challenges and potentialities: Report 2020*. Rome: Food and Agriculture Organization of the United Nations (FAO). Report No.: 9789251335826. (doi:10.4060/cb1928en)

220. Rojas RV, Caon L. 2016 The international year of soils revisited: promoting sustainable soil management beyond 2015. *Environ. Earth Sci.* **75**, 1–5. (doi:10.1007/s12665-016-5891-z)

221. FAO. 2015 *Revised World Soil Charter*. Rome: Food and Agriculture Organization of the United Nations (FAO). (https://www.fao.org/3/i4965e/i4965e.pdf)

222. FAO. 2017 *Voluntary Guidelines for Sustainable Soil Management*. Rome: Food and Agriculture Organization of the United Nations (FAO). (http://www.fao.org/documents/card/en/c/5544358d-f11f-4e9f-90ef-a37c3bf52db7/)

223. Orr B, Cowie A, Castillo Sanchez V, Chasek P, Crossman N, Erlewein A, Louwagie G, Maron M, Metternicht G, Minelli S. 2017 *Scientific conceptual framework for land degradation neutrality. A report of the science-policy interface*. Bonn, Germany: *United Nations Convention to Combat Desertification (UNCCD)*. (https://www.unccd.int/publications/scientific-conceptual-framework-land-degradation-neutrality-report-science-policy)

224. International Law Association. 2020 *ILA Guidelines on the Role of International Law in Sustainable Natural Resources Management for Development, Resolution 4/2020, 2020 Report of the Seventy Ninth Conference, Kyoto*. London: ILA. (https://www.ila-hq.org/images/ILA/docs/kyoto/Draft%20Resolution%204%20&%20Guidelines%20Int%20Law%20Sustainable%20Natural%20Resources%20Kyoto%202020.pdf)

225. Minasny B *et al.* 2017 Soil carbon 4 per mille. *Geoderma*. **292**, 59–86. (doi:10.1016/j.geoderma.2017.01.002)

226. Chabbi A *et al.* 2017 Aligning agriculture and climate policy. *Nature Climate Change*. **7**, 307–309. (doi:10.1038/nclimate3286)

227. Poulton P, Johnston J, Macdonald A, White R, Powlson D. 2018 Major limitations to achieving '4 per 1000' increases in soil organic carbon stock in temperate regions: evidence from long-term experiments at Rothamsted Research. United Kingdom. *Glob. Change Biol.* **24**, 2563–2584. (doi:10.1111/gcb.14066)

228. Minasny B *et al.* 2018 Rejoinder to Comments on Minasny *et al*., **2017** Soil carbon 4 per mille Geoderma 292, 59–86. Geoderma. **309**, 124-129. (10.1016/j.geoderma.2017.05.026)

229. Soussana J-F *et al.* 2019 Matching policy and science: Rationale for the '4 per 1000-soils for food security and climate' initiative. *Soil and*

*Tillage Research*. **188**, 3–15. (doi:10.1016/j.still.2017.12.002)

230. Kibugi R. 2018 Soil Health, Sustainable Land Management and Land Degradation in Africa: Legal Options on the Need for a Specific African Soil Convention or Protocol. In *International Yearbook of Soil Law and Policy 2017* (eds H Ginzky, E Dooley, IL Heuser, E Kasimbazi, T Markus, T Qin), pp. 387–411. Cham, Switzerland: Springer. (doi:10.1007/978-3-319-68885-5_21)

231. Markus T. 2017 The Alpine Convention's Soil Conservation Protocol: A Model Regime? In *International Yearbook of Soil Law and Policy 2016* (eds H Ginzky, IL Heuser, T Qin, OC Ruppel, P Wegerdt), pp. 149–164. Cham, Switzerland: Springer. (doi:10.1007/978-3-319-42508-5_12)

232. Committee for the activities of the Council of Europe in the field of biological and landscape diversity (CO-DBP). 2003 *Revised European Charter for the Protection and Sustainable Management of Soil*. Strasbourg: Council of Europe. (https://rm.coe.int/09000016805dfc89)

233. European Commission. 2006 *Soil protection - The story behind the Strategy*. Luxembourg: European Commission. (https://ec.europa.eu/environment/archives/soil/pdf/soillight.pdf)

234. Stankovics P, Tóth G, Tóth Z. 2018 Identifying gaps between the legislative tools of soil protection in the EU member states for a common European soil protection legislation. *Sustainability* **10**, 2886. (doi:10.3390/su10082886)

235. Montanarella L, Panagos P. 2021 The relevance of sustainable soil management within the European Green Deal. *Land Use Policy* **100**, 104950. (doi:10.1016/j.landusepol.2020.104950)

236. European Parliament. *European Parliament resolution of 28 April 2021 on soil protection*. Brussels; 2021. (https://oeil.secure.europarl.europa.eu/oeil/popups/ficheprocedure.do?lang=en&reference=2021/2548(RSP))

237. Keesstra SD *et al.* 2016 The significance of soils and soil science towards realization of the United Nations Sustainable Development Goals. *SOIL* **2**, 111–128. (doi:10.5194/soil-2-111-2016)

238. Tóth G, Hermann T, da Silva MR, Montanarella L. 2018 Monitoring soil for sustainable development and land degradation neutrality. *Environ. Monit. Assess.* **190**, 57. (doi:10.1007/s10661-017-6415-3)

239. Shen X, Wang X, Zhang Z, Lu Z, Lv T. 2019 Evaluating the effectiveness of land use plans in containing urban expansion: an integrated view. *Land Use Policy* **80**, 205–213. (doi:10.1016/j.landusepol.2018.10.001)

240. Roose A, Kull A, Gauk M, Tali T. 2013 Land use policy shocks in the post-communist urban fringe: a case study of Estonia. *Land Use Policy* **30**, 76–83. (doi:10.1016/j.landusepol.2012.02.008)

241. Lu Y *et al.* 2015 Impacts of soil and water pollution on food safety and health risks in China. *Environment International*. **77**, 5–15. (doi:10.1016/j.envint.2014.12.010)

242. Zhao F-J, Ma Y, Zhu Y-G, Tang Z, McGrath SP. 2015 Soil contamination in China: current status

and mitigation strategies. *Environ. Sci. Technol.* **49**, 750–759. (doi:10.1021/es5047099)

243. Chen R, de Sherbinin A, Ye C, Shi G. 2014 China's soil pollution: farms on the frontline. *Science* **344**, 691. (doi:10.1126/science.344. 6185.691-a)

244. Vogel H-J, Eberhardt E, Franko U, Lang B, Ließ M, Weller U, Wiesmeier M, Wollschläger U. 2019 Quantitative evaluation of soil functions: potential and state. *Front. Environ. Sci.* **7**, 164. (doi:10.3389/fenvs.2019.00164)

245. Vargas L, Willemen L, Hein L. 2019 Assessing the capacity of ecosystems to supply ecosystem services using remote sensing and an ecosystem accounting approach. *Environ. Manage.* **63**, 1–15. (doi:10.1007/s00267-018-1110-x)

246. Kibblewhite MG, Ritz K, Swift MJ. 2008 Soil health in agricultural systems. *Philos. Trans. R Soc. B* **363**, 685–701. (doi:10.1098/rstb.2007.2178)

247. Powlson DS, Gregory PJ, Whalley WR, Quinton JN, Hopkins DW, Whitmore AP, Hirsch PR, Goulding KWT. 2011 Soil management in relation to sustainable agriculture and ecosystem services. *Food Policy* **36**(Supplement 1), S72–S87. (doi:10. 1016/j.foodpol.2010.11.025)

248. Paustian K, Lehmann J, Ogle S, Reay D, Robertson GP, Smith P. 2016 Climate-smart soils. *Nature* **532**, 49. (doi:10.1038/ nature17174)

249. LaCanne CE, Lundgren JG. 2018 Regenerative agriculture: merging farming and natural resource conservation profitably. *PeerJ* **6**, e4428. (doi:10.7717/peerj.4428)

250. Lal R *et al.* 2021 Soils and sustainable development goals of the United Nations: An International Union of Soil Sciences perspective. *Geoderma Regional*. **25**, e00398. (https://doi. org/10.1016/j.geodrs.2021.e00398)

251. Amundson R. 2020 The policy challenges to managing global soil resources. *Geoderma* **379**, 114639. (doi:10.1016/j.geoderma.2020.114639)

252. Frelih-Larsen A *et al.* 2017 *Updated Inventory and Assessment of Soil Protection Policy Instruments in EU Member States. Final Report to DG Environment*. Berlin: Ecologic Institute. (https://www.ecologic.eu/14567)

253. Ronchi S, Salata S, Arcidiacono A, Piroli E, Montanarella L. 2019 Policy instruments for soil protection among the EU member states: a comparative analysis. *Land Use Policy* **82**, 763–780. (doi:10.1016/j.landusepol.2019.01.017)

254. Wingeyer AB, Amado TJC, Pérez-Bidegain M, Studdert GA, Varela CHP, Garcia FO, Karlen DL. 2015 Soil quality impacts of current South American agricultural practices. *Sustainability* **7**, 2213–2242. (doi:10.3390/su7022213)

255. Webb A, Kelly G, Dougherty W. 2015 Soil governance in the agricultural landscapes of New South Wales, Australia. *Int. J. Rural Law Policy Spec. Ed.* **1**, 1–16. (doi:10.5130/ijrlp.i1. 2015.4169)

256. Chukov SN, Yakovlev AS. 2019 Soil and land categories in the modern legislation of Russia. *Eurasian Soil Sci.* **52**, 865–870. (doi:10.1134/ S1064229319070020)

257. Fromherz NA. 2012 The case for a global treaty on soil conservation, sustainable farming, and the preservation of agrarian culture. *Ecol. Law Q.* **39**, 57. (doi:10.15779/Z38BC49)

258. Soil Desertification & Sustainable Agriculture Specialist Group. 2019 Mid-Year Report. Online.: IUCN World Commission on Environmental Law (WCEL). (https://www.iucn.org/sites/dev/files/ content/documents/soil_specialist_group_ mid_year_report_june_2019.pdf)

259. Hazelton PA. 2006 Australian examples of the role of soils in environmental problems. In *Function of Soils for Human Societies and the Environment* (eds E Frossard, WEH Blum, BP Warkentin), pp. 141–147. London: Geological Society. (doi:10.1144/gsl.Sp.2006.266.01.12)

260. de Franchis L. 2003 *Threats to Soils in Mediterranean Countries: Document Review. Plan Bleu Papers 2*. Sophia Antipolis, Valbonnes, France: Plan Bleu for UNEP. (https://citeseerx.ist. psu.edu/viewdoc/download?doi=10.1.1.460. 6771&rep=rep1&type=pdf)

261. Gonzalez Lago M, Plant R, Jacobs B. 2019 Re-politicising soils: what is the role of soil framings in setting the agenda? *Geoderma* **349**, 97–106. (doi:10.1016/j.geoderma.2019.04.021)

262. Heuser IL. 2018 Development of Soil Awareness in Europe and Other Regions: Historical and Ethical Reflections About European (and International) Soil Protection Law. In *International Yearbook of Soil Law and Policy 2017* (eds H Ginzky, E Dooley, IL Heuser, E Kasimbazi, T Markus, T Qin), pp. 451–474. Cham, Switzerland: Springer. (doi:10.1007/978-3-319-68885-5_24)

263. Moallemi EA *et al.* 2021 Evaluating Participatory Modeling Methods for Co-creating Pathways to Sustainability. *Earth's Future* **9**. (doi:10.1029/ 2020EF001843)

264. Koch A *et al.* 2013 Soil Security: Solving the Global Soil Crisis. *Global Policy*. **4**, 434–441. (doi:10.1111/1758-5899.12096)

265. McBratney A, Field DJ, Koch A. 2014 The dimensions of soil security. *Geoderma* **213**, 203–213. (doi:10.1016/j.geoderma.2013. 08.013)

266. Bouma J. 2020 Soil security as a roadmap focusing soil contributions on sustainable development agendas. *Soil Secur.* **1**, 100001. (doi:10.1016/j.soisec.2020.100001)

267. Hill R. 2017 The Place of Soil in International Government Policy. In *Global Soil Security* (eds DJ Field, CLS Morgan, AB McBratney), pp. 443–449. Cham, Switzerland: Springer. (doi:10. 1007/978-3-319-43394-3_41)

268. Field DJ, Morgan CLS, McBratney AB. 2017 *Global Soil Security*. Cham, Switzerland: Springer. (https://doi.org/10.1007/978-3-319-43394-3)

269. Lilburne L, Eger A, Mudge P, Ausseil A-G, Stevenson B, Herzig A, Beare M. 2020 The Land Resource Circle: supporting land-use decision making with an ecosystem-service-based framework of soil functions. *Geoderma* **363**, 114134. (doi:10.1016/j.geoderma.2019. 114134)

270. Juerges N, Hagemann N, Bartke S. 2018 A tool to analyse instruments for soil governance: the REEL-framework. *J. Environ. Policy Plan.* **20**, 617–631. (doi:10.1080/1523908X.2018. 1474731)

271. Ginzky H. 2020 Good Governance for "Sustainable Management of Soil" on National and International Level: How to Do It? In *Legal Instruments for Sustainable Soil Management in Africa* (eds H Yahyah, H Ginzky, E Kasimbazi, R Kibugi, OC Ruppel), pp. 35–54. Cham, Switzerland: Springer. (doi:10.1007/978-3-030-36004-7_3)
