## [Peer Review File · Royal Society Open Science]

Review History

RSOS-201994.R0 (Original submission)

Review form: Reviewer 1

Is the manuscript scientifically sound in its present form?

Yes

Are the interpretations and conclusions justified by the results?

Yes

Is the language acceptable?

Yes

Do you have any ethical concerns with this paper?

No

Have you any concerns about statistical analyses in this paper?

No

Recommendation?

Major revision is needed (please make suggestions in comments)

Comments to the Author(s)

See attached (Appendix A).

Review form: Reviewer 2

Is the manuscript scientifically sound in its present form?

No

Are the interpretations and conclusions justified by the results?

No

Is the language acceptable?

No

Do you have any ethical concerns with this paper?

No

Have you any concerns about statistical analyses in this paper?

No

Recommendation?

Reject

Comments to the Author(s)

This paper provides an engaging narrative on the evolution of soil and land governance and its cultural context. It is less successful in providing an objective critique of the current state and trajectory of soil governance. A recommendation is to either develop it as a straightforward polemical essay for publication in a more opinion-orientated journal, or to use it as the basis for a systematic and robust scientific review that presents data and analysis about soil degradation, its economic consequences and the success or otherwise of existing policy and associated measures. The current manuscript falls somewhere between these two options, albeit it is closer to the former at present.

Decision letter (RSOS-201994.R0)

Dear Dr Peake

The Editors assigned to your paper RSOS-201994 "Saving the ground beneath our feet, part 1: Establishing criteria and priorities for governing soil use and protection" have now received comments from reviewers and would like you to revise the paper in accordance with the

reviewer comments and any comments from the Editors. Please note this decision does not guarantee eventual acceptance.

Please submit your revised manuscript and required files (see below) no later than 21 days from today's (ie until the 27th of March). Note: the ScholarOne system will 'lock' if submission of the revision is attempted 21 or more days after the deadline. If you do not think you will be able to meet this deadline please contact the editorial office immediately.

on behalf of Professor Brian Reid (Associate Editor) and Agnieszka Latawiec (Subject Editor)
openscience@royalsociety.org

Associate Editor Comments to Author (Professor Brian Reid):

I apologies of the delay in the review process - this has been due to difficulty in finding reviewers. Reviewer comments have now been received from two reviewers and, based on these, I unfortunately need to recommend reject and allow resubmission. As you will see in the comments the issue is systemic to the style and rigour of the reporting. Personally, I really like the assemblage of information and encourage you to overhaul the writing and resubmit.

Reviewer 1:

Comments to the Author:

This manuscript presents the evolution and potential development of cultural and legal frameworks relevant to the valuation and governance of soil resources. It assembles and contextualises a wide-ranging set of perspectives and trends relating to soil management, from early agriculture to the present. It concludes by re-stating the need to close gaps in an effective governance of soil - emphasising that legal measures are likely insufficient without an accompanying and persuasive political and economic rationale for soil protection.

The style of the manuscript is more towards that of a polemical essay than a scientific review or perspective. There is a large and developing literature on soil governance, including historical and cultural aspects, which is not adequately explored. There is no systematic summary of existing legal frameworks or reference to those in the literature or that have been prepared by international and national bodies.

A weakness is the lack of a method statement for the review and analysis process. A systematic one would have better revealed the extent of existing theory about soil valuation and governance. Additionally, from a scientific perspective, there is a lack of summary data to, for example, support statements about rates of land and soil degradation.

Reviewer: 2

Comments to the Author:

This paper provides an engaging narrative on the evolution of soil and land governance and its cultural context. It is less successful in providing an objective critique of the current state and trajectory of soil governance. A recommendation is to either develop it as a straightforward polemical essay for publication in a more opinion-orientated journal, or to use it as the basis for a systematic and robust scientific review that presents data and analysis about soil degradation, its economic consequences and the success or otherwise of existing policy and associated measures. The current manuscript falls somewhere between these two options, albeit it is closer to the former at present.

===PREPARING YOUR MANUSCRIPT===

===PREPARING YOUR REVISION IN SCHOLARONE===

Author's Response to Decision Letter for (RSOS-201994.R0)

See Appendices B & C.

RSOS-201994.R1 (Revision)

Review form: Reviewer 2

Is the manuscript scientifically sound in its present form?

Yes

Are the interpretations and conclusions justified by the results?

Yes

Is the language acceptable?

Yes

Do you have any ethical concerns with this paper?

No

Have you any concerns about statistical analyses in this paper?

No

Recommendation?

Accept with minor revision (please list in comments)

Comments to the Author(s)

The manuscript successfully assembles and provides a commentary on the evolution of a fuller appreciation of the importance of soil as a natural resource that requires specific governance. There are some gaps in addressing the legal framework for soil governance: for example the importance and availability of soil information, soil education (of citizens generally as well as land managers), the design of soil protection measures and their implementation (fiscal and other incentives as well as regulatory requirements). However, the manuscript provides a valuable commentary on the aspects that it does address. The difficulty of disentangling soil from land in the policy and legal context is identified and grappled with helpfully, although and inevitably, the focus of the discussion on existing initiatives is mainly land-centric because this is where existing soil protection is mostly found.

I am slightly uneasy that the manuscript lacks the necessary scientific emphasis appropriate to the journal but leave that to the editors' discernment.

A few small edits are suggested as follows:

line 321 replace 'excluding' by exempting' and delete 'as ifdefault'. Insert new sentence: "The assumption being that the role of farmers in protecting the rural landscape made controls on agricultural land use unnecessary."

line 384 'aeons' not 'eons'

line 384 "chronic soil degradation tends to be associated with rural poverty". This statement does not seem quite correct considering that intensive production in regions of higher wealth, for example of vegetable and root crops, is associated with serious soil degradation.

Decision letter (RSOS-201994.R1)

Dear Dr Peake

The Editors assigned to your paper RSOS-201994.R1 "Saving the ground beneath our feet: Establishing priorities and criteria for governing soil use and protection." have now received comments from reviewers and would like you to revise the paper in accordance with the reviewer comments and any comments from the Editors. Please note this decision does not guarantee eventual acceptance.

Please submit your revised manuscript and required files (see below) no later than 21 days from today's (ie 06-Jul-2021) date. Note: the ScholarOne system will 'lock' if submission of the revision is attempted 21 or more days after the deadline. If you do not think you will be able to meet this deadline please contact the editorial office immediately.

Kind regards,
Royal Society Open Science Editorial Office

on behalf of Professor Brian Reid (Associate Editor) and Agnieszka Latawiec (Subject Editor)
 openscience@royalsociety.org

Associate Editor Comments to Author (Professor Brian Reid):

In keeping with the reviewer comments, the authors have made extensive revisions to improve the manuscript. However, two issues persist.

- Firstly, the use of emotive phrasing, for example, but not limited to: relentless loss; before getting to grips with the thorny issues; at the mercy of the creators and implementers; transformed and even devastated; this gradual fall from grace.
- Secondly, the use obtuse vocabulary, for example, but not limited to: embody the same axiomatic concept; one prescient promulgation of reduced tillage; vociferous advocates; eclectic think tank, the Club of Rome, published the now iconic.

Sometime both issues are conflated in parallel, for example, but not limited to: Something so eternal and fundamental as soil, which has even been revered as a deity in its own right [195], might have been expected to have acquired an aura of intrinsic value, but it was ostensibly this its extrinsic value that made it a prize for which it was worth fighting, or even dying. Why styling it out in this fashion might wash for a book it does not work for a journal manuscript. The authors need to take stock of their writing style, offer a more objective account and make the language more accessible to a global audience.

In addition, the manuscript is very long. There is 26 pages of preamble prefacing the core of the manuscript (i.e. the content that begins with jurisprudence). The authors need to scale back the first 26 pages. A reduction of a third can easily be achieved through pragmatism and simplifying long-winded statements. The later half of the manuscript, while better conceived, will similarly benefit from abridging. The authors need to cut back on the anecdotal content and repetition of points already made. The current length (clean version 44 pages excluding reference) needs to be reduced to a maximum of 32 pages (at this the paper would still be long, but manageable).

Reviewer comments to Author:

Reviewer: 2

Comments to the Author(s)

The manuscript successfully assembles and provides a commentary on the evolution of a fuller appreciation of the importance of soil as a natural resource that requires specific governance. There are some gaps in addressing the legal framework for soil governance: for example the importance and availability of soil information, soil education (of citizens generally as well as land managers), the design of soil protection measures and their implementation (fiscal and other incentives as well as regulatory requirements). However, the manuscript provides a valuable commentary on the aspects that it does address. The difficulty of disentangling soil from land in the policy and legal context is identified and grappled with helpfully, although and inevitably, the focus of the discussion on existing initiatives is mainly land-centric because this is where existing soil protection is mostly found.

I am slightly uneasy that the manuscript lacks the necessary scientific emphasis appropriate to the journal but leave that to the editors' discernment.

A few small edits are suggested as follows:

line 321 replace 'excluding' by exempting' and delete 'as ifdefault'. Insert new sentence: "The assumption being that the role of farmers in protecting the rural landscape made controls on agricultural land use unnecessary."

line 384 'aeons' not 'eons'

line 384 "chronic soil degradation tends to be associated with rural poverty". This statement does not seem quite correct considering that intensive production in regions of higher wealth, for example of vegetable and root crops, is associated with serious soil degradation.

===PREPARING YOUR MANUSCRIPT===

===PREPARING YOUR REVISION IN SCHOLARONE===

Author's Response to Decision Letter for (RSOS-201994.R1)

See Appendix D.

Decision letter (RSOS-201994.R2)

Dear Dr Peake

On behalf of the Editors, we are pleased to inform you that your Manuscript RSOS-201994.R2 "Saving the ground beneath our feet: Establishing priorities and criteria for governing soil use and protection." has been accepted for publication in Royal Society Open Science subject to minor revision in accordance with the referees' reports. Please find the referees' comments along with any feedback from the Editors below my signature.

Please submit your revised manuscript and required files (see below) no later than 7 days from today's (ie 22-Oct-2021) date. Note: the ScholarOne system will 'lock' if submission of the revision is attempted 7 or more days after the deadline. If you do not think you will be able to meet this deadline please contact the editorial office immediately.

on behalf of Professor Brian Reid (Associate Editor) and Agnieszka Latawiec (Subject Editor)
openscience@royalsociety.org

Associate Editor Comments to Author (Professor Brian Reid):
Comments to the Author:
The revised manuscript:

RSOS-201994.R2 - Saving the ground beneath our feet: Establishing priorities and criteria for governing soil use and protection.

is much improved. I reread the pages with a heightened vigour. The progression of the prose is simpler and slicker. This has transformed the readability of the work.

Please accept with minor corrections:

There are a few instances where Latin terms could be italicised, for example: *per se*, *ad hoc*,
Final proofread

===PREPARING YOUR MANUSCRIPT===

one version should clearly identify all the changes that have been made (for instance, in coloured highlight, in bold text, or tracked changes);

===PREPARING YOUR REVISION IN SCHOLARONE===

-- Ensure that your data access statement meets the requirements at https://royalsociety.org/journals/authors/author-guidelines/#data.

You should ensure that you cite the dataset in your reference list. If you have deposited data etc in the Dryad repository, please only include the 'For publication' link at this stage. You should remove the 'For review' link.

-- If you are requesting an article processing charge waiver, you must select the relevant waiver option (if requesting a discretionary waiver, the form should have been uploaded, see 'File upload' above).

-- If you have uploaded any electronic supplementary (ESM) files, please ensure you follow the guidance at <https://royalsociety.org/journals/authors/author-guidelines/#supplementary-material> to include a suitable title and informative caption. An example of appropriate titling and captioning may be found at https://figshare.com/articles/Table_S2_from_Is_there_a_trade-off_between_peak_performance_and_performance_breadth_across_temperatures_for_aerobic_scope_in_teleost_fishes_/3843624.

Author's Response to Decision Letter for (RSOS-201994.R2)

See Appendices E & F.

Decision letter (RSOS-201994.R3)

Dear Dr Peake,

I am pleased to inform you that your manuscript entitled "Saving the ground beneath our feet: Establishing priorities and criteria for governing soil use and protection." is now accepted for publication in Royal Society Open Science.

on behalf of Professor Brian Reid (Associate Editor) and Agnieszka Latawiec (Subject Editor)
openscience@royalsociety.org

Appendix A

Reviewer comments on:

RSOS-201994

Saving the ground beneath our feet, part 1: Establishing criteria and priorities for governing soil use and protection

An erudite and well written paper. Although in places, particularly the conclusion the language is obtuse for a non-native English speaker. At times the manuscript takes a robust and objective scientific line, while at others the writing is rooted in history, views and opinion. The combination of styles makes the piece readable and engaging. However, on many occasions, the writing strays and this reduces the credibility of the piece. Most of the comments below are recommendations to help the paper chart a truer course.

The assembled content of the paper is well synthesised and the drawing together of the many strands of evidence within each section is skilfully handled.

However, the conclusion is unwieldy and needs to be rewritten. Currently, the authors introduce a lot of new elements in the conclusion: asbestos, toxic waste dumps, Covid, cooler/safer environments etc. I appreciate these are counterpoise and/or context for the argument. But these are not supported by the text of the manuscript. The authors must be more objective in framing the conclusion based on the material *actually presented* in the paper.

The conclusion is not the place to introduce a raft of new perspectives. If you really want to make new points (e.g. see list above) put these earlier in the narrative. The conclusion should *conclude* (ideal with a handshake to the aims stated at the end of the introduction). Currently, the ship is rudderless, the aims and structure of the manuscript need to be stated clearly at the end of the introduction.

Specific comments:

Title	This currently runs as “Saving the ground beneath our feet, part 1: Establishing criteria and priorities for governing soil use and protection” As there is no part 2 – that I am aware of – under review “part 1” should be deleted. The authors might also want to review the need for such a long title. Either the first or second clause would be fine alone, in my opinion. This said it would seem logical to put priorities before criteria.	
pp	line	
2	25	“and protection of agricultural soils from conversion” – unclear, what do you mean “conversion”? I appreciate this is elaborated on later – but at this point the reader need clarity.
	31	“threatens food security, and soil functions and ecosystem services” reword: threatens food security, soil functions and ecosystem services.
	32	“Yet individual instances often go unnoticed or at least unaddressed, falling between the stalls of agricultural and environmental law, being left to the planners who don’t see themselves as responsible for soils”. Too complex – split into two statements.
	35	“This article is the first part of a two-part article that will review and analyse the issues presented. The second part of this article will focus on the issue of prime farmland preservation with detailed case studies and proposals to address some of the existing governance shortcomings.” As far as I am ware there is no part 2. Mention of a part 2 needs to be removed and abstract concluded with the aims

		of the paper. “review and analyse the issues presented” is too vague in this regard – more detail is needed.
	45	“Agricultural land conversion, Farmland preservation, Land take, Soil ecosystem services, Soil governance, Soil security” – review use of capitals.
3	Quotes	Move the quotes to below “introduction”.
	42	While creative: “In this hypothetical world that curious messy material we call soil would presumably exist, if at all, only in recreational theme parks or abandoned wilderness”, this statement is overly sensational and unnecessary. Delete or recast.
	46	“The existential reality is that despite extraordinary technological advances, especially in the fields of engineering, biotechnology, computation and communication, we are as dependent on the soil for our sustenance as we have ever been (1)” – odd time to start the citation. Given the lack of citation to what is opinion and outlook I don’t think this statement in isolation needs citation. Either go back to the start and tighten up the citation or omit starting it here.
	51	“Soil ecosystem services is a relatively new academic subdiscipline” – if anything this needs a citation more than the reference linked to other elements of the onward paragraph.
4	63	Insert “arguably: “advanced 3D tomography has in fact revealed that, arguably : “Soil is the 64 most complicated biomaterial on the planet.” (5).”
	68	“conversion of toxic sewage and decomposing matter into plant nutrients” – these are not good example of purely chemical processes. They are biologically driven. A better example of a chemical processes might be: regulating the adsorption and release of macro- and micro-nutrients.
5	89	“The bottom line however, is that we need soil, but it does not need us.” This is counter to the motivation of the paper. Surely.... governance and protection of soil is of the human realm? This is not a useful concluding line to the introduction. Furthermore, there is no statement of aims – this is needed.
	89	It would be useful for the introduction to conclude by explain the onward structure of the manuscript – give the reader a sense of the sections that lie ahead.
7	164	“USDA” in full at first use.
8	214	It would be worth mentioning links between agricultural policy and payments/incentives made to farmers/landowners. The UK, in post-Brexit adaptation, is currently developing a new Agriculture Bill and in its extension the use of public money to support the provision of public goods (under ELMS). I see this is acknowledged L229-
9	220	“including in” , just “in” would do?
	234	“, agriculture finds itself more implicated than ever as a cause of environmental crisis rather than its victim (65).” This might be better qualified with some facts on carbon emission linked to agriculture vs other sources. I realise you are conflating climate forcing with other environmental damage e.g. via pollution. Nonetheless, a clear statement on agriculture’s contribution to climate forcing would be helpful.
	238	“existential” – this must be the 4 th or 5 th time you have used this word. It’s wearing thin....
10	266	“his then leaves rural land use conversion at the mercy of the creators and implementers of planning laws, who may feel that soil per se is outside their jurisdiction, despite the soil-related implications of the land use changes that

		their legislation may encompass. There is in effect a blind spot whereby the legal instruments with the greatest power to transform soil, sometimes irreversibly, are often framed and worded with little or no reference to soil” - This passage and the preamble is succinct to the context of the paper and I would recommend these ideas are incorporated into the abstract to make it more informative.
11	298	“such costs remain intangible (77), and often irreversible” wording... are costs irreversible. Sentence needs review.
11	304	“of which some examples are provided in 304 Part 2 this article” - remove
14	398	“in the round” – not needed – remove.
14	402	“swallowed up by” – bit too much – reword.
15	432	“one of pure economics” – add encasing commas
16	461	“It is hard to imagine a less efficient use of land than growing rapeseed in fields around Romford.” This is rather parochial for the RSOS’s global readership - remove.
17	489	“ad hoc” in italics.
	494	“Food and Agriculture Organization (FAO)” I think you referred to FAO in full earlier. Please check.
18	510	Might be an idea to define jurisprudence for the reader. RSOS is a science platform...
	512	“is” – are?
	516	“for dealing with soils” – reword.
19	560	Here might not be the best place, but, there are instruments of biodiversity net gain e.g. in the UK where development (captured under planning) has an obligation to not only off-set biodiversity loss associated with the development but to make a net-gain in biodiversity (somewhere else). The authors might want to weave this into their discussion (somewhere).
20	575	“of soil” – in other instances you keep ecosystem services broad i.e. not limited to soil ecosystem services. Review use here and throughout – what do you mean to incorporate?
21	626	“understood” – not the best choice of word, not least given your nod to Da Vinci. Perhaps change to “appreciated” – or something else....
	638	“could point the way forward to how policymakers and planners could apply more” – too many coulds – reword.
22	642	“this” - these?
	645	“a” before national
24	709	“ecosystem services issue” – rephrase “issue” is not the right word.
		“which he clearly regarded as somewhat eccentric” – this is not clearly sustained in your narrative – remove.
	713	“ore a matter of degree rather than kind” – not making sense to me – rephrase.
	711	I am not sure this section is needed, nor helping the paper.
25	731	Life on land (SDG15) should be highlighted as a specific soiling facing SDG - then link soil to the wider gamut of SDGs.
Table 1		Consider font, layout and format. Do you really want the table to appear like this in print?
29	864	“Upland peat can also have a critical role in flood control” – this is a distraction to the flow – suggest remove.
30	890	REEL – in full.
Conclusion		This is rather long – perhaps not ideal to engage the “casual” reader. I would suggest it is abridged to make it more accessible.

31	910	“being flouted or subverted” this is not sustained by the content of the manuscript – remove.
	916	“status quo” italics
		“The greatest “winners” are typically speculative developers and any officials or agents who may be the recipients of their largesse (81), but even for the beneficiaries these gains are short-term and arguably Pyrrhic victories because of the cumulative negative impacts which ultimately tend to be visited on the whole community” – does this really need to be expressed with unusual vocabulary and in such a complex sentence? Are those you seek to influence limited to the best read/educated in the native English-speaking world?
		“holier-than-thou observers” – as above.
		“literal example of Hardin’s tragedy of the commons (222), that is a plethora of small land use infringements resulting in a larger loss to the community – death by a thousand cuts ” – as above, who are you serving with this statement?
		“A serendipitous outcome of the COVID-19 pandemic, with its attendant restrictions on travel and industrial activity, has been the widely acknowledged reduction in air pollution, and even increased incursions of 969 wildlife in urban or peri-urban areas around the world. These are subjective indicators (or co-variables) of a significant drop in global C emissions after a surprisingly short period of national lockdowns (223). However, climate scientists have emphasised that this reduction, though measurable, will do little or nothing in its own right to halt long-term climate change (224). On the contrary, if easing of lockdown has a “bounce-back” effect as governments and business redouble their efforts to re-energise their economies, this current fall in emissions will be cancelled out entirely and serve only as a brief reminder of what a slightly cleaner, cooler and safer world could look like.” – this has no place here, it has no roots in the main text and has nothing to do with soil governance.
		“The first part of this article has reviewed” – remove reference to part 1 and 2.

Saving the ground beneath our feet

Establishing priorities and criteria for governing soil use and protection

Lewis Peake¹ and Cairo Robb²

¹ University of East Anglia, School of Environmental Science, Norwich, NR4 7TJ, United Kingdom;
corresponding author: l.peake@uea.ac.uk; +447940240683, +441603702560

² Legal Research Fellow, Centre for International Sustainable Development Law, <https://www.cisd.org>

Abstract

The relentless loss and impairment of soil functions and ecosystems services across the globe calls for a fundamental reconsideration of soil governance mechanisms. This review charts the history and evolution of national and international soil law and seeks to unravel the challenges that have contributed to this failure in governance. It describes and categorises law and policy responses to different soil threats, and identifies a worrying widespread absence of legislation for oversight and protection of agricultural soils from urbanisation, as well as a lack of clear legal mechanisms to set and determine national priorities for soil protection. A reduction in the world's prime farmland threatens soil ecosystem services, including food security, carbon storage, and biodiversity. Falling between the stalls of agricultural and environmental law, the fate of farmland is often left to planners who do not see themselves as responsible for soils. Consequently, legal instruments with the greatest power to transform soil, sometimes irreversibly, are often framed and worded with little or no reference to soil. Nevertheless, emerging conceptual frameworks might offer positive outcomes. The authors advocate robust holistic policies of soil governance and land use planning that place soil ecosystem services and natural capital at the heart of decision making. (197 words)

Keywords

agricultural land conversion, farmland preservation, land take, soil ecosystem services, soil governance, soil security

Introduction

Land belongs to a vast family of whom many are dead, a few are living and a countless host are still unborn.

- Traditional Ashanti adage [1]

It is the value of the improvements only, and not the earth itself, that is individual property.

- Thomas Paine, *Agrarian Justice* (1795).

These stylistically contrasting statements from two different cultures and geographic zones essentially embody the same axiomatic concept: that the ground beneath our feet provides so much more than the ground beneath our feet, and that holistically it transcends title deeds, boundary fences and even national borders. Soil is a vital and enduring resource that constitutes the metaphorical as well as the literal foundations of human civilization. This multidimensional and multifunctional resource has always provided a range of benefits, of which food production is just the most obvious example. As the world's population has soared and almost every parcel of its terrestrial surface has been assigned to the various national states, and then further subdivided and transformed by governments, private corporations or individuals, the need to safeguard this increasingly degraded resource is more urgent than ever.

A recurring theme in futuristic scenarios is the prediction that humans will one day grow or make all of their food in laboratories and vertical farms, deploying soilless hydroponics. Technically this is perfectly possible, should humans survive long enough to achieve it but, as in the case of many societal endeavours, the challenges are primarily social, economic and political. So for the time being at least, this is science fiction. The existential reality is that despite extraordinary technological advances, especially in the fields of engineering, biotechnology, computation and communication, we are as dependent on the soil for our sustenance as we have ever been. In fact, we are arguably more dependent on the multifaceted and critical functions of soil than ever, as our dominance as a species has resulted in greater demands and stressors on our environment.

Soil ecosystem services (SES) is a relatively new academic subdiscipline [2], but human awareness of the holistic functionality of soil is as old as civilization, if not older. When the Romans recorded some of the oldest known soil conservation laws, in the 6th century *Digesta* of Justinian, they described a range of scenarios each with their own legal argument, not necessarily related to soil productivity, for example, that might lead to disputes between neighbouring farms, including the damage to a neighbour's soil from uprooting trees or channelling water, or the harm that might arise from eroding sediment [3].

A millennium later, coincidentally not far from Rome, Leonardo da Vinci directed a major land reclamation project which depended on the ability of soil to absorb and then release water, and also to trap and redirect it

as required. These two historical examples both touch on aspects of soil not directly related to primary production, but instead to its various roles now collectively referred to as ecosystem services. It was also da Vinci who famously declared: “*We know more about the movement of celestial bodies than about the soil underfoot.*” [4]. Indeed, while many scientists have famously claimed that the human brain is the most complex structure in the known universe [5], advanced 3D tomography has in fact revealed that, arguably: “*Soil is the most complicated biomaterial on the planet.*” [6].

The almost infinite utility of soil is universal. In addition to the familiar and very direct role that soil has in providing our food, fibre and timber, we often forget its many other interconnected functions, for example:

- Physical: provision of building or landscaping material; stable substrate for infrastructure;
- Chemical: regulating the adsorption and release of nutrients;
- Biological: constituting biodiverse ecosystems and communities of useful organisms, conversion of toxic sewage and decomposing matter into plant nutrients, genetic reservoir;
- Hydrological: water absorption, storage and filtration; flood control;
- Cultural: repository of cultural heritage and archaeological information, recreation;
- Climatic: carbon (C) storage.

The last of these would have been almost inconceivable to our forebears and has only recently assumed paramount importance, approximately since the beginning of the new millennium.

The other side of the coin is that in certain circumstances soil, like any other natural resource, can deliver negative impacts, such as emitting greenhouse gases (GHGs), harbouring pests and diseases, or contributing to the siltation of water, the accumulation of sediment, dust storms and landslides. These events are reminders that good governance of soil is also about minimising ecosystem “disservices” as well as safeguarding essential ecosystem services.

We live in a time of anthropogenic environmental crisis, of unprecedented climate change and species extinction. Hardly a day goes by without another meteorological record being broken or ecological threshold being breached somewhere. A report just released by WWF states that: “*Since 1970, our Ecological Footprint has exceeded the Earth’s rate of regeneration,*” and currently exceeds the world’s capacity to support humanity by 56%. In other words, for 50 years we have been exceeding our running costs and depleting the planet’s natural capital. This is despite a technological expansion of the world’s biocapacity of 28% since 1960 [7]. The irony of this crisis is that it is the direct result of human ecological “success” – and excess – but the impact is untold human suffering and ultimately perhaps our demise. Soil is an integral component within this system, both as an enabler and a victim of exponential human expansion.

In Hardin’s 1968 essay *Tragedy of the Commons* [8], he presents a parable of herders grazing their cattle on common land as an analogy for our accelerating use of the Earth’s finite natural resources. Every herder

knows that the shared pasture has limited carrying capacity, but each one may also decide that it will be in their interest to add more animals to their herd, because their personal gain will (they believe) exceed the collective harm. The inevitable result is overgrazing and general ruin for all (probably involving soil degradation, although Hardin did not mention that!). For the first time in history, we are facing a tragedy of the commons on a global scale. In his book *Collapse* Jared Diamond [9] highlights how most societies eventually disintegrated for combinations of partly avoidable reasons, including resource depletion, and chillingly suggests that all of these antecedents may simply be pilot studies for the ultimate collapse.

In this critical review we trace the long historical path of the status of soil, in terms of its value as a natural resource for society and with respect to its evolutionary governance. In the course of the analysis we present and describe categories of anthropogenic threat to soil and the types of legal instruments that can be applied to address these threats. There is also a discussion of the semantics surrounding legal aspects of soil and land, and an attempt to disambiguate and illuminate the use of terminology. The current state of the art of soil governance is explored in depth and a key finding is a failure to prevent or even acknowledge the worldwide loss of prime farmland and the consequential implications of this for society. The review concludes on a hopeful note by looking to emerging approaches such as the soil security conceptual framework, proposed as a way of translating critical soil knowledge into sustainable development policy, and to holistic land evaluation methodologies that incorporate SES into land use planning.

Methods

This study is underpinned by the role of soil governance in helping to achieve the UN Sustainable Development Goals (SDGs), mitigating and adapting to climate change and contributing to a safer future for humanity. Nearly 300 references have been cited out of a larger number inspected, but the literature search was intentionally not constrained to a narrowly directed keyword search. Moreover, our intention was to add new content to the debate. Our initial premise was that if SES, which we acknowledge are critical to human wellbeing, are being dangerously diminished, is this due in part to a failure of governance? If so, to what extent is this a failure of legislation, implementation or both? In terms of *priorities* and *criteria*, as mentioned in the subtitle, where are the gaps and weaknesses, and are there opportunities to address these? Furthermore, is this particular mix of problems and potential solutions entangled with wider issues of terminology, perception and how society has historically managed and valued soil? What would be the most effective contributions from soil science, scientists and practitioners?

From this starting point, we undertook an open-ended literature review. Particular words or phrases became key search terms, such as combinations of “agriculture”, “biodiversity”, “conservation”, “contamination”, “conversion”, “degradation”, “ecosystem services”, “erosion”, “farmland”, “governance”, “land”, “land take”, “land use planning”, “legislation”, “non-agricultural”, “preservation”, “protection”, “SDG”,

urbanis(z)ation”, “urban sprawl”, “soil”, “soil sealing”, and so on, depending on which section of the article was the focus. However, an exhaustive search with such a wide and holistic remit would have been impractical and ultimately yield diminishing returns. Instead, the approach taken was to home in on a few specific areas of interest, based on our own judgement and current trends, and to follow these narratives organically in promising directions. The ensuing strategy was that by selective close reading and analysis we synthesised key points, such as our own categorisations of the terminology and information. By presenting in this way, the intention was twofold: to provide the reader with some clear signposts within a large and complex domain; and to provide ourselves with a framework within which the context of any given issue could be better understood and deconstructed.

Data and cases studies from the literature are referred to throughout to support our arguments. In addition, a comprehensive search was conducted to abstract legislative data from FAOLEX, the UN Food and Agriculture Organization (FAO) online database of policies and laws. The FAOLEX online database documents as far as possible legal instruments related to agriculture and natural resources in force in the nearly 200 countries of the world. This is a hierarchical database organised, by country, into the following groups, of which those marked with an asterisk were reviewed by the lead author (where they existed for the country in question): ***Policies****; ***Legislation: Agricultural and rural development****, ***Climate change, Cultivated plants, Disaster risk management, Environment****, ***Fisheries, Food and nutrition, Forestry*, Land and soil*, Livestock, Sea, Water, Wild species and ecosystems****; ***International Agreements****. Of these groups, *Land and soil* was the most useful, but it was necessary to widen the search further to obtain a comprehensive picture, because in many cases this group was primarily devoted to the administration of land ownership and land reform. These groups are not mutually exclusive, so some legislation appears more than once. Not surprisingly, for some countries, e.g. the US or China, the list of entries runs to hundreds.

Very occasionally the name of the FAOLEX entry alone was broadly sufficient to explain its meaning, e.g. “Soil conservation law”, but in most cases it was necessary to select the embedded hyperlink to access its descriptive summary (usually in English). However, in many cases, ambiguous or over-simplified wording made it necessary to select a further hyperlink to the full text of the legal instrument, either within FAOLEX or on the websites of national governments. These documents were often in the local language, and sometimes also in the local script, e.g. Russian or Arabic. Where it was felt useful to do so, and where feasible, Google Translate was used to interpret the text. In a few instances further corroboration or clarification was obtained from academic articles in this article’s bibliography or via personal communication from individuals professionally familiar with the legislation. In a small number of interesting cases local experts were contacted to translate or explain some of the wording and, furthermore, asked whether the laws in question were implemented and effectively governed. Approximately 1% of countries had either created no such instruments or lacked any accessible data in FAOLEX. This

information was compiled during the summer and autumn months of 2020. A spreadsheet was created with one row per country and columns indicating categories of legal instrument, and from this it was possible to calculate percentages of countries with such measures in force. For our purposes the FAOLEX groups are merely retrieval mechanisms, unrelated to our categories that were derived on the basis of our analysis of soil governance. The resulting data is presented in Table 3 and discussed in the section *The modern governance of soil resources worldwide*, towards the end of the article.

While our research was being conducted FAO was in the process of creating a similar, but soil-specific, repository (SoiLEX) from the data in FAOLEX, but this was unavailable for our purposes at the time¹. In terms of structural organisation and ease of use, SoiLEX is superior to FAOLEX and would have speeded up our search. However, SoiLEX is limited to highly soil-specific categories of legislation, whereas FAOLEX is wider in scope and, crucially for our purposes, contains laws relating to the protection or preservation of categories of land such as prime farmland and natural ecosystems.

Some caveats are necessary. Even setting aside subsequent updates, such a summary can only ever be approximate, and is undoubtedly incomplete. Several polities listed in FAOLEX as countries are actually a number of countries with differing laws (e.g. the UK), and other countries consist of separate provinces or states with differing laws (e.g. the US). The laws of some countries are recorded only in the local language or script. Furthermore, many instruments, especially with respect to policies, are generalised, and in some cases it can be difficult to ascertain legislative intent. For example, a law stating that land will be classified as urban or rural could imply protected status or simply spatial planning data; the latter was assumed unless explicitly defined. Many legal statements are also terse to the point of ambiguity or even circularity, for example: “*The land is the main national wealth, which is under special protection of the state...ensuring rational use and protection of land,*” [11] iterated many times, but with little or no further qualification. Furthermore, this is purely a review of officially documented policies and laws, not an assessment of enforcement or effective governance.

Soil as a valued resource and legal entity

¹ SoiLEX is described as a global database on national legislation on soil protection, conservation and restoration to facilitate access to information on the existing legal instruments in force and bridge the gap between the various soil stakeholders. The online platform is intended to facilitate the search for national soil legal instruments, and the understanding of the different legal areas relevant to soil management and protection, as well as the exchange of experiences in soil governance between countries and regions. 10. Global Soil Partnership web page. 2020 Contribute to the GSP newest tool: SoiLEX..

The root and “commonwealth” of civilization

Intuitive human awareness of the relationship between soil type and natural resource availability is self-evident from history. Even before the invention of agriculture, land that retained moisture and plant nutrients was the place to find rich pickings which, as any historian will tell you, is why the first settled civilisations developed in wide flat river floodplains and deltas.

When Herodotus said, Egypt is the gift of the Nile [12], he was explicitly referring to the land-forming deposition of black silty clay by the annual river flood, which incidentally has been occurring at the rate of 1mm per annum (1m per millennium [13]). The ancient Egyptians set their calendar according to these floods and called their nation state *Kemet* which means black land or dark earth, in contrast to the barren *Deshret*, or red land, stretching out on either side. The polygonal state boundaries of modern Egypt would have been meaningless to its former inhabitants who knew that their country stopped where the irrigated feddans met the dry sand. The eponymous black soil was not simply a metaphor for their country; as far as they were concerned, it was their country.

Human dependence on productive soil and the impact of its degradation on civilisation throughout history is well attested [14-18]. Mesopotamia experienced at least two catastrophic regime changes due respectively to soil salinity and soil erosion [19, 20]. The Greek and Roman writers make multiple references both to the benefits of soil quality and the threat of soil degradation, and by the 1500s in Europe soil was regarded as the most important factor of an economy. History is also full of examples of land-related wars and conquests, often correlated with soil quality [21-23], and tragically the killing of environmental defenders, including those seeking to protect land and soil quality, continues today [24, 25].

The evolution of soil law

The earliest known attempts to put an economic value on soil, that is productive land, were Chinese soil classification and maps created 2000 years BCE for taxation purposes [26]. Other ancient peoples also independently derived similar systems, for example in Mesoamerica [27]. However, while there are many historical examples of land ownership rights and transactions [28], it is difficult to pin down when laws or policies were first introduced to protect soil or land as a vulnerable or scarce resource in its own right, that is part of the natural capital that belongs to society and is for the common good, assets formerly referred to as the *commonweal* or “commonwealth”. One prescient promulgation of reduced tillage appears in the *Manusmriti*, a collection of Hindu laws thought to date from 1000 BCE which advises against the use of iron-tipped ploughs “...because they injure the Earth and its creatures.” [29].

The *Statutes on Agriculture*, compiled during the Qin Dynasty of the late 4th century BCE is China’s, and possibly the world’s, earliest known set of soil-related laws which mandate, *inter alia*, field sizes and the

correct season for earth-moving [30]. Even before this period, Chinese rulers made it known that feeding the nation was paramount for its survival [31]. The *Statutes* were followed a century later, during the Han Dynasty (206 BCE-8 CE), by the *Book of Fan Shengzhi*, China's oldest surviving agronomic treatise which provides extraordinarily detailed and precise instructions on almost every aspect of soil management [32]. It would take perhaps another thousand years before this work was eclipsed by the Arab agronomic scholars of Andalusia [33]. Although this document is presented as advisory rather than legally binding, its author, *Fan Shengzhi*, was the equivalent of the Minister of Agriculture and its arrival coincided with a radical transformation of China's agriculture, so it seems probable that it was regarded by farmers as de facto law. Perhaps the oldest surviving documented legislation explicitly concerned with soil is that contained in the *Digesta* of the Eastern Roman emperor Justinian, published in the 6th century, but including many laws dating from centuries earlier. While there was no overt implication that soil per se was a scarce resource in the Roman Empire, these edicts leave the reader in no doubt that soil was of great importance to everyone involved in its use, from the question of who benefited from its bounty to who was responsible for its maintenance [3]. Over the subsequent centuries the value of soil or the need to protect it appears occasionally in the literature, but hardly the fear of an absolute scarcity of soil, or of vast areas of it becoming unusable. However, two millennia after those early Chinese and Roman ordinances were recorded, that mindset gradually began to change as time and time again human communities rudely bumped up against the limits to growth.

In the modern era, the world's first legally protected forest reserve, for the purpose of conservation, was created on the Caribbean island of Tobago in 1776 to preserve the "*fertility of lands*" [34]. However, the first specific instances of soil legislation were usually initiated in response to the threat of severe soil erosion. In many cases this was triggered by sudden catastrophic events which were typically the synergistic result of inappropriate land management combined with extreme climatic conditions. The most iconic example is the US Dust Bowl of the 1930s which dealt a massive psychological blow to world's wealthiest nation. Since that time, soil erosion has been perceived as an ever-present threat to American farmers and food production, so it is no surprise that the first major piece of soil-related legislation in the US was the 1935 Soil Conservation Act, which still survives in a modified form. This act was followed by a steady stream of further soil and land-related legislation in the US and elsewhere, but it was not the first such law in the modern world.

Possibly the first modern ordinance explicitly designed to prevent soil erosion, was the *General Order banning tree-cutting in Hawkesbury River valley*, NSW, Australia in 1803 [35] and by the 1870s there was widespread awareness of the problem of erosion, for example in colonial Africa [36]. The environmental movement which had emerged in Europe and America since the Industrial Revolution also had a growing influence on policy [37, 38] and forest protection laws that embodied the principle of soil conservation, if

not the wording, began to appear around the world, for example in the US from 1873 onwards [38] and New Zealand in 1874 [39]. In 1894 grazing was banned on federal reserves in the US to prevent land degradation [38] and in 1901 the Indian state of Punjab, then under British administration, passed what might be considered the first true soil and water conservation law, the *Punjab Land Preservation Act*, which essentially remains in force today [40]. However, it was Iceland, in 1907, that passed the first national parliamentary soil conservation act [41].

A significant result of the Dust Bowl and President Roosevelt's generous response to it, in parallel to the practical and legal assistance given to farmers, was the huge contribution made by the United States Department of Agriculture (USDA) and American scientists to the worldwide study and professional management of soils. With hindsight, the officially established causes of the crisis, and some applied aspects of the USDA's arguably overwhelming response, have come in for criticism [42-44], as well as the narrow focus of its globally influential research agenda [45]. However, the far-reaching impact of this intense period of activity, barely perceived outside the sphere of soil science, is perhaps unparalleled in human history. Building on a legacy of international scholarship, especially in Germany and Russia [46], in the space of a decade at least three trailblazing conceptual tools were created: the Soil Taxonomy, the Land Capability Classification (LCC) methodology and the Universal Soil Loss Equation (USLE). These would radically transform the international science, management and legislation of soils.

From the 1930s onwards other nations rapidly fell into step, passing soil conservation laws and other forms of soil and land legislation. While food security was a major driver, so was the loss of rural landscapes and wilderness. Other kinds of environmental protection were also implemented throughout the 20th century, such as the creation of green belts to protect countryside around cities, even as early as 1901 [47]. In the 1940s, however, underscoring the growing appreciation of protecting farmland, World War II had multiple impacts on food production and distribution, especially in Europe, forcing many countries to become more self-reliant. The postwar world was one in which agricultural productivity took on a new significance and the new technique of land evaluation was enthusiastically applied [48, 49].

In the UK, for example, even before the end of the war, the Scott Report on rural land use signalled a new approach to planning that incorporated both landscape amenity and food production: "*Land which is classified as good agricultural land should only be alienated from its present use if it can be shown to be in the general national interest.*" [50]. A member of the Scott Committee opined: "...*carving-out of a place for agriculture in the world of planning was perhaps the most significant achievement of the Report,*" and that, "*maps of land utilisation, land classification, and types of farming...became recognized tools of the planning trade.*" [51]. The Report has also been described as a prescient aspiration for sustainable development [52]. This report fed into the 1944 White Paper, *The Control of Land Use* [53] which paved the way for one of the earliest and most radical planning policies of its kind anywhere, the UK 1947 Town and

Country Planning Act which nationalised land use regulations and has been emulated worldwide [54]. Having ring-fenced farmland, the reforming Labour government bolstered this with financial support for farmers via the 1947 Agriculture Act.

It was therefore especially unfortunate that in the same year as these landmark pieces of legislation, when the British government embarked on the Groundnut Scheme in what was then Tanganyika (now Tanzania), in the final years of its overseas Empire, it eschewed the very land evaluation methods that might have prevented a promising project collapsing into a catastrophic misadventure [55]. Meanwhile, the land classification techniques adapted from the USDA LCC methodology and being put to work by soil surveyors in postwar Britain, were being applied not to find new land to cultivate, but to identify and protect existing prime farmland. While the driving force of the LCC had been on-site farm management and soil conservation, the British system was primarily a tool of the new planning system. The latest incarnation of this system in England and Wales is the 1988 Agricultural Land Classification (ALC) set of guidelines whereby prime farmland equates to what is designated the “best and most versatile” (BMV) land [56]. In 1984 the USDA Natural Resources Conservation Service (NRCS) developed its own system that is oriented towards planning legislation, *Land Evaluation and Site Assessment* (LESA) [57].

The postwar era is also notable for the creation of a number of global and regional institutions such as the FAO, the International Union for Conservation of Nature (IUCN), and the European Union (EU), all of which have been instrumental in relation to land and soil governance. Three years after its creation in 1945, the FAO embarked on the world’s first formal attempt by an international institution to address soil conservation as a global issue, initially as a published study which relied heavily on data from the US and China [58], and subsequently in the form of a conference held in Florence in 1948 [59]. Underpinning such initiatives, especially in the wake of the US Dust Bowl, was a growing awareness of the economic impact of soil degradation, not just to individual land users, but to society at large, with far-sighted hints at the future valuation of ecosystem services and implications for government policy [60]. Similar analyses followed in the ensuing decades [61, 62], and soil law continued to evolve, with the US taking the lead, particularly with respect to soil conservation [63].

In 1949 the French botanist Aubreville coined the term desertification to describe the transformation of productive agricultural land in arid or semiarid areas to a desert-like, uncultivable state which has lost ecological viability due to a combination of climatic factors and human activity [64]. By 1958 the Chinese government acknowledged the threat that desertification posed to the wellbeing of nearly 200 million people, and have initiated afforestation programmes since 1978 in response [65]. While desertification is a major cause of land degradation in so-called drylands, it became mired in controversy because of a misunderstanding both of its meaning (i.e. a conflation with the natural process of desert formation and alleged expansion) and of its root causes. Nevertheless, the concept of desertification became and continues

to be a significant driver of sustainable land management (SLM) initiatives designed to tackle land degradation [66], where other approaches had faltered [67] – see “Soil degradation” below.

The conflicted role of agriculture

To highlight one salient point that was a sign of things to come, the UK postwar agricultural policy came in for subsequent criticism for virtually excluding farmers from planning regulations as if in recognition of their role in protecting the rural landscape, of which they were an essential component, they were exempt from harming it by default [52, 68]. In the decades that followed WWII geopolitics stabilised and agricultural productivity soared due to a range of technological innovations including mechanisation, agrochemicals and biotechnology [43], which from the 1960s reached the low income economies in the form of the Green Revolution [69]. This period represents a radical reframing of society’s attitudes towards the environment and agriculture [70]. In some parts of the world, at least, food security was no longer regarded as a serious threat, whereas agriculture itself was increasingly perceived as the primary threat to the environment, and hence a mixed blessing. Both the expansion and intensification of agriculture led to unprecedented wildlife habitat loss, encompassing deforestation, wetland drainage, conversion of grassland to arable and hedgerow clearance. Added to this were many other environmental impacts associated with industrial agriculture, including pollution, soil degradation and fuel-intensive inefficiency [71-73].

The publication of *Silent Spring* by Carson [74] led to a major international turning point in changing attitudes and policies in response to the excessive use of pesticides and herbicides. In Europe in the 1970s and 1980s the Common Agricultural Policy (CAP) came to epitomise the ironic twin dilemma of taxpayers subsidising overproduction at the expense of environmental destruction [53, 68, 70]. With the emergence into the public domain of climate change science since the late 1980s [75] and growing evidence of the significant climate-forcing influence of GHG emissions resulting directly and indirectly from farming [76], alongside the degradation and destruction of carbon reservoirs, and the ongoing loss of biodiversity [77], agriculture finds itself more implicated than ever as a cause of environmental crisis rather than its victim.

It could be argued that the cumulative effect of this gradual fall from grace has been to diminish the value of arable land in the public perception, by association. More specifically, any potential value of such land tends to be eclipsed by its current condition, for example its reduced biodiversity and highly disturbed and often degraded soil. Meanwhile, as an academic discipline that straddled geology and biology, but on the periphery of both, soil science found itself virtually a subdiscipline of agriculture at least halfway into the 20th century [19]. Hence soil, by association and also because largely hidden, rarely evokes the kind of intrinsic concern that attaches to wildlife or landscape. This is a topic that will be revisited in subsequent sections.

The relationship between soil and land

It is not pedantic to clarify that soil and land are not the same thing, but in the context of governance they are inextricably linked. Indeed, many languages confuse the two in common parlance. For example, most French people (and dictionaries) would offer the word *terre* (land) as a translation of the English word soil, whereas French soil scientists are clear that the correct word is *sol*, commonly regarded as the French word for ground or floor. Weigelt *et al.* [78] stress the distinction between soil and land, but also identify a lacuna in the literature to address the critical importance of governing them together to achieve sustainability. It is also axiomatic that an absence of land rights goes hand-in-hand with poor soil management and land degradation [79-82]

From a legal perspective the concept of land is normally operative in matters of ownership and boundaries, and also in terms of spatial and territorial planning. Soil tends to enter the legal fray when a portion of it is transformed, harmed or threatened by a specific activity, which might be performed by the owner of the land containing the soil, or by another party. Recognition of this distinction in itself signals two very different aspects of soil law. The former perspective would generally be categorised as property rights, whereby the injured party is primarily the landowner, rather than the soil. The latter perspective offers protection to soil as a legal entity in its own right, even from its “owner”, for the common good.

Where the blurring of land and soil becomes problematic is in instances of proposed changes which are framed only in relation to land despite having significant impact on the soil within, or adjacent to, that land. This opacity can apply to legislation specific to agriculture, the environment or planning, and as a result salient aspects of land use conversion may fall between the gaps among all three. For example, the creators of agricultural laws may feel that their remit, as far as soil protection and conservation is concerned, extends only to the direct impact of farming on soil, while the creators of environmental laws may feel that their remit for soil protection and conservation extends only to an unspecified component within a larger ecosystem or protected landscape.

This then leaves rural land use conversion at the mercy of the creators and implementers of planning laws, who may feel that soil *per se* is outside their jurisdiction, despite the soil-related implications of the land use changes that their legislation may encompass. There is in effect a blind spot whereby the legal instruments with the greatest power to transform soil, sometimes irreversibly, are often framed and worded with little or no reference to soil. Raising this in a meeting with government soil scientists caused heads to nod in agreement and perusing comparative laws reveals that this is a recurring theme throughout the world [83]. Furthermore, the modern image of agriculture as an industry that pollutes soil and threatens ecosystems, and with a long history of consuming forests and other natural landscapes, risks devaluing the public concept of

agricultural land, as if the sealing of such land would not, in and of itself, result in a loss of environmental benefits.

Anthropogenic threats to soil and land

Natural events have routinely transformed and even devastated bodies of soil for eons, but such events are part of soil-forming processes, to which ecosystems adapt. Humans have learnt to transform soil to their advantage but the overall impact of human society on soil, both intended and unintended, has been profound. Short-term gains have often been at the expense of long-term harms. The picture is further complicated by the fact that many of these human impacts manifest themselves not as discrete effects but by exacerbating and unbalancing natural processes, such as soil erosion, salinization or flooding.

Broadly speaking, soil or land is typically vulnerable to three kinds of anthropogenic impact, set out in Table 1.

Table 1: Types of anthropogenic threat to soil or land

1. Soil degradation (here meaning any harm done to soil, other than ‘soil sealing’)
2. Conversion of natural ecosystems or other semi-natural and uncultivated land to another form of land use, such as agriculture, forestry, urbanisation, or industry, including mining or energy infrastructure (conversion is also known as ‘land take’)
3. Conversion of farmland to urban or industrial use ² [84] (‘agricultural land take’)

1. Soil degradation

The most extreme anthropogenic impact to soil is soil sealing, which is strongly related to urbanisation, but not equivalent in meaning. Soil sealing, the permanent covering of soil, for example by concrete or tarmac, occurs outside urban areas, for example on roads or farm infrastructure, while conversely urban zones include much land that is not sealed, such as gardens, parks and playing fields. Nevertheless, urbanisation inevitably goes hand-in-hand with soil sealing, which is a drastic change that is effectively permanent and overwhelmingly negative with regard to SES [85-87].

² The conversion of farmland to forestry or natural regeneration is not normally regarded as a “threat”, though there are scenarios where this may be the case with questions arising, for example, in relation to afforestation of peatlands, or replacement of biodiverse agroforestry systems with plantation monoculture.

In this context, degradation refers to any harm done to soil other than sealing (which is encompassed within categories 2 and 3), and which is therefore potentially ameliorated by remedial measures. Soil degradation may take the form of acute problems such as erosion, contamination, salinization, acidification, sodification and compaction or other forms of structural collapse. The term soil degradation has also been widely used to describe gradual and chronic loss of productivity, typically the result of over-exploitation and inadequate management, and invariably bound up with organic C depletion, reduction in soil biodiversity and low nutrient status. While acute soil degradation is often the result of breaches of regulations, chronic soil degradation tends to be associated with rural poverty, a less conspicuous but significant factor in the recent Indian farm protests [88], and generally requires supportive rather than punitive legislation. These problems can be complex and interrelated, and sometimes partly the result of, or exacerbated by, natural causes or intrinsic to certain soil types.

Soil degradation is a huge global issue, sometimes due to local legacies of malpractice, with unintended consequences. In the worst cases large tracts of land have been abandoned or declared unfit for food production with severe economic impacts. Yet this phenomenon and the wider concept of land degradation have been claimed, at times, to have been subject to “gross exaggeration and hyperbole” [67]. While stressing the severity of land degradation and its interlinkages with other environmental problems such as climate change and biodiversity loss, Gisladottir and Stocking [67] are equally assiduous in criticising several institutional studies that they suggest relied on subjective methods or unsound extrapolations. In the case of soil erosion, for example, a common mistake was to base regional estimates on plot experiments despite the fact that most “lost” soil will be deposited elsewhere. Veracity aside, the authors point out that far from marshalling support, such errors have led ultimately to scepticism and underfunding.

In the first study of its kind, UNEP commissioned the Global Assessment of Soil Degradation (GLASOD) which was conducted by the International Soil Reference and Information Centre (ISRIC). GLASOD addressed the period from World War II to 1990, relying on local expert opinion rather than direct measurement, and presented its results in areas of land as opposed to the less meaningful tonnes of soil eroded employed by many earlier studies [89]. Soil degradation was defined as: “...a process which lowers the current and/or future capacity of the soils to produce goods or services”. The results were broken down by region, types of land use (arable, pasture or woodland) and categories of degradation. In summary 15% of all land worldwide (almost 2 billion hectares) was classed as degraded, equating to 23% of inhabited land (14% seriously). Soil erosion was by far the most widespread form of soil degradation, affecting 83% of degraded land.

Approximately 20 years later ISRIC scientists reviewed GLASOD and used satellite imagery to measure an overlapping and subsequently ongoing period (1981-2003) of land degradation, which it defined as: “...a long-term decline in ecosystem function and productivity”. This was the first study of its kind to use

normalized difference vegetation index (NDVI) as a proxy for changes in net primary productivity (NPP), with appropriate adjustments for climatic variability and land-use change [90]. Without entirely rejecting the GLASOD results, which they declared unreliable and unreproducible, the authors detected a further 24% of the total land area that was degrading *mostly in addition to* GLASOD's 15% already degraded, i.e. a cumulative process. Although not explicitly stated by the authors, this would imply that at least a third of the world's land appeared to be degraded to some extent by 2003³. Additional key findings were that land degradation was not primarily a problem of drylands, as had long been proposed, but was in fact more widespread in humid areas, and that it was a worldwide phenomenon, albeit with distinct hotspots. However, it is important to bear in mind that NPP can sometimes be increased in the short-term by intensive land use which degrades the land [91].

In a 2015 follow-up study, including some of the same authors and updated to include data up to 2011, the global estimate of degrading land was revised down slightly to 22%, with 14% of land showing some improvement [92]. Also in 2015 FAO/ITPS⁴ claimed that 33% of land is degraded and that that figure could reach 90% by 2050 [93]. A 2016 study based on similar methods to the ISRIC analyses produced a global figure of 30% degraded land [94]. An analysis of terrestrial NDVI in 2017 revised the total down further still to 23% [95]. The 2017 UNCCD Global Land Outlook report states: "...over the last two decades, approximately 20 per cent of the Earth's vegetated surface shows persistent declining trends in productivity." [96]. Drawing on a wide range of studies, the 2018 IPBES⁵ *Assessment Report on Land Degradation and Restoration* concludes that 75% of the Earth's land surface is either transformed or degraded to some extent by human activity [97]. According to IPBES land degradation impairs 3.2 billion livelihoods, costing 10% of the annual global gross product, whereas the benefits of restoration are typically 10 times higher than the costs,

While global land degradation can appear overwhelming, awareness of soil degradation in the narrower sense is growing and attitudes are changing, and this subtle shift may be key to reversing the trend. Most countries now have some form of regulation addressing soil degradation, typically embedded within their agricultural legislation, and in a fair number of these countries the law is strictly enforced. In many cases,

³ The authors made no attempt to merge the data because of the contrasting methods of the separate studies and the fact that the GLASOD data was unverifiable, but one of the authors opines that: "It is not unreasonable to judge that all land now under anything less than natural climax vegetation is degraded in terms of biodiversity and stored carbon and nitrogen," (perhaps 66% of the world's land surface and rising) (David Dent, personal communication.)

⁴ The Intergovernmental Technical Panel on Soils (ITPS), made up of 27 soil experts representing all the regions of the world, was established at the first Plenary Assembly of the Global Soil Partnership held at FAO Headquarters in 2013.

⁵ The Intergovernmental Science-Policy Platform on Biodiversity and Ecosystem Services

soil policies go beyond prohibitive and punitive laws dealing with acute soil degradation, and incorporate support and incentives to foster improved long-term soil health and protection [98].

Compared to land use conversion (see below), soil degradation, even though it may be the result of deliberate law-breaking, is generally recognised as an unintended and unwelcome consequence, so there are fewer obvious social pressures to oppose such laws and the global trend has been for greater governance. However, the cost of remediation can be prohibitive, whether for those held responsible or those who ultimately pay the bill, for example taxpayers, so circumvention is an ever-present risk [99], even though evidence shows the virtual spiral of paybacks possible from rehabilitation and restoration [97]. China stands out as the country probably most afflicted by soil degradation, of every kind, but by contamination in particular, on a scale that appears irreparable and unaffordable, requiring more than the world's entire wealth to remediate, according to *The Economist* [100]. Nevertheless, no country seems more acutely aware of this or determined to remediate it than China itself [101, 102].

2. *Conversion of natural ecosystems*

Natural ecosystems or semi-natural landscapes, that is land that is unoccupied or sparsely occupied, is uncultivated and largely unmanaged (at least according to currently prevailing notions of cultivation and management, [103]), are ubiquitously valued for a variety of reasons: their aesthetic characteristics and recreational potential; their educational and scientific worth; and, with increasing urgency, their vital ecosystem service benefits, and importance for many Indigenous communities. Aside from unvegetated and largely uninhabitable areas, such as mountains or deserts, this category of land includes forest, wetland, native grassland (e.g. prairie or steppe) and heath-/moor-/peatland. Globally the primary threat to natural and semi natural lands is from agriculture, responsible for 80% of deforestation according to WWF [7], but the full picture is nuanced and dogged by claims of both overestimation and underestimation [104].

Furthermore, the dynamics of forest change are complicated by temporary deforestation and instances of forest gain, for instance via afforestation, reforestation or regrowth on abandoned farmland. A recent study [105] which used satellite imagery to distinguish various categories of forest disturbance, estimated that between 2001 and 2015, of a global total tree cover loss of 314 Mha, only 27% (5 Mha each year) was permanent land use change for commodity production, i.e. large-scale agriculture, mining and energy infrastructure. Urbanization accounted for less than 1%. Impermanent changes included forestry (26%), shifting agriculture (24%), and wildfire (23%). Regional contrasts were stark with permanent deforestation overwhelmingly accounting for most of the tree loss that occurred in Latin America and Southeast Asia (> 60%), while Latin America also had the world's largest total loss. Characteristic drivers of change are oil palm plantations in Southeast Asia and soybean and ranching in Latin America [105]. Indonesia and Malaysia stand out as areas of increased deforestation while the rate of loss had declined markedly in Brazil,

but since the change of regime in Brazil in 2018 forest destruction is reported to have increased again dramatically [106]. Most other regions have relatively low rates of permanent deforestation, with most temporary disturbance associated with managed forestry and, in North America, Russia and Australia, wildfire. A handful of countries have even achieved a net forest gain.

Several authors have long argued that the abundance of available land in sub-Saharan Africa (SSA) is a myth in much of the region and even where natural forest exists near settlements, there are significant disincentives for farmers to encroach on such land [107]. Clearing forest is difficult and expensive for small farmers, and often occupies poor soil. The Curtis study confirms that most forest disturbance in SSA is due to impermanent shifting agriculture, but the alarm has been raised concerning the impact of rising and often uncontrolled urbanization on biodiversity in Africa [108].

Natural ecosystems tend to be well protected in the higher income countries, but historically this has not so much been due to wisdom or sound governance, but for more mercantile reasons. Many wild and majestic landscapes have survived largely because they lack the kind of natural resources that facilitate intensive land use and often present severe obstacles to development. For the most part wherever wilderness occupied flat land with productive soil, it was converted long ago. The stark reality is that very little such land remains, and some parts of the tropics have reached or are close to this point [109]. In 2009 Rockström *et al.* [110] suggested we are approaching the limits of the planet's cultivable land. This makes the preservation of natural and semi-natural land across the globe, all the more urgent, especially given its far-reaching environmental benefits, such as those relating to C storage, biodiversity and soil and water conservation.

The accelerating consumption of natural and semi-natural land by agriculture is not an inevitable consequence of population pressure and progress in the Global South, but it is usually an inevitable consequence of uncontrolled development. Forest loss often follows farmland loss, which can be avoided without penalising the rural poor (see below). A combination of agricultural support and incentives, and rational territorial planning can raise farm incomes, improve national food security and obviate loss and expansion of farmland [111-115]. Byerlee *et al.* [111] also observe that while market-driven intensification of agriculture (e.g. greater demand for oil palm or soybean) tends to consume more land, technology-driven intensification tends to save land, i.e. raises productivity without expansion. However, they stress that sustainable intensification is possible only in tandem with environmental protection and incentives.

Nevertheless, conversion of natural lands to agriculture is rarely as drastic or permanent as urbanisation, and many countries are increasingly rewarding landowners, land managers or farmers, for example via payments for ecosystem services (PES), to implement more environmentally friendly farming and land management practices [116]. Putting aside the question of how seriously each national government responds to the loss of natural ecosystems, or governs it, that issue is relatively unambiguous and virtually every country on Earth,

theoretically at least, protects such land with environmental laws and policies. Furthermore, soil protection per se is rarely the driving force behind such legislation, which tends to be framed in terms of ecology, biodiversity or watershed protection. Having said that, some of the earliest environmental laws in the modern era were primarily a response to soil erosion caused by deforestation [35], which is also still frequently cited in such legislation.

3. Conversion of farmland to urban or industrial use

The vast majority of land consumed by urban and industrial expansion throughout the world is agricultural land, and this type of threat, which encompasses much soil sealing, is arguably the least reversible and the most profound of these three categories. Furthermore, for sound socioeconomic reasons, urban settlement has historically developed in close proximity to, and often surrounded by, prime farmland. Hence it is frequently the best quality land of regions that is most vulnerable to conversion [85, 117-119]. Peri-urban farmland is particularly attractive to developers because unless it is subject to strict green belt or zoning laws, it lacks the environmental protections of natural ecosystems⁶, is usually cheaper to develop than brownfield sites [120] and tends to be conveniently situated in terms of existing facilities and infrastructure. Furthermore, a study of 25 EU countries found that the land most at risk was that slightly further out from city boundaries which was typically on more fertile soil than that closer to the city [121], although interestingly the influence of CAP subsidies reduced the rate of land take in general.

The mere potential for future urban development, sometimes even without planning permission, typically increases the value of farmland by, for example fivefold in Korea [122], sixfold in Morocco [123], nearly tenfold in New Zealand [124], and even by orders of magnitude more than this: 100-fold in Ghana over ten years [125] and similar average multipliers for land earmarked for development in Britain and Japan [126]. These market distortions create enormous commercial pressure for landowners to sell, especially in lower income countries where agricultural livelihoods may be precarious and lack government support [127], but do not necessarily leave poorer farmers with any viable long-term benefits. With the exception of a few who may be able to exploit new urban markets, most farmers will use this temporary windfall to pay off debts and be forced to become landless farmers or seek other occupations [128]. The prospects of re-investing in cheaper land to cultivate are greatly diminished by the fragmentation of holdings [129] and the creeping outward shift of the agricultural zone onto lower quality land [128].

⁶ Even if natural ecosystems were an option, they often present many other obstacles such as remote location, uneven terrain and geological hazards, the need to clear vegetation or extensively drain, and an almost complete lack of existing infrastructure.

The view that urbanisation represents economic progress [130] has led many administrators and politicians, especially in lower income countries, to embrace urban expansion policies with enthusiasm [131], but this line of reasoning is embedded in a pre-millennial paradigm in which human betterment is predicated on economic growth without setting environmental limits on such growth. This largely market-driven worldview has been strongly criticised for at least 50 years [132, 133] and arguably much longer [134], as well as being relentlessly underscored by what is becoming known as the Anthropocene crisis [135]. At the local level there are serious concerns, even within government [136], that the wider implications of farmland loss are increasing food prices and imports, escalating rural poverty, land degradation and conflict [127, 128, 136-139], as well as the loss of critical ecosystem services [85, 136, 140].

However, it is worth addressing the arguments of authors who advocate farmland conversion, or are at least sanguine on the issue. Visser takes this argument to a purely economic extreme of treating natural resources as tradeable commodities or assets and even suggests that land markets would benefit from greater scarcity of farmland [141]. In 2010 Satterthwaite *et al.* [130] presented an ostensibly even-handed review of the positive and negative impacts of urban expansion, but ultimately argued that the world had abundant arable land and that urban populations were no longer dependent on their own agricultural hinterland because in a modern globalised world they can source food from far and wide. However, all else aside, this thesis presupposes reliable food supplies and reliable trading partners, missing the point that geopolitical instability and climate change are increasingly threatening global food security, and this applies even more so in the decade since that article was published [142], notwithstanding the additional risks associated with a global pandemic.

It might also be tempting to argue, in defence of conversion, that much farmland, often government-owned, has been abandoned or under-utilised for various reasons, such as de-population of rural areas or, as in the case of the former Soviet Union, incomplete land reform [143], and that this land could be recultivated. There may be limited capacity for rehabilitated farmland to compensate for conversion elsewhere [144], but this raises a number of issues. A common reason for abandoning farmland is the difficulty in extracting an adequate livelihood from that land, often because it is degraded [145] or marginal land [143]. Such land will tend to be intrinsically less productive so, even if farmers have the means to restore it, and can be persuaded to do so, a greater area might need to be cultivated to achieve requisite returns. Where productive land has been abandoned for socioeconomic reasons, such as a lack of infrastructure in remote areas, and steps are taken to remedy this, recultivation is feasible [146, 147].

However, abandoned farmland presents another important global narrative, for it is swathes of such land, in Russia, US, China, Australia, Latin America and elsewhere, that have inadvertently facilitated a global pattern of reforestation and rewilding [144] that goes some way towards buffering the effects of deforestation elsewhere. Recultivation of such land would be likely to reduce biodiversity and C storage

[144, 148], for it is hard to imagine farming systems that would achieve comparable environmental benefits on the same area of soil, other than carefully managed agroforestry systems or some other form of continuous perennial cover [149]. However, degraded land often has higher than average C sequestration potential so, while such land may be challenging for agricultural rehabilitation, it may be highly suitable for ecological restoration [150], although the slow process of natural regeneration may not necessarily be the most effective way to achieve this [151, 152].

There are also vociferous advocates of farmland conversion in high-income countries like the UK, for instance in relation to the London Metropolitan Green Belt (MGB). In some cases these are primarily socioeconomic arguments with respect to a scarcity of land for housing. [153]. However, as long ago as 1977 it was argued that the MGB policy was rigid and anachronistic in protecting all farmland regardless of its environmental value, while constraining suburban gardens which harboured more biodiversity [68]. This message, which has been echoed over the years by others [153-155], has some merit, but reprises the trope of industrial agriculture as the enemy of nature, and as such is contentious. Residential land, by definition, undergoes much sealing and topsoil removal, while the biodiversity of its gardens, though potentially of great value, is entirely at the mercy of the residents. Farmland, on the other hand, remains largely unsealed and, given appropriate policy incentives, such as PES, can become more biodiverse and provide greater ecosystem services, as has been a gradual trend in the UK since the late 1980s [156], and looks to continue with the proposed new Environmental Land Management scheme (ELMS) [157]. However, alongside this trend in the UK, and related to it, has been a gradual weakening of the protection afforded to BMV land [156, 158].

Where such authors have a much stronger case is in criticising the binary distinction between green belt land, where all farmland is protected regardless of its agricultural quality, and non-green belt rural land where prime farmland should be protected but is often developed [153]. Furthermore, Amati and Taylor [154] point out that city workers who cannot afford either city or green belt house prices are forced to live further out and commute in, increasing their C footprint, as well as exacerbating inequality. Brownfield sites are finite and if some land must be sacrificed for housing one can see why some argue that it could be preferable to release some low grade land within the green belt zone rather than build on BMV land beyond the MGB.

The MGB is one of the oldest green belt zones in the world and now finds itself constraining one of Europe's largest cities. With its protected open space continually under strain, primarily from pressure for housing and transit development, some of which does get planning permission in spite of the strict regulations, much more leapfrogs onto greenfield land beyond. Similar but younger zones have yet to experience such pressures, but in the case of the Ontario Greenbelt, for example, this is partly due to an approach which is arguably more integrated and enlightened, incorporating green infrastructure, principles

of environmental and economic sustainability, greater public participation and underpinned by a legal framework that is centrally owned and controlled by the provincial government, yet both robust and flexible [47, 154, 159]. To take the case in point, farmland preservation in Ontario is not simply a matter of locally ad hoc prohibition in the teeth of fierce opposition from planners and developers, but is integrated into a holistic policy that includes agricultural support, local food marketing, employment opportunities, recreation, tourism and ecosystem services in a regional context [154, 159-161].

For the reasons already stated, agricultural land take is rarely covered by either environmental or agricultural legislation. Evidence for this exists in FAOLEX [83]. One type of exception to this is where farmland is protected for its above-ground biodiversity value, e.g. high natural value (HNV) [121], but this is not the same thing as protecting land for its intrinsic SES value or, for that matter, its soil biodiversity. The last resort for farmland preservation, then, if it occurs at all, is usually some form of spatial or territorial planning. Still, the primary motive for such planning is typically urban expansion driven by population and commercial pressure, rather than rural protection [162, 163]. Not surprisingly, therefore, there is a distinct lack of robust legal instruments to prohibit or restrict agricultural land take and the approach taken varies considerably between and within countries, producing policies which are frequently complex and sometimes ambiguous or conflicting [53, 124, 136, 163-168]. Added to this is the prevalence of “informal” governance of planning regulations in many low- and middle -income countries [166, 169, 170].

Setting aside the national and wider global consequences, the greatest losers especially in the Global South tend to be the rural poor who become less able to afford to buy or lease land [139], and the wider population as food imports and prices rise due to a shrinking agricultural base [112]. Much of the land consumed by urban sprawl is common land or, in Indian parlance, common property resources (CPRs), on which many rural communities depend heavily [136]. Such communities also suffer disproportionately from the less appreciated degradation of SES that results from soil sealing, such as problems of sanitation, flooding or pollution [171, 172]. The greatest “winners” are typically speculative developers and any officials or agents who may benefit from their transactions [173], but the cumulative negative impacts will be felt by the whole community. Furthermore, the most strident whistle-blowers to these encroachments are not distant activists, but witnesses on the ground, from local academics [127, 139, 174] to journalists and blighted farmers taking their grievances to the courts, at least those who are able to do so [173, 175]. Social inequalities drive many to facilitate the very land use changes from which they gain the least, for example selling land simply to survive, and then converting natural ecosystems to replace what they have lost. There is much evidence that appropriate government support for farmers reduces the sale of farmland for non-agricultural development, as well as enhancing food security and biodiversity [121, 162, 176]. It is therefore imperative that any attempt to prohibit or restrict agricultural land conversion, must be accompanied by economic support and viable alternatives. No amount of legislation or governance will succeed without this.

While all three types of anthropogenic threats to land and soil listed in Table 1 above endanger ecosystem services, from a law and policy point of view, compared to soil degradation and natural ecosystem protection, land take is often “out of sight and out of mind”. For example, the Thompson Reuters Practical Law website (accessed September 2020) includes a very detailed global section summarizing the laws of 72 countries in a highly readable Q&A format on, for example, agricultural real estate or environmental impact. Yet none of the questions specifically address the prohibition or restriction of agricultural land take or preservation other than that of foreign ownership, i.e. the “loss” of land as a national asset.

4. Cross-cutting issues

A few academics argue that land conversion or degradation may be justified in some cases, on the basis that the economic benefits may outweigh the environmental costs [177], but despite the best efforts of environmental economists, such costs remain intangible [178], and the impacts often profound. Furthermore, as currently framed, economic benefits are not necessarily sustainable and are ultimately dependent on ecosystem services that economists have traditionally treated as “free” and, more to the point, as inexhaustible and unconstrained, for example, by planetary boundaries [110]. Economic benefits also accrue unevenly and not even necessarily in the national interest. The escalating potential profits from development can also foster an “unholy alliance” between the public and private sector that rewards elaborate circumvention of the law [162, 179] or outright infringements [80, 95-97, 167, 174]. However, it is also important to appreciate that what may appear to outside observers to be infringements may, in some cases, simply be the result of longstanding traditions of informal governance or customary tenure [173, 180], or simply a lack of institutional capacity [130]. In other situations the pressure to convert land may be political, for example the need for an administration to acquire land to meet housing or other developmental targets. Mining, energy infrastructure and other forms of industrial land use represent a very small proportion of land loss or degradation overall [181], although this is increasing [96] and cannot go unmentioned because of their extreme impact in some locations. In western Ghana, for example, gold and diamond mining has had devastating effects on rural communities and natural ecosystems [182, 183]. This activity includes both corporate mechanised extraction and illegal and artisanal hand-digging, as a form of low-input mining but also to provide material for construction. These impacts include all three types of anthropogenic threats to soil listed above, going far beyond the loss of land, and including local water contamination and depletion, illegal logging, extensive soil and subsoil removal, and sometimes violent land disputes. The highly lucrative and largely unregulated context in which these enterprises operate tolerates or even encourages infringements of the law, even by corporations operating legal concessions [183].

The jurisprudence⁷ of soil and land

The implementation of soil legislation

There are essentially two components to legally protecting or preserving soil resources: (i) applying a method of evaluating and prioritising areas or bodies of soil and (ii) implementing the requisite (and effective) legal instruments and governance structures. At the time of writing neither of these components is entirely absent from the world at large, but both exist only within a fragmentary and, in some cases, theoretical patchwork of initiatives.

So, before getting to grips with the thorny issues of establishing which forms of governance structures or legal instruments have been or might be most applicable to soils, one must consider what criteria to apply. This is essential whether focussing primarily on specified spatial areas of land or on the functional aspects of soils in relation to their current status or use. However, preceding even all of this is a minefield of ambiguous terminology to navigate.

Semantics and legal terminology

When authors or legal documents and policies refer to the way in which soil is affected by those managing the land it occupies, and hence the possible harms it may encounter, such as erosion, pollution, contamination, salinization, acidification, compaction and so forth, the word “protection” is the term most widely used. Many if not most countries have some form of explicit or implicit soil protection policy in place, but this is more often than not subsumed within other legislative and policy domains, for example dealing with the environment, agriculture or spatial planning. Soil legislation is commonly sub-divided further into categories of protection, reflecting the severity of particular problems within a given country or the gravity attached to the problem by its authorities. For whatever reason, certain soil laws are almost always framed in negative harmful terms and others in positive or remedial terms. Hence there are soil pollution or contamination laws, but hardly ever soil erosion laws. Instead there are soil conservation laws or occasionally, in a similar vein, soil improvement laws.

Harder to pin down is the terminology applied to the conversion (effectively the loss) of agricultural or other undeveloped rural land to non-agricultural use which typically incorporates substantial additions of infrastructure (e.g. housing). In this domain several terms are used, some in the negative or destructive sense, and others in the positive or protective sense. However, it could be argued that the term land conversion, though technically describing the act of change or loss, is neutral, because unlike the unintended

⁷ Jurisprudence here refers to both legal theory and legal systems..

consequences of the harms done to soil, this refers to a deliberate act with an intended outcome and various beneficiaries.

The many approximate synonyms for rural land conversion with a more negative connotation include urbanisation, urban spread/sprawl/expansion/encroachment, land take/consumption/loss, long-term land cover change and (in France) artificialisation [184]. Other phrases commonly used in connection with rural land conversion are land competition and fragmentation, of farmland in particular, because this is one of the ways in which urbanisation becomes self-perpetuating [185]. Soil sealing (defined previously, see *Soil Degradation*) is a phrase widely used, often in conjunction with land take, because they typically occur together, but the meanings are distinct. While land take refers to a change of use which, by definition, usually entails some loss of land resource from its former use, sealing constitutes a much more permanent and intrinsic alteration of the land surface, typically with more far-reaching consequences for soil functions, especially drainage, and potentially severe effects on biodiversity. Land take almost always involves some degree of sealing, and where it does, that constitutes a double impact – the spatial loss of land resource and the additional degradation of the ecosystem services that that resource provides in the round. However, sealing also occurs independently of more general land take, even in highly protected areas, for instance in the form of roads [86].

On the positive land-saving side of the negative land taking lexical coin, the term predominantly used in North America, and in many countries, is land preservation [186, 187], but the terms land conservation or land protection are used almost interchangeably with it in the US [188, 189] and throughout the world. In most cases these phrases are applied to specific bounded areas, at various scales, which could mean a protected zone or alternatively a cluster of holdings, a single estate or even one field. The expressions “no net land take” or “zero land take”, which have appeared in EU documents in recent years [120, 190], are similar in principle, but refer to overall quotas or targets, with the additional challenges relating to relevant spatial scale and overall quantification. In contrast ‘land sparing’ is used in juxtaposition to ‘land sharing’, referring to the concept of safeguarding biodiversity, either by sparing natural lands from agricultural use or by incentivising farmers to support more biodiversity [191].

Globally other words or phrases are employed to mean approximately the same thing as land preservation but some can have other connotations. Land conservation has historically been most strongly associated with nature conservation, in the sense of land that is not recognised as developed or cultivated and has some form of protected status. Such land is often threatened by agricultural incursion at least as much as by urban sprawl. The meaning is also not to be confused with soil conservation, a term normally used professionally and more widely to refer specifically to the prevention of erosion with respect to land management.

References to land protection are often synonymous with land preservation, but in some examples of legislation it is intended to mean soil protection, that is *in-situ* safeguarding. Confusingly, however, soil protection is even occasionally intended to mean land preservation. This semantic problem may sometimes be one of translation because the full texts of many laws appear only in their original language and script, so the average researcher must rely on translated summaries or just keyword lists [83]. Occasionally countries have all-embracing “soil protection” policies which include the concept of spatial land preservation in context [192, 193] or conversely all-embracing “land preservation (or protection)” policies which include the concept of soil protection [11, 193]. In some cases this blurring seems to be deliberate, such as when land preservation is promoted as a means of protecting SES [11, 86]. Pragmatically soil scientists sometimes need to accommodate themselves to the language of policy and spatial land use planning in order to engage with the process.

Putting a value on soil

At the root of our current environmental crisis is that humanity has traditionally treated abundant natural resources as free goods and services. Economists and accountants are no exception: cost-benefit analyses include tradeable assets and products, but routinely exclude the ever-present prerequisites of life by which we are surrounded. Furthermore, traditional cost-benefit analysis (CBA) is short-termism by definition because it usually does not even consider impacts outside a 25-year window [194]. When a tonne of soil or water is traded, the allocated price is normally derived from the aggregate cost of acquisition, processing and delivery. No attempt is made to assess the intrinsic value of the substance which therefore would appear to have no value by default. This applies to soil as an intrinsic good, but also to some extent to its economic potential, i.e. the land that contains it will have a market value and that value will by default be related to the land’s productivity, yet unquantified and uncoded SES have far-reaching economic implications.

Even in a society without monetary currency it would be relatively straightforward to calculate the economic value of a given volume of soil, in conjunction with the land it occupies, purely in terms of its capacity for life-sustaining primary production; ask any farmer. Something so eternal and fundamental as soil, which has even been revered as a deity in its own right [195], might be expected to have acquired an aura of intrinsic value, but it was ostensibly its extrinsic value that made it a prize for which it was worth fighting, or even dying. So although we now have modern techniques to assess the productive capacity of land resources, a conceptual process of “following the money” has been at the heart of most farm or settlement emplacement decisions since the Neolithic.

For at least 80 years soil survey and land evaluation has been the traditional tool at our disposal for assessing the productive and economic potential of an area of land, alongside any limitations or hazards it may present. A range of soil properties is recorded alongside other local environmental data and, where

appropriate, socioeconomic data. This data is combined to grade the capability or suitability of each parcel or zone of land according to specified use criteria. The modern terminology of multicriteria decision analysis (MCDA) is more a change of style than substance, to reflect the much greater reliance on information technology, especially GIS [196]. In the above approaches “criteria” is the operative word. Where these methodologies differ from analogous ad hoc historical techniques, is in their capacity to distil decades of practical and scientific observation to apply much greater predictive accuracy and precision to the decision-making process [197].

Land evaluation though is not the same thing as land valuation, though they have always been interrelated and both are relevant to spatial planning. Land evaluation methodologies were never intended to calculate the true, i.e. total and holistic, economic value of an area of land or a volume of soil (which are themselves two very different things), but rather the capability or suitability of distinct parcels of land for specified forms of primary production, for example crops, pasture or forestry. The purpose is for land use decisions, albeit often with profound economic implications. The purpose of land valuation, however, is generally to set prices for landowners and taxes for governing authorities, and has always primarily been based on location, which admittedly is historically bound up with soil quality and land use, but with a myriad of other factors too.

In contrast, soil has always provided a variety of extrinsic benefits beyond primary production which were recognised and appreciated long before we referred to them as ecosystem services. To what extent our ancestors valued (or devalued) specific areas of soil or land for reasons other than primary production, or obvious physical location, is not always tangible, though it is clear that many Indigenous communities have developed deep understanding of, and spiritual connections to, the land and soil on which they depend [103]. The holistic value of soil is increasingly becoming more widely apparent. A wider all-embracing appreciation of soil and a few attempts to evaluate it in that context date from the 1960s [198]. This concept has gathered momentum and, more recently above all, with respect to C.

While it may seem too neat to earmark the last decade of the last millennium as a critical turning point in time, it seems inescapable that this period represented an important paradigm shift in our collective understanding of the role of soils. The 1990s shines out as a lightbulb moment when soil science and climate science came together as never before. Although the role of C in the soil has been studied for at least 150 years and its place in the C cycle appreciated for much of that time, it was only in the 1990s that a flurry of papers explicitly linked soil C sequestration to GHG emissions and climate change [199-201] which, after several decades of growing evidence, had started attracting worldwide media attention since 1988.

At the onset of the new millennium soil scientists have more than ever before started to deconstruct and articulate the critical importance of SES [202, 203]. Experts and advisors on soil law have responded

accordingly, emphasising that the value and status of soil far exceeds its role in agriculture [204]. Taking this a step further, several groups of researchers, building on earlier attempts to evaluate ecosystem services [205] and characterising soil as a form of natural capital, have developed tentative frameworks and quantitative methods to enumerate and evaluate SES in the fullest sense [206-209], sometimes using case studies to apply monetary values [210]. To what extent this is entirely possible, feasible or even desirable, is debatable [198], especially with regard to qualitative or ethical issues, but these kinds of approach may point the way forward to how policymakers and planners could apply more meaningful criteria and priorities to the governance of soil and land protection.

Categories of soil governance

Juerges et al. [79] provide a summary of the types of institution applied in the service of soil governance (i.e. regulatory, economic, and so on) and the levels at which these operate, from global to local. In this section we present a very different cross-cutting breakdown, based on issues and intended outcomes. One can identify three broad categories of governance instruments applied to preserve or protect soil or land, usually at a national or subnational level, each with various subcategories, set out in Table 2 and elaborated further below:

Table 2: Categories of governance instruments applied to preserve or protect soil or land

CATEGORY	DESCRIPTION
1	Regulations to prohibit (or guidance to discourage)
(a)	- Enforced by penalties
(b)	- Facilitated by financial support
2	Restrictions on development or change of use
(a)	- Enforced by zoning laws
(b)	- Enforced by laws (or encouraged by guidance) based on ad hoc site evaluation
(c)	- Enforced by public acquisition of land
(d)	- Facilitated by financial support
3	Generic incentives to preserve land or to enhance its ecosystem services value, i.e. for
(a)	- Taking land out of agricultural production

(b)	- Converting intensively farmed land via extensification
(c)	- Declining either to develop land or to intensify its use

The first category comprises regulations and laws to prohibit (or guidance to discourage) certain unauthorised actions by landowners which may degrade land, e.g. pollution, sealing, construction, stubble burning, tree felling, or cultivation methods leading to soil erosion, acidification, salinization, compaction, and so on; whereby best practice may be either:

- a. Enforced by penalties for infringement or
- b. Facilitated by financial support, e.g. farming subsidies, PES or C credits

The second category comprises restrictions or guidance on development involving conversion of use, either to protect terrestrial natural resources (for the benefit of any or all of primary production, watershed management, biodiversity, landscape or cultural or historical value) or simply to retain a certain quota of agricultural or forested land within a state or designated region; whereby land conservation or preservation may be:

- c. Enforced by laws based on zoning, e.g. National Parks, nature reserves, green belt, urban growth areas, UNESCO World Heritage Site, UK Sites of Special Scientific Interest (SSSI) or heritage landscapes such as the UK Areas of Outstanding Natural Beauty (AONB);
- d. Enforced by more general laws (or encouraged by guidance) not specific to the protected areas described in 2(a), based on ad hoc site evaluation, such as an Environmental Impact Assessment (EIA), and thereby affording protection to sites of ecological or landscape value, or prime farmland (e.g. based on ALC or LESA criteria);⁸
- e. Enforced by public acquisition of land, such as land trusts;
- f. Facilitated by financial support, e.g. using LESA to obtain Purchase of Development Rights (PDR) conservation easements (US).

The third category encompasses generic incentives to preserve land or to enhance its ecosystem services value (or disincentives to develop, such as the removal of subsidies), in the form of:

⁸ A legally binding example is the Indian Prohibition on Conversion of Agricultural Land for Non-Agricultural Use (No. 16 of 2010).

- a. Taking land out of agricultural production, purely for the purpose of conservation, e.g. EU set-aside, US land retirement, rewilding;
- b. Converting an area of intensive agricultural production (arable cropping) to one of more extensive agricultural production (such as pasture, silvopasture or agroforestry) or forestry;
- c. Declining to develop land, or to intensify its use, in order to maintain ecological or ecosystem services value (such as woodland, meadow, prairie, wetland, moor or heath) that could otherwise be legally developed.

Overlapping categories

The above categories are presented as a useful framework for analysis of soil governance. There are of course many instances of overlap. For example, regimes that are not targeted specifically at soils can still fall within the general categories above. There are also numerous cases where soil governance instruments span a number of the mechanisms listed above. The examples below illustrate these points.

Environmental Impact Assessment (EIA) of projects, Strategic Environmental Assessment (SEA) of plans and programmes, and related types of assessment, may apply to some of these categories, especially categories 2(a) or 2(b), and can be helpful in drawing attention to threats to natural resources, but there can be limitations in their application to soils. In the US, for example, EIA is applied primarily for federally-funded projects [211] which is partly why the NRCS developed the LESA system for smaller projects targeted at farmland. In the UK EIA is more widely applicable, but generally only where part of the area intended for development is uncultivated or only partially cultivated and hence might not be invoked where only arable land is at risk [212]. EIA regulations vary slightly in Scotland, however, where the site being assessed may consist exclusively of farmland where it exceeds 200 ha. The Scottish EPA also provides specific guidance on application of SEA to soils [213].

The broad North American planning term of *Land Preservation* covers many of these categories with the main focus on zoning [186, 214]. Every state and city has its own subset or version of Land Preservation laws and regulations which are varied and complex. One example is a “conservation easement” whereby a landowner is bound by a covenant set by the government or some other organisation and in return receives an incentive, such as a tax rebate; this could fall under categories 1(b), 2(d), or 3(a-c).

Apple Valley City, Minnesota has adopted a Natural Resources Management Plan (NRMP) which essentially constitutes a 2(b) type mechanism via a permitting system, but incorporates elements of 1(a) because it could apply to a single aspect of a development. It differs from most local planning laws and regulations by applying the broad principle of environmental impact to every development [215]. The NRMP is summarised thus: “*This broad natural resource management ordinance attempts to ‘protect, preserve, and enhance the natural resources and environment.’ It forces the collection of data on the parcel*

in question and mandates low impact and best practices which preserve water quality, wetlands, trees, and reduce soil erosion among other issues.”

There is an interesting example of environmental scientists trying to set a legal precedent in a 2(b) type scenario in New Zealand in 2011 using soil natural capital and ecosystem services arguments to prevent urban development on horticultural land. The defence lawyer (representing the developer) argued that the only measured ecosystem services of this soil was agriculture. The judge upheld the local authority decision not to allow development but not because of the “holistic” ecosystem services defence, which he disregarded [216].

The modern governance of soil resources worldwide

While the piecemeal governance of soil use is almost as old as human history, formal supranational global oversight of soil resources is only approximately 50 years old. In terms of the natural environment, the early 1970s stand out as a period when national policies, international initiatives and academic analyses of global problems coalesced in an unprecedented step change in attitudes. 1970 saw the conception of Earth Day, initially in the US but later worldwide, and as a consequence the creation of the US Environmental Protection Agency, an institution which also became replicated worldwide. In 1972 the eclectic think tank, the Club of Rome, published the now iconic *Limits to Growth*, with its prediction of societal collapse in the 21st century, alongside frequent references to soil degradation [217]. This period also saw the emergence of the concept of knowledge co-production whereby multiple agencies or stakeholder groups, academic and non-academic, join forces to address complex multidisciplinary problems such as sustainable development [218].

Emergence of soil in modern international law and policy

The *African Convention on the Conservation of Nature and Natural Resources* (the “Algiers Convention”), which came into force in 1969, was the first multilateral continent-wide treaty devoted to natural resource protection, that included a reference to soil. The salient wording called for soil conservation and improvement, and land use planning to this end. There was also a stated desire to, “...ensure long-term productivity of the land,” or in today’s technical parlance, SLM. In practice, although binding, the Algiers Convention lacked the resources and institutional framework to be particularly effective, but was nevertheless initially regarded as a milestone in international environmental law [219]. The following year, in a preparatory document for the forthcoming 1972 UN Conference on the Human Environment (UNCHE) in Stockholm, Wilson and Matthews of the Massachusetts Institute of Technology (MIT) coined the term environmental services [220], which would later be renamed ecosystem services [221] or, a more recent proposed alternative, nature’s contribution to people (NCP) [97]. The UNCHE was effectively the

first major international conference on environmental protection and sustainable development, and has been referred to by Boer *et al.* as marking the: “...*first phase of international soil protection law*” [222, 223].

At around the same time FAO was publishing several of its soils bulletins every year, including two landmark studies in 1971, on land degradation [224] and on legislative principles of soil conservation [225]. It is worth reprinting the first sentence of the latter in its entirety because of the way it succinctly reflects the prevailing zeitgeist: “*With the universal concern for environmental protection on the increase, more attention is being given to the introduction of adequate legislation and institutions for preventing or controlling soil degradation.*” This document was a relatively concise and readable set of guidelines, with real-world examples, that any nation could use as a template for creating or improving its own legal framework for governing soil.

In 1972 the Council of Europe created and adopted the *European Soil Charter*, a concise and perhaps surprisingly holistic set of statements about soil and soil protection aspirations [226] which was non-binding, but regarded nevertheless as the first international legal instrument dedicated specifically to protecting soils [222, 227]. Ten years later, in 1981, FAO adopted the World Soil Charter [228] and in 1982 UNEP developed a global soils policy [229]. In the same year, the IUCN/UN World Charter for Nature included explicit reference to soil, emphasising the need to combat land degradation and to maintain or enhance productivity [230]. It was also during this period, in 1977, that the UN held its first Conference on Desertification (UNCOD), and produced its *Plan of Action to Combat Desertification* (PACD) [231]. However, despite the apparent paradigm shift that had occurred, and the many subsequent conferences and reports this generated, it would take several more years before binding international law addressing soils materialised in the next wave of developments.

Soil as a feature of common concern

In 1987 the UN commissioned the so-called Brundtland report, *Our Common Future*, which reinvigorated the debate, again stressing the severity of soil degradation, and included a summary of proposed legal principles for environmental protection and sustainable development [232]. In order to address these issues financially, in 1991 UN agencies and the World Bank created the Global Environment Facility (GEF), as a prerequisite for the 1992 UN Conference on Environment and Development (UNCED) or Rio Earth Summit [233], a milestone event. From this emerged the UN Commission on Sustainable Development (UNCSD) and three legally binding international treaties: the 1992 *UN Convention on Biological Diversity* (CBD), the 1992 *UN Framework Convention on Climate Change* (UNFCCC) and the 1994 *UN Convention to Combat*

Desertification (UNCCD)⁹, which came into force in 1993, 1994 and 1996, respectively (the Rio Conventions) [234, 235]. There was appreciation of the interrelationships linking all three conventions to land degradation [236], but funding for the latter was mainly tied to the UNCCD which was always the weakest and poorest of the three due to donor scepticism, although the situation improved slightly after the *Johannesburg Declaration on Sustainable Development* in 2002 [67]. The UNCCD has been called the first and only “*legally binding global agreement directly dealing with the promotion of bio-productive land*” [222], but it is also recognised that it is not an “*adequate instrument for the protection and sustainable use of soil*” because of its limited provisions and scope [237]. Although recognising desertification and drought as being problems of global dimension, and their adverse effects as being of urgent concern to the international community, UNCCD contains little in the way of substantive obligations, and formally applies only to “*arid, semi-arid and dry sub-humid areas*”. Nevertheless, UNCCD is gaining momentum as a focal point for efforts to address the global land degradation neutrality (LDN) target [231, 238].

The interconnected nature of different environmental problems, as well as their linkages to socioeconomics and SLM, was taken a step further at another key event, the 2012 UN Conference on Sustainable Development (Rio+20). The UNCCD brought the concept of zero net land degradation (ZNLN) for adoption at Rio+20, reflected in the outcome document, “*The Future We Want*”, paragraphs 205-209 [231] and this later became embedded into the 2015 SDGs as the global LDN target in SDG15.3 [231, 239]. Land degradation was finally being acknowledged as a global problem that was intrinsically linked with climate change, biodiversity loss and poverty. The concept of ZNLN/LDN was that every effort should be made either to prevent further degradation or, where this was not possible, to rehabilitate equivalent areas of degraded land elsewhere [239].

While the thrust of LDN is land conservation, the concept clearly envisages the possibility of offsetting. The concept of offsetting or a no net loss (NNL), and in some cases even a net gain, policy has been applied to biodiversity in various countries along similar lines to the idea of LDN, in that any development causing such loss is required to compensate with equivalent gains elsewhere, but how this is measured and its effectiveness in practice has been challenged [240]. In relation to LDN, many questions arise regarding the conditions for, and scale at which any offsetting may occur. The Environment Bill introduced by the UK government, and currently making its way through Parliament, mandates biodiversity net gain under the heading, “*Biodiversity gain as condition of planning permission.*” [241].

⁹ The original full title of the convention is: ‘*United Nations Convention to Combat Desertification in Countries Experiencing Serious Drought and/or Desertification, Particularly in Africa*’.

Acknowledgment of the fundamental importance and relevance of land and soils is gaining more prominence in the context of the CBD [234], and the UNFCCC [242] and related *Paris Agreement* [243]. All three identify their main treaty objectives as being of common concern to humankind. The objectives of the CBD include the conservation of biological diversity and the sustainable use of its components¹⁰ and would appear largely to enable comprehensive regulation of soil [222]. Although the CBD text does not specifically mention soil, soil biodiversity falls within the convention's objectives and requires conservation. Further, as a terrestrial ecosystem, and falling within the concept of a biotic component of ecosystems with actual or potential use or value for humanity, soil is subject to sustainable use requirements under the CBD. The parties to the CBD have a programme of work on agricultural biodiversity, have established an *International Initiative for the Conservation and Sustainable Use of Soil Biodiversity* [244] and recently initiated the preparation of a global report on *State of Knowledge of Soil Biodiversity: Status, Challenges and Potentialities* [245]. It remains to be seen how soils will be reflected in the Global Biodiversity Framework to be adopted by the CBD COP 15, currently scheduled to take place in October 2021, in China, especially given China's rapid advance in this field¹¹.

The UNFCCC and *Paris Agreement* requires States to take action to reduce GHG emissions and to conserve and enhance GHG sinks and reservoirs [243, 246]. Land and soils provide vital ecosystem regulation functions, can be a significant C sink and reservoir or emitter of GHGs and their precursors, and are vulnerable to the impacts of climate change. Forests aside, sustainable land use and soil management has received comparatively little attention in the UNFCCC regime. UNFCCC Art 4 and *Paris Agreement* Art 5 mandate Parties to promote and cooperate in conserving and enhancing GHG sinks and reservoirs, including biomass, forests and oceans, as well as other terrestrial, coastal and marine ecosystems. In 2017, the COP adopted decision 4/CP.23 on the 'Koronivia joint work on agriculture' to address issues related to agriculture, taking into consideration the vulnerabilities of agriculture to climate change and approaches to addressing food security, including improved soil carbon, soil health and soil fertility under grassland and cropland as well as integrated systems, including water management, and socioeconomic and food security dimensions of climate change in agriculture [247].

At its first Plenary Assembly, held at the FAO Headquarters in Rome in June 2013, the Global Soil Partnership (GSP) created the Intergovernmental Technical Panel on Soils (ITPS), made up of 27 soil experts representing all the regions of the world. At the same time soil and land was incorporated into the SDGs [93]. The GSP

¹⁰ *Convention on Biological Diversity*, 5 June 1992 (in force 29 December 1993), 31 ILM 822 ("CBD"), Art 1.

¹¹ According to the SJR International Science Ranking website China ranks second only to the US in soil science (<https://www.scimagojr.com>; accessed 28th April, 2021).

also proposed 2015 as the International Year of Soils (IYS) and the annual observance of a World Soil Day (on 5 December), both of which were adopted at the 68th UN General Assembly later that year. Though primarily symbolic in nature, a principle aim of these events was to raise awareness of the importance and finite nature of soil and to, “...*Support effective policies and actions for the sustainable management and protection of soil resources*” [248]. The activities of the GSP, and its Regional Partnerships represent an important voluntary forum for stimulating activity and momentum across a range of stakeholders in relation to sustainable soil management. The *Revised World Soil Charter* [249] which was adopted by FAO members in 2015, and the *Voluntary Guidelines for Sustainable Soil Management* (VGSSM) [250], which translate the Charter’s principles into practice, and were endorsed by the FAO Council in 2016,¹² were drafted and agreed under the aegis of the GSP. The GSP continues to facilitate the production of a number of other documents relevant to soil governance¹³.

The work of the ITPS, UNCCD Science-Policy Interface (SPI), IPBES, the IPCC, and in particular the publication of the 2015 State of the World Soil Resources [93], the 2017 Scientific Conceptual Framework for Land Degradation Neutrality [251], the 2018 Global Land Degradation Assessment [97], and the 2019 Report on Climate Change and Land [65], by each respectively, also stand out as key influences in the soil governance narrative. The significance of science, and of work at the science-policy interface, is increasingly important in determining the scope of and extent of legal obligations, as evident in the recently adopted 2020 International Law Association (ILA) *Guidelines on the Role of International Law in Sustainable Natural Resource Management for Development* [252].

In 2015, to coincide with UNFCCC COP21, the French government issued a bold entreaty to the global community to raise average soil organic C (SOC) levels by 0.4% (the “4 per mille Soils for Food Security and Climate” initiative), to offset annual global C emissions into the atmosphere, as well as improving food security [253, 254]. The choice of SOC as the target was fundamental and twofold, because it represents both a means of sequestering atmospheric C and the most widely accepted measure of soil health or productivity. Furthermore, increases in SOC can be achieved in many ways, for example, by soil management techniques or by the choice of what is grown and therefore, by implication, via complete land use conversion, for example from row crops to pasture or forestry. A desire to save land from conversion can also be implied, because any form of land take which seals soil or reduces SOC, would increase the overall need to enhance SOC on the

¹² <http://www.fao.org/3/bl813e/bl813e.pdf>

¹³ For example, FAO, 2019. *The International Code of Conduct for the Sustainable Use and Management of Fertilizers*. Rome. <https://doi.org/10.4060/CA5253EN> and FAO. 2020. *A protocol for measurement, monitoring, reporting and verification of soil organic carbon in agricultural landscapes – GSOC-MRV Protocol*. Rome. <https://doi.org/10.4060/cb0509en>

remaining land. The 4 per 1000 Initiative, to which many countries have signed up, has succeeded in changing the discourse to highlight the critical role of soil and agriculture in climate change mitigation and adaptation. There has also been criticism [255], especially of the feasibility of the initiative and the underpinning data. This is not the place to discuss the debate at length, but while the authors have robustly defended the science, with caveats, they also stress that 4 per 1000 was never intended to be a precisely calculated solution, but a positive, politically-driven and powerfully symbolic aspirational target [256]. Clearly some eminent soil scientists agree with them, judging that the initiative would be a feasible, “no-regret” and indispensable climate action [257].

Regional and sectoral developments

At the regional level the Algiers Convention was revised in 2003 to become the Maputo Convention, and includes an updated article on land and soil (Article VI), though it only entered into force in 2016 [258]. In 1998 the *Protocol on the Implementation of the Alpine Convention of 1991 in the Field of Soil Conservation* (Alpine Soil Protocol) was agreed, becoming the first legally binding treaty expressly devoted to soil, and entering into force in 2006 [259]. For both the Maputo Convention and the Alpine Soil Protocol, implementation has not yet matched aspiration, to differing degrees, and both serve as useful focal points for awareness raising regarding soil management and best practice. The ASEAN Agreement on the Conservation of Nature and Natural Resources, signed in 1985, included an Article specifically on soil, but it has never come into force [222].

The Council of Europe revised the *European Soil Charter* in 2003 to become the *Revised European Charter for the Protection and Sustainable Management of Soil* [260]. In contrast, in the European Union, agreement on a soil instrument has to date resembled a triumph of hope over experience. As also exist, to a more limited extent, at the international level [252], there are numerous pieces of sectoral, as well as procedural, EU legislative instruments that have a bearing on soil (e.g. in areas such as agriculture, water, waste, chemicals, prevention of industrial pollution, and environmental impact assessment). In 2006 the European Commission presented its *Thematic Strategy on Soil Protection* and put forward a *Proposal for a Soil Framework Directive* to place healthy soil on a par with clean water and air [261]. However, after years of negotiations, the proposal was rejected by five member states and eventually withdrawn in 2014, the sticking points being subsidiarity to centralised control and costs of implementation. The rejecting states were in broad agreement with the Directive and even exceeded its requirements in some areas. However, they were not willing to agree to strengthened EU-wide legislation addressing soil sealing and liability for contaminated land [262]. A new EU Soil Strategy is planned for adoption in 2021 as a key component of a European Green Deal (EGD), aimed at making Europe the first climate-neutral continent by 2050 [263].

Human and Peoples' rights, access to environmental information and justice

The connections between land and soil, human health and wellbeing, food security, biodiversity and climate adaptation and mitigation are increasingly being acknowledged in the context of human and Peoples' rights, as is the importance of secure land tenure. To that end the VGSSM incorporates the Voluntary Guidelines on the Responsible Governance of Tenure of Land, Fisheries and Forests in the Context of National Food Security (VGGT) [251]. The right to a healthy environment is included in a growing number of national constitutions [264], as well as in regional and international human rights instruments. For example, the 2019 *Declaration on the Rights of Peasants and Other People Living in Rural Areas* has proclaimed in Article 17 a 'right to land' of peasants and other people living in rural areas [265]. This principle of small-scale agrarian rights has been powerfully linked to the more familiar scientific rationale of SES in arguments for more robust soil governance [80, 266, 267]. We can expect that failures of governance in relation to land and soils will become more of a focus in future human rights cases. Procedural rights of citizens in relation to environmental information, public participation in decision making and access to justice in environmental matters, within sectoral treaty regimes, or in dedicated treaties such as the UNECE Aarhus Convention in Europe and the Escazú Agreement in Latin America and the Caribbean [268], also have a role to play in underpinning land and soil governance,

Sustainable Development Goals (SDGs)

All countries have signed up to the seventeen SDGs [269]. These have been formulated in a purposefully general manner, to allow for country led implementation, yet provide for specific targets and indicators with a view to facilitating fulfilment of minimum standards and comparison across countries [238, 270]. Land and soil have profound relevance for most, if not all, SDGs, though only a small number of SDG targets and indicators make specific reference to them [269, 271]. In particular SDG 15: *Life on land*, aims to protect, restore and promote sustainable use of terrestrial ecosystems, including in SDG 15.3 combatting desertification, restoring degraded land and soil, and striving to achieve a land degradation neutral (LDN) world [271]. While the SDGs themselves are not-legally binding, they do in some respects echo or amplify existing and emerging binding international commitments.

National soil legislation

Last year FAO and UNEP jointly published a key document, *Legislative approaches to sustainable agriculture and natural resources governance* [238]. The report is candid in its acknowledgement that implementation of overarching international commitments can be difficult to enforce and monitor, and that,

although certain underlying principles can be discerned, "good governance" is a "nebulous" concept; all the more so when it is associated with aspirational outcomes rather than tangible laws and regulations that are legally binding. Nevertheless, great strides have been made in some areas. Although expressly not commenting on enforcement, the report provides many examples of innovative and progressive legislation in relation to land and soil, though questions still remain in relation to enforcement, which was expressly not investigated in the report, and has been reported to come up short even for some of the potentially positive and legally binding measures [82].

Binding or non-binding, international commitments rely on effective national implementation and enforcement. As outlined above, broadly speaking, a selection of legal instruments exist in various countries, or provinces within countries, throughout the world to achieve all of the 'categories of governance' described above. However, this state of affairs is very different to one in which all countries have such instruments or, more to the point, enforce them effectively.

On a global scale the most urgent action needed may relate to category 2a (Table 2) - restrictions or guidance on development involving conversion of use, to protect terrestrial natural resources and enforced by laws based on zoning, e.g. National Parks, nature reserves, etc., to address issues such as deforestation in Amazonia or Southeast Asia. Nevertheless, as already suggested, the instruments to regulate this type of anthropogenic threat are relatively straightforward and consistently defined across the world. The obstacles are political and socioeconomic. This is also true of other existing laws that directly or indirectly protect soil, the success of which require that entrenched power imbalances are challenged [78].

Lack of enforcement and governance on the ground remain serious obstacles throughout much of the world. See, for example, the studies referenced at [80, 108, 112, 127, 167, 169, 174, 179, 272, 273]. Those ten studies cited, merely a subset of many more, draw on a range of data for their evidence, in addition to literature reviews, including land use and land evaluation records, census and demographic statistics, case studies, planning and legal decisions, stakeholder interviews and, perhaps most telling of all, spatial remote sensing of land use change, using GIS software tools such as CORINE. Furthermore, even some of the potentially positive and legally binding instruments highlighted in the FAO/UNEP report currently appear far from effective in practice [82].

Nevertheless, the process of enacting soil and land legislation is now ongoing and ubiquitous. Each country's soil legislation, if it has enacted any, reflects that country's specific circumstances and its government's priorities and, one might like to think, international commitments. As described in the sections *Semantics and legal terminology* and *Methods*, some countries subsume soil into other areas of legislation, while others

have one or more laws explicitly addressing soil. China, for example, with possibly the greatest absolute area of contaminated soil on Earth [274-276], as well as a long history of soil erosion [16], has not only soil-specific laws, but a separate law to address each of these two problems. The extent to which national soils governance, such as it may exist in any particular country, is a result of self-identified necessity, or as a consequence of international commitments, is an interesting one beyond the scope of this article.

Table 3 presents a condensed numerical summary of the documents held in FAOLEX analysed by the lead author and placed into derived categories considered to be salient to this article, presented as approximate percentages of countries that have created or adopted such instruments (as detailed in *Methods* above).

It is evident that soil and land protection laws are conspicuous by their relative absence on the world stage. Environmental legislation is not recorded here but, for comparison, it has been almost universally adopted. It is also conspicuous from the data below that for many countries the word *soil* would be entirely absent from their legal portfolio were it not for such environmental legislation.

Table 3 Proportions of countries with soil-related legislation

(a) SOIL-SPECIFIC POLICIES AND LAWS	COUNTRIES (%)
Explicit soil policy.	24
Soil conservation/erosion law(s).	37
Soil contamination/pollution law(s).	30
Soil sealing law(s).	2
Generic or other soil “improvement or protection” law(s).	36
Reference to soil embedded within environmental legislation.	88
Reference to soil embedded within agricultural or land rights legislation.	71
Reference to soil embedded within spatial/regional planning legislation.	49
Soil protection monitoring and/or targets.	14
(b) AGRICULTURAL LAND PRESERVATION POLICIES AND LAWS	
Policies designating zoning, including rural and/or agricultural land.	31
Policy advocating the preservation of farmland/ land take avoidance.	22
Land take avoidance targets.	5
Legal framework to facilitate farmland preservation schemes.	19

Farmland preservation guidance based on soil or land classification.	21
Law prohibiting or strongly restricting loss of all prime farmland.	15
(c) SOIL-RELATED ENVIRONMENTAL POLICIES	
Policies promoting organic or regenerative agriculture, agroecology or PES.	39
Land Degradation Neutrality (LDN) policy (commitments)	64
Policies referring specifically to the critical ecosystem services of soils.	15

Prime farmland preservation

The results presented in Table 3, along with a comprehensive literature review, leads the authors to conclude that one of the most serious failings with respect to the ambiguity or absence of legal instruments relates to what we have termed category 2(b) above, that is land take outside protected areas. At most risk and of most concern in this context is the conversion of high quality or prime farmland, which is usually afforded far less protection than natural ecosystems, or none at all, yet can in certain respects sometimes provide equivalent, or even greater, ecosystem services, depending on the criteria used. This is an important and often misunderstood point that cannot be overestimated. It is easy to fall into the trap of assuming that agricultural land cannot possibly provide environmental benefits comparable to those of natural ecosystems, either in terms of quantity or quality. In many situations this will be demonstrably true, especially when based on current land management practices, but certainly does not reflect the intrinsic value of such land nor necessarily even its existential value [277].

Undeniably, other things being equal, land which is densely covered in vegetation, with undisturbed topsoil, will tend to store more water and C, and foster greater biodiversity. However, with respect to land use, other things are rarely equal. Firstly, the principle reason land becomes prime farmland results from its highly valued ecosystem attributes: low altitude and gentle gradients; deep soil with favourable texture and structure that retains water, nutrients and C; benign biochemistry; and a favourable moisture regime that is not susceptible to extremes of drought or waterlogging. These are the desirable attributes that facilitate not just food production, but terrestrial life in general, which ecologists assess in aggregate as NPP. As crude as it might be in some respects, especially in relation to cultural or scientific value, NPP is widely regarded as one of the best proxy measures of total ecosystem service contribution [278].

In contrast to this, many of the varied soil landscapes that currently support natural ecosystems can be biologically marginal, exhibiting low NPP. Naturally eroding mountain slopes with shallow lithosols and leached podzols, dry sandy heaths, arid subtropical plains of alkaline, saline and sodic soils, thin rendzinas consisting of a few centimetres of soil over solid chalk or limestone, highly leached acidic and occasionally

lateritic soils of the humid tropics, and so on, are all important and fascinating in their own right. However, even putting aside the utilitarian criterion of agricultural potential, these peripheral zones can also be limited with respect to a range of critical ecosystem services. Even the biodiversity of such areas, rare and invaluable as it can be in absolute terms, is only as rich as its environment can support.

That said, millennia of evolution in marginal environments have also produced extraordinary abundance, such as the biodiversity of the tropical rainforests on soils that are typically of low agricultural value or the vast reserves of C in the peatlands of the boreal tundra and the temperate upland peat moors, where the climate has little to offer agriculture [279].

Prime farmland is not normally an obvious candidate for voluntary conversion to natural regeneration or rewilding, although a strong case can be made for taking lowland peat, that has been drained and converted to agriculture, out of arable production [280, 281]. Nevertheless, high grade arable land, even in its cultivated state, can provide abundant benefits in terms of flood control, water storage and filtration, wildlife habitat, landscape value and recreation. However, and critically, such benefits are greatly affected by land management. The negative impacts of modern farming, on the soil in particular, such as compaction, erosion, pollution and reduced SOC, much of this a result of aggressive cultivation methods, are precisely the factors that undermine the ability of the soil to provide ecosystem services [282, 283].

One very welcome trend in recent decades has been to roll back many of these harmful methods. A range of techniques that are loosely grouped under the term of conservation, or regenerative, agriculture, agroecology, or even “carbon farming” are increasingly being promoted and applied in many countries [98, 284-286]. Included within this approach are minimum or zero tillage, permanent soil cover with crop residues, live mulches, cover cropping, diversity of planting, intercropping and enhanced rotations including, for example, deep roots, leguminous leys and rotational livestock grazing, agroforestry and other forms of mixed cropping, perennial crops, integrated pest management (IPM) and biological control, and many other ways of conserving and enhancing soil quality and hence, by default, ameliorating environmental impacts. While important everywhere for food security and food sovereignty, sustainable soil management is especially important on the C-rich Chernozems (Mollisols) of eastern Europe and the Americas, arguably the world’s most productive soils [277, 287]. Hence, our advocacy of preserving our most valuable farmland is proposed in tandem with the adoption of such methods.

Emerging holistic conceptual frameworks

Amundson sums up soil governance in the US as a complex patchwork of transient interventions, as opposed to long-term solutions [288]. Other authors describe comparable situations elsewhere, for example, in the EU [289, 290], in South America [291], in Australia [292], in Russia [293] and in other examples already cited in this article. Fromherz summarises many of the existing soil governance initiatives, as well as

the failures, and makes an impassioned appeal for a dedicated, legally binding international soil governance instrument on the basis that, "...individual states lack both the power and the incentives to make these changes." [266]. This has been echoed by others [204, 223, 252, 294]. Fromherz also highlights a key point alluded to already in this article, the low profile (literal as well as metaphorical, one could say) of soil, which is often conspicuous in its absence from risk assessments of other natural resources [295, 296]. Gonzalez Lago *et al.* refer to a global soils policy vacuum and call for an urgent transdisciplinary framework approach to "re-politicise" soil [297]. This begs the question as to what extent soils have been "de-politicised", but what is emerging from a number of authors and institutions is the degree to which soil protection is inescapably enmeshed with ethics [298].

Participatory modelling and conceptual frameworks have been applied to complex cross-cutting problems such as addressing the SDGs, and these approaches continue to evolve [299]. The latest stages in the process of soil governance so far, in the last decade, are encouraging to some extent, at least with regard to what soil scientists are bringing to the table. An overarching conceptual framework which has been proposed by some of the world's leading soil scientists, broadly as a memorandum of understanding, is soil security [300]. This concept has five dimensions: capability, condition, capital, connectivity and codification, which encompasses the translation of soil knowledge into policy and legislation [301], e.g. aimed at achieving the SDGs [302], although an attempt to establish soil security as one of the SDGs was unsuccessful [303]. The soil security initiative is still at a high level, with the emphasis on policy rather than active governance, but progress is being made and reported on in some areas [304].

Also emerging are more formalised frameworks which place the concept of SES further than ever in the context of socioeconomic decisions, policy-making and governance [305, 306]. The most fully developed of these methodologies, the Land Resource Circle (LRC) framework, goes further than any other approach known to the authors in extending land evaluation to incorporate SES. The framework identifies in-depth environmental and socioeconomic implications of land-use decisions via a scoring system which avoids reducing all outcomes to simplistic monetary values [305]. Furthermore, the LRC paper includes a very detailed hypothetical example which perfectly demonstrates how quantified soil variables can be combined and converted into societal costs and benefits. By contrast, the the Resilience-Effectiveness-Efficiency-Legitimacy (REEL) framework [306], provides a purely qualitative means of comparing different approaches to soil governance according to the four criteria (dimensions) embedded within its title. Of these dimensions, legitimacy encompasses issues of equity, fairness, accountability, and participation, and is perhaps the least formulaic and the most elusive to achieve. The REEL framework also highlights interconnectedness and attempts to address situations where policy targets mismatch the causes of problems, either spatially or in terms of scale or even time. This is almost a meta-framework, with the emphasis on socioeconomics and due diligence.

As with other forms of natural resource governance, politics and socioeconomics are often obstacles to effective soil governance, as are weak governance structures, but a critical distinction is that there is much scope for creating more effective legal instruments to tackle this problem than currently exist in most countries, and also for improving clarity and consistency to bring prime farmland protection further in line with ecosystem and soil protection.

Conclusion

An appreciation of the tangible benefits of soil is woven into the fabric of history and is reflected by the value and protections afforded to productive soil by society over millennia. Some of these protections have taken the form of legislation, but this has been ad hoc, inadequately enforced and rarely if ever proportionate to the total ecosystem service value of soil. Two challenges identified as hindering progress to date are the terminological difficulties associated with soil and land, and the absence of a soil-centric policy framework to address wider issues that are intrinsically bound up with soil. Moreover, soil exists as a component of land which, in the face of population pressure, has been inexorably subject to the twin impacts of firstly, extra demands, e.g. of food production, and secondly, competing demands, such as urbanisation.

Agriculture has traditionally responded to these demands by both expansion and intensification, each approach engendering anthropogenic threats to soil, the former often encroaching on natural ecosystems and the latter potentially leading to soil degradation. These threats have profound impacts on the environment locally and worldwide, and both have spawned a great deal of national and international legislation. However, the problem of land take, in particular the conversion of prime agricultural land to non-agricultural use, including some irreversible soil sealing, is arguably an equally serious threat because of the disproportionate loss of SES this represents. Yet this threat tends to be relatively inconspicuous, and subject to ambiguous legislation and governance.

Putting aside the issue of whether soil and land governance is effectively implemented in general, threats to natural ecosystems are explicitly addressed by environmental laws, and soil degradation is typically addressed by agricultural or soil-specific laws. A few of these laws have been in place for centuries and many others have joined them in recent decades as environmental awareness and global concerns have grown. However, neither environmental nor agricultural legislation normally encompasses the loss of prime farmland, the fate of which tends to fall within the remit of spatial planning. Furthermore, planners and planning regulations must navigate many competing demands and interest groups, political as well as economic, and hence they rarely prioritise soil or its intrinsic value to future generations. As a result, legal instruments which are frequently designed with little if any reference to soil, often have greatest power to transform soil and land use. This situation is further compounded by the fact that intense commercial

pressure in tandem with conventional economics, which has traditionally ignored the benefits of long-term ecosystem services, promotes land use options that are more profitable than farming.

In a time of so much environmental concern, a further paradoxical factor that has emerged in the last half century to weaken the case for farmland preservation (as opposed to soil protection *per se*), is the conflicted nature of agriculture: on the one hand the producer of our sustenance and the custodian of rural landscapes, while on the other, a significant cause of the other two types of threat described above, as well as having other detrimental impacts such as climate forcing and water pollution. Lacking both the visual impact and the iconic status of natural landscapes, soil in an (often intensive) agricultural context does not constitute an obvious rallying point in the public consciousness. So the constant attrition of prime farmland occurs in something of a legal and ecological grey area, attracting far less attention than other environmental issues.

However, the case for protecting soil as a critical part of our natural capital is separately gaining ground. A key paradigm shift occurred in the 1990s when it became more widely appreciated that soil C is inextricably linked with atmospheric C. This development also approximately coincided with a broad gradual acceptance that land degradation was interrelated with climate change and biodiversity loss. This view of soil as a central component in limiting global warming, adapting to climate change and addressing the ecological crisis, is being framed unequivocally in the wider context of sustainability and the linking of science to policy. Emerging multidisciplinary frameworks, such as the concept of soil security and methodologies for holistically evaluating SES, may collectively contribute to this process.

Clearly a lack of effective soil governance is often more significant a problem than an absence of legal instruments. Indeed, failings or inequities in human behaviour, whether ethical or intellectual, have always been endemic, which is why laws exist. Nevertheless, the arc of history suggests that, despite some serious setbacks, the moral and ultimately self-preserving imperative of collective justice and consensus can and often does prevail over short-term vested interests. However, the status quo will not give way without the constant drip-feeding of international resolutions, critical reviews and articles, pressure groups, whistle-blowing and local protests, regardless of how ineffective each of these clarion calls may appear in isolation.

Many politicians and activists are calling for “green new deals” to emulate Roosevelt’s New Deal following the Great Depression, but with an emphasis on climate justice and ecosystem restoration. This would of course include sustainable land use, by default, and an important component of that needs to be soil protection and land preservation, consistent with climate justice, including the attendant social and economic incentives necessary to achieve local and global climate goals, biodiversity objectives and food security.

Profound reforms, such as democratic freedoms, become most unassailable when society perceives a binary moral choice that serves the common good. One of the most successful examples of global co-operation of this kind was the signing of the *Montreal Protocol* in 1986 to protect the ozone layer. Persuading the global

community to reduce GHG emissions in the *Paris Agreement*, though desperately hard-won and far from over, provides another exemplar of what can be achieved. Ginzky cites other examples and, despite a lack of progress to date, suggests that a binding international treaty on soils is both necessary and achievable [307]. Some of the details may be complex, but it can be achievable where the message is simple, and a window is open.

The lesson here, for saving our soil, is to strive for a clear consistent message, backed up by clear consistent policies and laws, and clear consistent criteria to decide how to prioritise one soil type or area of land over another. This alone cannot solve the vast problem, but it will make the process more streamlined and more transparent. It will also facilitate governance for those who genuinely want to govern and it will make infringements that bit more difficult to effect and pass unnoticed. The key aim is to couple a compelling case with an achievable solution.

Data accessibility

The only original data presented is the categorisation percentages abstracted from FAOLEX in Table 1.

Authors' contributions

L.R.P. conceived the idea, conducted much of the research and wrote the bulk of the text. C.R. made considerable contributions to the text, especially in relation to soil law, and performed much editing.

Competing interests

We declare we have no competing interests.

Funding

We received no funding for this study.

Acknowledgements

Several individuals and organisations provided valuable information via correspondence or in meetings. The following persons helped to explain their local soil and land laws, or planning regulations, in theory and practice: Dr Thomas Daniels (Univ. Pennsylvania), Dr Mark Lapping (Univ. Southern Maine), Ian Rugg and

colleagues (Natural Resources Wales and Natural England), Dr. Fernando García Prechac (Fac. of Agronomy, Univ. of Uruguay, and former Director General of Natural Resources, Ministry of Livestock, Agriculture and Fisheries, Uruguay), Marcel Achkar (Univ. República de Uruguay), Gundula Prokop (EU Research and Territorial Co-operation Projects, Federal Environment Agency, Austria), Anna Shortly (The Greenbelt Foundation, Ontario), Dr Elena Havlicek and Ruedi Stähli (Federal Office for the Environment, Switzerland), Ljubiša Bezbradica (Institute of Architecture and Urban & Spatial Planning of Serbia), Tom Corser (New Zealand Ministry for the Environment), and Dr. Suhail Ahmad (Cluster University Of Srinagar, Kashmir). In addition: Dr David Dent (former Director, ISRIC) gave advice on soil degradation assessments which he co-authored at ISRIC; CPRE posted a free copy of *Valuing the Land* which is unavailable online; and two anonymous reviewers helped us to improve the balance and style of the paper.

References

1. Asante, SKB. 1965 Interests in Land in the Customary Law of Ghana. A New Appraisal. *The Yale Law Journal* **74**, 848-885. (doi:doi: 10.2307/794709).
2. Brussaard, L. 2021 Biodiversity and ecosystem functioning in soil: The dark side of nature and the bright side of life. *Ambio*. (doi:10.1007/s13280-021-01507-z).
3. Milde, KF. 1950 Roman Contributions to the Law of Soil Conservation. *Fordham L. Rev.* **19**, 192-196.
4. Pfister, L, Savenije, HHG & Fenicia, F. 2009 *Leonardo Da Vinci's water theory : on the origin and fate of water*. Wallingford, International Association of Hydrological Sciences.
5. New Scientist. 2017 *How Your Brain Works: Inside the most complicated object in the known universe*, Quercus.
6. Young, IM & Crawford, JW. 2004 Interactions and self-organization in the soil-microbe complex. *Science* **304**, 1634-1637. (doi:10.1126/science.1097394).
7. Almond, REA, Grooten, M & Petersen, T. 2020 WWF (2020) Living Planet Report 2020 - Bending the curve of biodiversity loss. eds. REA Almond, M Grooten & T Petersen. Gland, Switzerland, WWF.
8. Hardin, G. 1968 The Tragedy of the Commons. *Science* **162**, 1243-1248. (doi:10.1126/science.162.3859.1243).
9. Diamond, JM. 2005 *Collapse: How Societies Choose to Fail Or Succeed*, Viking.
10. Global Soil Partnership web page. 2020 Contribute to the GSP newest tool: SoILEX.
11. 2020 Ukrainian Government, Land Code (No. 2786-III of 2001; revised 16.10.2020).
12. Herodotus. 1922 (Loeb Classical Library). Translated by A. D. Godley. Vol. I.: Books 1 and 2, pp. xxi + 504. Vol. II.: Books 3 and 4, pp. xviii + 416. London: W. Heinemann. 10s. a volume. *The Classical Review* **36**, 135-135. (doi:10.1017/S0009840X00016759).

13. Meguid, MA. 2019 Key Features of the Egypt's Water and Agricultural Resources. In *Conventional Water Resources and Agriculture in Egypt* (ed. AM Negm), pp. 39-99 Springer International Publishing).
14. Bennett, HH. 1939 *Soil conservation*. New York, London, McGraw-Hill.
15. Butzer, K. 2005 Environmental history in the Mediterranean world: cross-disciplinary investigation of cause-and-effect for degradation and soil erosion. *Journal of Archaeological Science* **32**, 1773-1800. (doi: 1710.1016/j.jas.2005.1706.1001).
16. Dotterweich, M. 2013 The history of human-induced soil erosion: Geomorphic legacies, early descriptions and research, and the development of soil conservation—A global synopsis. *Geomorphology* **201**, 1-34. (doi:10.1016/j.geomorph.2013.07.021).
17. Hyams, E. 1952 *Soil and Civilization*, Thames and Hudson.
18. Redman, CL. 1999 *Human impact on ancient environments*. Tucson, University of Arizona Press.
19. Brevik, EC & Hartemink, AE. 2010 Early soil knowledge and the birth and development of soil science. *CATENA* **83**, 23-33. (doi:10.1016/j.catena.2010.06.011).
20. Verheye, WH. 2009 *Land Use, Land Cover and Soil Sciences - Volume VII: Soils and Soil Sciences - 2*, EOLSS Publ.
21. Homer-Dixon, TF. 1994 Environmental Scarcities and Violent Conflict: Evidence from Cases. *International Security* **19**, 5-40. (doi:10.2307/2539147).
22. Berman, N, Couttenier, M & Soubeyran, R. 2019 Fertile Ground for Conflict. *Journal of the European Economic Association*. (doi:10.1093/jeea/jvz068).
23. Fearon, J & Laitin, D. 2014 Does Contemporary Armed Conflict Have 'Deep Historical Roots'? *SSRN Electronic Journal*. (doi:10.2139/ssrn.1922249).
24. Global Witness. 2020 Defending Tomorrow: The climate crisis and threats against land and environmental defenders.
25. Le Billon, P & Lujala, P. 2020 Environmental and land defenders: Global patterns and determinants of repression. *Global Environmental Change* **65**, 102163. (doi:10.1016/j.gloenvcha.2020.102163).
26. Gong, Z, Zhang, X, Chen, J & Zhang, G. 2003 Origin and development of soil science in ancient China. *Geoderma* **115**, 3-13. (doi:10.1016/S0016-7061(03)00071-5).
27. Barrera-Bassols, N, Alfred Zinck, J & Van Ranst, E. 2006 Symbolism, knowledge and management of soil and land resources in indigenous communities: Ethnopedology at global, regional and local scales. *CATENA* **65**, 118-137. (doi:10.1016/j.catena.2005.11.001).
28. Ellickson, RC & Thorland, CD. 1995 Ancient Land Law: Mesopotamia, Egypt, Israel. *Chicago-Kent Law Review* **71**, 321.
29. Wasson, RJ. 2010 Exploitation and conservation of soil in the 3000-year agricultural and forestry history of South Asia. (eds. JR McNeill & V Winiwarter), pp. 13-50. Isle of Harris, White Horse Press).
30. Barbieri-Low, AJ & Yates, RDS. 2015 *Law, State, and Society in Early Imperial China (2 vols)*, Brill.

31. Sterckx, R. 2020 Food and Agriculture. In *Routledge Handbook of Early Chinese History* (ed. P Goldin), pp. 300-318, Routledge).
32. Shi, S. 1959 *On "Fan Shêng-chih shu" : an agriculturist book of China written by Fan Shêng-chih in the first century B.C. / translated and commented upon by Shih Sheng-han = Fan Shengzhi shu jin shi / Shi Shenghan yi shi*. Peking, Science Press.
33. Watson, AM. 1983 *Agricultural innovation in the early Islamic world : the diffusion of crops and farming techniques, 700-1100*. Cambridge, Cambridge University Press.
34. Hailey, A & Cazabon-Mannette, M. 2011 Conservation of Caribbean Island Herpetofaunas Volume 1: Conservation Biology and the Wider Caribbean. In *Conservation Of Herpetofauna In The Republic Of Trinidad And Tobago* (pp. 183. (doi: 110.1163/ej.9789004183957.i-9789004183228.9789004183964), Brill).
35. Williams, J. 2015 Soils Governance in Australia: challenges of cooperative federalism. *International Journal of Rural Law and Policy*. (doi:10.5130/ijrlp.i1.2015.4173).
36. Stocking, M. 1985 Soil Conservation Policy in Colonial Africa. *Agricultural History* **59**, 148-161.
37. Sandbach, F. 1980 *Environment, Ideology, and Policy*, Allanheld, Osmun.
38. Williams, GW & United States. Forest Service. 2005 *The USDA Forest Service : the first century*. Washington, DC, USDA Forest Service; iv, 156 p. p.
39. Wynn, G. 1979 Pioneers, politicians and the conservation of forests in early New Zealand. *Journal of Historical Geography* **5**, 171-188. (doi:10.1016/0305-7488(79)90132-4).
40. Reddy, B, Hoag, D & Shobha, B. 2004 Economic incentives for soil conservation in India. In *Proceedings of the 13th International Soil Conservation Organization Conference, Brisbane*.
41. Crofts, R & Olgeirsson, FG. 2011 *Healing the Land: The Story of Land Reclamation and Soil Conservation in Iceland*, Soil conservation service.
42. McLeman, RA, Dupre, J, Berrang Ford, L, Ford, J, Gajewski, K & Marchildon, G. 2014 What we learned from the Dust Bowl: lessons in science, policy, and adaptation. *Population and Environment* **35**, 417-440. (doi:10.1007/s11111-013-0190-z).
43. Pretty, JN. 1995 *Regenerating agriculture : policies and practice for sustainability and self-reliance*. London, Earthscan.
44. Sylvester, KM & Rupley, ES. 2012 Revising the Dust Bowl: High Above the Kansas Grasslands. *Environ Hist Durh N C* **17**, 603-633. (doi:10.1093/envhis/ems047).
45. Peake, L. 1986 *Erosion, Crop Yields and Time: A Reassessment of Quantitative Relationships*, University of East Anglia School of Development Studies.
46. Konyushkov, D. 2014 Russian: Evolution and Examples. In *Reference Module in Earth Systems and Environmental Sciences* (Elsevier).
47. Carter-Whitney, M, Esakin, TC & Canadian Institute for Environmental Law and, P. 2010 Ontario's greenbelt in an international context.

48. Simonson, RW. 1989 *Historical Highlights of Soil Survey and Soil Classification with Emphasis on the United States, 1899-1970*. Wageningen, the Netherlands, International Soil Reference and Information Centre (ISRIC).
49. Young, A. 1980 *Tropical Soils and Soil Survey*, Cambridge University Press.
50. Robinson, A. 1943 The Scott and Uthwatt Reports on Land Utilisation. *The Economic Journal* **53**, 28-38. (doi:10.2307/2226286).
51. Engholm, B. 1985 The Scott report - a personal perspective. *Planning History Bulletin* **7**, 39-43.
52. Sheail, J. 1997 Scott revisited: Post-war agriculture, planning and the British countryside. *Journal of Rural Studies* **13**, 387-398. (doi:10.1016/S0743-0167(97)00028-4).
53. CPRE. 2017 LANDLINES: why we need a strategic approach to land. London, CPRE.
54. Monk, S, Whitehead, C, Burgess, G & Tang, C. 2013 International review of land supply and planning systems. York, UK, Joseph Rowntree Foundation.
55. Hogendorn, JS & Scott, KM. 1981 The East African Groundnut Scheme: Lessons of a Large-Scale Agricultural Failure. *African Economic History*, 81-115. (doi:10.2307/3601296).
56. Natural England. 2012 Agricultural Land Classification: protecting the best and most versatile agricultural land. Technical Information Note TIN049.
57. Daniels, T. 1990 Using LESA in a purchase of development rights program. *Journal of Soil and Water Conservation* **45**, 617-621.
58. Lee, ATM. 1948 Soil Conservation—An International Study, Food and Agriculture Organization of the United Nations, Washington, D. C., 1948. *American Journal of Agricultural Economics* **30**, 784-786. (doi:10.2307/1232795).
59. United Nations. 1949 *UN Yearbook 1947-48*. Washington.
60. Bunce, AC. 1942 *The economics of soil conservation*. Ames, Ia., Iowa State college Press.
61. Ciriacy-Wantrup, SV. 1952 *Resource Conservation: Economics and Policies*, University of California, Division of Agricultural Sciences, Agricultural Experiment Station.
62. Sauer, EL. 1967 *Economics of Soil Conservation, Reclamation and Rehabilitation*.
63. Morgan, RJ & Future, Rft. 1965 *Governing Soil Conservation: Thirty Years of the New Decentralization*, Resources for the Future.
64. Aubreville, A. 1949 *Climats, Forêts, et Désertification de l'Afrique Tropicale*. Société d'Editions Géographiques, Maritimes et Tropicales. Paris.
65. IPCC. 2019 *Climate Change and Land: an IPCC special report on climate change, desertification, land degradation, sustainable land management, food security, and greenhouse gas fluxes in terrestrial ecosystems*.
66. Thomas, DSG & Middleton, NJ. 1994 *Desertification: Exploding the Myth*. London, Wiley.

67. Gisladdottir, G & Stocking, M. 2005 Land degradation control and its global environmental benefits. *Land Degradation & Development* **16**, 99-112. (doi:10.1002/ldr.687).
68. Davidson, J & Wibberley, GP. 1977 Planning and the rural environment.
69. Pingali, PL. 2012 Green Revolution: Impacts, limits, and the path ahead. *Proceedings of the National Academy of Sciences* **109**, 12302-12308. (doi:10.1073/pnas.0912953109).
70. Mannion, AM. 1991 *Global environmental change : a natural and cultural environmental history*. Harlow, Longman Scientific & Technical.
71. Agricultural Advisory Council. 1970 *Modern Farming and the Soil: Report of the Agricultural Advisory Council on Soil Structure and Soil Fertility*. London, HM Stationery Office.
72. Biniek, JP. 1979 *Agricultural and environmental relationships issues and priorities: report*. Washington, Library of Congress, U.S. Govt. Print. Off.; xvi, [1], 696 p. p.
73. USDA. 1973 *Monoculture in Agriculture: Extent, Causes, and Problems-report of the Task Force on Spatial Heterogeneity in Agricultural Landscapes and Enterprises*.
74. Carson, R. 1962 *Silent spring*. Boston, Cambridge, Mass., Houghton Mifflin; Riverside Press; x, 368 p. p.
75. Schneider, SH. 1989 The greenhouse effect: science and policy. *Science* **243**, 771-781.
76. Leahy, S, Clark, H & Reisinger, A. 2020 Challenges and Prospects for Agricultural Greenhouse Gas Mitigation Pathways Consistent With the Paris Agreement. *Frontiers in Sustainable Food Systems* **4**. (doi:10.3389/fsufs.2020.00069).
77. Shivanna, KR. 2020 The Sixth Mass Extinction Crisis and its Impact on Biodiversity and Human Welfare. *Resonance* **25**, 93-109. (doi:10.1007/s12045-019-0924-z).
78. Weigelt, J, Müller, A, Janetschek, H & Töpfer, K. 2015 Land and soil governance towards a transformational post-2015 Development Agenda: an overview. *Current Opinion in Environmental Sustainability* **15**, 57-65. (doi:10.1016/j.cosust.2015.08.005).
79. Juerges, N & Hansjürgens, B. 2016 Soil governance in the transition towards a sustainable bioeconomy – A review. *Journal of Cleaner Production* **170**. (doi:10.1016/j.jclepro.2016.10.143).
80. Guereña, A. 2016 *Unearthed: Land, power and inequality in Latin America*. OXFAM.
81. van Schaik, L & Dinnissen, R. 2014 *Terra Incognita: Land degradation as underestimated threat amplifier. Clingendael report for the Netherlands Environmental Assessment Agency (PBL)*. The Hague, Clingendael.
82. Baba, SH, Wani, MH, Zargar, BA & Bhat, IF. 2020 Determinants of land degradation in Jammu and Kashmir: implications for land governance. *Agricultural Economics Research Review* **32**, 303645.
83. FAO. FAOLEX Database.
84. Sloan, T, Payne, R, Anderson, R, Bain, C, Chapman, S, Cowie, N, Gilbert, PJ, Lindsay, R, Mauquoy, D, Newton, A, et al. 2018 Peatland afforestation in the UK and consequences for carbon storage. *Mires and Peat* **23**. (doi:10.19189/MaP.2017.OMB.315).

85. Gardi, C. 2017 *Urban expansion, land cover and soil ecosystem services*, Taylor & Francis.
86. Prokop, G, Jobstmann, H & Schönbauer, A. 2011 *Report on Best Practices for Limiting Soil Sealing and Mitigating Its Effects*. European Communities.
87. Scalenghe, R & Marsan, FA. 2009 The anthropogenic sealing of soils in urban areas. *Landscape and Urban Planning* **90**, 1-10. (doi:10.1016/j.landurbplan.2008.10.011).
88. Bhogal, S & Sinha, S. 2021 India protests: farmers could switch to more climate-resilient crops – but they have been given no incentive. In *The Conversation*.
89. Oldeman, LR, Hakkeling, RTA & Sombroek, WG. 1991 World map of the status of human-induced soil degradation : an explanatory note, 2nd. rev. ed. Wageningen [etc.], ISRIC [etc.].
90. Bai, ZG, Dent, DL, Olsson, L & Schaepman, ME. 2008 Proxy global assessment of land degradation. *Soil Use and Management* **24**, 223-234. (doi:10.1111/j.1475-2743.2008.00169.x).
91. Sonneveld, BGJS & Dent, DL. 2009 How good is GLASOD? *J. Environ. Manage.* **90**, 274-283. (doi:10.1016/j.jenvman.2007.09.008).
92. Bai, ZG, Dent, DL, Olsson, L, Tengberg, A, Tucker, C & Yengoh, G. 2015 A longer, closer, look at land degradation. *Agriculture for Development* **24**, 3-9.
93. FAO & ITPS. 2015 *The Status of the World's Soil Resources*.
94. Nkonya, E, Mirzabaev, A & von Braun, J. 2016 *Economics of Land Degradation and Improvement – A Global Assessment for Sustainable Development*, Springer Nature.
95. Esch, S, Brink, B, Stehfest, E, Bakkenes, M, Sewell, A, Bouwman, A, Meijer, J, Westhoek, H & van den Berg, M. 2017 *Exploring future changes in land use and land condition and the impacts on food, water, climate change and biodiversity: Scenarios for the UNCCD Global Land Outlook*.
96. United Nations Convention to Combat Desertification. 2017 *The Global Land Outlook*, first edition. Bonn, Germany.
97. IPBES. 2018 *The IPBES assessment report on land degradation and restoration. Secretariat of the Intergovernmental Science-Policy Platform on Biodiversity and Ecosystem Services*. Bonn, Germany.
98. Zimmerer, KS. 2011 " Conservation booms" with agricultural growth? Sustainability and Shifting Environmental Governance in Latin America, 1985-2008 (Mexico, Costa Rica, Brazil, Peru, Bolivia). *Latin American Research Review*, 82-114.
99. Stubenrauch, J, Garske, B & Ekardt, F. 2018 Sustainable Land Use, Soil Protection and Phosphorus Management from a Cross-National Perspective. *Sustainability* **10**, 1988. (doi:10.3390/su10061988).
100. 2017 The most neglected threat to public health in China is toxic soil; and fixing it will be hard and costly. In *The Economist*.
101. Daily, C. 2019 New law on soil pollution will pinpoint responsibility. China.org.cn.
102. Li, T, Liu, Y, Lin, S, Liu, Y & Xie, Y. 2019 Soil Pollution Management in China: A Brief Introduction. *Sustainability* **11**, 556.

103. Hendlin, YH. 2014 From Terra Nullius to Terra Communis Reconsidering Wild Land in an Era of Conservation and Indigenous Rights. *Environmental Philosophy* **11**, 141-174. (doi:10.5281/zenodo.260245).
104. Pearce, F. 2018 Conflicting Data: How Fast Is the World Losing its Forests? Yale Environment 360, Yale School of the Environment.
105. Curtis, PG, Slay, CM, Harris, NL, Tyukavina, A & Hansen, MC. 2018 Classifying drivers of global forest loss. *Science* **361**, 1108-1111. (doi:10.1126/science.aau3445).
106. Escobar, H. 2020 Deforestation in the Brazilian Amazon is still rising sharply. *Science* **369**, 613-613. (doi:10.1126/science.369.6504.613).
107. Chamberlin, J, Jayne, TS & Headey, D. 2014 Scarcity amidst abundance? Reassessing the potential for cropland expansion in Africa. *Food Policy* **48**, 51-65. (doi:10.1016/j.foodpol.2014.05.002).
108. Güneralp, B, Lwasa, S, Masundire, H, Parnell, S & Seto, KC. 2017 Urbanization in Africa: challenges and opportunities for conservation. *Environmental Research Letters* **13**, 015002. (doi:10.1088/1748-9326/aa94fe).
109. Young, A. 1999 Is there Really Spare Land? A Critique of Estimates of Available Cultivable Land in Developing Countries. *Environment, Development and Sustainability* **1**, 3-18. (doi:10.1023/A:1010055012699).
110. Rockström, J, Steffen, W, Noone, K, Persson, Å, Chapin, FS, Lambin, EF, Lenton, TM, Scheffer, M, Folke, C, Schellnhuber, HJ, et al. 2009 A safe operating space for humanity. *Nature* **461**, 472-475. (doi:10.1038/461472a).
111. Byerlee, D, Stevenson, J & Villoria, N. 2014 Does intensification slow crop land expansion or encourage deforestation? *Global Food Security* **3**, 92-98. (doi:10.1016/j.gfs.2014.04.001).
112. Abu Hatab, A, Cavinato, MER, Lindemer, A & Lagerkvist, C-J. 2019 Urban sprawl, food security and agricultural systems in developing countries: A systematic review of the literature. *Cities* **94**, 129-142. (doi:10.1016/j.cities.2019.06.001).
113. Metternicht, G. 2018 *Land Use and Spatial Planning: Enabling Sustainable Management of Land Resources*, Springer International Publishing.
114. Seitzinger, SP, Svedin, U, Crumley, CL, Steffen, W, Abdullah, SA, Alfsen, C, Broadgate, WJ, Biermann, F, Bondre, NR, Dearing, JA, et al. 2012 Planetary Stewardship in an Urbanizing World: Beyond City Limits. *AMBIO* **41**, 787-794. (doi:10.1007/s13280-012-0353-7).
115. Smith, P, Gregory, PJ, Vuuren, Dv, Obersteiner, M, Havlík, P, Rounsevell, M, Woods, J, Stehfest, E & Bellarby, J. 2010 Competition for land. *Philosophical Transactions of the Royal Society B: Biological Sciences* **365**, 2941-2957. (doi:10.1098/rstb.2010.0127).
116. FAO. 2011 Payments for ecosystem services and food security.
117. van Vliet, J, Eitelberg, DA & Verburg, PH. 2017 A global analysis of land take in cropland areas and production displacement from urbanization. *Global Environmental Change* **43**, 107-115. (doi:10.1016/j.gloenvcha.2017.02.001).

118. Bren d'Amour, C, Reitsma, F, Baiocchi, G, Barthel, S, Güneralp, B, Erb, KH, Haberl, H, Creutzig, F & Seto, KC. 2017 Future urban land expansion and implications for global croplands. *Proc Natl Acad Sci U S A* **114**, 8939-8944. (doi:10.1073/pnas.1606036114).
119. Chen, G, Li, X, Liu, X, Chen, Y, Liang, X, Leng, J, Xu, X, Liao, W, Qiu, Y, Wu, Q, et al. 2020 Global projections of future urban land expansion under shared socioeconomic pathways. *Nature Communications* **11**.
120. European Environment Agency. 2017 Landscapes in transition: An account of 25 years of land cover change in Europe. In *EEA Report*. Copenhagen.
121. Ustaoglu, E & Williams, B. 2017 Determinants of Urban Expansion and Agricultural Land Conversion in 25 EU Countries. *Environmental Management* **60**, 717-746. (doi:10.1007/s00267-017-0908-2).
122. Cho, J. 2005 Urban Planning and Urban Sprawl in Korea. *Urban Policy and Research* **23**, 203-218. (doi:10.1080/08111470500143304).
123. Debolini, M, Valette, E, François, M & Chéry, J-P. 2015 Mapping land use competition in the rural–urban fringe and future perspectives on land policies: A case study of Meknès (Morocco). *Land Use Pol.* **47**, 373-381. (doi:10.1016/j.landusepol.2015.01.035).
124. Silva, C. 2019 Auckland's Urban Sprawl, Policy Ambiguities and the Peri-Urbanisation to Pukekohe. *Urban Science* **3**, 1.
125. Naab, FZ, Dinye, R & Kasanga, RK. 2013 Urbanisation and its impact on agricultural lands in growing cities in developing countries: a case study of Tamale in Ghana. *European scientific journal* **2**, 256-287.
126. Mori, H. 1998 Land Conversion at the Urban Fringe: A Comparative Study of Japan, Britain and the Netherlands. *Urban Studies* **35**, 1541-1558. (doi:10.1080/0042098984277).
127. Ladu, JLC, Athiba, AL & Ondogo, EC. 2019 An Assessment of the Impact of Urbanization on Agricultural Land Use in Juba City, Central Equatoria State, Republic of South Sudan. *Journal of Applied Agricultural Economics and Policy Analysis* **2**, 22-30.
128. Ullah, S, Khan, MA, Rahman, A & Mahmood, S. 2019 Evaluation of urban encroachment on farmland: a threat to urban agriculture in Peshawar City District, Pakistan. *Erdkunde* **73**, 127-142. (doi:10.3112/erdkunde.2019.02.04).
129. Paudel, B, Pandit, J & Reed, B. 2013 Fragmentation and conversion of agriculture land in Nepal and Land Use Policy 2012.
130. Satterthwaite, D, McGranahan, G & Tacoli, C. 2010 Urbanization and its implications for food and farming. *Philosophical Transactions of the Royal Society B: Biological Sciences* **365**, 2809-2820. (doi:10.1098/rstb.2010.0136).
131. Suu, NV. 2009 Agricultural land conversion and its effects on farmers in contemporary Vietnam. *Focaal* **2009**, 106. (doi:10.3167/fcl.2009.540109).
132. Meadows, D & Randers, J. 2004 *Limits to Growth: The 30-Year Update*, Chelsea Green Publishing.
133. Schumacher, EF. 1973 *Small is beautiful; economics as if people mattered*. New York, Harper & Row.

134. Kuper, S. 2019 Thoreau, Leopold, & Carson: Challenging Capitalist Conceptions of the Natural Environment. *Consilience* **0**. (doi:10.7916/consilience.v0i13.3927).
135. Kennel, CF. The gathering anthropocene crisis. *The Anthropocene Review* **2**, 81-98. (doi:10.1177/2053019620957355).
136. Indian Government. 2009 Report of the Committee on State Agrarian Relations and the Unfinished Task in Land Reforms.
137. Paudel, B, Pandit, J & Reed, B. 2013 Fragmentation and conversion of agriculture land in Nepal and Land Use Policy 2012.
138. Chen, A, He, H, Wang, J, Li, M, Guan, Q & Hao, J. 2019 A Study on the Arable Land Demand for Food Security in China. *Sustainability* **11**, 4769.
139. Govindaprasad, PK & Manikandan, K. 2016 Farm Land Conversion and Food Security: Empirical Evidences from Three Villages of Tamil Nadu. In *Indian Journal of Agricultural Economics*, pp. 493-503.
140. Islam, GMT, Islam, A, Shopan, AA, Rahman, MM, Lázár, AN & Mukhopadhyay, A. 2015 Implications of agricultural land use change to ecosystem services in the Ganges delta. *J Environ Manage* **161**, 443-452. (doi:10.1016/j.jenvman.2014.11.018).
141. Visser, O. 2017 Running out of farmland? Investment discourses, unstable land values and the sluggishness of asset making. *Agric Human Values* **34**, 185-198. (doi:10.1007/s10460-015-9679-7).
142. Cottrell, RS, Nash, KL, Halpern, BS, Remenyi, TA, Corney, SP, Fleming, A, Fulton, EA, Hornborg, S, Johne, A, Watson, RA, et al. 2019 Food production shocks across land and sea. *Nature Sustainability* **2**, 130-137. (doi:10.1038/s41893-018-0210-1).
143. Prishchepov, AV, Müller, D, Dubinin, M, Baumann, M & Radeloff, VC. 2013 Determinants of agricultural land abandonment in post-Soviet European Russia. *Land Use Pol.* **30**, 873-884. (doi:10.1016/j.landusepol.2012.06.011).
144. Lambin, EF & Meyfroidt, P. 2011 Global land use change, economic globalization, and the looming land scarcity. *Proceedings of the National Academy of Sciences* **108**, 3465-3472. (doi:10.1073/pnas.1100480108).
145. Kaz'min, MA. 2016 Transformation of agricultural land use in Russian regions in the course of modern socioeconomic reforms. *Regional Research of Russia* **6**, 87-94. (doi:10.1134/S2079970516010056).
146. Liefert, WM, Liefert, O, Vocke, G & Allen, EW. 2010 Former Soviet Union Region To Play Larger Role in Meeting World Wheat Needs. In *Amber Waves*, pp. 12-19.
147. Meyfroidt, P, Schierhorn, F, Prishchepov, AV, Müller, D & Kuemmerle, T. 2016 Drivers, constraints and trade-offs associated with recultivating abandoned cropland in Russia, Ukraine and Kazakhstan. *Global Environmental Change* **37**, 1-15. (doi:10.1016/j.gloenvcha.2016.01.003).
148. Schierhorn, F, Müller, D, Beringer, T, Prishchepov, AV, Kuemmerle, T & Balmann, A. 2013 Post-Soviet cropland abandonment and carbon sequestration in European Russia, Ukraine, and Belarus. *Global Biogeochemical Cycles* **27**, 1175-1185. (doi:10.1002/2013gb004654).

149. Rozakis, S & Borek, R. 2018 Evaluation of agricultural reactivation on abandoned lands in Poland. *AgBioForum* **21**, 135-152.
150. Hobbs, RJ & Cramer, VA. 2012 *Old Fields: Dynamics and Restoration of Abandoned Farmland*, Island Press.
151. Yang, Y, Hobbie, SE, Hernandez, RR, Fargione, J, Grodsky, SM, Tilman, D, Zhu, Y-G, Luo, Y, Smith, TM, Jungers, JM, et al. 2020 Restoring Abandoned Farmland to Mitigate Climate Change on a Full Earth. *One Earth* **3**, 176-186. (doi:10.1016/j.oneear.2020.07.019).
152. Bastin, J-F, Finegold, Y, Garcia, C, Mollicone, D, Rezende, M, Routh, D, Zohner, CM & Crowther, TW. 2019 The global tree restoration potential. *Science* **365**, 76-79. (doi:10.1126/science.aax0848).
153. Papworth, T. 2015 The green noose. *Adam Smith Institute* **14**.
154. Amati, M & Taylor, L. 2010 From Green Belts to Green Infrastructure. *Planning Practice & Research* **25**, 143-155. (doi:10.1080/02697451003740122).
155. Cheshire, P. 2014 Turning houses into gold: the failure of British planning. Centre for Economic Performance, LSE.
156. CPRE. 2000 Valuing the land: planning for the best and most versatile agricultural land.
157. Defra. 2020 Farming for the future: Policy and progress update. UK Government.
158. Defra. 2011 Defra Soil Research Programme: Review of the weight that should be given to the protection of best and most versatile (BMV) land.
159. Shortly & A. (Greenbelt Foundation, Toronto), Personal Communication.
160. Carter-Whitney, M, Esakin, TC & Canadian Institute for Environmental Law and, P. 2010 Ontario's greenbelt in an international context. Toronto, Ont., Canadian Institute for Environmental Law and Policy.
161. Loghrin, H. Forthcoming Global Greenbelts Updates since 2009 (2019). Toronto, Greenbelt Foundation.
162. OECD. 2009 Farmland Conversion – The Spatial Implications of Agricultural and Land-use Policies. ed. D Diakosavvas. Paris, OECD.
163. Perrin, C, Clément, C, Melot, R & Nougaredes, B. 2020 Preserving Farmland on the Urban Fringe: A Literature Review on Land Policies in Developed Countries. *Land* **9**, 223.
164. OECD. 2010 *Regional Development Policies in OECD Countries.*, OECD Publishing.
165. Nishi, M. 2019 Multi-Level Governance of Agricultural Land in Japan: Farmers' Perspectives and Responses to Farmland Banking. New York, Columbia.
166. Oberndorf, RB. 2012 Legal review of recently enacted farmland law and vacant, fallow and virgin lands management law: improving the legal & policy frameworks relating to land management in Myanmar. *Food Security Working Group and Land Core Group*.

167. Fernández-Maldonado, AM. 2019 Unboxing the Black Box of Peruvian Planning. *Planning Practice & Research* **34**, 368-386. (doi:10.1080/02697459.2019.1618596).
168. Slätmo, E. - Preservation of Agricultural Land as an Issue of Societal Importance. - **4**, -.
169. Jain, M. 2018 Contemporary urbanization as unregulated growth in India: The story of census towns. *Cities* **73**, 117-127. (doi:10.1016/j.cities.2017.10.017).
170. Sili, M & Soumoulou, L. 2011 The issue of land in Argentina: Conflicts and dynamics of use, holdings and concentration. Rome, IFAD.
171. Robson, JS, Ayad, HM, Wasfi, RA & El-Geneidy, AM. 2012 Spatial disintegration and arable land security in Egypt: A study of small- and moderate-sized urban areas. *Habitat International* **36**, 253-260. (doi:10.1016/j.habitatint.2011.10.001).
172. Dai, W & Dai, W. 2019 Effects of urban expansion on environment by morphological study. *IOP Conference Series: Earth and Environmental Science* **227**, 052004. (doi:10.1088/1755-1315/227/5/052004).
173. Agrimonde-Terra. 2018 *Land Use and Food Security in 2050: a Narrow Road*, éditions Quae.
174. Bezbradica, L, Pantić, M & Gajić, A. 2019 The land use and soil protection: Planning and legal regulations in Serbia. *Zemljište i biljka* **68**, 51-71. (doi:10.5937/ZemBilj1902051B).
175. Owusu Ansah, B & Chigbu, UE. 2020 The Nexus between Peri-Urban Transformation and Customary Land Rights Disputes: Effects on Peri-Urban Development in Trede, Ghana. *Land* **9**, 187.
176. Jiang, L & Zhang, Y. 2016 Modeling Urban Expansion and Agricultural Land Conversion in Henan Province, China: An Integration of Land Use and Socioeconomic Data. *Sustainability* **8**, 920.
177. Li, S. 2018 Change detection: how has urban expansion in Buenos Aires metropolitan region affected croplands. *International Journal of Digital Earth* **11**, 195-211. (doi:10.1080/17538947.2017.1311954).
178. Carreño, L, Frank, FC & Viglizzo, EF. 2012 Tradeoffs between economic and ecosystem services in Argentina during 50 years of land-use change. *Agriculture, Ecosystems & Environment* **154**, 68-77. (doi:10.1016/j.agee.2011.05.019).
179. Żróbek-Róžańska, A & Zielińska-Szczepkowska, J. 2019 National Land Use Policy against the Misuse of the Agricultural Land—Causes and Effects. Evidence from Poland. *Sustainability* **11**, 6403.
180. McPike, JL. 2015 Creating Space for the Formal Amongst the Informal: An Examination of Urban Housing Policies, State Power, and Multi-Scalar Politics in Indian Cities. In *RC21 International Conference on "The Ideal City: between myth and reality. Representations, policies, contradictions and challenges for tomorrow's urban life" Urbino (Italy) 27-29 August 2015*.
181. Bartz, D. 2015 *Soil atlas: Facts and figures about earth, land and fields*. Berlin.
182. Appiah, DO, Asante, F & Nketiah, B. 2019 Perspectives on Agricultural Land Use Conversion and Food Security in Rural Ghana. *Sci* **1**, 14.
183. Schueler, V, Kuemmerle, T & Schröder, H. 2011 Impacts of Surface Gold Mining on Land Use Systems in Western Ghana. *AMBIO* **40**, 528-539. (doi:10.1007/s13280-011-0141-9).

184. Franco, J & Borrás Jr, SM. 2013 Land concentration, land grabbing and people's struggles in Europe. *Transnational Institute, Amsterdam*.
185. Khalid, H & Yusuf, MD. 2012 Resource management: Fragmentation of land ownership and its impact on sustainability of agriculture.
186. Daniels, T & Lapping, M. 2005 Land Preservation: An Essential Ingredient in Smart Growth. *Journal of Planning Literature* **19**, 316-329. (doi:10.1177/0885412204271379).
187. Desrousseaux, M, Schmitt, B, Billet, P, Béchet, B, Le Bissonnais, Y & Ruas, A. 2019 Artificialised Land and Land Take: What Policies Will Limit Its Expansion and/or Reduce Its Impacts? In *International Yearbook of Soil Law and Policy 2018* (eds. H Ginzky, E Dooley, IL Heuser, E Kasimbazi, T Markus & T Qin), pp. 149-165. Cham, Springer International Publishing).
188. Hellerstein, DR. 2002 *Farmland protection : the role of public preferences for rural amenities*. Washington, D.C., U.S. Dept. of Agriculture, Economic Research Service.
189. Gosnell, H, Kline, JD, Chrostek, G & Duncan, J. 2011 Is Oregon's land use planning program conserving forest and farm land? A review of the evidence. *Land Use Pol.* **28**, 185-192. (doi:10.1016/j.landusepol.2010.05.012).
190. Ludlow, D, Falconi, M, Carmichael, L, Croft, N, Di Leginio, M, Fumanti, F, Sheppard, A & Smith, N. 2013 Land Planning and Soil Evaluation Instruments in EEA Member and Cooperating Countries (with inputs from Eionet NRC Land Use and Spatial Planning). Final Report for EEA from ETC/SIA (EEA project managers: G. Louwagie and G. Dige). Available at: <http://www.eea.europa.eu/themes/landuse/document-library>. EEA & ETC/SIA.
191. Grass, I, Loos, J, Baensch, S, Batáry, P, Librán-Embid, F, Ficiciyan, A, Klaus, F, Riechers, M, Rosa, J, Tiede, J, et al. 2019 Land-sharing/-sparing connectivity landscapes for ecosystem services and biodiversity conservation. *People and Nature* **1**, 262-272. (doi:10.1002/pan3.21).
192. Huang, Q, Lu, J, Li, M, Chen, Z & Li, F. 2015 Developing Planning Measures to Preserve Farmland: A Case Study from China. *Sustainability* **7**, 13011-13028.
193. 2003 Law for the preservation of the agricultural lands. ed. I Government.
194. Sartori, D, Catalano, G, Genco, M, Pancotti, C, Sirtori, E, Vignetti, S & Bo, C. 2014 Guide to Cost-benefit Analysis of Investment Projects. Economic appraisal tool for Cohesion Policy 2014-2020.
195. Frazer, JG. 1926 The Worship of Nature. *Nature* **118**, 4-5. (doi:10.1038/118004a0).
196. Malczewski, J. 2004 GIS-based land-use suitability analysis: a critical overview. *Progress in Planning* **62**, 3-65. (doi:10.1016/j.progress.2003.09.002).
197. Dent, D & Young, A. 1981 *Soil Survey and Land Evaluation*, Allen & Unwin.
198. Baveye, PC, Baveye, J & Gowdy, J. 2016 Soil "Ecosystem" Services and Natural Capital: Critical Appraisal of Research on Uncertain Ground. *Frontiers in Environmental Science* **4**. (doi:10.3389/fenvs.2016.00041).

199. Bouwman, AF & Sombroek, WG. 1990 Inputs to climatic change by soils and agriculture related activities: Present status and possible future trends. In *Soils on a Warmer Earth* (pp. 15 - 30. Amsterdam, Elsevier).
200. Bouwman, AF. 1990 Land use related sources of greenhouse gases: Present emissions and possible future trends. *Land Use Pol.* **7**, 154-164. (doi:10.1016/0264-8377(90)90006-K).
201. Batjes, NH & Bridges, EM. 1992 *A Review of Soil Factors and Processes that Control Fluxes of Heat, Moisture and Greenhouse Gases*. Wageningen, The Netherlands., ISRIC.
202. Wall, DH. 2004 *Sustaining Biodiversity and Ecosystem Services in Soils and Sediments*, Island Press.
203. Karlen, DL, Mausbach, MJ, Doran, JW, Cline, RG, Harris, RF & Schuman, GE. 1997 Soil Quality: A Concept, Definition, and Framework for Evaluation (A Guest Editorial). *Soil Science Society of America Journal* **61**, 4-10. (doi:10.2136/sssaj1997.03615995006100010001x).
204. Hannam, I & Boer, B. 2002 Legal and institutional frameworks for sustainable soils.
205. Costanza, R, d'Arge, R, de Groot, R, Farber, S, Grasso, M, Hannon, B, Limburg, K, Naeem, S, O'Neill, RV, Paruelo, J, et al. 1997 The value of the world's ecosystem services and natural capital. *Nature* **387**, 253-260. (doi:10.1038/387253a0).
206. Breure, AM, De Deyn, GB, Dominati, E, Eglin, T, Hedlund, K, Van Orshoven, J & Posthuma, L. 2012 Ecosystem services: a useful concept for soil policy making! *Current Opinion in Environmental Sustainability* **4**, 578-585. (doi:10.1016/j.cosust.2012.10.010).
207. Lescourret, F, Magda, D, Richard, G, Adam-Blondon, A-F, Bardy, M, Baudry, J, Doussan, I, Dumont, B, Lefèvre, F, Litrico, I, et al. 2015 A social–ecological approach to managing multiple agro-ecosystem services. *Current Opinion in Environmental Sustainability* **14**, 68-75. (doi:10.1016/j.cosust.2015.04.001).
208. Robinson, DA, Hockley, N, Cooper, DM, Emmett, BA, Keith, AM, Lebron, I, Reynolds, B, Tipping, E, Tye, AM, Watts, CW, et al. 2013 Natural capital and ecosystem services, developing an appropriate soils framework as a basis for valuation. *Soil Biology and Biochemistry* **57**, 1023-1033. (doi:10.1016/j.soilbio.2012.09.008).
209. Ellili-Bargaoui, Y, Walter, C, Lemercier, B & Michot, D. 2021 Assessment of six soil ecosystem services by coupling simulation modelling and field measurement of soil properties. *Ecological Indicators* **121**, 107211. (doi:10.1016/j.ecolind.2020.107211).
210. Dominati, E, Mackay, A, Green, S & Patterson, M. 2014 A soil change-based methodology for the quantification and valuation of ecosystem services from agro-ecosystems: A case study of pastoral agriculture in New Zealand. *Ecological Economics* **100**, 119-129. (doi:10.1016/j.ecolecon.2014.02.008).
211. Daniels, T. 2020 (Professor of City and Regional Planning, Weitzman School of Design, University of Pennsylvania), Personal communication.
212. Defra. 2020 EIA (Agriculture) regulations: apply to make changes to rural land.
213. 2019 Scottish Government, Guidance on consideration of soil in Strategic Environmental Assessment , v5, 5.4.19.

214. Caldwell, WJ, Hiltz, S & Wilton, B. 2017 *Farmland Preservation: Land for Future Generations*, University of Manitoba Press.
215. Nolon, J. 2003 Land Preservation. *Pace Law Faculty Publications*.
216. Churchman, GJ & Landa, E. 2014 *The soil underfoot : infinite possibilities for a finite resource*, CRC Press.
217. Meadows, DH. 1972 *The Limits to growth; a report for the Club of Rome's project on the predicament of mankind*, New York : Universe Books, [1972].
218. Norström, AV, Cvitanovic, C, Löf, MF, West, S, Wyborn, C, Balvanera, P, Bednarek, AT, Bennett, EM, Biggs, R, de Bremond, A, et al. 2020 Principles for knowledge co-production in sustainability research. *Nature Sustainability* **3**, 182-190. (doi:10.1038/s41893-019-0448-2).
219. Erinosh, BT. 2013 The Revised African Convention on the Conservation of Nature and Natural Resources: Prospects for a Comprehensive Treaty for the Management of Africa's Natural Resources. *African Journal of International and Comparative Law* **21**, 378-397. (doi:10.3366/ajicl.2013.0069).
220. Wilson, C & Matthews, W. 1970 Man's Impact on the Global Environment. Report of the Study of Critical Environmental Problems (SCEP). *MIT Press, Cambridge, Massachusetts* **16**, 19.
221. Lele, S, Springate-Baginski, O, Lakerveld, R, Deb, D & Dash, P. 2013 Ecosystem Services: Origins, Contributions, Pitfalls, and Alternatives. *Conservation and Society* **11**, 343-358. (doi:10.4103/0972-4923.125752).
222. Boer, B, Ginzky, H & Heuser, I. 2017 International Soil Protection Law: History, Concepts and Latest Developments. (pp. 49-72).
223. Bodle, R, Stockhaus, H, Wolff, F, Scherf, C-S & Oberthür, S. 2019 *Improving international soil governance - Analysis and recommendations*.
224. FAO. 1971 *Land degradation*. Rome, Food and Agriculture Organization of the United Nations.
225. FAO. 1971 *Legislative principles of soil conservation*.
226. Europe, Co. 1972 European Soil Charter. ed. Co Europe. Brussels.
227. Tóth, Z. 2018 International dimensions of EU soil policy – The main binding and non-binding legal instruments. *Hungarian Journal of Legal Studies Acta Juridica Hungarica* **59**, 290. (doi:10.1556/2052.2018.59.3.4).
228. FAO. 1981 World Soil Charter.
229. UNEP. 1982 World Soils Policy.
230. Wood & Harold, W. 1985 The United Nations World Charter for Nature: The Developing Nations' Initiative to Establish Protections for the Environment. *Ecology Law Quarterly* **12**, 977.
231. Byron-Cox, R. 2020 From Desertification to Land Degradation Neutrality: The UNCCD and the Development of Legal Instruments for Protection of Soils. In *Legal Instruments for Sustainable Soil*

- Management in Africa* (eds. H Yahyah, H Ginzky, E Kasimbazi, R Kibugi & OC Ruppel), pp. 1-13. Cham, Springer International Publishing).
232. World Commission on Environment and Development. 1987 *Our common future*. Oxford; New York, Oxford University Press.
233. World Bank. 2002 *The First Decade of the GEF: second overall performance study*. Washington DC, World Bank.
234. Wolff, F & Kaphengst, T. 2017 The UN Convention on Biological Diversity and Soils: Status and Future Options. In *International Yearbook of Soil Law and Policy 2016* (eds. H Ginzky, IL Heuser, T Qin, OC Ruppel & P Wegerdt), pp. 129-148. Cham, Springer International Publishing).
235. United Nations Framework Convention on Climate Change, 9 May 1992 (in force 21 March 1994), 31 ILM 849; 1771 UNTS 10 7. (“UNFCCC”). Available at: <https://www.cbd.int/doc/c/f25f/ac08/fac2443375cab303ef45c22/sbstta-24-07-en.pdf>.
236. Global Environment Facility (GEF). 1999 *Clarifying Linkages Between Land Degradation And The GEF Focal Areas: An Action Plan For Enhancing GEF Support: An Action Plan for Enhancing GEF Support*, GEF/C.14/4, November 17, 1999. Washington DC, World Bank Group.
237. Hannam, I & Boer, B. 2019 Land degradation and international environmental law. (pp. 429-440).
238. FAO and UNEP. 2020 *Legislative approaches to sustainable agriculture and natural resources governance. FAO Legislative Study No. 114*. Rome, Food and Agriculture Organization of the United Nations.
239. Chasek, P, Safriel, U, Shikongo, S & Fuhrman, VF. 2015 Operationalizing Zero Net Land Degradation: The next stage in international efforts to combat desertification? *Journal of Arid Environments* **112**, 5-13. (doi:10.1016/j.jaridenv.2014.05.020).
240. zu Ermgassen, SOSE, Baker, J, Griffiths, RA, Strange, N, Struebig, MJ & Bull, JW. 2019 The ecological outcomes of biodiversity offsets under “no net loss” policies: A global review. *Conservation Letters* **12**, e12664. (doi:10.1111/conl.12664).
241. UK Government. 2020 Environment Act 2020. UK.
242. United Nations Framework Convention on Climate Change, 9 May 1992 (in force 21 March 1994), 31 ILM 849 ; 1771 UNTS 10 7. (“UNFCCC”).
243. Fee, E. 2019 Implementing the Paris Climate Agreement: Risks and Opportunities for Sustainable Land Use. In *International Yearbook of Soil Law and Policy 2018* (eds. H Ginzky, E Dooley, IL Heuser, E Kasimbazi, T Markus & T Qin), pp. 249-270. Cham, Springer International Publishing).
244. Convention on Biological Diversity. 2020 *Convention on Biological Diversity: Review of the International Initiative for the Conservation and Sustainable Use of Soil Biodiversity and Updated Plan of Action*.
245. FAO, ITPS, GSBI, SCBD & EC. 2020 *State of knowledge of soil biodiversity - Status, challenges and potentialities: Report 2020*. FAO.

246. Streck, C & Gay, A. 2017 The Role of Soils in International Climate Change Policy. (pp. 105-128).
247. UNFCCC. 2020 Issues related to agriculture.
248. Rojas, RV & Caon, L. 2016 The international year of soils revisited: promoting sustainable soil management beyond 2015. *Environmental Earth Sciences* **75**. (doi:10.1007/s12665-016-5891-z).
249. FAO. 2015 Revised World Soil Charter. Rome, Italy.
250. FAO. 2017 Voluntary Guidelines for Sustainable Soil Management.
251. Orr, B, Cowie, A, Castillo Sanchez, V, Chasek, P, Crossman, N, Erlewein, A, Louwagie, G, Maron, M, Metternicht, G & Minelli, S. 2017 Scientific conceptual framework for land degradation neutrality. In *A report of the science-policy interface. United Nations Convention to Combat Desertification (UNCCD), Bonn, Germany*, pp. 1-98.
252. International Law Association. 2020 ILA Guidelines on the Role of International Law in Sustainable Natural Resources Management for Development, Resolution 4/2020, 2020 Report of the Seventy Ninth Conference, Kyoto (forthcoming).
253. Minasny, B, Malone, BP, McBratney, AB, Angers, DA, Arrouays, D, Chambers, A, Chaplot, V, Chen, Z-S, Cheng, K, Das, BS, et al. 2017 Soil carbon 4 per mille. *Geoderma* **292**, 59-86. (doi:10.1016/j.geoderma.2017.01.002).
254. Chabbi, A, Lehmann, J, Ciais, P, Loescher, HW, Cotrufo, MF, Don, A, SanClements, M, Schipper, L, Six, J, Smith, P, et al. 2017 Aligning agriculture and climate policy. *Nature Climate Change* **7**, 307-309. (doi:10.1038/nclimate3286).
255. Poulton, P, Johnston, J, Macdonald, A, White, R & Powlson, D. 2018 Major limitations to achieving “4 per 1000” increases in soil organic carbon stock in temperate regions: Evidence from long-term experiments at Rothamsted Research, United Kingdom. *Glob. Change Biol.* **24**, 2563-2584. (doi:10.1111/gcb.14066).
256. Minasny, B, Arrouays, D, McBratney, AB, Angers, DA, Chambers, A, Chaplot, V, Chen, Z-S, Cheng, K, Das, BS, Field, DJ, et al. 2018 Rejoinder to Comments on Minasny et al., 2017 Soil carbon 4 per mille *Geoderma* **292**, 59–86. *Geoderma* **309**, 124-129. (doi:10.1016/j.geoderma.2017.05.026).
257. Soussana, J-F, Lutfalla, S, Ehrhardt, F, Rosenstock, T, Lamanna, C, Havlík, P, Richards, M, Lini, E, Wollenberg, E, Chotte, J-L, et al. 2019 Matching policy and science: Rationale for the '4 per 1000-soils for food security and climate' initiative.
258. Kibugi, R. 2018 Soil Health, Sustainable Land Management and Land Degradation in Africa: Legal Options on the Need for a Specific African Soil Convention or Protocol. In *International Yearbook of Soil Law and Policy 2017* (eds. H Ginzky, E Dooley, IL Heuser, E Kasimbazi, T Markus & T Qin), pp. 387-411. Cham, Springer International Publishing).
259. Markus, T. 2017 The Alpine Convention’s Soil Conservation Protocol: A Model Regime? In *International Yearbook of Soil Law and Policy 2016* (eds. H Ginzky, IL Heuser, T Qin, OC Ruppel & P Wegerdt), pp. 149-164. Cham, Springer International Publishing).

260. 2003 *Revised European Charter for the Protection and Sustainable Management of Soil*, Strasbourg, 17 July 2003 CO-DBP/documents/codbp2003/10e.
261. European Commission. 2006 Soil protection - The story behind the Strategy.
262. Stankovics, P, Tóth, G & Tóth, Z. 2018 Identifying Gaps between the Legislative Tools of Soil Protection in the EU Member States for a Common European Soil Protection Legislation. *Sustainability* **10**, 2886. (doi:10.3390/su10082886).
263. Montanarella, L & Panagos, P. 2021 The relevance of sustainable soil management within the European Green Deal. *Land Use Pol.* **100**, 104950. (doi:10.1016/j.landusepol.2020.104950).
264. Knox, JH. 2017 Report of the Special Rapporteur on the Issue of Human Rights Obligations Relating to the Enjoyment of a Safe, Clean, Healthy and Sustainable Environment :note. Geneva, UN.
265. UN. Human Rights Council (39th sess.: 2018: Geneva). 2018 United Nations Declaration on the Rights of Peasants and Other People Working in Rural Areas :resolution. Geneva, UN.
266. Fromherz, NA. 2012 The Case for a Global Treaty on Soil Conservation, Sustainable Farming, and the Preservation of Agrarian Culture. **39**. (doi:10.15779/Z38BC49).
267. De Schutter, O. 2010 The Emerging Human Right to Land. *International Community Law Review* **12**, 303-334. (doi:10.1163/187197310X513725).
268. Stec, S & Jendrośka, J. 2019 The Escazú Agreement and the Regional Approach to Rio Principle 10: Process, Innovation, and Shortcomings. *Journal of Environmental Law* **31**, 533-545. (doi:10.1093/jel/eqz027).
269. Keesstra, SD, Bouma, J, Wallinga, J, Tittonell, P, Smith, P, Cerdà, A, Montanarella, L, Quinton, JN, Pachepsky, Y, van der Putten, WH, et al. 2016 The significance of soils and soil science towards realization of the United Nations Sustainable Development Goals. *SOIL* **2**, 111-128. (doi:10.5194/soil-2-111-2016).
270. Gil, JDB, Reidsma, P, Giller, K, Todman, L, Whitmore, A & van Ittersum, M. 2019 Sustainable development goal 2: Improved targets and indicators for agriculture and food security. *Ambio* **48**, 685-698. (doi:10.1007/s13280-018-1101-4).
271. Tóth, G, Hermann, T, da Silva, MR & Montanarella, L. 2018 Monitoring soil for sustainable development and land degradation neutrality. *Environmental Monitoring and Assessment* **190**, 57. (doi:10.1007/s10661-017-6415-3).
272. Shen, X, Wang, X, Zhang, Z, Lu, Z & Lv, T. 2019 Evaluating the effectiveness of land use plans in containing urban expansion: An integrated view. *Land Use Pol.* **80**, 205-213. (doi:10.1016/j.landusepol.2018.10.001).
273. Roose, A, Kull, A, Gauk, M & Tali, T. 2013 Land use policy shocks in the post-communist urban fringe: A case study of Estonia. *Land Use Pol.* **30**, 76-83. (doi:10.1016/j.landusepol.2012.02.008).
274. Lu, Y, Song, S, Wang, R, Liu, Z, Meng, J, Sweetman, AJ, Jenkins, A, Ferrier, RC, Li, H, Luo, W, et al. 2015 Impacts of soil and water pollution on food safety and health risks in China. *Environment International* **77**, 5-15. (doi:10.1016/j.envint.2014.12.010).

275. Zhao, F-J, Ma, Y, Zhu, Y-G, Tang, Z & McGrath, SP. 2015 Soil Contamination in China: Current Status and Mitigation Strategies. *Environmental Science & Technology* **49**, 750-759. (doi:10.1021/es5047099).
276. Chen, R, de Sherbinin, A, Ye, C & Shi, G. 2014 China's Soil Pollution: Farms on the Frontline. *Science* **344**, 691-691. (doi:10.1126/science.344.6185.691-a).
277. Vogel, H-J, Eberhardt, E, Franko, U, Lang, B, Ließ, M, Weller, U, Wiesmeier, M & Wollschläger, U. 2019 Quantitative Evaluation of Soil Functions: Potential and State. *Frontiers in Environmental Science* **7**. (doi:10.3389/fenvs.2019.00164).
278. Vargas, L, Willemsen, L & Hein, L. 2019 Assessing the Capacity of Ecosystems to Supply Ecosystem Services Using Remote Sensing and An Ecosystem Accounting Approach. *Environmental Management* **63**, 1-15. (doi:10.1007/s00267-018-1110-x).
279. Ostle, NJ, Levy, PE, Evans, CD & Smith, P. 2009 UK land use and soil carbon sequestration. *Land Use Pol.* **26**, S274-S283. (doi:10.1016/j.landusepol.2009.08.006).
280. Buschmann, C, Röder, N, Berglund, K, Berglund, Ö, Lærke, PE, Maddison, M, Mander, Ü, Myllys, M, Osterburg, B & van den Akker, JJH. 2020 Perspectives on agriculturally used drained peat soils: Comparison of the socioeconomic and ecological business environments of six European regions. *Land Use Pol.* **90**, 104181. (doi:10.1016/j.landusepol.2019.104181).
281. Committee on Climate Change. 2020 Land use: Policies for a Net Zero UK.
282. Kibblewhite, MG, Ritz, K & Swift, MJ. 2008 Soil health in agricultural systems. *Philos Trans R Soc Lond B Biol Sci* **363**, 685-701. (doi:10.1098/rstb.2007.2178).
283. Powlson, DS, Gregory, PJ, Whalley, WR, Quinton, JN, Hopkins, DW, Whitmore, AP, Hirsch, PR & Goulding, KWT. 2011 Soil management in relation to sustainable agriculture and ecosystem services. *Food Policy* **36**, Supplement 1, S72-S87. (doi:10.1016/j.foodpol.2010.11.025).
284. Paustian, K, Lehmann, J, Ogle, S, Reay, D, Robertson, GP & Smith, P. 2016 Climate-smart soils. *Nature* **532**, 49. (doi:10.1038/nature17174).
285. LaCanne, CE & Lundgren, JG. 2018 Regenerative agriculture: merging farming and natural resource conservation profitably. *PeerJ* **6**, e4428-e4428. (doi:10.7717/peerj.4428).
286. Lal, R & Kosaki, T. 2018 *Soils and Sustainable Development Goals. GeoEcology Essay*, Schweizerbart Science Publishers.
287. Strouhalová - Vysloužilová, B, Ertlen, D, Schwartz, D & Šefrna, L. 2016 Chernozem. From concept to classification: a review. *AUC GEOGRAPHICA* **51**, 85-95. (doi:10.14712/23361980.2016.8).
288. Amundson, R. 2020 The policy challenges to managing global soil resources. *Geoderma* **379**, 114639. (doi:10.1016/j.geoderma.2020.114639).
289. Frelih-Larsen, A, Bowyer, C, Albrecht, S, Keenleyside, C, Kemper, M, Nanni, S, Naumann, S, Mottershead, RD, Langrebe, R, Andersen, E, et al. 2017 *Updated Inventory and Assessment of Soil Protection Policy Instruments in EU Member States*.

290. Ronchi, S, Salata, S, Arcidiacono, A, Piroli, E & Montanarella, L. 2019 Policy instruments for soil protection among the EU member states: A comparative analysis. *Land Use Pol.* **82**, 763-780. (doi:10.1016/j.landusepol.2019.01.017).
291. Wingeyer, AB, Amado, TJC, Pérez-Bidegain, M, Studdert, GA, Varela, CHP, Garcia, FO & Karlen, DL. 2015 Soil Quality Impacts of Current South American Agricultural Practices. *Sustainability* **7**, 2213-2242.
292. Webb, A, Kelly, G & Dougherty, W. 2015 Soil governance in the agricultural landscapes of New South Wales, Australia. *International Journal of Rural Law and Policy Special Edition* **1**, 1-16. (doi:10.5130/ijrlp.i1.2015.4169).
293. Chukov, SN & Yakovlev, AS. 2019 Soil and Land Categories in the Modern Legislation of Russia. *Eurasian Soil Science* **52**, 865-870. (doi:10.1134/S1064229319070020).
294. IUCN. 2019 World Commission on Environmental Law, Soil Desertification & Sustainable Agriculture Specialist Group, 2019 Mid-Year Report. Online.
295. Hazelton, PA, Frossard, E, Blum, WEH & Warkentin, BP. 2006 Australian examples of the role of soils in environmental problems. In *Function of Soils for Human Societies and the Environment* (p. 0, Geological Society of London).
296. Sophia Antipolis. 2003 Threats to Soils in Mediterranean Countries: Document Review. In *Plan Bleu Papers*. Valbonne, France, Plan Bleu.
297. Gonzalez Lago, M, Plant, R & Jacobs, B. 2019 Re-politicising soils: What is the role of soil framings in setting the agenda? *Geoderma* **349**, 97-106. (doi:10.1016/j.geoderma.2019.04.021).
298. Heuser, I. 2018 Development of Soil Awareness in Europe and Other Regions: Historical and Ethical Reflections About European (and International) Soil Protection Law. (pp. 451-474).
299. Moallemi, EA, Haan, F, Hadjidakou, M, Khatami, S, Malekpour, S, Smajgl, A, Stafford Smith, M, Voinov, A, Bandari, R, Lamichhane, P, et al. 2021 Evaluating Participatory Modeling Methods for Co-creating Pathways to Sustainability. *Earth's Future* **9**. (doi:10.1029/2020EF001843).
300. Koch, A, McBratney, A, Adams, M, Field, D, Hill, R, Crawford, J, Minasny, B, Lal, R, Abbott, L, O'Donnell, A, et al. 2013 Soil Security: Solving the Global Soil Crisis. *Global Policy* **4**, 434-441. (doi:10.1111/1758-5899.12096).
301. McBratney, A, Field, DJ & Koch, A. 2014 The dimensions of soil security. *Geoderma* **213**, 203-213. (doi:10.1016/j.geoderma.2013.08.013).
302. Bouma, J. 2020 Soil security as a roadmap focusing soil contributions on sustainable development agendas. *Soil Security* **1**, 100001. (doi:10.1016/j.soisec.2020.100001).
303. Hill, R. 2017 The Place of Soil in International Government Policy. In *Global Soil Security* (eds. DJ Field, CLS Morgan & AB McBratney), pp. 443-449. Cham, Springer International Publishing).
304. Field, DJ, Morgan, CLS & McBratney, AB. 2017 *Global Soil Security*, Springer International Publishing, A. G.

305. Lilburne, L, Eger, A, Mudge, P, Ausseil, A-G, Stevenson, B, Herzig, A & Beare, M. 2020 The Land Resource Circle: Supporting land-use decision making with an ecosystem-service-based framework of soil functions. *Geoderma* **363**, 114134. (doi:10.1016/j.geoderma.2019.114134).
306. Juerges, N, Hagemann, N & Bartke, S. 2018 A tool to analyse instruments for soil governance: the REEL-framework. *Journal of Environmental Policy & Planning* **20**, 617-631. (doi:10.1080/1523908X.2018.1474731).
307. Ginzky, H. 2020 Good Governance for Sustainable Management of Soil on National and International Level: How to Do It?

Saving the ground beneath our feet

Establishing priorities and criteria for governing soil use and protection

Lewis Peake¹ and Cairo Robb²

¹ University of East Anglia, School of Environmental Science, Norwich, NR4 7TJ, United Kingdom;
corresponding author: l.peake@uea.ac.uk; +447940240683, +441603702560

² Legal Research Fellow, Centre for International Sustainable Development Law, <https://www.cisd.org>

Abstract

The relentless loss and impairment of soil functions and ecosystems services across the globe calls for a fundamental reconsideration of soil governance mechanisms. This review charts the history and evolution of national and international soil law and seeks to unravel the challenges that have contributed to this failure in governance. It describes and categorises law and policy responses to different soil threats, and identifies a worrying widespread absence of legislation for oversight and protection of agricultural soils from urbanisation, as well as a lack of clear legal mechanisms to set and determine national priorities for soil protection. A reduction in the world's prime farmland threatens soil ecosystem services, including food security, carbon storage, and biodiversity. Falling between the stalls of agricultural and environmental law, the fate of farmland is often left to planners who do not see themselves as responsible for soils. Consequently, legal instruments with the greatest power to transform soil, sometimes irreversibly, are often framed and worded with little or no reference to soil. Nevertheless, emerging conceptual frameworks might offer positive outcomes. The authors advocate robust holistic policies of soil governance and land use planning that place soil ecosystem services and natural capital at the heart of decision making. (197 words)

Keywords

agricultural land conversion, farmland preservation, land take, soil ecosystem services, soil governance, soil security

Introduction

Land belongs to a vast family of whom many are dead, a few are living and a countless host are still unborn.

- Traditional Ashanti adage [1]

It is the value of the improvements only, and not the earth itself, that is individual property.

- Thomas Paine, *Agrarian Justice* (1795).

These stylistically contrasting statements from two different cultures and geographic zones essentially embody the same axiomatic concept: that the ground beneath our feet provides so much more than the ground beneath our feet, and that holistically it transcends title deeds, boundary fences and even national borders. Soil is a vital and enduring resource that constitutes the metaphorical as well as the literal foundations of human civilization. This multidimensional and multifunctional resource has always provided a range of benefits, of which food production is just the most obvious example. As the world's population has soared and almost every parcel of its terrestrial surface has been assigned to the various national states, and then further subdivided and transformed by governments, private corporations or individuals, the need to safeguard this increasingly degraded resource is more urgent than ever.

A recurring theme in futuristic scenarios is the prediction that humans will one day grow or make all of their food in laboratories and vertical farms, deploying soilless hydroponics. Technically this is perfectly possible, should humans survive long enough to achieve it but, as in the case of many societal endeavours, the challenges are primarily social, economic and political. So for the time being at least, this is science fiction. The existential reality is that despite extraordinary technological advances, especially in the fields of engineering, biotechnology, computation and communication, we are as dependent on the soil for our sustenance as we have ever been. In fact, we are arguably more dependent on the multifaceted and critical functions of soil than ever, as our dominance as a species has resulted in greater demands and stressors on our environment.

Soil ecosystem services (SES) is a relatively new academic subdiscipline [2], but human awareness of the holistic functionality of soil is as old as civilization, if not older. When the Romans recorded some of the oldest known soil conservation laws, in the 6th century *Digesta* of Justinian, they described a range of scenarios each with their own legal argument, not necessarily related to soil productivity, for example, that might lead to disputes between neighbouring farms, including the damage to a neighbour's soil from uprooting trees or channelling water, or the harm that might arise from eroding sediment [3].

A millennium later, coincidentally not far from Rome, Leonardo da Vinci directed a major land reclamation project which depended on the ability of soil to absorb and then release water, and also to trap and redirect it

as required. These two historical examples both touch on aspects of soil not directly related to primary production, but instead to its various roles now collectively referred to as ecosystem services. It was also da Vinci who famously declared: “*We know more about the movement of celestial bodies than about the soil underfoot.*” [4]. Indeed, while many scientists have famously claimed that the human brain is the most complex structure in the known universe [5], advanced 3D tomography has in fact revealed that, arguably: “*Soil is the most complicated biomaterial on the planet.*” [6].

The almost infinite utility of soil is universal. In addition to the familiar and very direct role that soil has in providing our food, fibre and timber, we often forget its many other interconnected functions, for example:

- Physical: provision of building or landscaping material; stable substrate for infrastructure;
- Chemical: regulating the adsorption and release of nutrients;
- Biological: constituting biodiverse ecosystems and communities of useful organisms, conversion of toxic sewage and decomposing matter into plant nutrients, genetic reservoir;
- Hydrological: water absorption, storage and filtration; flood control;
- Cultural: repository of cultural heritage and archaeological information, recreation;
- Climatic: carbon (C) storage.

The last of these would have been almost inconceivable to our forebears and has only recently assumed paramount importance, approximately since the beginning of the new millennium.

The other side of the coin is that in certain circumstances soil, like any other natural resource, can deliver negative impacts, such as emitting greenhouse gases (GHGs), harbouring pests and diseases, or contributing to the siltation of water, the accumulation of sediment, dust storms and landslides. These events are reminders that good governance of soil is also about minimising ecosystem “disservices” as well as safeguarding essential ecosystem services.

We live in a time of anthropogenic environmental crisis, of unprecedented climate change and species extinction. Hardly a day goes by without another meteorological record being broken or ecological threshold being breached somewhere. A report just released by WWF states that: “*Since 1970, our Ecological Footprint has exceeded the Earth’s rate of regeneration,*” and currently exceeds the world’s capacity to support humanity by 56%. In other words, for 50 years we have been exceeding our running costs and depleting the planet’s natural capital. This is despite a technological expansion of the world’s biocapacity of 28% since 1960 [7]. The irony of this crisis is that it is the direct result of human ecological “success” – and excess – but the impact is untold human suffering and ultimately perhaps our demise. Soil is an integral component within this system, both as an enabler and a victim of exponential human expansion.

In Hardin’s 1968 essay *Tragedy of the Commons* [8], he presents a parable of herders grazing their cattle on common land as an analogy for our accelerating use of the Earth’s finite natural resources. Every herder

knows that the shared pasture has limited carrying capacity, but each one may also decide that it will be in their interest to add more animals to their herd, because their personal gain will (they believe) exceed the collective harm. The inevitable result is overgrazing and general ruin for all (probably involving soil degradation, although Hardin did not mention that!). For the first time in history, we are facing a tragedy of the commons on a global scale. In his book *Collapse* Jared Diamond [9] highlights how most societies eventually disintegrated for combinations of partly avoidable reasons, including resource depletion, and chillingly suggests that all of these antecedents may simply be pilot studies for the ultimate collapse.

In this critical review we trace the long historical path of the status of soil, in terms of its value as a natural resource for society and with respect to its evolutionary governance. In the course of the analysis we present and describe categories of anthropogenic threat to soil and the types of legal instruments that can be applied to address these threats. There is also a discussion of the semantics surrounding legal aspects of soil and land, and an attempt to disambiguate and illuminate the use of terminology. The current state of the art of soil governance is explored in depth and a key finding is a failure to prevent or even acknowledge the worldwide loss of prime farmland and the consequential implications of this for society. The review concludes on a hopeful note by looking to emerging approaches such as the soil security conceptual framework, proposed as a way of translating critical soil knowledge into sustainable development policy, and to holistic land evaluation methodologies that incorporate SES into land use planning.

Methods

This study is underpinned by the role of soil governance in helping to achieve the UN Sustainable Development Goals (SDGs), mitigating and adapting to climate change and contributing to a safer future for humanity. Nearly 300 references have been cited out of a larger number inspected, but the literature search was intentionally not constrained to a narrowly directed keyword search. Moreover, our intention was to add new content to the debate. Our initial premise was that if SES, which we acknowledge are critical to human wellbeing, are being dangerously diminished, is this due in part to a failure of governance? If so, to what extent is this a failure of legislation, implementation or both? In terms of *priorities* and *criteria*, as mentioned in the subtitle, where are the gaps and weaknesses, and are there opportunities to address these? Furthermore, is this particular mix of problems and potential solutions entangled with wider issues of terminology, perception and how society has historically managed and valued soil? What would be the most effective contributions from soil science, scientists and practitioners?

From this starting point, we undertook an open-ended literature review. Particular words or phrases became key search terms, such as combinations of “agriculture”, “biodiversity”, “conservation”, “contamination”, “conversion”, “degradation”, “ecosystem services”, “erosion”, “farmland”, “governance”, “land”, “land take”, “land use planning”, “legislation”, “non-agricultural”, “preservation”, “protection”, “SDG”,

urbanis(z)ation”, “urban sprawl”, “soil”, “soil sealing”, and so on, depending on which section of the article was the focus. However, an exhaustive search with such a wide and holistic remit would have been impractical and ultimately yield diminishing returns. Instead, the approach taken was to home in on a few specific areas of interest, based on our own judgement and current trends, and to follow these narratives organically in promising directions. The ensuing strategy was that by selective close reading and analysis we synthesised key points, such as our own categorisations of the terminology and information. By presenting in this way, the intention was twofold: to provide the reader with some clear signposts within a large and complex domain; and to provide ourselves with a framework within which the context of any given issue could be better understood and deconstructed.

Data and cases studies from the literature are referred to throughout to support our arguments. In addition, a comprehensive search was conducted to abstract legislative data from FAOLEX, the UN Food and Agriculture Organization (FAO) online database of policies and laws. The FAOLEX online database documents as far as possible legal instruments related to agriculture and natural resources in force in the nearly 200 countries of the world. This is a hierarchical database organised, by country, into the following groups, of which those marked with an asterisk were reviewed by the lead author (where they existed for the country in question): ***Policies****; ***Legislation: Agricultural and rural development****, ***Climate change, Cultivated plants, Disaster risk management, Environment****, ***Fisheries, Food and nutrition, Forestry*, Land and soil****, ***Livestock, Sea, Water, Wild species and ecosystems****; ***International Agreements****. Of these groups, *Land and soil* was the most useful, but it was necessary to widen the search further to obtain a comprehensive picture, because in many cases this group was primarily devoted to the administration of land ownership and land reform. These groups are not mutually exclusive, so some legislation appears more than once. Not surprisingly, for some countries, e.g. the US or China, the list of entries runs to hundreds.

Very occasionally the name of the FAOLEX entry alone was broadly sufficient to explain its meaning, e.g. “Soil conservation law”, but in most cases it was necessary to select the embedded hyperlink to access its descriptive summary (usually in English). However, in many cases, ambiguous or over-simplified wording made it necessary to select a further hyperlink to the full text of the legal instrument, either within FAOLEX or on the websites of national governments. These documents were often in the local language, and sometimes also in the local script, e.g. Russian or Arabic. Where it was felt useful to do so, and where feasible, Google Translate was used to interpret the text. In a few instances further corroboration or clarification was obtained from academic articles in this article’s bibliography or via personal communication from individuals professionally familiar with the legislation. In a small number of interesting cases local experts were contacted to translate or explain some of the wording and, furthermore, asked whether the laws in question were implemented and effectively governed. Approximately 1% of countries had either created no such instruments or lacked any accessible data in FAOLEX. This

information was compiled during the summer and autumn months of 2020. A spreadsheet was created with one row per country and columns indicating categories of legal instrument, and from this it was possible to calculate percentages of countries with such measures in force. For our purposes the FAOLEX groups are merely retrieval mechanisms, unrelated to our categories that were derived on the basis of our analysis of soil governance. The resulting data is presented in Table 3 and discussed in the section *The modern governance of soil resources worldwide*, towards the end of the article.

While our research was being conducted FAO was in the process of creating a similar, but soil-specific, repository (SoiLEX) from the data in FAOLEX, but this was unavailable for our purposes at the time¹. In terms of structural organisation and ease of use, SoiLEX is superior to FAOLEX and would have speeded up our search. However, SoiLEX is limited to highly soil-specific categories of legislation, whereas FAOLEX is wider in scope and, crucially for our purposes, contains laws relating to the protection or preservation of categories of land such as prime farmland and natural ecosystems.

Some caveats are necessary. Even setting aside subsequent updates, such a summary can only ever be approximate, and is undoubtedly incomplete. Several polities listed in FAOLEX as countries are actually a number of countries with differing laws (e.g. the UK), and other countries consist of separate provinces or states with differing laws (e.g. the US). The laws of some countries are recorded only in the local language or script. Furthermore, many instruments, especially with respect to policies, are generalised, and in some cases it can be difficult to ascertain legislative intent. For example, a law stating that land will be classified as urban or rural could imply protected status or simply spatial planning data; the latter was assumed unless explicitly defined. Many legal statements are also terse to the point of ambiguity or even circularity, for example: “*The land is the main national wealth, which is under special protection of the state...ensuring rational use and protection of land,*“ [11] iterated many times, but with little or no further qualification. Furthermore, this is purely a review of officially documented policies and laws, not an assessment of enforcement or effective governance.

Soil as a valued resource and legal entity

¹ SoiLEX is described as a global database on national legislation on soil protection, conservation and restoration to facilitate access to information on the existing legal instruments in force and bridge the gap between the various soil stakeholders. The online platform is intended to facilitate the search for national soil legal instruments, and the understanding of the different legal areas relevant to soil management and protection, as well as the exchange of experiences in soil governance between countries and regions. 10. Global Soil Partnership web page. 2020 Contribute to the GSP newest tool: SoiLEX..

The root and “commonwealth” of civilization

Intuitive human awareness of the relationship between soil type and natural resource availability is self-evident from history. Even before the invention of agriculture, land that retained moisture and plant nutrients was the place to find rich pickings which, as any historian will tell you, is why the first settled civilisations developed in wide flat river floodplains and deltas.

When Herodotus said, Egypt is the gift of the Nile [12], he was explicitly referring to the land-forming deposition of black silty clay by the annual river flood, which incidentally has been occurring at the rate of 1mm per annum (1m per millennium [13]). The ancient Egyptians set their calendar according to these floods and called their nation state *Kemet* which means black land or dark earth, in contrast to the barren *Deshret*, or red land, stretching out on either side. The polygonal state boundaries of modern Egypt would have been meaningless to its former inhabitants who knew that their country stopped where the irrigated feddans met the dry sand. The eponymous black soil was not simply a metaphor for their country; as far as they were concerned, it was their country.

Human dependence on productive soil and the impact of its degradation on civilisation throughout history is well attested [14-18]. Mesopotamia experienced at least two catastrophic regime changes due respectively to soil salinity and soil erosion [19, 20]. The Greek and Roman writers make multiple references both to the benefits of soil quality and the threat of soil degradation, and by the 1500s in Europe soil was regarded as the most important factor of an economy. History is also full of examples of land-related wars and conquests, often correlated with soil quality [21-23], and tragically the killing of environmental defenders, including those seeking to protect land and soil quality, continues today [24, 25].

The evolution of soil law

The earliest known attempts to put an economic value on soil, that is productive land, were Chinese soil classification and maps created 2000 years BCE for taxation purposes [26]. Other ancient peoples also independently derived similar systems, for example in Mesoamerica [27]. However, while there are many historical examples of land ownership rights and transactions [28], it is difficult to pin down when laws or policies were first introduced to protect soil or land as a vulnerable or scarce resource in its own right, that is part of the natural capital that belongs to society and is for the common good, assets formerly referred to as the *commonweal* or “commonwealth”. One prescient promulgation of reduced tillage appears in the *Manusmriti*, a collection of Hindu laws thought to date from 1000 BCE which advises against the use of iron-tipped ploughs “...because they injure the Earth and its creatures.” [29].

The *Statutes on Agriculture*, compiled during the Qin Dynasty of the late 4th century BCE is China’s, and possibly the world’s, earliest known set of soil-related laws which mandate, *inter alia*, field sizes and the

correct season for earth-moving [30]. Even before this period, Chinese rulers made it known that feeding the nation was paramount for its survival [31]. The *Statutes* were followed a century later, during the Han Dynasty (206 BCE-8 CE), by the *Book of Fan Shengzhi*, China's oldest surviving agronomic treatise which provides extraordinarily detailed and precise instructions on almost every aspect of soil management [32]. It would take perhaps another thousand years before this work was eclipsed by the Arab agronomic scholars of Andalusia [33]. Although this document is presented as advisory rather than legally binding, its author, *Fan Shengzhi*, was the equivalent of the Minister of Agriculture and its arrival coincided with a radical transformation of China's agriculture, so it seems probable that it was regarded by farmers as de facto law. Perhaps the oldest surviving documented legislation explicitly concerned with soil is that contained in the *Digesta* of the Eastern Roman emperor Justinian, published in the 6th century, but including many laws dating from centuries earlier. While there was no overt implication that soil per se was a scarce resource in the Roman Empire, these edicts leave the reader in no doubt that soil was of great importance to everyone involved in its use, from the question of who benefited from its bounty to who was responsible for its maintenance [3]. Over the subsequent centuries the value of soil or the need to protect it appears occasionally in the literature, but hardly the fear of an absolute scarcity of soil, or of vast areas of it becoming unusable. However, two millennia after those early Chinese and Roman ordinances were recorded, that mindset gradually began to change as time and time again human communities rudely bumped up against the limits to growth.

In the modern era, the world's first legally protected forest reserve, for the purpose of conservation, was created on the Caribbean island of Tobago in 1776 to preserve the "*fertility of lands*" [34]. However, the first specific instances of soil legislation were usually initiated in response to the threat of severe soil erosion. In many cases this was triggered by sudden catastrophic events which were typically the synergistic result of inappropriate land management combined with extreme climatic conditions. The most iconic example is the US Dust Bowl of the 1930s which dealt a massive psychological blow to world's wealthiest nation. Since that time, soil erosion has been perceived as an ever-present threat to American farmers and food production, so it is no surprise that the first major piece of soil-related legislation in the US was the 1935 Soil Conservation Act, which still survives in a modified form. This act was followed by a steady stream of further soil and land-related legislation in the US and elsewhere, but it was not the first such law in the modern world.

Possibly the first modern ordinance explicitly designed to prevent soil erosion, was the *General Order banning tree-cutting in Hawkesbury River valley*, NSW, Australia in 1803 [35] and by the 1870s there was widespread awareness of the problem of erosion, for example in colonial Africa [36]. The environmental movement which had emerged in Europe and America since the Industrial Revolution also had a growing influence on policy [37, 38] and forest protection laws that embodied the principle of soil conservation, if

not the wording, began to appear around the world, for example in the US from 1873 onwards [38] and New Zealand in 1874 [39]. In 1894 grazing was banned on federal reserves in the US to prevent land degradation [38] and in 1901 the Indian state of Punjab, then under British administration, passed what might be considered the first true soil and water conservation law, the *Punjab Land Preservation Act*, which essentially remains in force today [40]. However, it was Iceland, in 1907, that passed the first national parliamentary soil conservation act [41].

A significant result of the Dust Bowl and President Roosevelt's generous response to it, in parallel to the practical and legal assistance given to farmers, was the huge contribution made by the United States Department of Agriculture (USDA) and American scientists to the worldwide study and professional management of soils. With hindsight, the officially established causes of the crisis, and some applied aspects of the USDA's arguably overwhelming response, have come in for criticism [42-44], as well as the narrow focus of its globally influential research agenda [45]. However, the far-reaching impact of this intense period of activity, barely perceived outside the sphere of soil science, is perhaps unparalleled in human history. Building on a legacy of international scholarship, especially in Germany and Russia [46], in the space of a decade at least three trailblazing conceptual tools were created: the Soil Taxonomy, the Land Capability Classification (LCC) methodology and the Universal Soil Loss Equation (USLE). These would radically transform the international science, management and legislation of soils.

From the 1930s onwards other nations rapidly fell into step, passing soil conservation laws and other forms of soil and land legislation. While food security was a major driver, so was the loss of rural landscapes and wilderness. Other kinds of environmental protection were also implemented throughout the 20th century, such as the creation of green belts to protect countryside around cities, even as early as 1901 [47]. In the 1940s, however, underscoring the growing appreciation of protecting farmland, World War II had multiple impacts on food production and distribution, especially in Europe, forcing many countries to become more self-reliant. The postwar world was one in which agricultural productivity took on a new significance and the new technique of land evaluation was enthusiastically applied [48, 49].

In the UK, for example, even before the end of the war, the Scott Report on rural land use signalled a new approach to planning that incorporated both landscape amenity and food production: "*Land which is classified as good agricultural land should only be alienated from its present use if it can be shown to be in the general national interest.*" [50]. A member of the Scott Committee opined: "...*carving-out of a place for agriculture in the world of planning was perhaps the most significant achievement of the Report,*" and that, "*maps of land utilisation, land classification, and types of farming...became recognized tools of the planning trade.*" [51]. The Report has also been described as a prescient aspiration for sustainable development [52]. This report fed into the 1944 White Paper, *The Control of Land Use* [53] which paved the way for one of the earliest and most radical planning policies of its kind anywhere, the UK 1947 Town and

Country Planning Act which nationalised land use regulations and has been emulated worldwide [54]. Having ring-fenced farmland, the reforming Labour government bolstered this with financial support for farmers via the 1947 Agriculture Act.

It was therefore especially unfortunate that in the same year as these landmark pieces of legislation, when the British government embarked on the Groundnut Scheme in what was then Tanganyika (now Tanzania), in the final years of its overseas Empire, it eschewed the very land evaluation methods that might have prevented a promising project collapsing into a catastrophic misadventure [55]. Meanwhile, the land classification techniques adapted from the USDA LCC methodology and being put to work by soil surveyors in postwar Britain, were being applied not to find new land to cultivate, but to identify and protect existing prime farmland. While the driving force of the LCC had been on-site farm management and soil conservation, the British system was primarily a tool of the new planning system. The latest incarnation of this system in England and Wales is the 1988 Agricultural Land Classification (ALC) set of guidelines whereby prime farmland equates to what is designated the “best and most versatile” (BMV) land [56]. In 1984 the USDA Natural Resources Conservation Service (NRCS) developed its own system that is oriented towards planning legislation, *Land Evaluation and Site Assessment* (LESA) [57].

The postwar era is also notable for the creation of a number of global and regional institutions such as the FAO, the International Union for Conservation of Nature (IUCN), and the European Union (EU), all of which have been instrumental in relation to land and soil governance. Three years after its creation in 1945, the FAO embarked on the world’s first formal attempt by an international institution to address soil conservation as a global issue, initially as a published study which relied heavily on data from the US and China [58], and subsequently in the form of a conference held in Florence in 1948 [59]. Underpinning such initiatives, especially in the wake of the US Dust Bowl, was a growing awareness of the economic impact of soil degradation, not just to individual land users, but to society at large, with far-sighted hints at the future valuation of ecosystem services and implications for government policy [60]. Similar analyses followed in the ensuing decades [61, 62], and soil law continued to evolve, with the US taking the lead, particularly with respect to soil conservation [63].

In 1949 the French botanist Aubreville coined the term desertification to describe the transformation of productive agricultural land in arid or semiarid areas to a desert-like, uncultivable state which has lost ecological viability due to a combination of climatic factors and human activity [64]. By 1958 the Chinese government acknowledged the threat that desertification posed to the wellbeing of nearly 200 million people, and have initiated afforestation programmes since 1978 in response [65]. While desertification is a major cause of land degradation in so-called drylands, it became mired in controversy because of a misunderstanding both of its meaning (i.e. a conflation with the natural process of desert formation and alleged expansion) and of its root causes. Nevertheless, the concept of desertification became and continues

to be a significant driver of sustainable land management (SLM) initiatives designed to tackle land degradation [66], where other approaches had faltered [67] – see “Soil degradation” below.

The conflicted role of agriculture

To highlight one salient point that was a sign of things to come, the UK postwar agricultural policy came in for subsequent criticism for virtually excluding farmers from planning regulations as if in recognition of their role in protecting the rural landscape, of which they were an essential component, they were exempt from harming it by default [52, 68]. In the decades that followed WWII geopolitics stabilised and agricultural productivity soared due to a range of technological innovations including mechanisation, agrochemicals and biotechnology [43], which from the 1960s reached the low income economies in the form of the Green Revolution [69]. This period represents a radical reframing of society’s attitudes towards the environment and agriculture [70]. In some parts of the world, at least, food security was no longer regarded as a serious threat, whereas agriculture itself was increasingly perceived as the primary threat to the environment, and hence a mixed blessing. Both the expansion and intensification of agriculture led to unprecedented wildlife habitat loss, encompassing deforestation, wetland drainage, conversion of grassland to arable and hedgerow clearance. Added to this were many other environmental impacts associated with industrial agriculture, including pollution, soil degradation and fuel-intensive inefficiency [71-73].

The publication of *Silent Spring* by Carson [74] led to a major international turning point in changing attitudes and policies in response to the excessive use of pesticides and herbicides. In Europe in the 1970s and 1980s the Common Agricultural Policy (CAP) came to epitomise the ironic twin dilemma of taxpayers subsidising overproduction at the expense of environmental destruction [53, 68, 70]. With the emergence into the public domain of climate change science since the late 1980s [75] and growing evidence of the significant climate-forcing influence of GHG emissions resulting directly and indirectly from farming [76], alongside the degradation and destruction of carbon reservoirs, and the ongoing loss of biodiversity [77], agriculture finds itself more implicated than ever as a cause of environmental crisis rather than its victim.

It could be argued that the cumulative effect of this gradual fall from grace has been to diminish the value of arable land in the public perception, by association. More specifically, any potential value of such land tends to be eclipsed by its current condition, for example its reduced biodiversity and highly disturbed and often degraded soil. Meanwhile, as an academic discipline that straddled geology and biology, but on the periphery of both, soil science found itself virtually a subdiscipline of agriculture at least halfway into the 20th century [19]. Hence soil, by association and also because largely hidden, rarely evokes the kind of intrinsic concern that attaches to wildlife or landscape. This is a topic that will be revisited in subsequent sections.

The relationship between soil and land

It is not pedantic to clarify that soil and land are not the same thing, but in the context of governance they are inextricably linked. Indeed, many languages confuse the two in common parlance. For example, most French people (and dictionaries) would offer the word *terre* (land) as a translation of the English word soil, whereas French soil scientists are clear that the correct word is *sol*, commonly regarded as the French word for ground or floor. Weigelt *et al.* [78] stress the distinction between soil and land, but also identify a lacuna in the literature to address the critical importance of governing them together to achieve sustainability. It is also axiomatic that an absence of land rights goes hand-in-hand with poor soil management and land degradation [79-82]

From a legal perspective the concept of land is normally operative in matters of ownership and boundaries, and also in terms of spatial and territorial planning. Soil tends to enter the legal fray when a portion of it is transformed, harmed or threatened by a specific activity, which might be performed by the owner of the land containing the soil, or by another party. Recognition of this distinction in itself signals two very different aspects of soil law. The former perspective would generally be categorised as property rights, whereby the injured party is primarily the landowner, rather than the soil. The latter perspective offers protection to soil as a legal entity in its own right, even from its “owner”, for the common good.

Where the blurring of land and soil becomes problematic is in instances of proposed changes which are framed only in relation to land despite having significant impact on the soil within, or adjacent to, that land. This opacity can apply to legislation specific to agriculture, the environment or planning, and as a result salient aspects of land use conversion may fall between the gaps among all three. For example, the creators of agricultural laws may feel that their remit, as far as soil protection and conservation is concerned, extends only to the direct impact of farming on soil, while the creators of environmental laws may feel that their remit for soil protection and conservation extends only to an unspecified component within a larger ecosystem or protected landscape.

This then leaves rural land use conversion at the mercy of the creators and implementers of planning laws, who may feel that soil *per se* is outside their jurisdiction, despite the soil-related implications of the land use changes that their legislation may encompass. There is in effect a blind spot whereby the legal instruments with the greatest power to transform soil, sometimes irreversibly, are often framed and worded with little or no reference to soil. Raising this in a meeting with government soil scientists caused heads to nod in agreement and perusing comparative laws reveals that this is a recurring theme throughout the world [83]. Furthermore, the modern image of agriculture as an industry that pollutes soil and threatens ecosystems, and with a long history of consuming forests and other natural landscapes, risks devaluing the public concept of

agricultural land, as if the sealing of such land would not, in and of itself, result in a loss of environmental benefits.

Anthropogenic threats to soil and land

Natural events have routinely transformed and even devastated bodies of soil for eons, but such events are part of soil-forming processes, to which ecosystems adapt. Humans have learnt to transform soil to their advantage but the overall impact of human society on soil, both intended and unintended, has been profound. Short-term gains have often been at the expense of long-term harms. The picture is further complicated by the fact that many of these human impacts manifest themselves not as discrete effects but by exacerbating and unbalancing natural processes, such as soil erosion, salinization or flooding.

Broadly speaking, soil or land is typically vulnerable to three kinds of anthropogenic impact, set out in Table 1.

Table 1: Types of anthropogenic threat to soil or land

1. Soil degradation (here meaning any harm done to soil, other than ‘soil sealing’)
2. Conversion of natural ecosystems or other semi-natural and uncultivated land to another form of land use, such as agriculture, forestry, urbanisation, or industry, including mining or energy infrastructure (conversion is also known as ‘land take’)
3. Conversion of farmland to urban or industrial use ² [84] (‘agricultural land take’)

1. Soil degradation

The most extreme anthropogenic impact to soil is soil sealing, which is strongly related to urbanisation, but not equivalent in meaning. Soil sealing, the permanent covering of soil, for example by concrete or tarmac, occurs outside urban areas, for example on roads or farm infrastructure, while conversely urban zones include much land that is not sealed, such as gardens, parks and playing fields. Nevertheless, urbanisation inevitably goes hand-in-hand with soil sealing, which is a drastic change that is effectively permanent and overwhelmingly negative with regard to SES [85-87].

In this context, degradation refers to any harm done to soil other than sealing (which is encompassed within categories 2 and 3), and which is therefore potentially ameliorated by remedial measures. Soil degradation

² The conversion of farmland to forestry or natural regeneration is not normally regarded as a “threat”, though there are scenarios where this may be the case with questions arising, for example, in relation to afforestation of peatlands, or replacement of biodiverse agroforestry systems with plantation monoculture.

may take the form of acute problems such as erosion, contamination, salinization, acidification, sodification and compaction or other forms of structural collapse. The term soil degradation has also been widely used to describe gradual and chronic loss of productivity, typically the result of over-exploitation and inadequate management, and invariably bound up with organic C depletion, reduction in soil biodiversity and low nutrient status. While acute soil degradation is often the result of breaches of regulations, chronic soil degradation tends to be associated with rural poverty, a less conspicuous but significant factor in the recent Indian farm protests [88], and generally requires supportive rather than punitive legislation. These problems can be complex and interrelated, and sometimes partly the result of, or exacerbated by, natural causes or intrinsic to certain soil types.

Soil degradation is a huge global issue, sometimes due to local legacies of malpractice, with unintended consequences. In the worst cases large tracts of land have been abandoned or declared unfit for food production with severe economic impacts. Yet this phenomenon and the wider concept of land degradation have been claimed, at times, to have been subject to “gross exaggeration and hyperbole” [67]. While stressing the severity of land degradation and its interlinkages with other environmental problems such as climate change and biodiversity loss, Gisladottir and Stocking [67] are equally assiduous in criticising several institutional studies that they suggest relied on subjective methods or unsound extrapolations. In the case of soil erosion, for example, a common mistake was to base regional estimates on plot experiments despite the fact that most “lost” soil will be deposited elsewhere. Veracity aside, the authors point out that far from marshalling support, such errors have led ultimately to scepticism and underfunding.

In the first study of its kind, UNEP commissioned the Global Assessment of Soil Degradation (GLASOD) which was conducted by the International Soil Reference and Information Centre (ISRIC). GLASOD addressed the period from World War II to 1990, relying on local expert opinion rather than direct measurement, and presented its results in areas of land as opposed to the less meaningful tonnes of soil eroded employed by many earlier studies [89]. Soil degradation was defined as: “...a process which lowers the current and/or future capacity of the soils to produce goods or services”. The results were broken down by region, types of land use (arable, pasture or woodland) and categories of degradation. In summary 15% of all land worldwide (almost 2 billion hectares) was classed as degraded, equating to 23% of inhabited land (14% seriously). Soil erosion was by far the most widespread form of soil degradation, affecting 83% of degraded land.

Approximately 20 years later ISRIC scientists reviewed GLASOD and used satellite imagery to measure an overlapping and subsequently ongoing period (1981-2003) of land degradation, which it defined as: “...a long-term decline in ecosystem function and productivity”. This was the first study of its kind to use normalized difference vegetation index (NDVI) as a proxy for changes in net primary productivity (NPP), with appropriate adjustments for climatic variability and land-use change [90]. Without entirely rejecting the

GLASOD results, which they declared unreliable and unreproducible, the authors detected a further 24% of the total land area that was degrading *mostly in addition to* GLASOD's 15% already degraded, i.e. a cumulative process. Although not explicitly stated by the authors, this would imply that at least a third of the world's land appeared to be degraded to some extent by 2003³. Additional key findings were that land degradation was not primarily a problem of drylands, as had long been proposed, but was in fact more widespread in humid areas, and that it was a worldwide phenomenon, albeit with distinct hotspots. However, it is important to bear in mind that NPP can sometimes be increased in the short-term by intensive land use which degrades the land [91].

In a 2015 follow-up study, including some of the same authors and updated to include data up to 2011, the global estimate of degrading land was revised down slightly to 22%, with 14% of land showing some improvement [92]. Also in 2015 FAO/ITPS⁴ claimed that 33% of land is degraded and that that figure could reach 90% by 2050 [93]. A 2016 study based on similar methods to the ISRIC analyses produced a global figure of 30% degraded land [94]. An analysis of terrestrial NDVI in 2017 revised the total down further still to 23% [95]. The 2017 UNCCD Global Land Outlook report states: "...over the last two decades, approximately 20 per cent of the Earth's vegetated surface shows persistent declining trends in productivity." [96]. Drawing on a wide range of studies, the 2018 IPBES⁵ *Assessment Report on Land Degradation and Restoration* concludes that 75% of the Earth's land surface is either transformed or degraded to some extent by human activity [97]. According to IPBES land degradation impairs 3.2 billion livelihoods, costing 10% of the annual global gross product, whereas the benefits of restoration are typically 10 times higher than the costs,

While global land degradation can appear overwhelming, awareness of soil degradation in the narrower sense is growing and attitudes are changing, and this subtle shift may be key to reversing the trend. Most countries now have some form of regulation addressing soil degradation, typically embedded within their agricultural legislation, and in a fair number of these countries the law is strictly enforced. In many cases, soil policies go beyond prohibitive and punitive laws dealing with acute soil degradation, and incorporate support and incentives to foster improved long-term soil health and protection [98].

³ The authors made no attempt to merge the data because of the contrasting methods of the separate studies and the fact that the GLASOD data was unverifiable, but one of the authors opines that: "It is not unreasonable to judge that all land now under anything less than natural climax vegetation is degraded in terms of biodiversity and stored carbon and nitrogen," (perhaps 66% of the world's land surface and rising) (David Dent, personal communication.)

⁴ The Intergovernmental Technical Panel on Soils (ITPS), made up of 27 soil experts representing all the regions of the world, was established at the first Plenary Assembly of the Global Soil Partnership held at FAO Headquarters in 2013.

⁵ The Intergovernmental Science-Policy Platform on Biodiversity and Ecosystem Services

Compared to land use conversion (see below), soil degradation, even though it may be the result of deliberate law-breaking, is generally recognised as an unintended and unwelcome consequence, so there are fewer obvious social pressures to oppose such laws and the global trend has been for greater governance. However, the cost of remediation can be prohibitive, whether for those held responsible or those who ultimately pay the bill, for example taxpayers, so circumvention is an ever-present risk [99], even though evidence shows the virtual spiral of paybacks possible from rehabilitation and restoration [97]. China stands out as the country probably most afflicted by soil degradation, of every kind, but by contamination in particular, on a scale that appears irreparable and unaffordable, requiring more than the world's entire wealth to remediate, according to *The Economist* [100]. Nevertheless, no country seems more acutely aware of this or determined to remediate it than China itself [101, 102].

2. *Conversion of natural ecosystems*

Natural ecosystems or semi-natural landscapes, that is land that is unoccupied or sparsely occupied, is uncultivated and largely unmanaged (at least according to currently prevailing notions of cultivation and management, [103]), are ubiquitously valued for a variety of reasons: their aesthetic characteristics and recreational potential; their educational and scientific worth; and, with increasing urgency, their vital ecosystem service benefits, and importance for many Indigenous communities. Aside from unvegetated and largely uninhabitable areas, such as mountains or deserts, this category of land includes forest, wetland, native grassland (e.g. prairie or steppe) and heath-/moor-/peatland. Globally the primary threat to natural and semi natural lands is from agriculture, responsible for 80% of deforestation according to WWF [7], but the full picture is nuanced and dogged by claims of both overestimation and underestimation [104].

Furthermore, the dynamics of forest change are complicated by temporary deforestation and instances of forest gain, for instance via afforestation, reforestation or regrowth on abandoned farmland. A recent study [105] which used satellite imagery to distinguish various categories of forest disturbance, estimated that between 2001 and 2015, of a global total tree cover loss of 314 Mha, only 27% (5 Mha each year) was permanent land use change for commodity production, i.e. large-scale agriculture, mining and energy infrastructure. Urbanization accounted for less than 1%. Impermanent changes included forestry (26%), shifting agriculture (24%), and wildfire (23%). Regional contrasts were stark with permanent deforestation overwhelmingly accounting for most of the tree loss that occurred in Latin America and Southeast Asia (> 60%), while Latin America also had the world's largest total loss. Characteristic drivers of change are oil palm plantations in Southeast Asia and soybean and ranching in Latin America [105]. Indonesia and Malaysia stand out as areas of increased deforestation while the rate of loss had declined markedly in Brazil, but since the change of regime in Brazil in 2018 forest destruction is reported to have increased again dramatically [106]. Most other regions have relatively low rates of permanent deforestation, with most

temporary disturbance associated with managed forestry and, in North America, Russia and Australia, wildfire. A handful of countries have even achieved a net forest gain.

Several authors have long argued that the abundance of available land in sub-Saharan Africa (SSA) is a myth in much of the region and even where natural forest exists near settlements, there are significant disincentives for farmers to encroach on such land [107]. Clearing forest is difficult and expensive for small farmers, and often occupies poor soil. The Curtis study confirms that most forest disturbance in SSA is due to impermanent shifting agriculture, but the alarm has been raised concerning the impact of rising and often uncontrolled urbanization on biodiversity in Africa [108].

Natural ecosystems tend to be well protected in the higher income countries, but historically this has not so much been due to wisdom or sound governance, but for more mercantile reasons. Many wild and majestic landscapes have survived largely because they lack the kind of natural resources that facilitate intensive land use and often present severe obstacles to development. For the most part wherever wilderness occupied flat land with productive soil, it was converted long ago. The stark reality is that very little such land remains, and some parts of the tropics have reached or are close to this point [109]. In 2009 Rockström *et al.* [110] suggested we are approaching the limits of the planet's cultivable land. This makes the preservation of natural and semi-natural land across the globe, all the more urgent, especially given its far-reaching environmental benefits, such as those relating to C storage, biodiversity and soil and water conservation.

The accelerating consumption of natural and semi-natural land by agriculture is not an inevitable consequence of population pressure and progress in the Global South, but it is usually an inevitable consequence of uncontrolled development. Forest loss often follows farmland loss, which can be avoided without penalising the rural poor (see below). A combination of agricultural support and incentives, and rational territorial planning can raise farm incomes, improve national food security and obviate loss and expansion of farmland [111-115]. Byerlee *et al.* [111] also observe that while market-driven intensification of agriculture (e.g. greater demand for oil palm or soybean) tends to consume more land, technology-driven intensification tends to save land, i.e. raises productivity without expansion. However, they stress that sustainable intensification is possible only in tandem with environmental protection and incentives.

Nevertheless, conversion of natural lands to agriculture is rarely as drastic or permanent as urbanisation, and many countries are increasingly rewarding landowners, land managers or farmers, for example via payments for ecosystem services (PES), to implement more environmentally friendly farming and land management practices [116]. Putting aside the question of how seriously each national government responds to the loss of natural ecosystems, or governs it, that issue is relatively unambiguous and virtually every country on Earth, theoretically at least, protects such land with environmental laws and policies. Furthermore, soil protection per se is rarely the driving force behind such legislation, which tends to be framed in terms of ecology,

biodiversity or watershed protection. Having said that, some of the earliest environmental laws in the modern era were primarily a response to soil erosion caused by deforestation [35], which is also still frequently cited in such legislation.

3. *Conversion of farmland to urban or industrial use*

The vast majority of land consumed by urban and industrial expansion throughout the world is agricultural land, and this type of threat, which encompasses much soil sealing, is arguably the least reversible and the most profound of these three categories. Furthermore, for sound socioeconomic reasons, urban settlement has historically developed in close proximity to, and often surrounded by, prime farmland. Hence it is frequently the best quality land of regions that is most vulnerable to conversion [85, 117-119]. Peri-urban farmland is particularly attractive to developers because unless it is subject to strict green belt or zoning laws, it lacks the environmental protections of natural ecosystems⁶, is usually cheaper to develop than brownfield sites [120] and tends to be conveniently situated in terms of existing facilities and infrastructure. Furthermore, a study of 25 EU countries found that the land most at risk was that slightly further out from city boundaries which was typically on more fertile soil than that closer to the city [121], although interestingly the influence of CAP subsidies reduced the rate of land take in general.

The mere potential for future urban development, sometimes even without planning permission, typically increases the value of farmland by, for example fivefold in Korea [122], sixfold in Morocco [123], nearly tenfold in New Zealand [124], and even by orders of magnitude more than this: 100-fold in Ghana over ten years [125] and similar average multipliers for land earmarked for development in Britain and Japan [126]. These market distortions create enormous commercial pressure for landowners to sell, especially in lower income countries where agricultural livelihoods may be precarious and lack government support [127], but do not necessarily leave poorer farmers with any viable long-term benefits. With the exception of a few who may be able to exploit new urban markets, most farmers will use this temporary windfall to pay off debts and be forced to become landless farmers or seek other occupations [128]. The prospects of re-investing in cheaper land to cultivate are greatly diminished by the fragmentation of holdings [129] and the creeping outward shift of the agricultural zone onto lower quality land [128].

The view that urbanisation represents economic progress [130] has led many administrators and politicians, especially in lower income countries, to embrace urban expansion policies with enthusiasm [131], but this line of reasoning is embedded in a pre-millennial paradigm in which human betterment is predicated on

⁶ Even if natural ecosystems were an option, they often present many other obstacles such as remote location, uneven terrain and geological hazards, the need to clear vegetation or extensively drain, and an almost complete lack of existing infrastructure.

economic growth without setting environmental limits on such growth. This largely market-driven worldview has been strongly criticised for at least 50 years [132, 133] and arguably much longer [134], as well as being relentlessly underscored by what is becoming known as the Anthropocene crisis [135]. At the local level there are serious concerns, even within government [136], that the wider implications of farmland loss are increasing food prices and imports, escalating rural poverty, land degradation and conflict [127, 128, 136-139], as well as the loss of critical ecosystem services [85, 136, 140].

However, it is worth addressing the arguments of authors who advocate farmland conversion, or are at least sanguine on the issue. Visser takes this argument to a purely economic extreme of treating natural resources as tradeable commodities or assets and even suggests that land markets would benefit from greater scarcity of farmland [141]. In 2010 Satterthwaite *et al.* [130] presented an ostensibly even-handed review of the positive and negative impacts of urban expansion, but ultimately argued that the world had abundant arable land and that urban populations were no longer dependent on their own agricultural hinterland because in a modern globalised world they can source food from far and wide. However, all else aside, this thesis presupposes reliable food supplies and reliable trading partners, missing the point that geopolitical instability and climate change are increasingly threatening global food security, and this applies even more so in the decade since that article was published [142], notwithstanding the additional risks associated with a global pandemic.

It might also be tempting to argue, in defence of conversion, that much farmland, often government-owned, has been abandoned or under-utilised for various reasons, such as de-population of rural areas or, as in the case of the former Soviet Union, incomplete land reform [143], and that this land could be recultivated. There may be limited capacity for rehabilitated farmland to compensate for conversion elsewhere [144], but this raises a number of issues. A common reason for abandoning farmland is the difficulty in extracting an adequate livelihood from that land, often because it is degraded [145] or marginal land [143]. Such land will tend to be intrinsically less productive so, even if farmers have the means to restore it, and can be persuaded to do so, a greater area might need to be cultivated to achieve requisite returns. Where productive land has been abandoned for socioeconomic reasons, such as a lack of infrastructure in remote areas, and steps are taken to remedy this, recultivation is feasible [146, 147].

However, abandoned farmland presents another important global narrative, for it is swathes of such land, in Russia, US, China, Australia, Latin America and elsewhere, that have inadvertently facilitated a global pattern of reforestation and rewilding [144] that goes some way towards buffering the effects of deforestation elsewhere. Recultivation of such land would be likely to reduce biodiversity and C storage [144, 148], for it is hard to imagine farming systems that would achieve comparable environmental benefits on the same area of soil, other than carefully managed agroforestry systems or some other form of continuous perennial cover [149]. However, degraded land often has higher than average C sequestration

potential so, while such land may be challenging for agricultural rehabilitation, it may be highly suitable for ecological restoration [150], although the slow process of natural regeneration may not necessarily be the most effective way to achieve this [151, 152].

There are also vociferous advocates of farmland conversion in high-income countries like the UK, for instance in relation to the London Metropolitan Green Belt (MGB). In some cases these are primarily socioeconomic arguments with respect to a scarcity of land for housing. [153]. However, as long ago as 1977 it was argued that the MGB policy was rigid and anachronistic in protecting all farmland regardless of its environmental value, while constraining suburban gardens which harboured more biodiversity [68]. This message, which has been echoed over the years by others [153-155], has some merit, but reprises the trope of industrial agriculture as the enemy of nature, and as such is contentious. Residential land, by definition, undergoes much sealing and topsoil removal, while the biodiversity of its gardens, though potentially of great value, is entirely at the mercy of the residents. Farmland, on the other hand, remains largely unsealed and, given appropriate policy incentives, such as PES, can become more biodiverse and provide greater ecosystem services, as has been a gradual trend in the UK since the late 1980s [156], and looks to continue with the proposed new Environmental Land Management scheme (ELMS) [157]. However, alongside this trend in the UK, and related to it, has been a gradual weakening of the protection afforded to BMV land [156, 158].

Where such authors have a much stronger case is in criticising the binary distinction between green belt land, where all farmland is protected regardless of its agricultural quality, and non-green belt rural land where prime farmland should be protected but is often developed [153]. Furthermore, Amati and Taylor [154] point out that city workers who cannot afford either city or green belt house prices are forced to live further out and commute in, increasing their C footprint, as well as exacerbating inequality. Brownfield sites are finite and if some land must be sacrificed for housing one can see why some argue that it could be preferable to release some low grade land within the green belt zone rather than build on BMV land beyond the MGB.

The MGB is one of the oldest green belt zones in the world and now finds itself constraining one of Europe's largest cities. With its protected open space continually under strain, primarily from pressure for housing and transit development, some of which does get planning permission in spite of the strict regulations, much more leapfrogs onto greenfield land beyond. Similar but younger zones have yet to experience such pressures, but in the case of the Ontario Greenbelt, for example, this is partly due to an approach which is arguably more integrated and enlightened, incorporating green infrastructure, principles of environmental and economic sustainability, greater public participation and underpinned by a legal framework that is centrally owned and controlled by the provincial government, yet both robust and flexible [47, 154, 159]. To take the case in point, farmland preservation in Ontario is not simply a matter of locally

ad hoc prohibition in the teeth of fierce opposition from planners and developers, but is integrated into a holistic policy that includes agricultural support, local food marketing, employment opportunities, recreation, tourism and ecosystem services in a regional context [154, 159-161].

For the reasons already stated, agricultural land take is rarely covered by either environmental or agricultural legislation. Evidence for this exists in FAOLEX [83]. One type of exception to this is where farmland is protected for its above-ground biodiversity value, e.g. high natural value (HNV) [121], but this is not the same thing as protecting land for its intrinsic SES value or, for that matter, its soil biodiversity. The last resort for farmland preservation, then, if it occurs at all, is usually some form of spatial or territorial planning. Still, the primary motive for such planning is typically urban expansion driven by population and commercial pressure, rather than rural protection [162, 163]. Not surprisingly, therefore, there is a distinct lack of robust legal instruments to prohibit or restrict agricultural land take and the approach taken varies considerably between and within countries, producing policies which are frequently complex and sometimes ambiguous or conflicting [53, 124, 136, 163-168]. Added to this is the prevalence of “informal” governance of planning regulations in many low- and middle -income countries [166, 169, 170].

Setting aside the national and wider global consequences, the greatest losers especially in the Global South tend to be the rural poor who become less able to afford to buy or lease land [139], and the wider population as food imports and prices rise due to a shrinking agricultural base [112]. Much of the land consumed by urban sprawl is common land or, in Indian parlance, common property resources (CPRs), on which many rural communities depend heavily [136]. Such communities also suffer disproportionately from the less appreciated degradation of SES that results from soil sealing, such as problems of sanitation, flooding or pollution [171, 172]. The greatest “winners” are typically speculative developers and any officials or agents who may benefit from their transactions [173], but the cumulative negative impacts will be felt by the whole community. Furthermore, the most strident whistle-blowers to these encroachments are not distant activists, but witnesses on the ground, from local academics [127, 139, 174] to journalists and blighted farmers taking their grievances to the courts, at least those who are able to do so [173, 175]. Social inequalities drive many to facilitate the very land use changes from which they gain the least, for example selling land simply to survive, and then converting natural ecosystems to replace what they have lost. There is much evidence that appropriate government support for farmers reduces the sale of farmland for non-agricultural development, as well as enhancing food security and biodiversity [121, 162, 176]. It is therefore imperative that any attempt to prohibit or restrict agricultural land conversion, must be accompanied by economic support and viable alternatives. No amount of legislation or governance will succeed without this.

While all three types of anthropogenic threats to land and soil listed in Table 1 above endanger ecosystem services, from a law and policy point of view, compared to soil degradation and natural ecosystem protection, land take is often “out of sight and out of mind”. For example, the Thompson Reuters Practical

Law website (accessed September 2020) includes a very detailed global section summarizing the laws of 72 countries in a highly readable Q&A format on, for example, agricultural real estate or environmental impact. Yet none of the questions specifically address the prohibition or restriction of agricultural land take or preservation other than that of foreign ownership, i.e. the “loss” of land as a national asset.

4. *Cross-cutting issues*

A few academics argue that land conversion or degradation may be justified in some cases, on the basis that the economic benefits may outweigh the environmental costs [177], but despite the best efforts of environmental economists, such costs remain intangible [178], and the impacts often profound. Furthermore, as currently framed, economic benefits are not necessarily sustainable and are ultimately dependent on ecosystem services that economists have traditionally treated as “free” and, more to the point, as inexhaustible and unconstrained, for example, by planetary boundaries [110]. Economic benefits also accrue unevenly and not even necessarily in the national interest. The escalating potential profits from development can also foster an “unholy alliance” between the public and private sector that rewards elaborate circumvention of the law [162, 179] or outright infringements [80, 95-97, 167, 174]. However, it is also important to appreciate that what may appear to outside observers to be infringements may, in some cases, simply be the result of longstanding traditions of informal governance or customary tenure [173, 180], or simply a lack of institutional capacity [130]. In other situations the pressure to convert land may be political, for example the need for an administration to acquire land to meet housing or other developmental targets. Mining, energy infrastructure and other forms of industrial land use represent a very small proportion of land loss or degradation overall [181], although this is increasing [96] and cannot go unmentioned because of their extreme impact in some locations. In western Ghana, for example, gold and diamond mining has had devastating effects on rural communities and natural ecosystems [182, 183]. This activity includes both corporate mechanised extraction and illegal and artisanal hand-digging, as a form of low-input mining but also to provide material for construction. These impacts include all three types of anthropogenic threats to soil listed above, going far beyond the loss of land, and including local water contamination and depletion, illegal logging, extensive soil and subsoil removal, and sometimes violent land disputes. The highly lucrative and largely unregulated context in which these enterprises operate tolerates or even encourages infringements of the law, even by corporations operating legal concessions [183].

The jurisprudence⁷ of soil and land

⁷ Jurisprudence here refers to both legal theory and legal systems..

The implementation of soil legislation

There are essentially two components to legally protecting or preserving soil resources: (i) applying a method of evaluating and prioritising areas or bodies of soil and (ii) implementing the requisite (and effective) legal instruments and governance structures. At the time of writing neither of these components is entirely absent from the world at large, but both exist only within a fragmentary and, in some cases, theoretical patchwork of initiatives.

So, before getting to grips with the thorny issues of establishing which forms of governance structures or legal instruments have been or might be most applicable to soils, one must consider what criteria to apply. This is essential whether focussing primarily on specified spatial areas of land or on the functional aspects of soils in relation to their current status or use. However, preceding even all of this is a minefield of ambiguous terminology to navigate.

Semantics and legal terminology

When authors or legal documents and policies refer to the way in which soil is affected by those managing the land it occupies, and hence the possible harms it may encounter, such as erosion, pollution, contamination, salinization, acidification, compaction and so forth, the word “protection” is the term most widely used. Many if not most countries have some form of explicit or implicit soil protection policy in place, but this is more often than not subsumed within other legislative and policy domains, for example dealing with the environment, agriculture or spatial planning. Soil legislation is commonly sub-divided further into categories of protection, reflecting the severity of particular problems within a given country or the gravity attached to the problem by its authorities. For whatever reason, certain soil laws are almost always framed in negative harmful terms and others in positive or remedial terms. Hence there are soil pollution or contamination laws, but hardly ever soil erosion laws. Instead there are soil conservation laws or occasionally, in a similar vein, soil improvement laws.

Harder to pin down is the terminology applied to the conversion (effectively the loss) of agricultural or other undeveloped rural land to non-agricultural use which typically incorporates substantial additions of infrastructure (e.g. housing). In this domain several terms are used, some in the negative or destructive sense, and others in the positive or protective sense. However, it could be argued that the term land conversion, though technically describing the act of change or loss, is neutral, because unlike the unintended consequences of the harms done to soil, this refers to a deliberate act with an intended outcome and various beneficiaries.

The many approximate synonyms for rural land conversion with a more negative connotation include urbanisation, urban spread/sprawl/expansion/encroachment, land take/consumption/loss, long-term land

cover change and (in France) artificialisation [184]. Other phrases commonly used in connection with rural land conversion are land competition and fragmentation, of farmland in particular, because this is one of the ways in which urbanisation becomes self-perpetuating [185]. Soil sealing (defined previously, see *Soil Degradation*) is a phrase widely used, often in conjunction with land take, because they typically occur together, but the meanings are distinct. While land take refers to a change of use which, by definition, usually entails some loss of land resource from its former use, sealing constitutes a much more permanent and intrinsic alteration of the land surface, typically with more far-reaching consequences for soil functions, especially drainage, and potentially severe effects on biodiversity. Land take almost always involves some degree of sealing, and where it does, that constitutes a double impact – the spatial loss of land resource and the additional degradation of the ecosystem services that that resource provides in the round. However, sealing also occurs independently of more general land take, even in highly protected areas, for instance in the form of roads [86].

On the positive land-saving side of the negative land taking lexical coin, the term predominantly used in North America, and in many countries, is land preservation [186, 187], but the terms land conservation or land protection are used almost interchangeably with it in the US [188, 189] and throughout the world. In most cases these phrases are applied to specific bounded areas, at various scales, which could mean a protected zone or alternatively a cluster of holdings, a single estate or even one field. The expressions “no net land take” or “zero land take”, which have appeared in EU documents in recent years [120, 190], are similar in principle, but refer to overall quotas or targets, with the additional challenges relating to relevant spatial scale and overall quantification. In contrast ‘land sparing’ is used in juxtaposition to ‘land sharing’, referring to the concept of safeguarding biodiversity, either by sparing natural lands from agricultural use or by incentivising farmers to support more biodiversity [191].

Globally other words or phrases are employed to mean approximately the same thing as land preservation but some can have other connotations. Land conservation has historically been most strongly associated with nature conservation, in the sense of land that is not recognised as developed or cultivated and has some form of protected status. Such land is often threatened by agricultural incursion at least as much as by urban sprawl. The meaning is also not to be confused with soil conservation, a term normally used professionally and more widely to refer specifically to the prevention of erosion with respect to land management.

References to land protection are often synonymous with land preservation, but in some examples of legislation it is intended to mean soil protection, that is *in-situ* safeguarding. Confusingly, however, soil protection is even occasionally intended to mean land preservation. This semantic problem may sometimes be one of translation because the full texts of many laws appear only in their original language and script, so the average researcher must rely on translated summaries or just keyword lists [83]. Occasionally countries have all-embracing “soil protection” policies which include the concept of spatial land preservation in

context [192, 193] or conversely all-embracing “land preservation (or protection)” policies which include the concept of soil protection [11, 193]. In some cases this blurring seems to be deliberate, such as when land preservation is promoted as a means of protecting SES [11, 86]. Pragmatically soil scientists sometimes need to accommodate themselves to the language of policy and spatial land use planning in order to engage with the process.

Putting a value on soil

At the root of our current environmental crisis is that humanity has traditionally treated abundant natural resources as free goods and services. Economists and accountants are no exception: cost-benefit analyses include tradeable assets and products, but routinely exclude the ever-present prerequisites of life by which we are surrounded. Furthermore, traditional cost-benefit analysis (CBA) is short-termism by definition because it usually does not even consider impacts outside a 25-year window [194]. When a tonne of soil or water is traded, the allocated price is normally derived from the aggregate cost of acquisition, processing and delivery. No attempt is made to assess the intrinsic value of the substance which therefore would appear to have no value by default. This applies to soil as an intrinsic good, but also to some extent to its economic potential, i.e. the land that contains it will have a market value and that value will by default be related to the land’s productivity, yet unquantified and uncoded SES have far-reaching economic implications.

Even in a society without monetary currency it would be relatively straightforward to calculate the economic value of a given volume of soil, in conjunction with the land it occupies, purely in terms of its capacity for life-sustaining primary production; ask any farmer. Something so eternal and fundamental as soil, which has even been revered as a deity in its own right [195], might be expected to have acquired an aura of intrinsic value, but it was ostensibly its extrinsic value that made it a prize for which it was worth fighting, or even dying. So although we now have modern techniques to assess the productive capacity of land resources, a conceptual process of “following the money” has been at the heart of most farm or settlement emplacement decisions since the Neolithic.

For at least 80 years soil survey and land evaluation has been the traditional tool at our disposal for assessing the productive and economic potential of an area of land, alongside any limitations or hazards it may present. A range of soil properties is recorded alongside other local environmental data and, where appropriate, socioeconomic data. This data is combined to grade the capability or suitability of each parcel or zone of land according to specified use criteria. The modern terminology of multicriteria decision analysis (MCDA) is more a change of style than substance, to reflect the much greater reliance on information technology, especially GIS [196]. In the above approaches “criteria” is the operative word. Where these methodologies differ from analogous ad hoc historical techniques, is in their capacity to distil

decades of practical and scientific observation to apply much greater predictive accuracy and precision to the decision-making process [197].

Land evaluation though is not the same thing as land valuation, though they have always been interrelated and both are relevant to spatial planning. Land evaluation methodologies were never intended to calculate the true, i.e. total and holistic, economic value of an area of land or a volume of soil (which are themselves two very different things), but rather the capability or suitability of distinct parcels of land for specified forms of primary production, for example crops, pasture or forestry. The purpose is for land use decisions, albeit often with profound economic implications. The purpose of land valuation, however, is generally to set prices for landowners and taxes for governing authorities, and has always primarily been based on location, which admittedly is historically bound up with soil quality and land use, but with a myriad of other factors too.

In contrast, soil has always provided a variety of extrinsic benefits beyond primary production which were recognised and appreciated long before we referred to them as ecosystem services. To what extent our ancestors valued (or devalued) specific areas of soil or land for reasons other than primary production, or obvious physical location, is not always tangible, though it is clear that many Indigenous communities have developed deep understanding of, and spiritual connections to, the land and soil on which they depend [103]. The holistic value of soil is increasingly becoming more widely apparent. A wider all-embracing appreciation of soil and a few attempts to evaluate it in that context date from the 1960s [198]. This concept has gathered momentum and, more recently above all, with respect to C.

While it may seem too neat to earmark the last decade of the last millennium as a critical turning point in time, it seems inescapable that this period represented an important paradigm shift in our collective understanding of the role of soils. The 1990s shines out as a lightbulb moment when soil science and climate science came together as never before. Although the role of C in the soil has been studied for at least 150 years and its place in the C cycle appreciated for much of that time, it was only in the 1990s that a flurry of papers explicitly linked soil C sequestration to GHG emissions and climate change [199-201] which, after several decades of growing evidence, had started attracting worldwide media attention since 1988.

At the onset of the new millennium soil scientists have more than ever before started to deconstruct and articulate the critical importance of SES [202, 203]. Experts and advisors on soil law have responded accordingly, emphasising that the value and status of soil far exceeds its role in agriculture [204]. Taking this a step further, several groups of researchers, building on earlier attempts to evaluate ecosystem services [205] and characterising soil as a form of natural capital, have developed tentative frameworks and quantitative methods to enumerate and evaluate SES in the fullest sense [206-209], sometimes using case studies to apply monetary values [210]. To what extent this is entirely possible, feasible or even desirable, is

debatable [198], especially with regard to qualitative or ethical issues, but these kinds of approach may point the way forward to how policymakers and planners could apply more meaningful criteria and priorities to the governance of soil and land protection.

Categories of soil governance

Juerges et al. [79] provide a summary of the types of institution applied in the service of soil governance (i.e. regulatory, economic, and so on) and the levels at which these operate, from global to local. In this section we present a very different cross-cutting breakdown, based on issues and intended outcomes. One can identify three broad categories of governance instruments applied to preserve or protect soil or land, usually at a national or subnational level, each with various subcategories, set out in Table 2 and elaborated further below:

Table 2: Categories of governance instruments applied to preserve or protect soil or land

CATEGORY	DESCRIPTION
1	Regulations to prohibit (or guidance to discourage)
(a)	- Enforced by penalties
(b)	- Facilitated by financial support
2	Restrictions on development or change of use
(a)	- Enforced by zoning laws
(b)	- Enforced by laws (or encouraged by guidance) based on ad hoc site evaluation
(c)	- Enforced by public acquisition of land
(d)	- Facilitated by financial support
3	Generic incentives to preserve land or to enhance its ecosystem services value, i.e. for
(a)	- Taking land out of agricultural production
(b)	- Converting intensively farmed land via extensification
(c)	- Declining either to develop land or to intensify its use

The first category comprises regulations and laws to prohibit (or guidance to discourage) certain unauthorised actions by landowners which may degrade land, e.g. pollution, sealing, construction, stubble

burning, tree felling, or cultivation methods leading to soil erosion, acidification, salinization, compaction, and so on; whereby best practice may be either:

- a. Enforced by penalties for infringement or
- b. Facilitated by financial support, e.g. farming subsidies, PES or C credits

The second category comprises restrictions or guidance on development involving conversion of use, either to protect terrestrial natural resources (for the benefit of any or all of primary production, watershed management, biodiversity, landscape or cultural or historical value) or simply to retain a certain quota of agricultural or forested land within a state or designated region; whereby land conservation or preservation may be:

- c. Enforced by laws based on zoning, e.g. National Parks, nature reserves, green belt, urban growth areas, UNESCO World Heritage Site, UK Sites of Special Scientific Interest (SSSI) or heritage landscapes such as the UK Areas of Outstanding Natural Beauty (AONB);
- d. Enforced by more general laws (or encouraged by guidance) not specific to the protected areas described in 2(a), based on ad hoc site evaluation, such as an Environmental Impact Assessment (EIA), and thereby affording protection to sites of ecological or landscape value, or prime farmland (e.g. based on ALC or LESA criteria);⁸
- e. Enforced by public acquisition of land, such as land trusts;
- f. Facilitated by financial support, e.g. using LESA to obtain Purchase of Development Rights (PDR) conservation easements (US).

The third category encompasses generic incentives to preserve land or to enhance its ecosystem services value (or disincentives to develop, such as the removal of subsidies), in the form of:

- a. Taking land out of agricultural production, purely for the purpose of conservation, e.g. EU set-aside, US land retirement, rewilding;
- b. Converting an area of intensive agricultural production (arable cropping) to one of more extensive agricultural production (such as pasture, silvopasture or agroforestry) or forestry;
- c. Declining to develop land, or to intensify its use, in order to maintain ecological or ecosystem services value (such as woodland, meadow, prairie, wetland, moor or heath) that could otherwise be legally developed.

⁸ A legally binding example is the Indian Prohibition on Conversion of Agricultural Land for Non-Agricultural Use (No. 16 of 2010).

Overlapping categories

The above categories are presented as a useful framework for analysis of soil governance. There are of course many instances of overlap. For example, regimes that are not targeted specifically at soils can still fall within the general categories above. There are also numerous cases where soil governance instruments span a number of the mechanisms listed above. The examples below illustrate these points.

Environmental Impact Assessment (EIA) of projects, Strategic Environmental Assessment (SEA) of plans and programmes, and related types of assessment, may apply to some of these categories, especially categories 2(a) or 2(b), and can be helpful in drawing attention to threats to natural resources, but there can be limitations in their application to soils. In the US, for example, EIA is applied primarily for federally-funded projects [211] which is partly why the NRCS developed the LESA system for smaller projects targeted at farmland. In the UK EIA is more widely applicable, but generally only where part of the area intended for development is uncultivated or only partially cultivated and hence might not be invoked where only arable land is at risk [212]. EIA regulations vary slightly in Scotland, however, where the site being assessed may consist exclusively of farmland where it exceeds 200 ha. The Scottish EPA also provides specific guidance on application of SEA to soils [213].

The broad North American planning term of *Land Preservation* covers many of these categories with the main focus on zoning [186, 214]. Every state and city has its own subset or version of Land Preservation laws and regulations which are varied and complex. One example is a “conservation easement” whereby a landowner is bound by a covenant set by the government or some other organisation and in return receives an incentive, such as a tax rebate; this could fall under categories 1(b), 2(d), or 3(a-c).

Apple Valley City, Minnesota has adopted a Natural Resources Management Plan (NRMP) which essentially constitutes a 2(b) type mechanism via a permitting system, but incorporates elements of 1(a) because it could apply to a single aspect of a development. It differs from most local planning laws and regulations by applying the broad principle of environmental impact to every development [215]. The NRMP is summarised thus: “*This broad natural resource management ordinance attempts to 'protect, preserve, and enhance the natural resources and environment.' It forces the collection of data on the parcel in question and mandates low impact and best practices which preserve water quality, wetlands, trees, and reduce soil erosion among other issues.*”

There is an interesting example of environmental scientists trying to set a legal precedent in a 2(b) type scenario in New Zealand in 2011 using soil natural capital and ecosystem services arguments to prevent urban development on horticultural land. The defence lawyer (representing the developer) argued that the only measured ecosystem services of this soil was agriculture. The judge upheld the local authority decision

not to allow development but not because of the “holistic” ecosystem services defence, which he disregarded [216].

The modern governance of soil resources worldwide

While the piecemeal governance of soil use is almost as old as human history, formal supranational global oversight of soil resources is only approximately 50 years old. In terms of the natural environment, the early 1970s stand out as a period when national policies, international initiatives and academic analyses of global problems coalesced in an unprecedented step change in attitudes. 1970 saw the conception of Earth Day, initially in the US but later worldwide, and as a consequence the creation of the US Environmental Protection Agency, an institution which also became replicated worldwide. In 1972 the eclectic think tank, the Club of Rome, published the now iconic *Limits to Growth*, with its prediction of societal collapse in the 21st century, alongside frequent references to soil degradation [217]. This period also saw the emergence of the concept of knowledge co-production whereby multiple agencies or stakeholder groups, academic and non-academic, join forces to address complex multidisciplinary problems such as sustainable development [218].

Emergence of soil in modern international law and policy

The *African Convention on the Conservation of Nature and Natural Resources* (the “Algiers Convention”), which came into force in 1969, was the first multilateral continent-wide treaty devoted to natural resource protection, that included a reference to soil. The salient wording called for soil conservation and improvement, and land use planning to this end. There was also a stated desire to, “...ensure long-term productivity of the land,” or in today’s technical parlance, SLM. In practice, although binding, the Algiers Convention lacked the resources and institutional framework to be particularly effective, but was nevertheless initially regarded as a milestone in international environmental law [219]. The following year, in a preparatory document for the forthcoming 1972 UN Conference on the Human Environment (UNCHE) in Stockholm, Wilson and Matthews of the Massachusetts Institute of Technology (MIT) coined the term environmental services [220], which would later be renamed ecosystem services [221] or, a more recent proposed alternative, nature’s contribution to people (NCP) [97]. The UNCHE was effectively the first major international conference on environmental protection and sustainable development, and has been referred to by Boer *et al.* as marking the: “...first phase of international soil protection law” [222, 223].

At around the same time FAO was publishing several of its soils bulletins every year, including two landmark studies in 1971, on land degradation [224] and on legislative principles of soil conservation [225]. It is worth reprinting the first sentence of the latter in its entirety because of the way it succinctly reflects the prevailing zeitgeist: “*With the universal concern for environmental protection on the increase, more*

attention is being given to the introduction of adequate legislation and institutions for preventing or controlling soil degradation.” This document was a relatively concise and readable set of guidelines, with real-world examples, that any nation could use as a template for creating or improving its own legal framework for governing soil.

In 1972 the Council of Europe created and adopted the *European Soil Charter*, a concise and perhaps surprisingly holistic set of statements about soil and soil protection aspirations [226] which was non-binding, but regarded nevertheless as the first international legal instrument dedicated specifically to protecting soils [222, 227]. Ten years later, in 1981, FAO adopted the World Soil Charter [228] and in 1982 UNEP developed a global soils policy [229]. In the same year, the IUCN/UN World Charter for Nature included explicit reference to soil, emphasising the need to combat land degradation and to maintain or enhance productivity [230]. It was also during this period, in 1977, that the UN held its first Conference on Desertification (UNCOD), and produced its *Plan of Action to Combat Desertification* (PACD) [231]. However, despite the apparent paradigm shift that had occurred, and the many subsequent conferences and reports this generated, it would take several more years before binding international law addressing soils materialised in the next wave of developments.

Soil as a feature of common concern

In 1987 the UN commissioned the so-called Brundtland report, *Our Common Future*, which reinvigorated the debate, again stressing the severity of soil degradation, and included a summary of proposed legal principles for environmental protection and sustainable development [232]. In order to address these issues financially, in 1991 UN agencies and the World Bank created the Global Environment Facility (GEF), as a prerequisite for the 1992 UN Conference on Environment and Development (UNCED) or Rio Earth Summit [233], a milestone event. From this emerged the UN Commission on Sustainable Development (UNCSD) and three legally binding international treaties: the 1992 *UN Convention on Biological Diversity* (CBD), the 1992 *UN Framework Convention on Climate Change* (UNFCCC) and the 1994 *UN Convention to Combat Desertification* (UNCCD)⁹, which came into force in 1993, 1994 and 1996, respectively (the Rio Conventions) [234, 235]. There was appreciation of the interrelationships linking all three conventions to land degradation [236], but funding for the latter was mainly tied to the UNCCD which was always the weakest and poorest of the three due to donor scepticism, although the situation improved slightly after the *Johannesburg Declaration on Sustainable Development* in 2002 [67]. The UNCCD has been called the first and only “*legally binding global agreement directly dealing with the promotion of bio-productive land*”

⁹ The original full title of the convention is: ‘*United Nations Convention to Combat Desertification in Countries Experiencing Serious Drought and/or Desertification, Particularly in Africa*’.

[222], but it is also recognised that it is not an “adequate instrument for the protection and sustainable use of soil” because of its limited provisions and scope [237]. Although recognising desertification and drought as being problems of global dimension, and their adverse effects as being of urgent concern to the international community, UNCCD contains little in the way of substantive obligations, and formally applies only to “arid, semi-arid and dry sub-humid areas”. Nevertheless, UNCCD is gaining momentum as a focal point for efforts to address the global land degradation neutrality (LDN) target [231, 238].

The interconnected nature of different environmental problems, as well as their linkages to socioeconomics and SLM, was taken a step further at another key event, the 2012 UN Conference on Sustainable Development (Rio+20). The UNCCD brought the concept of zero net land degradation (ZNLN) for adoption at Rio+20, reflected in the outcome document, “The Future We Want”, paragraphs 205-209 [231] and this later became embedded into the 2015 SDGs as the global LDN target in SDG15.3 [231, 239]. Land degradation was finally being acknowledged as a global problem that was intrinsically linked with climate change, biodiversity loss and poverty. The concept of ZNLN/LDN was that every effort should be made either to prevent further degradation or, where this was not possible, to rehabilitate equivalent areas of degraded land elsewhere [239].

While the thrust of LDN is land conservation, the concept clearly envisages the possibility of offsetting. The concept of offsetting or a no net loss (NNL), and in some cases even a net gain, policy has been applied to biodiversity in various countries along similar lines to the idea of LDN, in that any development causing such loss is required to compensate with equivalent gains elsewhere, but how this is measured and its effectiveness in practice has been challenged [240]. In relation to LDN, many questions arise regarding the conditions for, and scale at which any offsetting may occur. The Environment Bill introduced by the UK government, and currently making its way through Parliament, mandates biodiversity net gain under the heading, “Biodiversity gain as condition of planning permission.” [241].

Acknowledgment of the fundamental importance and relevance of land and soils is gaining more prominence in the context of the CBD [234], and the UNFCCC [242] and related *Paris Agreement* [243]. All three identify their main treaty objectives as being of common concern to humankind. The objectives of the CBD include the conservation of biological diversity and the sustainable use of its components¹⁰ and would appear largely to enable comprehensive regulation of soil [222]. Although the CBD text does not specifically mention soil, soil biodiversity falls within the convention’s objectives and requires conservation. Further, as a terrestrial ecosystem, and falling within the concept of a biotic component of ecosystems with actual or potential use or value for humanity, soil is subject to sustainable use requirements under the CBD. The parties to the CBD

¹⁰ *Convention on Biological Diversity*, 5 June 1992 (in force 29 December 1993), 31 ILM 822 (“CBD”), Art 1.

have a programme of work on agricultural biodiversity, have established an *International Initiative for the Conservation and Sustainable Use of Soil Biodiversity* [244] and recently initiated the preparation of a global report on *State of Knowledge of Soil Biodiversity: Status, Challenges and Potentialities* [245]. It remains to be seen how soils will be reflected in the Global Biodiversity Framework to be adopted by the CBD COP 15, currently scheduled to take place in October 2021, in China, especially given China's rapid advance in this field ¹¹.

The UNFCCC and *Paris Agreement* requires States to take action to reduce GHG emissions and to conserve and enhance GHG sinks and reservoirs [243, 246]. Land and soils provide vital ecosystem regulation functions, can be a significant C sink and reservoir or emitter of GHGs and their precursors, and are vulnerable to the impacts of climate change. Forests aside, sustainable land use and soil management has received comparatively little attention in the UNFCCC regime. UNFCCC Art 4 and *Paris Agreement* Art 5 mandate Parties to promote and cooperate in conserving and enhancing GHG sinks and reservoirs, including biomass, forests and oceans, as well as other terrestrial, coastal and marine ecosystems. In 2017, the COP adopted decision 4/CP.23 on the 'Koronivia joint work on agriculture' to address issues related to agriculture, taking into consideration the vulnerabilities of agriculture to climate change and approaches to addressing food security, including improved soil carbon, soil health and soil fertility under grassland and cropland as well as integrated systems, including water management, and socioeconomic and food security dimensions of climate change in agriculture [247].

At its first Plenary Assembly, held at the FAO Headquarters in Rome in June 2013, the Global Soil Partnership (GSP) created the Intergovernmental Technical Panel on Soils (ITPS), made up of 27 soil experts representing all the regions of the world. At the same time soil and land was incorporated into the SDGs [93]. The GSP also proposed 2015 as the International Year of Soils (IYS) and the annual observance of a World Soil Day (on 5 December), both of which were adopted at the 68th UN General Assembly later that year. Though primarily symbolic in nature, a principle aim of these events was to raise awareness of the importance and finite nature of soil and to, "...*Support effective policies and actions for the sustainable management and protection of soil resources*" [248]. The activities of the GSP, and its Regional Partnerships represent an important voluntary forum for stimulating activity and momentum across a range of stakeholders in relation to sustainable soil management. The *Revised World Soil Charter* [249] which was adopted by FAO members in 2015, and the *Voluntary Guidelines for Sustainable Soil Management* (VGSSM) [250], which translate the

¹¹ According to the SJR International Science Ranking website China ranks second only to the US in soil science (<https://www.scimagojr.com>; accessed 28th April, 2021).

Charter's principles into practice, and were endorsed by the FAO Council in 2016,¹² were drafted and agreed under the aegis of the GSP. The GSP continues to facilitate the production of a number of other documents relevant to soil governance¹³.

The work of the ITPS, UNCCD Science-Policy Interface (SPI), IPBES, the IPCC, and in particular the publication of the 2015 State of the World Soil Resources [93], the 2017 Scientific Conceptual Framework for Land Degradation Neutrality [251], the 2018 Global Land Degradation Assessment [97], and the 2019 Report on Climate Change and Land [65], by each respectively, also stand out as key influences in the soil governance narrative. The significance of science, and of work at the science-policy interface, is increasingly important in determining the scope of and extent of legal obligations, as evident in the recently adopted 2020 International Law Association (ILA) *Guidelines on the Role of International Law in Sustainable Natural Resource Management for Development* [252].

In 2015, to coincide with UNFCCC COP21, the French government issued a bold entreaty to the global community to raise average soil organic C (SOC) levels by 0.4% (the “4 per mille Soils for Food Security and Climate” initiative), to offset annual global C emissions into the atmosphere, as well as improving food security [253, 254]. The choice of SOC as the target was fundamental and twofold, because it represents both a means of sequestering atmospheric C and the most widely accepted measure of soil health or productivity. Furthermore, increases in SOC can be achieved in many ways, for example, by soil management techniques or by the choice of what is grown and therefore, by implication, via complete land use conversion, for example from row crops to pasture or forestry. A desire to save land from conversion can also be implied, because any form of land take which seals soil or reduces SOC, would increase the overall need to enhance SOC on the remaining land. The 4 per 1000 Initiative, to which many countries have signed up, has succeeded in changing the discourse to highlight the critical role of soil and agriculture in climate change mitigation and adaptation. There has also been criticism [255], especially of the feasibility of the initiative and the underpinning data. This is not the place to discuss the debate at length, but while the authors have robustly defended the science, with caveats, they also stress that 4 per 1000 was never intended to be a precisely calculated solution, but a positive, politically-driven and powerfully symbolic aspirational target [256]. Clearly some eminent soil scientists agree with them, judging that the initiative would be a feasible, “no-regret” and indispensable climate action [257].

¹² <http://www.fao.org/3/bl813e/bl813e.pdf>

¹³ For example, FAO, 2019. *The International Code of Conduct for the Sustainable Use and Management of Fertilizers*. Rome. <https://doi.org/10.4060/CA5253EN> and FAO. 2020. *A protocol for measurement, monitoring, reporting and verification of soil organic carbon in agricultural landscapes – GSOC-MRV Protocol*. Rome. <https://doi.org/10.4060/cb0509en>

Regional and sectoral developments

At the regional level the Algiers Convention was revised in 2003 to become the Maputo Convention, and includes an updated article on land and soil (Article VI), though it only entered into force in 2016 [258]. In 1998 the *Protocol on the Implementation of the Alpine Convention of 1991 in the Field of Soil Conservation* (Alpine Soil Protocol) was agreed, becoming the first legally binding treaty expressly devoted to soil, and entering into force in 2006 [259]. For both the Maputo Convention and the Alpine Soil Protocol, implementation has not yet matched aspiration, to differing degrees, and both serve as useful focal points for awareness raising regarding soil management and best practice. The ASEAN Agreement on the Conservation of Nature and Natural Resources, signed in 1985, included an Article specifically on soil, but it has never come into force [222].

The Council of Europe revised the *European Soil Charter* in 2003 to become the *Revised European Charter for the Protection and Sustainable Management of Soil* [260]. In contrast, in the European Union, agreement on a soil instrument has to date resembled a triumph of hope over experience. As also exist, to a more limited extent, at the international level [252], there are numerous pieces of sectoral, as well as procedural, EU legislative instruments that have a bearing on soil (e.g. in areas such as agriculture, water, waste, chemicals, prevention of industrial pollution, and environmental impact assessment). In 2006 the European Commission presented its *Thematic Strategy on Soil Protection* and put forward a *Proposal for a Soil Framework Directive* to place healthy soil on a par with clean water and air [261]. However, after years of negotiations, the proposal was rejected by five member states and eventually withdrawn in 2014, the sticking points being subsidiarity to centralised control and costs of implementation. The rejecting states were in broad agreement with the Directive and even exceeded its requirements in some areas. However, they were not willing to agree to strengthened EU-wide legislation addressing soil sealing and liability for contaminated land [262]. A new EU Soil Strategy is planned for adoption in 2021 as a key component of a European Green Deal (EGD), aimed at making Europe the first climate-neutral continent by 2050 [263].

Human and Peoples' rights, access to environmental information and justice

The connections between land and soil, human health and wellbeing, food security, biodiversity and climate adaptation and mitigation are increasingly being acknowledged in the context of human and Peoples' rights, as is the importance of secure land tenure. To that end the VGSSM incorporates the Voluntary Guidelines on the Responsible Governance of Tenure of Land, Fisheries and Forests in the Context of National Food Security (VGGT) [251]. The right to a healthy environment is included in a growing number of national constitutions [264], as well as in regional and international human rights instruments. For example, the 2019 *Declaration*

on the Rights of Peasants and Other People Living in Rural Areas has proclaimed in Article 17 a ‘right to land’ of peasants and other people living in rural areas [265]. This principle of small-scale agrarian rights has been powerfully linked to the more familiar scientific rationale of SES in arguments for more robust soil governance [80, 266, 267]. We can expect that failures of governance in relation to land and soils will become more of a focus in future human rights cases. Procedural rights of citizens in relation to environmental information, public participation in decision making and access to justice in environmental matters, within sectoral treaty regimes, or in dedicated treaties such as the UNECE Aarhus Convention in Europe and the Escazú Agreement in Latin America and the Caribbean [268], also have a role to play in underpinning land and soil governance,

Sustainable Development Goals (SDGs)

All countries have signed up to the seventeen SDGs [269]. These have been formulated in a purposefully general manner, to allow for country led implementation, yet provide for specific targets and indicators with a view to facilitating fulfilment of minimum standards and comparison across countries [238, 270]. Land and soil have profound relevance for most, if not all, SDGs, though only a small number of SDG targets and indicators make specific reference to them [269, 271]. In particular SDG 15: *Life on land*, aims to protect, restore and promote sustainable use of terrestrial ecosystems, including in SDG 15.3 combatting desertification, restoring degraded land and soil, and striving to achieve a land degradation neutral (LDN) world [271]. While the SDGs themselves are not-legally binding, they do in some respects echo or amplify existing and emerging binding international commitments.

National soil legislation

Last year FAO and UNEP jointly published a key document, *Legislative approaches to sustainable agriculture and natural resources governance* [238]. The report is candid in its acknowledgement that implementation of overarching international commitments can be difficult to enforce and monitor, and that, although certain underlying principles can be discerned, ”good governance” is a “nebulous” concept; all the more so when it is associated with aspirational outcomes rather than tangible laws and regulations that are legally binding. Nevertheless, great strides have been made in some areas. Although expressly not commenting on enforcement, the report provides many examples of innovative and progressive legislation in relation to land and soil, though questions still remain in relation to enforcement, which was expressly not investigated in the report, and has been reported to come up short even for some of the potentially positive and legally binding measures [82]..

Binding or non-binding, international commitments rely on effective national implementation and enforcement. As outlined above, broadly speaking, a selection of legal instruments exist in various

countries, or provinces within countries, throughout the world to achieve all of the ‘categories of governance’ described above. However, this state of affairs is very different to one in which all countries have such instruments or, more to the point, enforce them effectively.

On a global scale the most urgent action needed may relate to category 2a (Table 2) - restrictions or guidance on development involving conversion of use, to protect terrestrial natural resources and enforced by laws based on zoning, e.g. National Parks, nature reserves, etc., to address issues such as deforestation in Amazonia or Southeast Asia. Nevertheless, as already suggested, the instruments to regulate this type of anthropogenic threat are relatively straightforward and consistently defined across the world. The obstacles are political and socioeconomic. This is also true of other existing laws that directly or indirectly protect soil, the success of which require that entrenched power imbalances are challenged [78].

Lack of enforcement and governance on the ground remain serious obstacles throughout much of the world. See, for example, the studies referenced at [80, 108, 112, 127, 167, 169, 174, 179, 272, 273]. Those ten studies cited, merely a subset of many more, draw on a range of data for their evidence, in addition to literature reviews, including land use and land evaluation records, census and demographic statistics, case studies, planning and legal decisions, stakeholder interviews and, perhaps most telling of all, spatial remote sensing of land use change, using GIS software tools such as CORINE. Furthermore, even some of the potentially positive and legally binding instruments highlighted in the FAO/UNEP report currently appear far from effective in practice [82].

Nevertheless, the process of enacting soil and land legislation is now ongoing and ubiquitous. Each country’s soil legislation, if it has enacted any, reflects that country’s specific circumstances and its government’s priorities and, one might like to think, international commitments. As described in the sections *Semantics and legal terminology* and *Methods*, some countries subsume soil into other areas of legislation, while others have one or more laws explicitly addressing soil. China, for example, with possibly the greatest absolute area of contaminated soil on Earth [274-276], as well as a long history of soil erosion [16], has not only soil-specific laws, but a separate law to address each of these two problems. The extent to which national soils governance, such as it may exist in any particular country, is a result of self-identified necessity, or as a consequence of international commitments, is an interesting one beyond the scope of this article.

Table 3 presents a condensed numerical summary of the documents held in FAOLEX analysed by the lead author and placed into derived categories considered to be salient to this article, presented as approximate percentages of countries that have created or adopted such instruments (as detailed in *Methods* above).

It is evident that soil and land protection laws are conspicuous by their relative absence on the world stage. Environmental legislation is not recorded here but, for comparison, it has been almost universally adopted. It is also conspicuous from the data below that for many countries the word *soil* would be entirely absent from their legal portfolio were it not for such environmental legislation.

Table 3 Proportions of countries with soil-related legislation

(a) SOIL-SPECIFIC POLICIES AND LAWS	COUNTRIES (%)
Explicit soil policy.	24
Soil conservation/erosion law(s).	37
Soil contamination/pollution law(s).	30
Soil sealing law(s).	2
Generic or other soil “improvement or protection” law(s).	36
Reference to soil embedded within environmental legislation.	88
Reference to soil embedded within agricultural or land rights legislation.	71
Reference to soil embedded within spatial/regional planning legislation.	49
Soil protection monitoring and/or targets.	14
(b) AGRICULTURAL LAND PRESERVATION POLICIES AND LAWS	
Policies designating zoning, including rural and/or agricultural land.	31
Policy advocating the preservation of farmland/ land take avoidance.	22
Land take avoidance targets.	5
Legal framework to facilitate farmland preservation schemes.	19
Farmland preservation guidance based on soil or land classification.	21
Law prohibiting or strongly restricting loss of all prime farmland.	15
(c) SOIL-RELATED ENVIRONMENTAL POLICIES	
Policies promoting organic or regenerative agriculture, agroecology or PES.	39
Land Degradation Neutrality (LDN) policy (commitments)	64
Policies referring specifically to the critical ecosystem services of soils.	15

Prime farmland preservation

The results presented in Table 3, along with a comprehensive literature review, leads the authors to conclude that one of the most serious failings with respect to the ambiguity or absence of legal instruments relates to what we have termed category 2(b) above, that is land take outside protected areas. At most risk and of most concern in this context is the conversion of high quality or prime farmland, which is usually afforded far less protection than natural ecosystems, or none at all, yet can in certain respects sometimes provide equivalent, or even greater, ecosystem services, depending on the criteria used. This is an important and often misunderstood point that cannot be overestimated. It is easy to fall into the trap of assuming that agricultural land cannot possibly provide environmental benefits comparable to those of natural ecosystems, either in terms of quantity or quality. In many situations this will be demonstrably true, especially when based on current land management practices, but certainly does not reflect the intrinsic value of such land nor necessarily even its existential value [277].

Undeniably, other things being equal, land which is densely covered in vegetation, with undisturbed topsoil, will tend to store more water and C, and foster greater biodiversity. However, with respect to land use, other things are rarely equal. Firstly, the principle reason land becomes prime farmland results from its highly valued ecosystem attributes: low altitude and gentle gradients; deep soil with favourable texture and structure that retains water, nutrients and C; benign biochemistry; and a favourable moisture regime that is not susceptible to extremes of drought or waterlogging. These are the desirable attributes that facilitate not just food production, but terrestrial life in general, which ecologists assess in aggregate as NPP. As crude as it might be in some respects, especially in relation to cultural or scientific value, NPP is widely regarded as one of the best proxy measures of total ecosystem service contribution [278].

In contrast to this, many of the varied soil landscapes that currently support natural ecosystems can be biologically marginal, exhibiting low NPP. Naturally eroding mountain slopes with shallow lithosols and leached podzols, dry sandy heaths, arid subtropical plains of alkaline, saline and sodic soils, thin rendzinas consisting of a few centimetres of soil over solid chalk or limestone, highly leached acidic and occasionally lateritic soils of the humid tropics, and so on, are all important and fascinating in their own right. However, even putting aside the utilitarian criterion of agricultural potential, these peripheral zones can also be limited with respect to a range of critical ecosystem services. Even the biodiversity of such areas, rare and invaluable as it can be in absolute terms, is only as rich as its environment can support.

That said, millennia of evolution in marginal environments have also produced extraordinary abundance, such as the biodiversity of the tropical rainforests on soils that are typically of low agricultural value or the vast reserves of C in the peatlands of the boreal tundra and the temperate upland peat moors, where the climate has little to offer agriculture [279].

Prime farmland is not normally an obvious candidate for voluntary conversion to natural regeneration or rewilding, although a strong case can be made for taking lowland peat, that has been drained and converted to agriculture, out of arable production [280, 281]. Nevertheless, high grade arable land, even in its cultivated state, can provide abundant benefits in terms of flood control, water storage and filtration, wildlife habitat, landscape value and recreation. However, and critically, such benefits are greatly affected by land management. The negative impacts of modern farming, on the soil in particular, such as compaction, erosion, pollution and reduced SOC, much of this a result of aggressive cultivation methods, are precisely the factors that undermine the ability of the soil to provide ecosystem services [282, 283].

One very welcome trend in recent decades has been to roll back many of these harmful methods. A range of techniques that are loosely grouped under the term of conservation, or regenerative, agriculture, agroecology, or even “carbon farming” are increasingly being promoted and applied in many countries [98, 284-286]. Included within this approach are minimum or zero tillage, permanent soil cover with crop residues, live mulches, cover cropping, diversity of planting, intercropping and enhanced rotations including, for example, deep roots, leguminous leys and rotational livestock grazing, agroforestry and other forms of mixed cropping, perennial crops, integrated pest management (IPM) and biological control, and many other ways of conserving and enhancing soil quality and hence, by default, ameliorating environmental impacts. While important everywhere for food security and food sovereignty, sustainable soil management is especially important on the C-rich Chernozems (Mollisols) of eastern Europe and the Americas, arguably the world’s most productive soils [277, 287]. Hence, our advocacy of preserving our most valuable farmland is proposed in tandem with the adoption of such methods.

Emerging holistic conceptual frameworks

Amundson sums up soil governance in the US as a complex patchwork of transient interventions, as opposed to long-term solutions [288]. Other authors describe comparable situations elsewhere, for example, in the EU [289, 290], in South America [291], in Australia [292], in Russia [293] and in other examples already cited in this article. Fromherz summarises many of the existing soil governance initiatives, as well as the failures, and makes an impassioned appeal for a dedicated, legally binding international soil governance instrument on the basis that, “...individual states lack both the power and the incentives to make these changes.” [266]. This has been echoed by others [204, 223, 252, 294]. Fromherz also highlights a key point alluded to already in this article, the low profile (literal as well as metaphorical, one could say) of soil, which is often conspicuous in its absence from risk assessments of other natural resources [295, 296]. Gonzalez Lago *et al.* refer to a global soils policy vacuum and call for an urgent transdisciplinary framework approach to “re-politicise” soil [297]. This begs the question as to what extent soils have been “de-

politicised”, but what is emerging from a number of authors and institutions is the degree to which soil protection is inescapably enmeshed with ethics [298].

Participatory modelling and conceptual frameworks have been applied to complex cross-cutting problems such as addressing the SDGs, and these approaches continue to evolve [299]. The latest stages in the process of soil governance so far, in the last decade, are encouraging to some extent, at least with regard to what soil scientists are bringing to the table. An overarching conceptual framework which has been proposed by some of the world’s leading soil scientists, broadly as a memorandum of understanding, is soil security [300]. This concept has five dimensions: capability, condition, capital, connectivity and codification, which encompasses the translation of soil knowledge into policy and legislation [301], e.g. aimed at achieving the SDGs [302], although an attempt to establish soil security as one of the SDGs was unsuccessful [303]. The soil security initiative is still at a high level, with the emphasis on policy rather than active governance, but progress is being made and reported on in some areas [304].

Also emerging are more formalised frameworks which place the concept of SES further than ever in the context of socioeconomic decisions, policy-making and governance [305, 306]. The most fully developed of these methodologies, the Land Resource Circle (LRC) framework, goes further than any other approach known to the authors in extending land evaluation to incorporate SES. The framework identifies in-depth environmental and socioeconomic implications of land-use decisions via a scoring system which avoids reducing all outcomes to simplistic monetary values [305]. Furthermore, the LRC paper includes a very detailed hypothetical example which perfectly demonstrates how quantified soil variables can be combined and converted into societal costs and benefits. By contrast, the the Resilience-Effectiveness-Efficiency-Legitimacy (REEL) framework [306], provides a purely qualitative means of comparing different approaches to soil governance according to the four criteria (dimensions) embedded within its title. Of these dimensions, legitimacy encompasses issues of equity, fairness, accountability, and participation, and is perhaps the least formulaic and the most elusive to achieve. The REEL framework also highlights interconnectedness and attempts to address situations where policy targets mismatch the causes of problems, either spatially or in terms of scale or even time. This is almost a meta-framework, with the emphasis on socioeconomics and due diligence.

As with other forms of natural resource governance, politics and socioeconomics are often obstacles to effective soil governance, as are weak governance structures, but a critical distinction is that there is much scope for creating more effective legal instruments to tackle this problem than currently exist in most countries, and also for improving clarity and consistency to bring prime farmland protection further in line with ecosystem and soil protection.

Conclusion

An appreciation of the tangible benefits of soil is woven into the fabric of history and is reflected by the value and protections afforded to productive soil by society over millennia. Some of these protections have taken the form of legislation, but this has been ad hoc, inadequately enforced and rarely if ever proportionate to the total ecosystem service value of soil. Two challenges identified as hindering progress to date are the terminological difficulties associated with soil and land, and the absence of a soil-centric policy framework to address wider issues that are intrinsically bound up with soil. Moreover, soil exists as a component of land which, in the face of population pressure, has been inexorably subject to the twin impacts of firstly, extra demands, e.g. of food production, and secondly, competing demands, such as urbanisation.

Agriculture has traditionally responded to these demands by both expansion and intensification, each approach engendering anthropogenic threats to soil, the former often encroaching on natural ecosystems and the latter potentially leading to soil degradation. These threats have profound impacts on the environment locally and worldwide, and both have spawned a great deal of national and international legislation. However, the problem of land take, in particular the conversion of prime agricultural land to non-agricultural use, including some irreversible soil sealing, is arguably an equally serious threat because of the disproportionate loss of SES this represents. Yet this threat tends to be relatively inconspicuous, and subject to ambiguous legislation and governance.

Putting aside the issue of whether soil and land governance is effectively implemented in general, threats to natural ecosystems are explicitly addressed by environmental laws, and soil degradation is typically addressed by agricultural or soil-specific laws. A few of these laws have been in place for centuries and many others have joined them in recent decades as environmental awareness and global concerns have grown. However, neither environmental nor agricultural legislation normally encompasses the loss of prime farmland, the fate of which tends to fall within the remit of spatial planning. Furthermore, planners and planning regulations must navigate many competing demands and interest groups, political as well as economic, and hence they rarely prioritise soil or its intrinsic value to future generations. As a result, legal instruments which are frequently designed with little if any reference to soil, often have greatest power to transform soil and land use. This situation is further compounded by the fact that intense commercial pressure in tandem with conventional economics, which has traditionally ignored the benefits of long-term ecosystem services, promotes land use options that are more profitable than farming.

In a time of so much environmental concern, a further paradoxical factor that has emerged in the last half century to weaken the case for farmland preservation (as opposed to soil protection *per se*), is the conflicted nature of agriculture: on the one hand the producer of our sustenance and the custodian of rural landscapes, while on the other, a significant cause of the other two types of threat described above, as well as having other detrimental impacts such as climate forcing and water pollution. Lacking both the visual impact and the iconic status of natural landscapes, soil in an (often intensive) agricultural context does not constitute an

obvious rallying point in the public consciousness. So the constant attrition of prime farmland occurs in something of a legal and ecological grey area, attracting far less attention than other environmental issues.

However, the case for protecting soil as a critical part of our natural capital is separately gaining ground. A key paradigm shift occurred in the 1990s when it became more widely appreciated that soil C is inextricably linked with atmospheric C. This development also approximately coincided with a broad gradual acceptance that land degradation was interrelated with climate change and biodiversity loss. This view of soil as a central component in limiting global warming, adapting to climate change and addressing the ecological crisis, is being framed unequivocally in the wider context of sustainability and the linking of science to policy. Emerging multidisciplinary frameworks, such as the concept of soil security and methodologies for holistically evaluating SES, may collectively contribute to this process.

Clearly a lack of effective soil governance is often more significant a problem than an absence of legal instruments. Indeed, failings or inequities in human behaviour, whether ethical or intellectual, have always been endemic, which is why laws exist. Nevertheless, the arc of history suggests that, despite some serious setbacks, the moral and ultimately self-preserving imperative of collective justice and consensus can and often does prevail over short-term vested interests. However, the status quo will not give way without the constant drip-feeding of international resolutions, critical reviews and articles, pressure groups, whistle-blowing and local protests, regardless of how ineffective each of these clarion calls may appear in isolation.

Many politicians and activists are calling for “green new deals” to emulate Roosevelt’s New Deal following the Great Depression, but with an emphasis on climate justice and ecosystem restoration. This would of course include sustainable land use, by default, and an important component of that needs to be soil protection and land preservation, consistent with climate justice, including the attendant social and economic incentives necessary to achieve local and global climate goals, biodiversity objectives and food security. Profound reforms, such as democratic freedoms, become most unassailable when society perceives a binary moral choice that serves the common good. One of the most successful examples of global co-operation of this kind was the signing of the *Montreal Protocol* in 1986 to protect the ozone layer. Persuading the global community to reduce GHG emissions in the *Paris Agreement*, though desperately hard-won and far from over, provides another exemplar of what can be achieved. Ginzky cites other examples and, despite a lack of progress to date, suggests that a binding international treaty on soils is both necessary and achievable [307]. Some of the details may be complex, but it can be achievable where the message is simple, and a window is open.

The lesson here, for saving our soil, is to strive for a clear consistent message, backed up by clear consistent policies and laws, and clear consistent criteria to decide how to prioritise one soil type or area of land over another. This alone cannot solve the vast problem, but it will make the process more streamlined and more

transparent. It will also facilitate governance for those who genuinely want to govern and it will make infringements that bit more difficult to effect and pass unnoticed. The key aim is to couple a compelling case with an achievable solution.

Data accessibility

The only original data presented is the categorisation percentages abstracted from FAOLEX in Table 1.

Authors' contributions

L.R.P. conceived the idea, conducted much of the research and wrote the bulk of the text. C.R. made considerable contributions to the text, especially in relation to soil law, and performed much editing.

Competing interests

We declare we have no competing interests.

Funding

We received no funding for this study.

Acknowledgements

Several individuals and organisations provided valuable information via correspondence or in meetings. The following persons helped to explain their local soil and land laws, or planning regulations, in theory and practice: Dr Thomas Daniels (Univ. Pennsylvania), Dr Mark Lapping (Univ. Southern Maine), Ian Rugg and colleagues (Natural Resources Wales and Natural England), Dr. Fernando García Prechac (Fac. of Agronomy, Univ. of Uruguay, and former Director General of Natural Resources, Ministry of Livestock, Agriculture and Fisheries, Uruguay), Marcel Achkar (Univ. República de Uruguay), Gundula Prokop (EU Research and Territorial Co-operation Projects, Federal Environment Agency, Austria), Anna Shortly (The Greenbelt Foundation, Ontario), Dr Elena Havlicek and Ruedi Stähli (Federal Office for the Environment, Switzerland), Ljubiša Bezbradica (Institute of Architecture and Urban & Spatial Planning of Serbia), Tom Corser (New Zealand Ministry for the Environment), and Dr. Suhail Ahmad (Cluster University Of Srinagar, Kashmir). In addition: Dr David Dent (former Director, ISRIC) gave advice on soil degradation

assessments which he co-authored at ISRIC; CPRE posted a free copy of *Valuing the Land* which is unavailable online; and two anonymous reviewers helped us to improve the balance and style of the paper.

References

1. Asante, SKB. 1965 Interests in Land in the Customary Law of Ghana. A New Appraisal. *The Yale Law Journal* **74**, 848-885. (doi:doi: 10.2307/794709).
2. Brussaard, L. 2021 Biodiversity and ecosystem functioning in soil: The dark side of nature and the bright side of life. *Ambio*. (doi:10.1007/s13280-021-01507-z).
3. Milde, KF. 1950 Roman Contributions to the Law of Soil Conservation. *Fordham L. Rev.* **19**, 192-196.
4. Pfister, L, Savenije, HHG & Fenicia, F. 2009 *Leonardo Da Vinci's water theory : on the origin and fate of water*. Wallingford, International Association of Hydrological Sciences.
5. New Scientist. 2017 *How Your Brain Works: Inside the most complicated object in the known universe*, Quercus.
6. Young, IM & Crawford, JW. 2004 Interactions and self-organization in the soil-microbe complex. *Science* **304**, 1634-1637. (doi:10.1126/science.1097394).
7. Almond, REA, Grooten, M & Petersen, T. 2020 WWF (2020) Living Planet Report 2020 - Bending the curve of biodiversity loss. eds. REA Almond, M Grooten & T Petersen. Gland, Switzerland, WWF.
8. Hardin, G. 1968 The Tragedy of the Commons. *Science* **162**, 1243-1248. (doi:10.1126/science.162.3859.1243).
9. Diamond, JM. 2005 *Collapse: How Societies Choose to Fail Or Succeed*, Viking.
10. Global Soil Partnership web page. 2020 Contribute to the GSP newest tool: SoILEX.
11. 2020 Ukrainian Government, Land Code (No. 2786-III of 2001; revised 16.10.2020).
12. Herodotus. 1922 (Loeb Classical Library). Translated by A. D. Godley. Vol. I.: Books 1 and 2, pp. xxi + 504. Vol. II.: Books 3 and 4, pp. xviii + 416. London: W. Heinemann. 10s. a volume. *The Classical Review* **36**, 135-135. (doi:10.1017/S0009840X00016759).
13. Meguid, MA. 2019 Key Features of the Egypt's Water and Agricultural Resources. In *Conventional Water Resources and Agriculture in Egypt* (ed. AM Negm), pp. 39-99 Springer International Publishing).
14. Bennett, HH. 1939 *Soil conservation*. New York, London, McGraw-Hill.
15. Butzer, K. 2005 Environmental history in the Mediterranean world: cross-disciplinary investigation of cause-and-effect for degradation and soil erosion. *Journal of Archaeological Science* **32**, 1773-1800. (doi: 1710.1016/j.jas.2005.1706.1001).
16. Dotterweich, M. 2013 The history of human-induced soil erosion: Geomorphic legacies, early descriptions and research, and the development of soil conservation—A global synopsis. *Geomorphology* **201**, 1-34. (doi:10.1016/j.geomorph.2013.07.021).

17. Hyams, E. 1952 *Soil and Civilization*, Thames and Hudson.
18. Redman, CL. 1999 *Human impact on ancient environments*. Tucson, University of Arizona Press.
19. Brevik, EC & Hartemink, AE. 2010 Early soil knowledge and the birth and development of soil science. *CATENA* **83**, 23-33. (doi:10.1016/j.catena.2010.06.011).
20. Verheye, WH. 2009 *Land Use, Land Cover and Soil Sciences - Volume VII: Soils and Soil Sciences - 2*, EOLSS Publ.
21. Homer-Dixon, TF. 1994 Environmental Scarcities and Violent Conflict: Evidence from Cases. *International Security* **19**, 5-40. (doi:10.2307/2539147).
22. Berman, N, Couttenier, M & Soubeyran, R. 2019 Fertile Ground for Conflict. *Journal of the European Economic Association*. (doi:10.1093/jeea/jvz068).
23. Fearon, J & Laitin, D. 2014 Does Contemporary Armed Conflict Have 'Deep Historical Roots'? *SSRN Electronic Journal*. (doi:10.2139/ssrn.1922249).
24. Global Witness. 2020 Defending Tomorrow: The climate crisis and threats against land and environmental defenders.
25. Le Billon, P & Lujala, P. 2020 Environmental and land defenders: Global patterns and determinants of repression. *Global Environmental Change* **65**, 102163. (doi:10.1016/j.gloenvcha.2020.102163).
26. Gong, Z, Zhang, X, Chen, J & Zhang, G. 2003 Origin and development of soil science in ancient China. *Geoderma* **115**, 3-13. (doi:10.1016/S0016-7061(03)00071-5).
27. Barrera-Bassols, N, Alfred Zinck, J & Van Ranst, E. 2006 Symbolism, knowledge and management of soil and land resources in indigenous communities: Ethnopedology at global, regional and local scales. *CATENA* **65**, 118-137. (doi:10.1016/j.catena.2005.11.001).
28. Ellickson, RC & Thorland, CD. 1995 Ancient Land Law: Mesopotamia, Egypt, Israel. *Chicago-Kent Law Review* **71**, 321.
29. Wasson, RJ. 2010 Exploitation and conservation of soil in the 3000-year agricultural and forestry history of South Asia. (eds. JR McNeill & V Winiwarter), pp. 13-50. Isle of Harris, White Horse Press).
30. Barbieri-Low, AJ & Yates, RDS. 2015 *Law, State, and Society in Early Imperial China (2 vols)*, Brill.
31. Sterckx, R. 2020 Food and Agriculture. In *Routledge Handbook of Early Chinese History* (ed. P Goldin), pp. 300-318, Routledge).
32. Shi, S. 1959 *On "Fan Shêng-chih shu" : an agriculturist book of China written by Fan Shêng-chih in the first century B.C. / translated and commented upon by Shih Sheng-han = Fan Shengzhi shu jin shi / Shi Shenghan yi shi*. Peking, Science Press.
33. Watson, AM. 1983 *Agricultural innovation in the early Islamic world : the diffusion of crops and farming techniques, 700-1100*. Cambridge, Cambridge University Press.
34. Hailey, A & Cazabon-Mannette, M. 2011 Conservation of Caribbean Island Herpetofaunas Volume 1: Conservation Biology and the Wider Caribbean. In *Conservation Of Herpetofauna In The Republic Of*

- Trinidad And Tobago* (pp. 183. (doi: 110.1163/ej.9789004183957.i-9789004183228.9789004183964), Brill).
35. Williams, J. 2015 Soils Governance in Australia: challenges of cooperative federalism. *International Journal of Rural Law and Policy*. (doi:10.5130/ijrlp.i1.2015.4173).
 36. Stocking, M. 1985 Soil Conservation Policy in Colonial Africa. *Agricultural History* **59**, 148-161.
 37. Sandbach, F. 1980 *Environment, Ideology, and Policy*, Allanheld, Osmun.
 38. Williams, GW & United States. Forest Service. 2005 *The USDA Forest Service : the first century*. Washington, DC, USDA Forest Service; iv, 156 p. p.
 39. Wynn, G. 1979 Pioneers, politicians and the conservation of forests in early New Zealand. *Journal of Historical Geography* **5**, 171-188. (doi:10.1016/0305-7488(79)90132-4).
 40. Reddy, B, Hoag, D & Shobha, B. 2004 Economic incentives for soil conservation in India. In *Proceedings of the 13th International Soil Conservation Organization Conference, Brisbane*.
 41. Crofts, R & Olgeirsson, FG. 2011 *Healing the Land: The Story of Land Reclamation and Soil Conservation in Iceland*, Soil conservation service.
 42. McLeman, RA, Dupre, J, Berrang Ford, L, Ford, J, Gajewski, K & Marchildon, G. 2014 What we learned from the Dust Bowl: lessons in science, policy, and adaptation. *Population and Environment* **35**, 417-440. (doi:10.1007/s11111-013-0190-z).
 43. Pretty, JN. 1995 *Regenerating agriculture : policies and practice for sustainability and self-reliance*. London, Earthscan.
 44. Sylvester, KM & Rupley, ES. 2012 Revising the Dust Bowl: High Above the Kansas Grasslands. *Environ Hist Durh N C* **17**, 603-633. (doi:10.1093/envhis/ems047).
 45. Peake, L. 1986 *Erosion, Crop Yields and Time: A Reassessment of Quantitative Relationships*, University of East Anglia School of Development Studies.
 46. Konyushkov, D. 2014 Russian: Evolution and Examples. In *Reference Module in Earth Systems and Environmental Sciences* (Elsevier).
 47. Carter-Whitney, M, Esakin, TC & Canadian Institute for Environmental Law and, P. 2010 Ontario's greenbelt in an international context.
 48. Simonson, RW. 1989 *Historical Highlights of Soil Survey and Soil Classification with Emphasis on the United States, 1899-1970*. Wageningen, the Netherlands, International Soil Reference and Information Centre (ISRIC).
 49. Young, A. 1980 *Tropical Soils and Soil Survey*, Cambridge University Press.
 50. Robinson, A. 1943 The Scott and Uthwatt Reports on Land Utilisation. *The Economic Journal* **53**, 28-38. (doi:10.2307/2226286).
 51. Engholm, B. 1985 The Scott report - a personal perspective. *Planning History Bulletin* **7**, 39-43.

52. Sheail, J. 1997 Scott revisited: Post-war agriculture, planning and the British countryside. *Journal of Rural Studies* **13**, 387-398. (doi:10.1016/S0743-0167(97)00028-4).
53. CPRE. 2017 LANDLINES: why we need a strategic approach to land. London, CPRE.
54. Monk, S, Whitehead, C, Burgess, G & Tang, C. 2013 International review of land supply and planning systems. York, UK, Joseph Rowntree Foundation.
55. Hogendorn, JS & Scott, KM. 1981 The East African Groundnut Scheme: Lessons of a Large-Scale Agricultural Failure. *African Economic History*, 81-115. (doi:10.2307/3601296).
56. Natural England. 2012 Agricultural Land Classification: protecting the best and most versatile agricultural land. Technical Information Note TIN049.
57. Daniels, T. 1990 Using LESA in a purchase of development rights program. *Journal of Soil and Water Conservation* **45**, 617-621.
58. Lee, ATM. 1948 Soil Conservation—An International Study, Food and Agriculture Organization of the United Nations, Washington, D. C., 1948. *American Journal of Agricultural Economics* **30**, 784-786. (doi:10.2307/1232795).
59. United Nations. 1949 *UN Yearbook 1947-48*. Washington.
60. Bunce, AC. 1942 *The economics of soil conservation*. Ames, Ia., Iowa State college Press.
61. Ciriacy-Wantrup, SV. 1952 *Resource Conservation: Economics and Policies*, University of California, Division of Agricultural Sciences, Agricultural Experiment Station.
62. Sauer, EL. 1967 *Economics of Soil Conservation, Reclamation and Rehabilitation*.
63. Morgan, RJ & Future, Rft. 1965 *Governing Soil Conservation: Thirty Years of the New Decentralization*, Resources for the Future.
64. Aubreville, A. 1949 *Climats, Forêts, et Désertification de l'Afrique Tropicale*. Société d'Editions Géographiques, Maritimes et Tropicales. Paris.
65. IPCC. 2019 *Climate Change and Land: an IPCC special report on climate change, desertification, land degradation, sustainable land management, food security, and greenhouse gas fluxes in terrestrial ecosystems*.
66. Thomas, DSG & Middleton, NJ. 1994 *Desertification: Exploding the Myth*. London, Wiley.
67. Gisladdottir, G & Stocking, M. 2005 Land degradation control and its global environmental benefits. *Land Degradation & Development* **16**, 99-112. (doi:10.1002/ldr.687).
68. Davidson, J & Wibberley, GP. 1977 *Planning and the rural environment*.
69. Pingali, PL. 2012 Green Revolution: Impacts, limits, and the path ahead. *Proceedings of the National Academy of Sciences* **109**, 12302-12308. (doi:10.1073/pnas.0912953109).
70. Mannion, AM. 1991 *Global environmental change : a natural and cultural environmental history*. Harlow, Longman Scientific & Technical.

71. Agricultural Advisory Council. 1970 *Modern Farming and the Soil: Report of the Agricultural Advisory Council on Soil Structure and Soil Fertility*. London, HM Stationery Office.
72. Biniek, JP. 1979 *Agricultural and environmental relationships issues and priorities: report*. Washington, Library of Congress, U.S. Govt. Print. Off.; xvi, [1], 696 p. p.
73. USDA. 1973 *Monoculture in Agriculture: Extent, Causes, and Problems-report of the Task Force on Spatial Heterogeneity in Agricultural Landscapes and Enterprises*.
74. Carson, R. 1962 *Silent spring*. Boston, Cambridge, Mass., Houghton Mifflin; Riverside Press; x, 368 p. p.
75. Schneider, SH. 1989 The greenhouse effect: science and policy. *Science* **243**, 771-781.
76. Leahy, S, Clark, H & Reisinger, A. 2020 Challenges and Prospects for Agricultural Greenhouse Gas Mitigation Pathways Consistent With the Paris Agreement. *Frontiers in Sustainable Food Systems* **4**. (doi:10.3389/fsufs.2020.00069).
77. Shivanna, KR. 2020 The Sixth Mass Extinction Crisis and its Impact on Biodiversity and Human Welfare. *Resonance* **25**, 93-109. (doi:10.1007/s12045-019-0924-z).
78. Weigelt, J, Müller, A, Janetschek, H & Töpfer, K. 2015 Land and soil governance towards a transformational post-2015 Development Agenda: an overview. *Current Opinion in Environmental Sustainability* **15**, 57-65. (doi:10.1016/j.cosust.2015.08.005).
79. Juerges, N & Hansjürgens, B. 2016 Soil governance in the transition towards a sustainable bioeconomy – A review. *Journal of Cleaner Production* **170**. (doi:10.1016/j.jclepro.2016.10.143).
80. Guereña, A. 2016 *Unearthed: Land, power and inequality in Latin America*. OXFAM.
81. van Schaik, L & Dinnissen, R. 2014 *Terra Incognita: Land degradation as underestimated threat amplifier. Clingendael report for the Netherlands Environmental Assessment Agency (PBL)*. The Hague, Clingendael.
82. Baba, SH, Wani, MH, Zargar, BA & Bhat, IF. 2020 Determinants of land degradation in Jammu and Kashmir: implications for land governance. *Agricultural Economics Research Review* **32**, 303645.
83. FAO. FAOLEX Database.
84. Sloan, T, Payne, R, Anderson, R, Bain, C, Chapman, S, Cowie, N, Gilbert, PJ, Lindsay, R, Mauquoy, D, Newton, A, et al. 2018 Peatland afforestation in the UK and consequences for carbon storage. *Mires and Peat* **23**. (doi:10.19189/MaP.2017.OMB.315).
85. Gardi, C. 2017 *Urban expansion, land cover and soil ecosystem services*, Taylor & Francis.
86. Prokop, G, Jobstmann, H & Schönbauer, A. 2011 *Report on Best Practices for Limiting Soil Sealing and Mitigating Its Effects*. European Communities.
87. Scalenghe, R & Marsan, FA. 2009 The anthropogenic sealing of soils in urban areas. *Landscape and Urban Planning* **90**, 1-10. (doi:10.1016/j.landurbplan.2008.10.011).
88. Bhogal, S & Sinha, S. 2021 India protests: farmers could switch to more climate-resilient crops – but they have been given no incentive. In *The Conversation*.

89. Oldeman, LR, Hakkeling, RTA & Sombroek, WG. 1991 World map of the status of human-induced soil degradation : an explanatory note, 2nd. rev. ed. Wageningen [etc.], ISRIC [etc.].
90. Bai, ZG, Dent, DL, Olsson, L & Schaepman, ME. 2008 Proxy global assessment of land degradation. *Soil Use and Management* **24**, 223-234. (doi:10.1111/j.1475-2743.2008.00169.x).
91. Sonneveld, BGJS & Dent, DL. 2009 How good is GLASOD? *J. Environ. Manage.* **90**, 274-283. (doi:10.1016/j.jenvman.2007.09.008).
92. Bai, ZG, Dent, DL, Olsson, L, Tengberg, A, Tucker, C & Yengoh, G. 2015 A longer, closer, look at land degradation. *Agriculture for Development* **24**, 3-9.
93. FAO & ITPS. 2015 *The Status of the World's Soil Resources*.
94. Nkonya, E, Mirzabaev, A & von Braun, J. 2016 *Economics of Land Degradation and Improvement – A Global Assessment for Sustainable Development*, Springer Nature.
95. Esch, S, Brink, B, Stehfest, E, Bakkenes, M, Sewell, A, Bouwman, A, Meijer, J, Westhoek, H & van den Berg, M. 2017 *Exploring future changes in land use and land condition and the impacts on food, water, climate change and biodiversity: Scenarios for the UNCCD Global Land Outlook*.
96. United Nations Convention to Combat Desertification. 2017 *The Global Land Outlook*, first edition. Bonn, Germany.
97. IPBES. 2018 *The IPBES assessment report on land degradation and restoration. Secretariat of the Intergovernmental Science-Policy Platform on Biodiversity and Ecosystem Services*. Bonn, Germany.
98. Zimmerer, KS. 2011 " Conservation booms" with agricultural growth? Sustainability and Shifting Environmental Governance in Latin America, 1985-2008 (Mexico, Costa Rica, Brazil, Peru, Bolivia). *Latin American Research Review*, 82-114.
99. Stubenrauch, J, Garske, B & Ekardt, F. 2018 Sustainable Land Use, Soil Protection and Phosphorus Management from a Cross-National Perspective. *Sustainability* **10**, 1988. (doi:10.3390/su10061988).
100. 2017 The most neglected threat to public health in China is toxic soil; and fixing it will be hard and costly. In *The Economist*.
101. Daily, C. 2019 New law on soil pollution will pinpoint responsibility. China.org.cn.
102. Li, T, Liu, Y, Lin, S, Liu, Y & Xie, Y. 2019 Soil Pollution Management in China: A Brief Introduction. *Sustainability* **11**, 556.
103. Hendlin, YH. 2014 From Terra Nullius to Terra Communis Reconsidering Wild Land in an Era of Conservation and Indigenous Rights. *Environmental Philosophy* **11**, 141-174. (doi:10.5281/zenodo.260245).
104. Pearce, F. 2018 *Conflicting Data: How Fast Is the World Losing its Forests?* Yale Environment 360, Yale School of the Environment.
105. Curtis, PG, Slay, CM, Harris, NL, Tyukavina, A & Hansen, MC. 2018 Classifying drivers of global forest loss. *Science* **361**, 1108-1111. (doi:10.1126/science.aau3445).

106. Escobar, H. 2020 Deforestation in the Brazilian Amazon is still rising sharply. *Science* **369**, 613-613. (doi:10.1126/science.369.6504.613).
107. Chamberlin, J, Jayne, TS & Headey, D. 2014 Scarcity amidst abundance? Reassessing the potential for cropland expansion in Africa. *Food Policy* **48**, 51-65. (doi:10.1016/j.foodpol.2014.05.002).
108. Güneralp, B, Lwasa, S, Masundire, H, Parnell, S & Seto, KC. 2017 Urbanization in Africa: challenges and opportunities for conservation. *Environmental Research Letters* **13**, 015002. (doi:10.1088/1748-9326/aa94fe).
109. Young, A. 1999 Is there Really Spare Land? A Critique of Estimates of Available Cultivable Land in Developing Countries. *Environment, Development and Sustainability* **1**, 3-18. (doi:10.1023/A:1010055012699).
110. Rockström, J, Steffen, W, Noone, K, Persson, Å, Chapin, FS, Lambin, EF, Lenton, TM, Scheffer, M, Folke, C, Schellnhuber, HJ, et al. 2009 A safe operating space for humanity. *Nature* **461**, 472-475. (doi:10.1038/461472a).
111. Byerlee, D, Stevenson, J & Villoria, N. 2014 Does intensification slow crop land expansion or encourage deforestation? *Global Food Security* **3**, 92-98. (doi:10.1016/j.gfs.2014.04.001).
112. Abu Hatab, A, Cavinato, MER, Lindemer, A & Lagerkvist, C-J. 2019 Urban sprawl, food security and agricultural systems in developing countries: A systematic review of the literature. *Cities* **94**, 129-142. (doi:10.1016/j.cities.2019.06.001).
113. Metternicht, G. 2018 *Land Use and Spatial Planning: Enabling Sustainable Management of Land Resources*, Springer International Publishing.
114. Seitzinger, SP, Svedin, U, Crumley, CL, Steffen, W, Abdullah, SA, Alfsen, C, Broadgate, WJ, Biermann, F, Bondre, NR, Dearing, JA, et al. 2012 Planetary Stewardship in an Urbanizing World: Beyond City Limits. *AMBIO* **41**, 787-794. (doi:10.1007/s13280-012-0353-7).
115. Smith, P, Gregory, PJ, Vuuren, Dv, Obersteiner, M, Havlík, P, Rounsevell, M, Woods, J, Stehfest, E & Bellarby, J. 2010 Competition for land. *Philosophical Transactions of the Royal Society B: Biological Sciences* **365**, 2941-2957. (doi:10.1098/rstb.2010.0127).
116. FAO. 2011 Payments for ecosystem services and food security.
117. van Vliet, J, Eitelberg, DA & Verburg, PH. 2017 A global analysis of land take in cropland areas and production displacement from urbanization. *Global Environmental Change* **43**, 107-115. (doi:10.1016/j.gloenvcha.2017.02.001).
118. Bren d'Amour, C, Reitsma, F, Baiocchi, G, Barthel, S, Güneralp, B, Erb, KH, Haberl, H, Creutzig, F & Seto, KC. 2017 Future urban land expansion and implications for global croplands. *Proc Natl Acad Sci U S A* **114**, 8939-8944. (doi:10.1073/pnas.1606036114).
119. Chen, G, Li, X, Liu, X, Chen, Y, Liang, X, Leng, J, Xu, X, Liao, W, Qiu, Y, Wu, Q, et al. 2020 Global projections of future urban land expansion under shared socioeconomic pathways. *Nature Communications* **11**.

120. European Environment Agency. 2017 Landscapes in transition: An account of 25 years of land cover change in Europe. In *EEA Report*. Copenhagen.
121. Ustaoglu, E & Williams, B. 2017 Determinants of Urban Expansion and Agricultural Land Conversion in 25 EU Countries. *Environmental Management* **60**, 717-746. (doi:10.1007/s00267-017-0908-2).
122. Cho, J. 2005 Urban Planning and Urban Sprawl in Korea. *Urban Policy and Research* **23**, 203-218. (doi:10.1080/08111470500143304).
123. Debolini, M, Valette, E, François, M & Chéry, J-P. 2015 Mapping land use competition in the rural–urban fringe and future perspectives on land policies: A case study of Meknès (Morocco). *Land Use Pol.* **47**, 373-381. (doi:10.1016/j.landusepol.2015.01.035).
124. Silva, C. 2019 Auckland’s Urban Sprawl, Policy Ambiguities and the Peri-Urbanisation to Pukekohe. *Urban Science* **3**, 1.
125. Naab, FZ, Dinye, R & Kasanga, RK. 2013 Urbanisation and its impact on agricultural lands in growing cities in developing countries: a case study of Tamale in Ghana. *European scientific journal* **2**, 256-287.
126. Mori, H. 1998 Land Conversion at the Urban Fringe: A Comparative Study of Japan, Britain and the Netherlands. *Urban Studies* **35**, 1541-1558. (doi:10.1080/0042098984277).
127. Ladu, JLC, Athiba, AL & Ondogo, EC. 2019 An Assessment of the Impact of Urbanization on Agricultural Land Use in Juba City, Central Equatoria State, Republic of South Sudan. *Journal of Applied Agricultural Economics and Policy Analysis* **2**, 22-30.
128. Ullah, S, Khan, MA, Rahman, A & Mahmood, S. 2019 Evaluation of urban encroachment on farmland: a threat to urban agriculture in Peshawar City District, Pakistan. *Erdkunde* **73**, 127-142. (doi:10.3112/erdkunde.2019.02.04).
129. Paudel, B, Pandit, J & Reed, B. 2013 Fragmentation and conversion of agriculture land in Nepal and Land Use Policy 2012.
130. Satterthwaite, D, McGranahan, G & Tacoli, C. 2010 Urbanization and its implications for food and farming. *Philosophical Transactions of the Royal Society B: Biological Sciences* **365**, 2809-2820. (doi:10.1098/rstb.2010.0136).
131. Suu, NV. 2009 Agricultural land conversion and its effects on farmers in contemporary Vietnam. *Focaal* **2009**, 106. (doi:10.3167/fcl.2009.540109).
132. Meadows, D & Randers, J. 2004 *Limits to Growth: The 30-Year Update*, Chelsea Green Publishing.
133. Schumacher, EF. 1973 *Small is beautiful; economics as if people mattered*. New York, Harper & Row.
134. Kuper, S. 2019 Thoreau, Leopold, & Carson: Challenging Capitalist Conceptions of the Natural Environment. *Consilience* **0**. (doi:10.7916/consilience.v0i13.3927).
135. Kennel, CF. The gathering anthropocene crisis. *The Anthropocene Review* **2**, 81-98. (doi:10.1177/2053019620957355).
136. Indian Government. 2009 Report of the Committee on State Agrarian Relations and the Unfinished Task in Land Reforms.

137. Paudel, B, Pandit, J & Reed, B. 2013 Fragmentation and conversion of agriculture land in Nepal and Land Use Policy 2012.
138. Chen, A, He, H, Wang, J, Li, M, Guan, Q & Hao, J. 2019 A Study on the Arable Land Demand for Food Security in China. *Sustainability* **11**, 4769.
139. Govindaprasad, PK & Manikandan, K. 2016 Farm Land Conversion and Food Security: Empirical Evidences from Three Villages of Tamil Nadu. In *Indian Journal of Agricultural Economics*, pp. 493-503.
140. Islam, GMT, Islam, A, Shopan, AA, Rahman, MM, Lázár, AN & Mukhopadhyay, A. 2015 Implications of agricultural land use change to ecosystem services in the Ganges delta. *J Environ Manage* **161**, 443-452. (doi:10.1016/j.jenvman.2014.11.018).
141. Visser, O. 2017 Running out of farmland? Investment discourses, unstable land values and the sluggishness of asset making. *Agric Human Values* **34**, 185-198. (doi:10.1007/s10460-015-9679-7).
142. Cottrell, RS, Nash, KL, Halpern, BS, Remenyi, TA, Corney, SP, Fleming, A, Fulton, EA, Hornborg, S, Johne, A, Watson, RA, et al. 2019 Food production shocks across land and sea. *Nature Sustainability* **2**, 130-137. (doi:10.1038/s41893-018-0210-1).
143. Prishchepov, AV, Müller, D, Dubinin, M, Baumann, M & Radeloff, VC. 2013 Determinants of agricultural land abandonment in post-Soviet European Russia. *Land Use Pol.* **30**, 873-884. (doi:10.1016/j.landusepol.2012.06.011).
144. Lambin, EF & Meyfroidt, P. 2011 Global land use change, economic globalization, and the looming land scarcity. *Proceedings of the National Academy of Sciences* **108**, 3465-3472. (doi:10.1073/pnas.1100480108).
145. Kaz'min, MA. 2016 Transformation of agricultural land use in Russian regions in the course of modern socioeconomic reforms. *Regional Research of Russia* **6**, 87-94. (doi:10.1134/S2079970516010056).
146. Liefert, WM, Liefert, O, Vocke, G & Allen, EW. 2010 Former Soviet Union Region To Play Larger Role in Meeting World Wheat Needs. In *Amber Waves*, pp. 12-19.
147. Meyfroidt, P, Schierhorn, F, Prishchepov, AV, Müller, D & Kuemmerle, T. 2016 Drivers, constraints and trade-offs associated with recultivating abandoned cropland in Russia, Ukraine and Kazakhstan. *Global Environmental Change* **37**, 1-15. (doi:10.1016/j.gloenvcha.2016.01.003).
148. Schierhorn, F, Müller, D, Beringer, T, Prishchepov, AV, Kuemmerle, T & Balmann, A. 2013 Post-Soviet cropland abandonment and carbon sequestration in European Russia, Ukraine, and Belarus. *Global Biogeochemical Cycles* **27**, 1175-1185. (doi:10.1002/2013gb004654).
149. Rozakis, S & Borek, R. 2018 Evaluation of agricultural reactivation on abandoned lands in Poland. *AgBioForum* **21**, 135-152.
150. Hobbs, RJ & Cramer, VA. 2012 *Old Fields: Dynamics and Restoration of Abandoned Farmland*, Island Press.
151. Yang, Y, Hobbie, SE, Hernandez, RR, Fargione, J, Grodsky, SM, Tilman, D, Zhu, Y-G, Luo, Y, Smith, TM, Jungers, JM, et al. 2020 Restoring Abandoned Farmland to Mitigate Climate Change on a Full Earth. *One Earth* **3**, 176-186. (doi:10.1016/j.oneear.2020.07.019).

152. Bastin, J-F, Finegold, Y, Garcia, C, Mollicone, D, Rezende, M, Routh, D, Zohner, CM & Crowther, TW. 2019 The global tree restoration potential. *Science* **365**, 76-79. (doi:10.1126/science.aax0848).
153. Papworth, T. 2015 The green noose. *Adam Smith Institute* **14**.
154. Amati, M & Taylor, L. 2010 From Green Belts to Green Infrastructure. *Planning Practice & Research* **25**, 143-155. (doi:10.1080/02697451003740122).
155. Cheshire, P. 2014 Turning houses into gold: the failure of British planning. Centre for Economic Performance, LSE.
156. CPRE. 2000 Valuing the land: planning for the best and most versatile agricultural land.
157. Defra. 2020 Farming for the future: Policy and progress update. UK Government.
158. Defra. 2011 Defra Soil Research Programme: Review of the weight that should be given to the protection of best and most versatile (BMV) land.
159. Shortly & A. (Greenbelt Foundation, Toronto), Personal Communication.
160. Carter-Whitney, M, Esakin, TC & Canadian Institute for Environmental Law and, P. 2010 Ontario's greenbelt in an international context. Toronto, Ont., Canadian Institute for Environmental Law and Policy.
161. Loghrin, H. Forthcoming Global Greenbelts Updates since 2009 (2019). Toronto, Greenbelt Foundation.
162. OECD. 2009 Farmland Conversion – The Spatial Implications of Agricultural and Land-use Policies. ed. D Diakosavvas. Paris, OECD.
163. Perrin, C, Clément, C, Melot, R & Nougaredes, B. 2020 Preserving Farmland on the Urban Fringe: A Literature Review on Land Policies in Developed Countries. *Land* **9**, 223.
164. OECD. 2010 *Regional Development Policies in OECD Countries.*, OECD Publishing.
165. Nishi, M. 2019 Multi-Level Governance of Agricultural Land in Japan: Farmers' Perspectives and Responses to Farmland Banking. New York, Columbia.
166. Oberndorf, RB. 2012 Legal review of recently enacted farmland law and vacant, fallow and virgin lands management law: improving the legal & policy frameworks relating to land management in Myanmar. *Food Security Working Group and Land Core Group*.
167. Fernández-Maldonado, AM. 2019 Unboxing the Black Box of Peruvian Planning. *Planning Practice & Research* **34**, 368-386. (doi:10.1080/02697459.2019.1618596).
168. Slätmo, E. - Preservation of Agricultural Land as an Issue of Societal Importance. - **4**, -.
169. Jain, M. 2018 Contemporary urbanization as unregulated growth in India: The story of census towns. *Cities* **73**, 117-127. (doi:10.1016/j.cities.2017.10.017).
170. Sili, M & Soumoulou, L. 2011 The issue of land in Argentina: Conflicts and dynamics of use, holdings and concentration. Rome, IFAD.

171. Robson, JS, Ayad, HM, Wasfi, RA & El-Geneidy, AM. 2012 Spatial disintegration and arable land security in Egypt: A study of small- and moderate-sized urban areas. *Habitat International* **36**, 253-260. (doi:10.1016/j.habitatint.2011.10.001).
172. Dai, W & Dai, W. 2019 Effects of urban expansion on environment by morphological study. *IOP Conference Series: Earth and Environmental Science* **227**, 052004. (doi:10.1088/1755-1315/227/5/052004).
173. Agrimonde-Terra. 2018 *Land Use and Food Security in 2050: a Narrow Road*, éditions Quae.
174. Bezbradica, L, Pantić, M & Gajić, A. 2019 The land use and soil protection: Planning and legal regulations in Serbia. *Zemljište i biljka* **68**, 51-71. (doi:10.5937/ZemBilj1902051B).
175. Owusu Ansah, B & Chigbu, UE. 2020 The Nexus between Peri-Urban Transformation and Customary Land Rights Disputes: Effects on Peri-Urban Development in Trede, Ghana. *Land* **9**, 187.
176. Jiang, L & Zhang, Y. 2016 Modeling Urban Expansion and Agricultural Land Conversion in Henan Province, China: An Integration of Land Use and Socioeconomic Data. *Sustainability* **8**, 920.
177. Li, S. 2018 Change detection: how has urban expansion in Buenos Aires metropolitan region affected croplands. *International Journal of Digital Earth* **11**, 195-211. (doi:10.1080/17538947.2017.1311954).
178. Carreño, L, Frank, FC & Viglizzo, EF. 2012 Tradeoffs between economic and ecosystem services in Argentina during 50 years of land-use change. *Agriculture, Ecosystems & Environment* **154**, 68-77. (doi:10.1016/j.agee.2011.05.019).
179. Żróbek-Różańska, A & Zielińska-Szczepkowska, J. 2019 National Land Use Policy against the Misuse of the Agricultural Land—Causes and Effects. Evidence from Poland. *Sustainability* **11**, 6403.
180. McPike, JL. 2015 Creating Space for the Formal Amongst the Informal: An Examination of Urban Housing Policies, State Power, and Multi-Scalar Politics in Indian Cities. In *RC21 International Conference on "The Ideal City: between myth and reality. Representations, policies, contradictions and challenges for tomorrow's urban life" Urbino (Italy) 27-29 August 2015*.
181. Bartz, D. 2015 *Soil atlas: Facts and figures about earth, land and fields*. Berlin.
182. Appiah, DO, Asante, F & Nketiah, B. 2019 Perspectives on Agricultural Land Use Conversion and Food Security in Rural Ghana. *Sci* **1**, 14.
183. Schueler, V, Kuemmerle, T & Schröder, H. 2011 Impacts of Surface Gold Mining on Land Use Systems in Western Ghana. *AMBIO* **40**, 528-539. (doi:10.1007/s13280-011-0141-9).
184. Franco, J & Borras Jr, SM. 2013 *Land concentration, land grabbing and people's struggles in Europe*. Transnational Institute, Amsterdam.
185. Khalid, H & Yusuf, MD. 2012 Resource management: Fragmentation of land ownership and its impact on sustainability of agriculture.
186. Daniels, T & Lapping, M. 2005 Land Preservation: An Essential Ingredient in Smart Growth. *Journal of Planning Literature* **19**, 316-329. (doi:10.1177/0885412204271379).

187. Desrousseaux, M, Schmitt, B, Billet, P, Béchet, B, Le Bissonnais, Y & Ruas, A. 2019 Artificialised Land and Land Take: What Policies Will Limit Its Expansion and/or Reduce Its Impacts? In *International Yearbook of Soil Law and Policy 2018* (eds. H Ginzky, E Dooley, IL Heuser, E Kasimbazi, T Markus & T Qin), pp. 149-165. Cham, Springer International Publishing).
188. Hellerstein, DR. 2002 *Farmland protection : the role of public preferences for rural amenities*. Washington, D.C., U.S. Dept. of Agriculture, Economic Research Service.
189. Gosnell, H, Kline, JD, Chrostek, G & Duncan, J. 2011 Is Oregon's land use planning program conserving forest and farm land? A review of the evidence. *Land Use Pol.* **28**, 185-192. (doi:10.1016/j.landusepol.2010.05.012).
190. Ludlow, D, Falconi, M, Carmichael, L, Croft, N, Di Leginio, M, Fumanti, F, Sheppard, A & Smith, N. 2013 Land Planning and Soil Evaluation Instruments in EEA Member and Cooperating Countries (with inputs from Eionet NRC Land Use and Spatial Planning). Final Report for EEA from ETC/SIA (EEA project managers: G. Louwagie and G. Dige). Available at: <http://www.eea.europa.eu/themes/landuse/document-library>. EEA & ETC/SIA.
191. Grass, I, Loos, J, Baensch, S, Batáry, P, Librán-Embíd, F, Ficiciyan, A, Klaus, F, Riechers, M, Rosa, J, Tiede, J, et al. 2019 Land-sharing/-sparing connectivity landscapes for ecosystem services and biodiversity conservation. *People and Nature* **1**, 262-272. (doi:10.1002/pan3.21).
192. Huang, Q, Lu, J, Li, M, Chen, Z & Li, F. 2015 Developing Planning Measures to Preserve Farmland: A Case Study from China. *Sustainability* **7**, 13011-13028.
193. 2003 Law for the preservation of the agricultural lands. ed. I Government.
194. Sartori, D, Catalano, G, Genco, M, Pancotti, C, Sirtori, E, Vignetti, S & Bo, C. 2014 Guide to Cost-benefit Analysis of Investment Projects. Economic appraisal tool for Cohesion Policy 2014-2020.
195. Frazer, JG. 1926 The Worship of Nature. *Nature* **118**, 4-5. (doi:10.1038/118004a0).
196. Malczewski, J. 2004 GIS-based land-use suitability analysis: a critical overview. *Progress in Planning* **62**, 3-65. (doi:10.1016/j.progress.2003.09.002).
197. Dent, D & Young, A. 1981 *Soil Survey and Land Evaluation*, Allen & Unwin.
198. Baveye, PC, Baveye, J & Gowdy, J. 2016 Soil "Ecosystem" Services and Natural Capital: Critical Appraisal of Research on Uncertain Ground. *Frontiers in Environmental Science* **4**. (doi:10.3389/fenvs.2016.00041).
199. Bouwman, AF & Sombroek, WG. 1990 Inputs to climatic change by soils and agriculture related activities: Present status and possible future trends. In *Soils on a Warmer Earth* (pp. 15 - 30. Amsterdam, Elsevier).
200. Bouwman, AF. 1990 Land use related sources of greenhouse gases: Present emissions and possible future trends. *Land Use Pol.* **7**, 154-164. (doi:10.1016/0264-8377(90)90006-K).
201. Batjes, NH & Bridges, EM. 1992 *A Review of Soil Factors and Processes that Control Fluxes of Heat, Moisture and Greenhouse Gases*. Wageningen, The Netherlands., ISRIC.

202. Wall, DH. 2004 *Sustaining Biodiversity and Ecosystem Services in Soils and Sediments*, Island Press.
203. Karlen, DL, Mausbach, MJ, Doran, JW, Cline, RG, Harris, RF & Schuman, GE. 1997 Soil Quality: A Concept, Definition, and Framework for Evaluation (A Guest Editorial). *Soil Science Society of America Journal* **61**, 4-10. (doi:10.2136/sssaj1997.03615995006100010001x).
204. Hannam, I & Boer, B. 2002 Legal and institutional frameworks for sustainable soils.
205. Costanza, R, d'Arge, R, de Groot, R, Farber, S, Grasso, M, Hannon, B, Limburg, K, Naeem, S, O'Neill, RV, Paruelo, J, et al. 1997 The value of the world's ecosystem services and natural capital. *Nature* **387**, 253-260. (doi:10.1038/387253a0).
206. Breure, AM, De Deyn, GB, Dominati, E, Eglin, T, Hedlund, K, Van Orshoven, J & Posthuma, L. 2012 Ecosystem services: a useful concept for soil policy making! *Current Opinion in Environmental Sustainability* **4**, 578-585. (doi:10.1016/j.cosust.2012.10.010).
207. Lescourret, F, Magda, D, Richard, G, Adam-Blondon, A-F, Bardy, M, Baudry, J, Doussan, I, Dumont, B, Lefèvre, F, Litrico, I, et al. 2015 A social–ecological approach to managing multiple agro-ecosystem services. *Current Opinion in Environmental Sustainability* **14**, 68-75. (doi:10.1016/j.cosust.2015.04.001).
208. Robinson, DA, Hockley, N, Cooper, DM, Emmett, BA, Keith, AM, Lebron, I, Reynolds, B, Tipping, E, Tye, AM, Watts, CW, et al. 2013 Natural capital and ecosystem services, developing an appropriate soils framework as a basis for valuation. *Soil Biology and Biochemistry* **57**, 1023-1033. (doi:10.1016/j.soilbio.2012.09.008).
209. Ellili-Bargaoui, Y, Walter, C, Lemercier, B & Michot, D. 2021 Assessment of six soil ecosystem services by coupling simulation modelling and field measurement of soil properties. *Ecological Indicators* **121**, 107211. (doi:10.1016/j.ecolind.2020.107211).
210. Dominati, E, Mackay, A, Green, S & Patterson, M. 2014 A soil change-based methodology for the quantification and valuation of ecosystem services from agro-ecosystems: A case study of pastoral agriculture in New Zealand. *Ecological Economics* **100**, 119-129. (doi:10.1016/j.ecolecon.2014.02.008).
211. Daniels, T. 2020 (Professor of City and Regional Planning, Weitzman School of Design, University of Pennsylvania), Personal communication.
212. Defra. 2020 EIA (Agriculture) regulations: apply to make changes to rural land.
213. 2019 Scottish Government, Guidance on consideration of soil in Strategic Environmental Assessment , v5, 5.4.19.
214. Caldwell, WJ, Hilts, S & Wilton, B. 2017 *Farmland Preservation: Land for Future Generations*, University of Manitoba Press.
215. Nolon, J. 2003 Land Preservation. *Pace Law Faculty Publications*.
216. Churchman, GJ & Landa, E. 2014 *The soil underfoot : infinite possibilities for a finite resource*, CRC Press.

217. Meadows, DH. 1972 *The Limits to growth; a report for the Club of Rome's project on the predicament of mankind*, New York : Universe Books, [1972].
218. Norström, AV, Cvitanovic, C, Löf, MF, West, S, Wyborn, C, Balvanera, P, Bednarek, AT, Bennett, EM, Biggs, R, de Bremond, A, et al. 2020 Principles for knowledge co-production in sustainability research. *Nature Sustainability* **3**, 182-190. (doi:10.1038/s41893-019-0448-2).
219. Erinosh, BT. 2013 The Revised African Convention on the Conservation of Nature and Natural Resources: Prospects for a Comprehensive Treaty for the Management of Africa's Natural Resources. *African Journal of International and Comparative Law* **21**, 378-397. (doi:10.3366/ajicl.2013.0069).
220. Wilson, C & Matthews, W. 1970 Man's Impact on the Global Environment. Report of the Study of Critical Environmental Problems (SCEP). *MIT Press, Cambridge, Massachusetts* **16**, 19.
221. Lele, S, Springate-Baginski, O, Lakerveld, R, Deb, D & Dash, P. 2013 Ecosystem Services: Origins, Contributions, Pitfalls, and Alternatives. *Conservation and Society* **11**, 343-358. (doi:10.4103/0972-4923.125752).
222. Boer, B, Ginzky, H & Heuser, I. 2017 International Soil Protection Law: History, Concepts and Latest Developments. (pp. 49-72).
223. Bodle, R, Stockhaus, H, Wolff, F, Scherf, C-S & Oberthür, S. 2019 *Improving international soil governance - Analysis and recommendations*.
224. FAO. 1971 *Land degradation*. Rome, Food and Agriculture Organization of the United Nations.
225. FAO. 1971 *Legislative principles of soil conservation*.
226. Europe, Co. 1972 European Soil Charter. ed. Co Europe. Brussels.
227. Tóth, Z. 2018 International dimensions of EU soil policy – The main binding and non-binding legal instruments. *Hungarian Journal of Legal Studies Acta Juridica Hungarica* **59**, 290. (doi:10.1556/2052.2018.59.3.4).
228. FAO. 1981 World Soil Charter.
229. UNEP. 1982 World Soils Policy.
230. Wood & Harold, W. 1985 The United Nations World Charter for Nature: The Developing Nations' Initiative to Establish Protections for the Environment. *Ecology Law Quarterly* **12**, 977.
231. Byron-Cox, R. 2020 From Desertification to Land Degradation Neutrality: The UNCCD and the Development of Legal Instruments for Protection of Soils. In *Legal Instruments for Sustainable Soil Management in Africa* (eds. H Yahyah, H Ginzky, E Kasimbazi, R Kibugi & OC Ruppel), pp. 1-13. Cham, Springer International Publishing).
232. World Commission on Environment and Development. 1987 Our common future. Oxford; New York, Oxford University Press.
233. World Bank. 2002 *The First Decade of the GEF: second overall performance study*. Washington DC, World Bank.

234. Wolff, F & Kaphengst, T. 2017 The UN Convention on Biological Diversity and Soils: Status and Future Options. In *International Yearbook of Soil Law and Policy 2016* (eds. H Ginzky, IL Heuser, T Qin, OC Ruppel & P Wegerdt), pp. 129-148. Cham, Springer International Publishing).
235. United Nations Framework Convention on Climate Change, 9 May 1992 (in force 21 March 1994), 31 ILM 849; 1771 UNTS 10 7. (“UNFCCC”). Available at: <https://www.cbd.int/doc/c/f25f/ac08/fac2443375cab303ef45c22/sbstta-24-07-en.pdf>.
236. Global Environment Facility (GEF). 1999 Clarifying Linkages Between Land Degradation And The GEF Focal Areas: An Action Plan For Enhancing GEF Support: An Action Plan for Enhancing GEF Support, GEF/C.14/4, November 17, 1999. Washington DC, World Bank Group.
237. Hannam, I & Boer, B. 2019 Land degradation and international environmental law. (pp. 429-440).
238. FAO and UNEP. 2020 *Legislative approaches to sustainable agriculture and natural resources governance*. FAO Legislative Study No. 114. Rome, Food and Agriculture Organization of the United Nations.
239. Chasek, P, Safriel, U, Shikongo, S & Fuhrman, VF. 2015 Operationalizing Zero Net Land Degradation: The next stage in international efforts to combat desertification? *Journal of Arid Environments* **112**, 5-13. (doi:10.1016/j.jaridenv.2014.05.020).
240. zu Ermgassen, SOSE, Baker, J, Griffiths, RA, Strange, N, Struebig, MJ & Bull, JW. 2019 The ecological outcomes of biodiversity offsets under “no net loss” policies: A global review. *Conservation Letters* **12**, e12664. (doi:10.1111/conl.12664).
241. UK Government. 2020 Environment Act 2020. UK.
242. United Nations Framework Convention on Climate Change, 9 May 1992 (in force 21 March 1994), 31 ILM 849 ; 1771 UNTS 10 7. (“UNFCCC”).
243. Fee, E. 2019 Implementing the Paris Climate Agreement: Risks and Opportunities for Sustainable Land Use. In *International Yearbook of Soil Law and Policy 2018* (eds. H Ginzky, E Dooley, IL Heuser, E Kasimbazi, T Markus & T Qin), pp. 249-270. Cham, Springer International Publishing).
244. Convention on Biological Diversity. 2020 Convention on Biological Diversity: Review of the International Initiative for the Conservation and Sustainable Use of Soil Biodiversity and Updated Plan of Action.
245. FAO, ITPS, GSBI, SCBD & EC. 2020 State of knowledge of soil biodiversity - Status, challenges and potentialities: Report 2020. FAO.
246. Streck, C & Gay, A. 2017 The Role of Soils in International Climate Change Policy. (pp. 105-128).
247. UNFCCC. 2020 Issues related to agriculture.
248. Rojas, RV & Caon, L. 2016 The international year of soils revisited: promoting sustainable soil management beyond 2015. *Environmental Earth Sciences* **75**. (doi:10.1007/s12665-016-5891-z).
249. FAO. 2015 Revised World Soil Charter. Rome, Italy.
250. FAO. 2017 Voluntary Guidelines for Sustainable Soil Management.

251. Orr, B, Cowie, A, Castillo Sanchez, V, Chasek, P, Crossman, N, Erlewein, A, Louwagie, G, Maron, M, Metternicht, G & Minelli, S. 2017 Scientific conceptual framework for land degradation neutrality. In *A report of the science-policy interface. United Nations Convention to Combat Desertification (UNCCD), Bonn, Germany*, pp. 1-98.
252. International Law Association. 2020 ILA Guidelines on the Role of International Law in Sustainable Natural Resources Management for Development, Resolution 4/2020, 2020 Report of the Seventy Ninth Conference, Kyoto (forthcoming).
253. Minasny, B, Malone, BP, McBratney, AB, Angers, DA, Arrouays, D, Chambers, A, Chaplot, V, Chen, Z-S, Cheng, K, Das, BS, et al. 2017 Soil carbon 4 per mille. *Geoderma* **292**, 59-86. (doi:10.1016/j.geoderma.2017.01.002).
254. Chabbi, A, Lehmann, J, Ciais, P, Loescher, HW, Cotrufo, MF, Don, A, SanClements, M, Schipper, L, Six, J, Smith, P, et al. 2017 Aligning agriculture and climate policy. *Nature Climate Change* **7**, 307-309. (doi:10.1038/nclimate3286).
255. Poulton, P, Johnston, J, Macdonald, A, White, R & Powlson, D. 2018 Major limitations to achieving “4 per 1000” increases in soil organic carbon stock in temperate regions: Evidence from long-term experiments at Rothamsted Research, United Kingdom. *Glob. Change Biol.* **24**, 2563-2584. (doi:10.1111/gcb.14066).
256. Minasny, B, Arrouays, D, McBratney, AB, Angers, DA, Chambers, A, Chaplot, V, Chen, Z-S, Cheng, K, Das, BS, Field, DJ, et al. 2018 Rejoinder to Comments on Minasny et al., 2017 Soil carbon 4 per mille *Geoderma* **292**, 59–86. *Geoderma* **309**, 124-129. (doi:10.1016/j.geoderma.2017.05.026).
257. Soussana, J-F, Lutfalla, S, Ehrhardt, F, Rosenstock, T, Lamanna, C, Havlík, P, Richards, M, Lini, E, Wollenberg, E, Chotte, J-L, et al. 2019 Matching policy and science: Rationale for the '4 per 1000-soils for food security and climate' initiative.
258. Kibugi, R. 2018 Soil Health, Sustainable Land Management and Land Degradation in Africa: Legal Options on the Need for a Specific African Soil Convention or Protocol. In *International Yearbook of Soil Law and Policy 2017* (eds. H Ginzky, E Dooley, IL Heuser, E Kasimbazi, T Markus & T Qin), pp. 387-411. Cham, Springer International Publishing).
259. Markus, T. 2017 The Alpine Convention’s Soil Conservation Protocol: A Model Regime? In *International Yearbook of Soil Law and Policy 2016* (eds. H Ginzky, IL Heuser, T Qin, OC Ruppel & P Wegerdt), pp. 149-164. Cham, Springer International Publishing).
260. 2003 *Revised European Charter for the Protection and Sustainable Management of Soil*, Strasbourg, 17 July 2003 CO-DBP/documents/codbp2003/10e.
261. European Commission. 2006 Soil protection - The story behind the Strategy.
262. Stankovics, P, Tóth, G & Tóth, Z. 2018 Identifying Gaps between the Legislative Tools of Soil Protection in the EU Member States for a Common European Soil Protection Legislation. *Sustainability* **10**, 2886. (doi:10.3390/su10082886).
263. Montanarella, L & Panagos, P. 2021 The relevance of sustainable soil management within the European Green Deal. *Land Use Pol.* **100**, 104950. (doi:10.1016/j.landusepol.2020.104950).

264. Knox, JH. 2017 Report of the Special Rapporteur on the Issue of Human Rights Obligations Relating to the Enjoyment of a Safe, Clean, Healthy and Sustainable Environment :note. Geneva, UN.
265. UN. Human Rights Council (39th sess.: 2018: Geneva). 2018 United Nations Declaration on the Rights of Peasants and Other People Working in Rural Areas :resolution. Geneva, UN.
266. Fromherz, NA. 2012 The Case for a Global Treaty on Soil Conservation, Sustainable Farming, and the Preservation of Agrarian Culture. **39**. (doi:10.15779/Z38BC49).
267. De Schutter, O. 2010 The Emerging Human Right to Land. *International Community Law Review* **12**, 303-334. (doi:10.1163/187197310X513725).
268. Stec, S & Jendroška, J. 2019 The Escazú Agreement and the Regional Approach to Rio Principle 10: Process, Innovation, and Shortcomings. *Journal of Environmental Law* **31**, 533-545. (doi:10.1093/jel/eqz027).
269. Keesstra, SD, Bouma, J, Wallinga, J, Tittonell, P, Smith, P, Cerdà, A, Montanarella, L, Quinton, JN, Pachepsky, Y, van der Putten, WH, et al. 2016 The significance of soils and soil science towards realization of the United Nations Sustainable Development Goals. *SOIL* **2**, 111-128. (doi:10.5194/soil-2-111-2016).
270. Gil, JDB, Reidsma, P, Giller, K, Todman, L, Whitmore, A & van Ittersum, M. 2019 Sustainable development goal 2: Improved targets and indicators for agriculture and food security. *Ambio* **48**, 685-698. (doi:10.1007/s13280-018-1101-4).
271. Tóth, G, Hermann, T, da Silva, MR & Montanarella, L. 2018 Monitoring soil for sustainable development and land degradation neutrality. *Environmental Monitoring and Assessment* **190**, 57. (doi:10.1007/s10661-017-6415-3).
272. Shen, X, Wang, X, Zhang, Z, Lu, Z & Lv, T. 2019 Evaluating the effectiveness of land use plans in containing urban expansion: An integrated view. *Land Use Pol.* **80**, 205-213. (doi:10.1016/j.landusepol.2018.10.001).
273. Roose, A, Kull, A, Gauk, M & Tali, T. 2013 Land use policy shocks in the post-communist urban fringe: A case study of Estonia. *Land Use Pol.* **30**, 76-83. (doi:10.1016/j.landusepol.2012.02.008).
274. Lu, Y, Song, S, Wang, R, Liu, Z, Meng, J, Sweetman, AJ, Jenkins, A, Ferrier, RC, Li, H, Luo, W, et al. 2015 Impacts of soil and water pollution on food safety and health risks in China. *Environment International* **77**, 5-15. (doi:10.1016/j.envint.2014.12.010).
275. Zhao, F-J, Ma, Y, Zhu, Y-G, Tang, Z & McGrath, SP. 2015 Soil Contamination in China: Current Status and Mitigation Strategies. *Environmental Science & Technology* **49**, 750-759. (doi:10.1021/es5047099).
276. Chen, R, de Sherbinin, A, Ye, C & Shi, G. 2014 China's Soil Pollution: Farms on the Frontline. *Science* **344**, 691-691. (doi:10.1126/science.344.6185.691-a).
277. Vogel, H-J, Eberhardt, E, Franko, U, Lang, B, Ließ, M, Weller, U, Wiesmeier, M & Wollschläger, U. 2019 Quantitative Evaluation of Soil Functions: Potential and State. *Frontiers in Environmental Science* **7**. (doi:10.3389/fenvs.2019.00164).

278. Vargas, L, Willemen, L & Hein, L. 2019 Assessing the Capacity of Ecosystems to Supply Ecosystem Services Using Remote Sensing and An Ecosystem Accounting Approach. *Environmental Management* **63**, 1-15. (doi:10.1007/s00267-018-1110-x).
279. Ostle, NJ, Levy, PE, Evans, CD & Smith, P. 2009 UK land use and soil carbon sequestration. *Land Use Pol.* **26**, S274-S283. (doi:10.1016/j.landusepol.2009.08.006).
280. Buschmann, C, Röder, N, Berglund, K, Berglund, Ö, Lærke, PE, Maddison, M, Mander, Ü, Myllys, M, Osterburg, B & van den Akker, JJH. 2020 Perspectives on agriculturally used drained peat soils: Comparison of the socioeconomic and ecological business environments of six European regions. *Land Use Pol.* **90**, 104181. (doi:10.1016/j.landusepol.2019.104181).
281. Committee on Climate Change. 2020 Land use: Policies for a Net Zero UK.
282. Kibblewhite, MG, Ritz, K & Swift, MJ. 2008 Soil health in agricultural systems. *Philos Trans R Soc Lond B Biol Sci* **363**, 685-701. (doi:10.1098/rstb.2007.2178).
283. Powlson, DS, Gregory, PJ, Whalley, WR, Quinton, JN, Hopkins, DW, Whitmore, AP, Hirsch, PR & Goulding, KWT. 2011 Soil management in relation to sustainable agriculture and ecosystem services. *Food Policy* **36, Supplement 1**, S72-S87. (doi:10.1016/j.foodpol.2010.11.025).
284. Paustian, K, Lehmann, J, Ogle, S, Reay, D, Robertson, GP & Smith, P. 2016 Climate-smart soils. *Nature* **532**, 49. (doi:10.1038/nature17174).
285. LaCanne, CE & Lundgren, JG. 2018 Regenerative agriculture: merging farming and natural resource conservation profitably. *PeerJ* **6**, e4428-e4428. (doi:10.7717/peerj.4428).
286. Lal, R & Kosaki, T. 2018 *Soils and Sustainable Development Goals. GeoEcology Essay*, Schweizerbart Science Publishers.
287. Strouhalová - Vysloužilová, B, Ertlen, D, Schwartz, D & Šefrna, L. 2016 Chernozem. From concept to classification: a review. *AUC GEOGRAPHICA* **51**, 85-95. (doi:10.14712/23361980.2016.8).
288. Amundson, R. 2020 The policy challenges to managing global soil resources. *Geoderma* **379**, 114639. (doi:10.1016/j.geoderma.2020.114639).
289. Frelh-Larsen, A, Bowyer, C, Albrecht, S, Keenleyside, C, Kemper, M, Nanni, S, Naumann, S, Mottershead, RD, Langrebe, R, Andersen, E, et al. 2017 *Updated Inventory and Assessment of Soil Protection Policy Instruments in EU Member States*.
290. Ronchi, S, Salata, S, Arcidiacono, A, Piroli, E & Montanarella, L. 2019 Policy instruments for soil protection among the EU member states: A comparative analysis. *Land Use Pol.* **82**, 763-780. (doi:10.1016/j.landusepol.2019.01.017).
291. Wingeyer, AB, Amado, TJC, Pérez-Bidegain, M, Studdert, GA, Varela, CHP, Garcia, FO & Karlen, DL. 2015 Soil Quality Impacts of Current South American Agricultural Practices. *Sustainability* **7**, 2213-2242.
292. Webb, A, Kelly, G & Dougherty, W. 2015 Soil governance in the agricultural landscapes of New South Wales, Australia. *International Journal of Rural Law and Policy Special Edition* **1**, 1-16. (doi:10.5130/ijrlp.i1.2015.4169).

293. Chukov, SN & Yakovlev, AS. 2019 Soil and Land Categories in the Modern Legislation of Russia. *Eurasian Soil Science* **52**, 865-870. (doi:10.1134/S1064229319070020).
294. IUCN. 2019 World Commission on Environmental Law, Soil Desertification & Sustainable Agriculture Specialist Group, 2019 Mid-Year Report. Online.
295. Hazelton, PA, Frossard, E, Blum, WEH & Warkentin, BP. 2006 Australian examples of the role of soils in environmental problems. In *Function of Soils for Human Societies and the Environment* (p. 0, Geological Society of London).
296. Sophia Antipolis. 2003 Threats to Soils in Mediterranean Countries: Document Review. In *Plan Bleu Papers*. Valbonne, France, Plan Bleu.
297. Gonzalez Lago, M, Plant, R & Jacobs, B. 2019 Re-politicising soils: What is the role of soil framings in setting the agenda? *Geoderma* **349**, 97-106. (doi:10.1016/j.geoderma.2019.04.021).
298. Heuser, I. 2018 Development of Soil Awareness in Europe and Other Regions: Historical and Ethical Reflections About European (and International) Soil Protection Law. (pp. 451-474).
299. Moallemi, EA, Haan, F, Hadjidakou, M, Khatami, S, Malekpour, S, Smajgl, A, Stafford Smith, M, Voinov, A, Bandari, R, Lamichhane, P, et al. 2021 Evaluating Participatory Modeling Methods for Co-creating Pathways to Sustainability. *Earth's Future* **9**. (doi:10.1029/2020EF001843).
300. Koch, A, McBratney, A, Adams, M, Field, D, Hill, R, Crawford, J, Minasny, B, Lal, R, Abbott, L, O'Donnell, A, et al. 2013 Soil Security: Solving the Global Soil Crisis. *Global Policy* **4**, 434-441. (doi:10.1111/1758-5899.12096).
301. McBratney, A, Field, DJ & Koch, A. 2014 The dimensions of soil security. *Geoderma* **213**, 203-213. (doi:10.1016/j.geoderma.2013.08.013).
302. Bouma, J. 2020 Soil security as a roadmap focusing soil contributions on sustainable development agendas. *Soil Security* **1**, 100001. (doi:10.1016/j.soisec.2020.100001).
303. Hill, R. 2017 The Place of Soil in International Government Policy. In *Global Soil Security* (eds. DJ Field, CLS Morgan & AB McBratney), pp. 443-449. Cham, Springer International Publishing).
304. Field, DJ, Morgan, CLS & McBratney, AB. 2017 *Global Soil Security*, Springer International Publishing, A. G.
305. Lilburne, L, Eger, A, Mudge, P, Ausseil, A-G, Stevenson, B, Herzig, A & Beare, M. 2020 The Land Resource Circle: Supporting land-use decision making with an ecosystem-service-based framework of soil functions. *Geoderma* **363**, 114134. (doi:10.1016/j.geoderma.2019.114134).
306. Juerges, N, Hagemann, N & Bartke, S. 2018 A tool to analyse instruments for soil governance: the REEL-framework. *Journal of Environmental Policy & Planning* **20**, 617-631. (doi:10.1080/1523908X.2018.1474731).
307. Ginzky, H. 2020 Good Governance for Sustainable Management of Soil on National and International Level: How to Do It?

Appendix D

Associate Editor Comments to Author (Professor Brian Reid):

In keeping with the reviewer comments, the authors have made extensive revisions to improve the manuscript. However, two issues persist.

- Firstly, the use of emotive phrasing, for example, but not limited to: relentless loss; before getting to grips with the thorny issues; at the mercy of the creators and implementers; transformed and even devastated; this gradual fall from grace.
- Secondly, the use obtuse vocabulary, for example, but not limited to: embody the same axiomatic concept; one prescient promulgation of reduced tillage; vociferous advocates; eclectic think tank, the Club of Rome, published the now iconic.

Sometime both issues are conflated in parallel, for example, but not limited to: Something so eternal and fundamental as soil, which has even been revered as a deity in its own right [195], might have been expected to have acquired an aura of intrinsic value, but it was ostensibly this its extrinsic value that made it a prize for which it was worth fighting, or even dying. Why styling it out in this fashion might wash for a book it does not work for a journal manuscript. The authors need to take stock of their writing style, offer a more objective account and make the language more accessible to a global audience.

Authors' response: Many of these phrases have disappeared in the process of reducing the article size and we believe that all examples of obscure or emotive terminology have now been removed or replaced.

In addition, the manuscript is very long. There is 26 pages of preamble prefacing the core of the manuscript (i.e. the content that begins with jurisprudence). The authors need to scale back the first 26 pages. A reduction of a third can easily be achieved through pragmatism and simplifying long-winded statements. The later half of the manuscript, while better conceived, will similarly benefit from abridging. The authors need to cut back on the anecdotal content and repetition of points already made. The current length (clean version 44 pages excluding reference) needs to be reduced to a maximum of 32 pages (at this the paper would still be long, but manageable).

The manuscript has been radically cut and is now just under 32 pages.

Reviewer comments to Author:

Reviewer: 2

Comments to the Author(s)

The manuscript successfully assembles and provides a commentary on the evolution of a fuller appreciation of the importance of soil as a natural resource that requires specific governance. There are some gaps in addressing the legal framework for soil governance: for example the importance and availability of soil information, soil education (of citizens generally as well as land managers), the design of soil protection measures and their implementation (fiscal and other incentives as well as regulatory requirements). However, the manuscript provides a valuable commentary on the aspects that it does address. The difficulty of disentangling soil from land in the policy and legal context is identified and grappled with helpfully, although and inevitably, the focus of the discussion on existing initiatives is mainly land-centric because this is where existing soil protection is mostly found.

We do not feel that this comment requires further change, partly because the reviewer is making no specific request, but in particular because to fill the suggested "gaps" would require the article to expand in length. We also feel that these wider/more detailed aspects of governance are beyond the scope of the article, which is already very wide-ranging. However, we have added an acknowledgement of the importance of soil information and soil education for effective soil governance.

I am slightly uneasy that the manuscript lacks the necessary scientific emphasis appropriate to the journal but leave that to the editors' discernment.

We believe that the radical editing in this latest revision, including the removal of some emotive phrasing, has resulted in an article that is less discursive and anecdotal, and more directly and concisely addresses the scientific issues.

A few small edits are suggested as follows:

line 321 replace 'excluding' by exempting' and delete 'as ifdefault'. Insert new sentence: "The assumption being that the role of farmers in protecting the rural landscape made controls on agricultural land use unnecessary." **Completed**

line 384 'aeons' not 'eons' **This sentence has been deleted.**

line 384 "chronic soil degradation tends to be associated with rural poverty". This statement does not seem quite correct considering that intensive production in regions of higher wealth, for example of vegetable and root crops, is associated with serious soil degradation.

The reviewer is referring to line 406. There is evidence for degradation in both settings. The sentence now reads: "While acute soil degradation is often the result of breaches of regulations, chronic soil degradation tends to be associated with either intensive agriculture or rural poverty, and generally requires supportive rather than punitive legislation."

Saving the ground beneath our feet

Establishing priorities and criteria for governing soil use and protection

Lewis Peake^{1,2} and Cairo Robb³

¹ School of Environmental Science, University of East Anglia, Norwich, UK;

² Tyndall Centre for Climate Change Research, University of East Anglia, Norwich, UK;

³ Legal Research Fellow, Centre for International Sustainable Development Law, <https://www.cisd.org>;

corresponding author: l.peake@uea.ac.uk

Abstract

The continual loss and impairment of soil ecosystem services across the globe calls for a fundamental reconsideration of soil governance mechanisms. This critical synthesis charts the history and evolution of national and international soil law and seeks to unravel certain challenges that have contributed to this failure in governance. It describes and categorises law and policy responses to different soil threats, and identifies a worrying widespread absence of legislation for oversight and protection of agricultural soils from urbanisation, as well as a lack of clear legal mechanisms to determine national priorities for soil protection. A reduction in the world's prime farmland threatens soil ecosystem services, including food security, carbon storage, and biodiversity. Falling between the stalls of agricultural and environmental law, the fate of farmland is often left to planners who do not see themselves as responsible for soils. Consequently, legal instruments with the greatest power to affect soil, sometimes irreversibly, are often framed and worded with little or no reference to soil. Nevertheless, emerging conceptual frameworks might offer positive outcomes. The authors advocate robust holistic policies of soil governance and land use planning that place soil ecosystem services and natural capital at the heart of decision making.

Keywords

agricultural land conversion, farmland preservation, land take, soil ecosystem services, soil governance, soil security

Introduction

Soil is a vital multifunctional resource that could be regarded as the metaphorical as well as the literal foundation of human civilization. Soil ecosystem services (SES) [1], and the broader concept of soil-derived Nature's Contributions to People (NCP) [2], is a relatively new academic subdiscipline, but human awareness of the life-supporting role of soil is as old as civilization, if not older. Clearly, the ground beneath our feet provides so much more than the ground beneath our feet, and transcends title deeds, boundary fences and even national borders. Arguably, we are more dependent on soil than ever, as our dominance as a species has resulted in greater demands and stressors on our environment. As the world's population has soared and almost every parcel of its terrestrial surface has been assigned to the various national states, and then further subdivided and transformed by governments, private corporations or individuals, the need to safeguard this increasingly degraded resource is more urgent than ever.

In addition to the familiar and very direct role that soil has in providing our food, fibre and timber, we often forget its many other interconnected functions, for example:

- Physical: providing building or landscaping material, and a stable substrate for infrastructure;
- Chemical: regulating the supply of plant nutrients;
- Biological: constituting biodiverse ecosystems and communities of useful organisms, converting toxic sewage and decomposing matter into plant nutrients, and providing a genetic reservoir;
- Hydrological: facilitating water absorption, storage and filtration, and attenuating flooding;
- Cultural: preserving cultural heritage and archaeological information, and providing recreation;
- Climatic: storing carbon (C), regulating greenhouse gas (GHG) fluxes and heat buffering.

The last of these would have been almost inconceivable to our forebears, only assuming critical importance approximately since the beginning of the new millennium.

However, in certain circumstances soil, like any other natural resource, can deliver negative impacts, such as emitting GHGs, harbouring pests and diseases, or contributing to the accumulation of sediment, dust storms and landslides. These events are reminders that good governance of soil is also about minimising ecosystem "disservices" as well as safeguarding essential ecosystem services. We live in a time of anthropogenic environmental crisis, unprecedented climate change and species extinction. A report recently released by WWF states that: "Since 1970, our Ecological Footprint has exceeded the Earth's rate of regeneration," and currently exceeds the world's capacity to support humanity by 56% [3]. Soil is an integral component within this system, both as an enabler and a victim of exponential human expansion.

This interdisciplinary critical synthesis reviews the role of soil governance in helping to achieve the objectives of the UN Sustainable Development Goals (SDGs), to mitigate and adapt to climate change and

contribute to a more secure future for humanity. We trace the historical path of the status and governance of soil; categorise anthropogenic threats to soil, and legal mechanisms that can be applied to address them; and discuss the semantics surrounding legal aspects of soil and land, aiming to disambiguate and illuminate the terminology. The current state of art of soil governance is explored in depth and a key finding is a failure to prevent or barely acknowledge the worldwide loss of prime farmland and the implications of this for society. The review concludes by looking to emerging approaches such as the soil security conceptual framework, proposed as a way of translating critical soil knowledge into sustainable development policy, and to holistic land evaluation methodologies that incorporate SES into land use planning.

Methods

An open-ended literature review was conducted, using combinations of the following search terms: “agriculture”, “biodiversity”, “conservation”, “contamination”, “conversion”, “degradation”, “ecosystem services”, “erosion”, “farmland”, “governance”, “land”, “land take”, “land use planning”, “law”, “legislation”, “non-agricultural”, “policy”, “preservation”, “prime”, “protection”, “SDG”, urbanis(z)ation”, “urban sprawl”, “soil”, “soil sealing”, etc.. By selective analysis we synthesised key points, such as our own categorisations, thus adding new content. By presenting in this way, the intention was twofold: to provide the reader with some clear signposts within a large and complex domain; and to provide ourselves with a framework within which the context of any given issue could be better understood and deconstructed.

To complement data and cases studies from the literature, a comprehensive search was conducted to abstract data from FAOLEX, the UN Food and Agriculture Organization (FAO) online database of policies and laws [4]. FAOLEX documents legal instruments in force in nearly 200 countries. This database is organised by country, into the following groups, of which those marked with an asterisk were reviewed by the lead author (where they existed for the country in question): **Policies***; **Legislation: Agricultural and rural development***, *Climate change*, *Cultivated plants*, *Disaster risk management*, *Environment**, *Fisheries*, *Food and nutrition*, *Forestry**, *Land and soil**, *Livestock*, *Sea*, *Water*, *Wild species and ecosystems**; **International Agreements***. *Land and soil* was the most relevant, but it was necessary to search further, because in many cases this group is primarily devoted to the administration of land ownership and land reform. The groups are not mutually exclusive, so some legislation appears more than once.

The name of the FAOLEX entry alone was sometimes broadly sufficient to explain its meaning, e.g. “Soil conservation law”, but in many cases it was necessary to select the embedded hyperlink to access its summary (usually in English). Ambiguous or simplified wording often made it necessary to select a further link to the full text, either within FAOLEX or on the websites of national governments. These documents were often in the local language, sometimes in local script, and where necessary Google Translate was used. In some cases, individuals professionally familiar with the legislation were contacted for clarification and

asked whether the laws in question were implemented and effectively governed. Approximately 1% of countries had either created no such instruments or lacked any accessible data in FAOLEX. This information was compiled from the summer of 2020 until the summer of 2021. A spreadsheet was created with one row per country and columns indicating categories of legal instrument, and from this it was possible to calculate percentages of countries with such measures in force (Table 3). The FAOLEX groups are powerful retrieval mechanisms, but unrelated to the categories we derived. While our research was being conducted FAO was in the process of creating and populating a similar but soil-specific repository, SoiLEX, including data from FAOLEX¹. In terms of structural organisation and ease of use, SoiLEX is superior to FAOLEX but limited to highly soil-specific categories of legislation, whereas FAOLEX is wider in scope and, crucially for our purposes, contains laws relating to the protection or preservation of categories of land such as prime farmland and natural ecosystems.

Some caveats are necessary. Such a synthesis can only ever be approximate and is undoubtedly incomplete. Several polities listed in FAOLEX as countries actually comprise a number of countries (e.g. the UK), or separate provinces or states (e.g. the US), with differing laws. Many instruments, especially policies, are generalised, e.g.: “The land is the main national wealth, which is under special protection of the state...ensuring rational use and protection of land,” [5] iterated many times, but with little or no further qualification. A law stating that land will be classified as urban or rural could imply protected status or simply a degree of spatial planning; the latter was assumed unless explicitly otherwise. Furthermore, this is primarily a review of officially documented policies and laws, not a comprehensive assessment of enforcement or effective governance. While an overview of international soil law and policy context is provided, the synthesis does not deal in detail with states’ external relations or human and peoples’ rights relating to land and soil, nor does it address global or national soil information systems or soil education and literacy measures, all of which are also important aspects of comprehensive soil governance.

Soil as a valued resource and legal entity

The evolution of soil law

The ancient Egyptians called their nation state *Kemet* which means black land or dark earth, in contrast to the barren *Deshret*, or red land, stretching out on either side. The black silty clay of the Nile valley was not simply a metaphor for their country; in their minds, it *was* their country. Human dependence on productive

¹ SoiLEX is described as a global database on national legislation on soil protection, conservation and restoration to facilitate access to information on the existing legal instruments in force and bridge the gap between the various soil stakeholders.:

www.fao.org/soils-portal/soilex/en/

soil and the impact of its degradation on civilisation throughout history is well attested [6-10]. Mesopotamia experienced at least two catastrophic regime changes due respectively to soil salinity and soil erosion [11, 12]. The Greek and Roman writers make multiple references both to the benefits of soil quality and the threat of soil degradation, and by the 1500s in Europe soil was regarded as the key factor of an economy [11]. History is full of examples of land-related wars and conquests, often correlated with soil quality [13-15], and repression and killing of those seeking to protect land and soil quality continues today [16, 17].

The earliest known attempts to put an economic value on productive land were Chinese soil classification and maps created 2000 years BCE for taxation purposes [18]. Other ancient peoples also independently derived similar systems, for example in Mesoamerica [19]. However, while there are many historical examples of land ownership rights and transactions [20], it is difficult to pin down when laws or policies were first introduced to protect soil or land as a vulnerable or scarce resource in its own right, that is, part of the natural capital for the common good. The *Statutes on Agriculture*, compiled during the Qin Dynasty of the late 4th century BCE is China's, and possibly the world's, earliest known set of soil-related laws [21]. The *Statutes* were followed a century later, during the Han Dynasty (206 BCE-8 CE), by the *Book of Fan Shengzhi*, China's oldest surviving agronomic treatise which provides extraordinarily detailed instructions on most aspects of soil management and may well have been regarded by farmers as de facto law [22].

Perhaps the oldest surviving documented legislation explicitly addressing soil is that within the *Digesta* of the Eastern Roman emperor Justinian, published in the 6th century, but including laws dating from centuries earlier. While there was no overt implication that soil *per se* was a scarce resource in the Roman Empire, these edicts leave the reader in no doubt that soil was of great importance to everyone involved in its use, from the question of who benefited from its bounty to who was responsible for its maintenance [23].

In the modern era, the first specific instances of soil legislation were usually initiated in response to the threat of severe soil erosion, typically the result of inappropriate land management combined with extreme climatic conditions. The most iconic example is the US Dust Bowl of the 1930s, leading to the first major piece of soil-related legislation in the US, the 1935 Soil Conservation Act, but even before then soil erosion was perceived as an ever-present threat to American farmers and food production. By the 1870s there was widespread awareness of the problem of erosion [24]. The emerging environmental movement in Europe and America also had a growing influence on policy [25, 26], and forest protection laws incorporating the principle of soil conservation appeared, for example in the US from 1873 [26] and New Zealand in 1874 [27]. In 1894 the US banned grazing in federal reserves to prevent land degradation [26] and in 1901 the Indian state of Punjab, under British administration, passed what might be the first soil and water conservation law, the *Punjab Land Preservation Act*, still in force today [28]. In 1907 Iceland passed the first national parliamentary soil conservation act [29].

A significant result of the Dust Bowl and President Roosevelt's generous response to it was the huge contribution made by the United States Department of Agriculture (USDA) and American scientists to the worldwide study and management of soils. Despite some criticism [30-33], the far-reaching impact of this intense period of activity, barely perceived outside the sphere of soil science, is perhaps unparalleled in human history. Building on a legacy of international scholarship, especially in Germany and Russia [34], in the space of a decade at least three trailblazing conceptual tools were created: the Soil Taxonomy, the Land Capability Classification (LCC) methodology and the Universal Soil Loss Equation (USLE). These would radically transform the international science, management and legislation of soils.

From the 1930s other nations rapidly passed their own soil and land legislation. Food security was a major driver, but so was the loss of rural landscapes and wilderness. Other kinds of environmental protection were also implemented throughout the 20th century, such as the creation of green belts around cities, even as early as 1901 [35]. In the 1940s, however, World War II impacted food production and distribution, especially in Europe, fostering more self-reliance. In the postwar world agricultural productivity took on a new significance and the new technique of land evaluation was enthusiastically applied [36, 37].

In the UK, even before the end of the war, the Scott Report on rural land use recommended the protection of: "good agricultural land," but also signalled a new approach to planning that incorporated landscape amenity [38], and has been cited as prescient in its ethos of sustainable development [39]. The report fed into the 1944 White Paper, *The Control of Land Use* [40] which paved the way for one of the earliest and most radical planning policies of its kind anywhere, enshrined in the UK 1947 Town and Country Planning Act which nationalised land use regulations and has been emulated worldwide [41]. While the driving force of the USDA LCC had been farm management and soil conservation, the British system was primarily a tool of the new planning system, being put to work by soil surveyors in Britain not to find new land to cultivate, but to identify and protect existing prime farmland. The latest incarnation of this system in England and Wales is the 1988 Agricultural Land Classification (ALC) system for identifying the "best and most versatile" (BMV) land [42]. In 1984 the USDA Natural Resources Conservation Service (NRCS) developed its own system oriented towards planning legislation, Land Evaluation and Site Assessment (LESA) [43].

The postwar era is notable for the creation of global and regional institutions such as the FAO, the International Union for Conservation of Nature (IUCN), and the European Union (EU), all of which have been instrumental in relation to land and soil governance. Shortly after its creation in 1945, the FAO embarked on the world's first formal international attempt to address soil conservation as a global issue, via a report [44] and a conference in 1948 [45]. Underpinning such initiatives was a growing awareness of the economic impact of soil degradation, to society at large as well as individual land users, with far-sighted hints at the valuation of ecosystem services and implications for government policy [46]. Similar analyses followed in the ensuing decades [47, 48], and soil law continued to evolve, with the US taking the lead [49].

In 1949 the botanist Aubreville coined the term desertification to describe the transition of agricultural land in arid or semiarid areas to an uncultivable state lacking ecological viability due to a combination of climatic and human factors [50]. By 1958 the Chinese government acknowledged the threat of desertification to the wellbeing of nearly 200 million people, and has initiated afforestation programmes since 1978 [51]. Despite controversy over its meaning, the concept of desertification became and continues to be a significant driver of sustainable land management (SLM) initiatives designed to tackle land degradation [52, 53].

The conflicted role of agriculture

In a sign of things to come, the UK postwar agricultural policy came in for subsequent criticism for virtually exempting farmers from planning regulations, the assumption being that the role of farmers in protecting the rural landscape made controls on agricultural land use unnecessary [39, 54]. After WWII, geopolitics stabilised and agricultural productivity soared, due to a range of technological innovations [31], which from the 1960s reached lower income economies as the Green Revolution [55]. This period represents a radical reframing of society's attitudes towards the environment and agriculture [56]. In some parts of the world food security was no longer regarded as a serious threat, whereas agriculture itself was increasingly perceived as the primary threat to the environment, and hence a mixed blessing. Both the expansion and intensification of agriculture led to unprecedented wildlife habitat loss, encompassing deforestation, wetland drainage, conversion of grassland to arable and hedgerow clearance. Added to this were many other environmental impacts associated with industrial agriculture, including pollution, soil degradation and fuel inefficiency [57-59].

The publication of *Silent Spring* by Carson [60] led to a major international turning point in changing attitudes and policies in response to the excessive use of pesticides and herbicides. In Europe in the 1970s and 1980s the Common Agricultural Policy (CAP) came to epitomise the ironic twin dilemma of taxpayers subsidising overproduction at the expense of environmental destruction [40, 54, 56]. With the public emergence of climate change science since the late 1980s [61], and growing evidence of the significant climate-forcing influence of GHG emissions from farming [62], alongside the ongoing loss of biodiversity [63], agriculture finds itself more implicated than ever as a cause of environmental crisis as well as its victim. The negative image of modern agriculture as an industry that degrades soil and threatens ecosystems risks devaluing the public perception of agricultural land, as if the sealing of such land would not result in a loss of environmental benefits. Meanwhile soil science, straddling geology and biology, became virtually a subdiscipline of agriculture at least halfway into the 20th century [11]. Hence soil, by association and also because largely hidden, rarely evokes the kind of intrinsic concern that attaches to wildlife or landscape.

The relationship between soil and land

Soil and land are inextricably linked in the context of governance and, indeed, many languages conflate the two in common parlance. Weigelt *et al.* [64] stress a distinction between soil and land, but also identify a gap in the literature to address the critical importance of governing them together to achieve sustainability. An absence of land rights also often encourages poor soil management and land degradation [65-68]. From a legal perspective the concept of land is usually operative in matters of ownership and boundaries, and also in terms of spatial and territorial planning. Soil tends to enter the legal fray when it is transformed, harmed or threatened by a specific activity, whether by the owner of the land containing the soil, or by another party. This distinction signals two very different aspects of soil law. The former perspective generally encompasses property rights, whereby the injured party is primarily the landowner. The latter perspective offers protection to soil as a legal entity in its own right, even from its “owner”, for the common good.

Where the blurring of land and soil becomes problematic is in instances of proposed changes that are framed only in relation to land despite having significant impact on the soil within, or adjacent to, that land. This opacity can apply to legislation specific to agriculture, the environment or planning, and as a result important soil implications of land use conversion may fall into the gaps between all three. The creators of agricultural laws may feel that their remit regarding soil protection extends only to the direct impact of farming on soil, while the creators of environmental laws may feel that their remit for soil protection and conservation extends only to an unspecified component within a larger ecosystem or protected landscape. This then leaves rural land use conversion in the hands of the creators and implementers of planning laws, who may feel that soil *per se* is outside their jurisdiction, despite the soil-related implications of the land use changes that their legislation may encompass. There is in effect a blind spot whereby the legal instruments with the greatest power to affect soil, sometimes irreversibly, are often framed and worded with little or no reference to soil. Raising this in a meeting with government soil scientists caused heads to nod in agreement and perusing comparative laws reveals that this is a recurring theme throughout the world [4].

Anthropogenic threats to soil and land

Humans have learnt to transform soil to their advantage, and the overall impact of human society on soil, both intended and unintended, has been profound. Short-term gains have often been at the expense of long-term harms. The picture is further complicated by the fact that human impact exacerbates or unbalances natural processes such as soil erosion, salinization or flooding. Soil or land is typically vulnerable to three types of anthropogenic threat, set out in Table 1, and described further below.

Table 1: Types of anthropogenic threat to soil or land

1. **Soil degradation** (i.e. harm to soil, other than soil sealing, which is covered in 2 and 3 below)
2. **Conversion of natural ecosystems** or other semi-natural and uncultivated land to another form of land use, such as agriculture, forestry, urbanisation, or industry, including mining or energy infrastructure (conversion is also known as ‘land take’)
3. **Conversion of farmland** to urban or industrial use² [69] (‘agricultural land take’)

1. *Soil degradation*

In this context, degradation refers to any harm done to soil which is potentially ameliorated by remedial measures and includes acute problems such as contamination, salinization, acidification, sodification and compaction or other forms of structural collapse³. The term soil degradation has also been widely used to describe gradual and chronic loss of productivity, typically the result of over-exploitation and inadequate management, invariably bound up with erosion, organic C depletion, reduction in soil biodiversity and low nutrient status. While acute soil degradation is often the result of breaches of regulations, chronic soil degradation tends to be associated with either intensive agriculture or rural poverty, and generally requires supportive rather than punitive legislation. These problems can be complex and linked to natural causes or intrinsic to certain soil types. In the worst cases large tracts of land have been abandoned or declared unfit for food production with severe economic impacts. Gisladdottir and Stocking articulate the interlinkages of land degradation with other environmental problems such as climate change and biodiversity loss [52].

In the first study of its kind, UNEP commissioned the International Soil Reference and Information Centre (ISRIC) to conduct the Global Assessment of Soil Degradation (GLASOD). GLASOD concluded that from World War II to 1990 15% of all land worldwide (almost 2 billion hectares) was degraded, equating to 23% of inhabited land (14% seriously). Soil erosion was by far the most widespread form of soil degradation, affecting 83% of degraded land. Approximately 20 years later ISRIC scientists reviewed GLASOD and used satellite imagery to measure an overlapping and subsequently ongoing period (1981-2003 [70, 71]). The authors detected a further 24% of the total land area that was degrading *mostly in addition to* GLASOD’s 15% already degraded, i.e. a cumulative process, implying that at least a third of the world’s land appeared

² Converting farmland to tree cover is not usually regarded as a “threat”, though there are scenarios where this may be the case, for example, in relation to inappropriate afforestation of peatlands, or replacement of biodiverse systems with plantation monoculture.

³ The term “soil degradation” can also, more generally, be understood to encompass the other two threats listed here, including soil sealing.

to be degraded to some extent by 2003⁴. Additional key findings were that land degradation was not primarily a problem of drylands, as had long been proposed, but was a worldwide phenomenon.

In a 2015 follow-up study the global estimate of degrading land was revised down slightly to 22%, with 14% of land showing some improvement [72]. Also in 2015 FAO/ITPS⁵ claimed that 33% of land was degraded, and could reach 90% by 2050 [73]. A 2016 study produced a global figure of 30% degraded land [74]. A UNCCD analysis in 2017 revised the total to 23% [75], but states that: "...over the last two decades, approximately 20 per cent of the Earth's vegetated surface shows persistent declining trends in productivity." [76]. In 2018 IPBES⁶ concluded that 75% of the Earth's land surface is either transformed or degraded to some extent by human activity, impairing 3.2 billion livelihoods, and costing 10% of the annual global gross product, whereas the benefits of restoration are typically 10 times higher than the costs [77].

Most countries now have some form of regulation addressing soil degradation, typically embedded within their agricultural legislation, and in a number of these countries the law is strictly enforced. In many cases, soil policies go beyond prohibitive and punitive laws dealing with acute soil degradation, and incorporate support and incentives to foster improved long-term soil health and protection [78]. Compared to land use conversion, soil degradation, even though it may be the result of deliberate law-breaking, is generally recognised as unintended and unwelcome, so there are fewer obvious social pressures to oppose such laws. The global trend has been greater governance, but the cost of remediation can be prohibitive, whether for those held responsible or those who bear the cost, e.g. taxpayers, so circumvention is a constant risk [79]. China stands out as the country probably most afflicted by soil degradation, of every kind, but by contamination in particular, on a scale that appears irreparable and unaffordable, requiring more than the world's entire wealth to remediate, according to *The Economist* [80]. Nevertheless, no country seems more acutely aware of this or determined to address it than China itself [81, 82].

2. *Conversion of natural ecosystems*

Natural ecosystems or semi-natural landscapes, that is land that is uncultivated and largely unmanaged [83], are ubiquitously valued for a variety of reasons: their importance for Indigenous peoples and other local communities; their aesthetic characteristics and recreational potential; their educational and scientific worth;

⁴ The authors made no attempt to merge the data because of the contrasting methods of the separate studies and the fact that the GLASOD data was unverifiable, but one of the authors opines that: "It is not unreasonable to judge that all land now under anything less than natural climax vegetation is degraded in terms of biodiversity and stored carbon and nitrogen, perhaps 66% of the world's land surface and rising." (David Dent, personal communication.)

⁵ The Intergovernmental Technical Panel on Soils.

⁶ The Intergovernmental Science-Policy Platform on Biodiversity and Ecosystem Services.

and, with increasing urgency, their vital ecosystem service benefits. Aside from largely uninhabitable areas, this category of land includes forest, wetland, native grassland (e.g. prairie or steppe) and heath-/moor-/peatland. Globally the primary threat to such land is from conversion to agriculture, responsible for 80% of deforestation according to WWF [3], but the full picture is confused by claims of both overestimation and underestimation [84].

The dynamics of forest change are also complicated by temporary deforestation and forest gain, e.g. via afforestation, reforestation or regrowth on abandoned farmland. A recent study [85] using satellite imagery estimated that between 2001 and 2015, only 27% of global tree cover loss was permanent land use change for commodity production, i.e. large-scale agriculture, mining and energy infrastructure. Urbanisation accounted for less than 1%. Impermanent changes included forestry (26%), shifting agriculture (24%), and wildfire (23%). Regional contrasts were stark with permanent deforestation accounting for most of the tree loss that occurred in Latin America and Southeast Asia, mainly driven respectively by ranching and oil palm plantations [85]. Indonesia and Malaysia stand out as areas of increased deforestation. The rate of loss had declined markedly in Brazil, but since the change of regime in Brazil in 2018 forest destruction is reported to have increased again dramatically [86]. Other regions have relatively low rates of permanent deforestation, with temporary disturbance associated with managed forestry and, in North America, Russia and Australia, wildfire. A few countries have achieved a net forest gain.

Natural ecosystems tend to be better protected in the higher income countries, but historically this was largely because they lack the natural resources that facilitate intensive land use and often present severe obstacles to development. Very little "productive" wilderness remains, and some parts of the tropics have reached or are close to this point [87]. In sub-Saharan Africa, even where natural forest exists near settlements, it is difficult and expensive for farmers to encroach on such land, which also often has poor soil [88]. The alarm has also been raised concerning the impact of rising and uncontrolled urbanisation on biodiversity in Africa [89]. In 2009 Rockström *et al.* [90] suggested we are approaching the limits of planetary land use conversion. This makes the preservation of natural ecosystems worldwide all the more urgent, especially given their far-reaching environmental benefits, such as C storage, biodiversity and soil and water conservation. A combination of agricultural support and incentives, and rational territorial planning can raise farm incomes, improve food security and reduce loss and expansion of farmland [91-95]. Nevertheless, conversion of natural lands to agriculture is rarely as drastic or permanent as urbanisation, and many countries are increasingly rewarding landowners, land managers or farmers, for example via payments for ecosystem services (PES), to implement more environmentally benign land use and practices [96]. Putting aside the question of how effectively each government responds to the loss of natural ecosystems, the issue is relatively unambiguous and virtually every country on Earth, theoretically at least, protects such land with environmental laws and policies. Furthermore, soil protection *per se* is rarely the driving force

behind such legislation, which tends to be framed in terms of ecology, biodiversity or watershed protection, or occasionally preservation of landscape or culture.

3. Conversion of farmland to urban or industrial use

The vast majority of land consumed by urban and industrial expansion throughout the world is agricultural land and this type of threat, which encompasses much soil sealing, is arguably the least reversible and the most profound of these three types of anthropogenic threats. Furthermore, for sound socioeconomic reasons, urban settlement has historically developed near, and often surrounded by, prime farmland. Hence it is frequently the best quality land that is most vulnerable to conversion [97-100]. Peri-urban farmland is particularly attractive to developers because unless it is subject to strict green belt or zoning laws, it lacks the environmental protections of natural ecosystems, is usually cheaper to develop than brownfield sites [101] and tends to be conveniently situated in terms of existing facilities and infrastructure. In the UK there has been a gradual weakening of the protection afforded to BMV land, despite an official policy to the contrary [102, 103]. Furthermore, a study of 25 EU countries found that the land most at risk was that slightly further out from city boundaries, which typically has more fertile soil than that closer to the city [104], while, interestingly, the influence of CAP subsidies reduced the rate of land take in general.

Although agricultural land take is rarely covered by either environmental or agricultural legislation, an exception to this is where farmland is protected for its above-ground biodiversity value, e.g. high natural value (HNV) [104], but this is not the same thing as protecting land for its intrinsic SES value. The last resort for farmland preservation, if it occurs at all, is usually some form of spatial or territorial planning, but the primary motive for such planning regulation is typically urban expansion driven by population and commercial pressure, rather than rural protection [105, 106]. There is a distinct lack of robust legal instruments to prohibit or restrict agricultural land take and the approach taken varies considerably between and within countries, producing policies that are frequently complex and sometimes ambiguous or conflicting [40, 106-113]. Added to this is the prevalence of “informal” governance of planning regulations in many low- and middle-income countries [111, 114, 115].

The mere potential for urban development, even without planning permission, typically increases the value of farmland by, for example, fivefold in South Korea [116], sixfold in Morocco [117], nearly tenfold in New Zealand [107], and even by orders of magnitude more than this: 100-fold in Ghana over ten years [118] and similar multipliers in Britain and Japan [119]. This creates enormous commercial pressure for landowners to sell, especially in lower income countries where agricultural livelihoods may be precarious and lack government support [120], but do not necessarily leave poorer farmers with any long-term benefits. With the exception of a few who may be able to exploit new urban markets, most farmers will use this temporary windfall to pay off debts and become landless farmers or seek other occupations [121]. The prospects of re-

investing in cheaper land to cultivate are greatly diminished by the fragmentation of holdings and the creeping outward shift of the agricultural zone onto lower quality land [121].

It is worth addressing arguments advocating, or not opposed to, farmland conversion. Visser takes this logic to a purely economic extreme of treating natural resources as tradeable commodities or assets and even suggests that land markets would benefit from greater scarcity of farmland [122]. Satterthwaite *et al.* [123] argue that the world has abundant arable land and a global economy from which urban populations can import food without depending on their own agricultural hinterland. This suggestion presupposes reliable food supplies and reliable trading partners, yet geopolitical instability and climate change increasingly threaten global food security [124], not to mention the additional risks associated with a global pandemic.

In defence of conversion one might argue that much farmland, often government-owned, has been abandoned or under-utilised for various reasons, such as de-population of rural areas or, as in the case of the former Soviet Union, incomplete land reform [125], and that this land could be recultivated. There may be limited capacity for rehabilitated farmland to compensate for conversion elsewhere [126], but this raises other issues. One reason for abandoning farmland is the difficulty in extracting an adequate livelihood because the land is degraded [127] or marginal [125]. Such land will be intrinsically less productive so, even if farmers have the means and incentive to restore it, a greater area might be needed to achieve requisite returns. Where productive land has been abandoned for socioeconomic reasons, such as a lack of infrastructure in remote areas which could be remedied, recultivation is feasible [128, 129]. However, abandoned farmland presents another global narrative, for it is swathes of such land, in Russia, US, China, Australia, Latin America and elsewhere, that have inadvertently facilitated a global pattern of reforestation and rewilding [126] that goes some way towards buffering the effects of deforestation elsewhere [126, 130].

Some advocates of farmland conversion in high-income countries like the UK employ primarily socioeconomic arguments, for example with respect to a scarcity of land for housing in the London Metropolitan Green Belt (MGB), arguing that the MGB policy is rigid and anachronistic in protecting all farmland regardless of its environmental value, while constraining suburban gardens which can harbour more biodiversity [54, 131-133]. However, residential land undergoes much sealing and topsoil removal, while the biodiversity of its gardens, though potentially of great value, is entirely arbitrary. Farmland, on the other hand, remains largely unsealed and, given appropriate policy incentives, such as PES, can become more biodiverse and provide greater ecosystem services, as has been a gradual trend in the UK since the late 1980s [102], and looks to continue with the proposed new Environmental Land Management (ELM) schemes [134]. Where such authors have a much stronger case is in criticising the binary distinction between green belt land, where all farmland is protected regardless of its agricultural quality, and non-green belt rural land where prime farmland should be protected but is often developed [132].

The MGB, one of the oldest green belt zones in the world which constrains one of Europe's largest cities, is continually under strain from pressure for housing and transit development. In spite of strict regulations, planning permission is devolved and inconsistent, and much development leapfrogs onto agricultural land beyond. Similar but younger zones have yet to experience such pressures, but in the case of the Ontario Greenbelt, for example, this is partly due to an approach which is arguably more integrated and enlightened, incorporating green infrastructure, principles of environmental, economic and social sustainability, greater public participation and underpinned by a legal framework, centrally owned and controlled by the provincial government, yet both robust and flexible [35, 131, 135]. Farmland preservation in Ontario is not simply a matter of locally *ad hoc* prohibition in the teeth of fierce opposition from planners and developers, but is integrated into a holistic policy that includes agricultural support, local food marketing, employment opportunities, recreation, tourism and ecosystem services in a regional context [131, 135-137].

The view that urbanisation represents economic progress [123] has led many administrators and politicians, especially in lower income countries, to embrace urban expansion policies with enthusiasm [138]. However, at the local level there are serious concerns, even within government [108], that the wider implications of farmland loss are increasing food prices and imports, escalating rural poverty, land degradation and conflict [108, 120, 121, 139-141], as well as the less appreciated impact of soil sealing [100, 108, 142], including poor sanitation, flooding and pollution, which disproportionately affect rural communities [143, 144]. Much of the land consumed by urban sprawl is common land on which many rural communities depend [108]. Developers and officials or agents may benefit from these transactions [145], but the cumulative negative impacts are felt by the whole community. Those highlighting these issues include local academics [120, 141, 146], journalists and blighted farmers taking their grievances to the courts, if they can [145, 147]. Social inequalities drive many to facilitate the very land use changes from which they gain the least, sometimes converting natural ecosystems to replace what they have lost. There is evidence that appropriate government support reduces farmland sale for non-agricultural development, and enhances food security and biodiversity [104, 105, 148] so any attempt to prohibit or restrict agricultural land conversion, must be accompanied by economic support and viable alternatives. No amount of legislation or governance will succeed without this. While all three types of anthropogenic threats to land and soil listed in Table 1 above endanger ecosystem services, from a law and policy point of view, compared to soil degradation and natural ecosystem protection, agricultural land take is often "out of sight and out of mind".

4. *Cross-cutting issues*

A few academics argue that land conversion or degradation may be justified in some cases, on the basis that the economic benefits may outweigh the environmental costs [149], but despite the best efforts of environmental economists, such costs remain intangible [150] and often profound. Furthermore, such

benefits are not necessarily sustainable and are ultimately dependent on ecosystem services that economists have traditionally treated as “free” and, more to the point, inexhaustible [90]. This market-driven worldview has been strongly criticised for at least 50 years [151, 152] and arguably much longer [153], underscoring what is becoming known as the Anthropocene crisis [154]. Economic benefits also accrue unevenly and not necessarily in the national interest. The escalating profits from development can foster an “unholy alliance” between the public and private sector that rewards elaborate circumvention of the law [105, 155] or outright infringements [66, 75-77, 112, 146]. However, it is also important to appreciate that what may appear to outside observers to be infringements may, in some cases, simply be the result of longstanding traditions of informal governance or customary tenure [145, 156], or simply a lack of institutional capacity [123].

Mining, energy infrastructure and other forms of industrial land use represent a very small proportion of land loss or degradation overall [157], although this is increasing [76] and cannot go unmentioned because of their extreme impact in some locations. In western Ghana, for example, gold and diamond mining has had devastating effects on rural communities and natural ecosystems [158, 159]. This activity includes both corporate mechanised extraction and illegal and artisanal hand-digging, a form of low-input mining which also provides construction material. These can involve all three types of anthropogenic threats to soil listed above, going far beyond the loss of land, and including local water contamination and depletion, illegal logging, extensive soil and subsoil removal, and sometimes violent land disputes. The highly lucrative and largely unregulated context in which these enterprises operate tolerates and even encourages infringements of the law, from which corporations operating legal concessions are not exempt [159].

The jurisprudence⁷ of soil and land

The implementation of soil legislation

There are essentially two components to legally protecting or preserving soil resources: (i) applying a method of evaluating and prioritising areas or bodies of soil and (ii) implementing the requisite (and effective) legal instruments and governance structures. Both of these components exist throughout the world, but only within a fragmented and, in some cases, theoretical patchwork of initiatives. Before dealing effectively with the issues of which forms of governance structures or legal mechanisms might be most applicable to soils, one must consider what criteria to apply. This is essential whether focusing primarily on specified spatial areas of land or on the functional aspects of soils in relation to their current status or use. However, preceding even all of this is a minefield of ambiguous terminology to navigate.

⁷ Jurisprudence here refers to both legal theory and legal systems.

Semantics and legal terminology

When legal documents and policies refer to the way in which soil is affected by those managing the land it occupies, and hence the possible harms it may encounter, the word “protection” is the term most widely used. Many if not most countries have some form of explicit or implicit soil protection policy in place, but this is more often than not subsumed within other legislative and policy domains, for example the environment, agriculture or spatial planning. Soil legislation is commonly sub-divided further into categories of protection, reflecting the severity of particular problems within a given country or the gravity attached to the problem by its authorities. For whatever reason, certain soil laws are almost always framed in negative harmful terms and others in positive or remedial terms. Hence there are soil pollution or contamination laws, but hardly ever soil erosion laws. Instead, there are soil conservation laws or occasionally, in a similar vein, soil improvement laws.

Harder to pin down is the terminology applied to the conversion (effectively the loss) of agricultural or other undeveloped rural land to non-agricultural use which typically incorporates substantial additions of infrastructure (e.g. housing). In this domain several terms are used, some in the negative or destructive sense, and others in the positive or protective sense. It can be argued that the term land conversion, though technically describing the act of change or loss, is neutral, because unlike the unintended consequences of the harms done to soil, this refers to a deliberate act with an intended outcome and various beneficiaries.

The many approximate synonyms for rural land conversion with a more negative connotation include urbanisation, urban spread/sprawl/expansion/encroachment, land take/consumption/loss, long-term land cover change and (in France) artificialisation [160]. Other phrases commonly used in connection with rural land conversion are land competition and fragmentation, of farmland in particular, because this is one of the ways in which urbanisation becomes self-perpetuating [161]. Soil sealing, the permanent covering of soil, for example by concrete or tarmac, is a phrase widely used in conjunction with land take, because they typically occur together, but the meanings are distinct. While land take refers to a change of use which, by definition, usually entails some loss of land resource from its former use, sealing constitutes a much more permanent and intrinsic alteration of the land surface, typically with more far-reaching consequences for soil functions, especially drainage, and potentially severe effects on biodiversity [100, 162, 163]. Nevertheless, land take almost always involves some degree of sealing, and where it does, that constitutes a double impact – the spatial loss of land resource and the additional degradation of the ecosystem services that that resource provides in the round. However, sealing also occurs independently of more general land take, even in highly protected areas, for instance in the form of roads [162]. There are a few examples of unsealing, as a form of compensation for sealing elsewhere, but it is notable that these did not equate to total restoration [164].

On the positive land-saving side of the same lexical coin, the term predominantly used in North America, and in many countries, is land preservation [165, 166], but the terms land conservation or land protection are used almost interchangeably with it in the US [167, 168] and throughout the world. In most cases these phrases are applied to specific bounded areas, at various scales, which could mean a protected zone or alternatively a cluster of holdings, a single estate or even one field. The expressions “no net land take” or “zero land take”, which have appeared in EU documents in recent years [101, 169], are similar in principle, but refer to overall quotas or targets, with the additional challenges relating to relevant spatial scale and overall quantification. In contrast ‘land sparing’ is used in juxtaposition to ‘land sharing’, referring to the concept of safeguarding biodiversity, either by sparing natural lands from agricultural use (land sparing) or by incentivising farmers to support more biodiversity (land sharing) [170].

Globally words or phrases used to mean approximately the same thing as land preservation can sometimes have other connotations. Land conservation has historically been most strongly associated with nature conservation, in the sense of land that is not recognised as developed or cultivated and has some form of protected status. Such land is often threatened by agricultural incursion at least as much as by urbanisation. Land conservation is not necessarily synonymous with ‘soil conservation’, a term used professionally and more widely to refer specifically to the prevention of soil erosion with respect to land management practices.

Land protection is often synonymous with land preservation, but in some examples of legislation it is intended to mean soil protection, that is *in-situ* safeguarding. Soil protection is sometimes intended to mean or encompass land preservation. These semantic issues may sometimes result from translation because the full texts of many laws appear only in their original language and script [4]. Occasionally countries have all-embracing “soil protection” policies which include the concept of spatial land preservation in context [171, 172] or conversely all-embracing “land preservation (or protection)” policies which include the concept of soil protection [5, 172]. In some cases this blurring seems to be deliberate, such as when land preservation is promoted as a means to protect SES [5, 162]. Pragmatically soil scientists may need to accommodate themselves to the language of policy and spatial land use planning in order to engage with the process.

Finally, it is important to note that while many legal instruments have been designed specifically to protect only prime (i.e. highly productive) farmland, this is not always the main driver of land preservation or protection, which may prioritise other factors such as landscape, heritage or SES. In this article this distinction is highlighted by using the term prime farmland preservation (PFP) where appropriate.

Putting a value on soil

At the root of our current environmental crisis is that humanity has traditionally treated abundant natural resources as free goods and services. Economists and accountants are no exception: cost-benefit analyses include tradeable assets and products, but routinely exclude the ever-present prerequisites of life.

Furthermore, traditional cost-benefit analysis (CBA) is short-termism by definition because it usually does not even consider impacts outside a 25-year window [173]. When a tonne of soil or water is traded, the allocated price is normally derived from the aggregate cost of acquisition, processing and delivery, with no attempt to assess the intrinsic value of the substance. This applies not only to soil as an intrinsic good, but also to some extent to its economic potential, i.e. the land that contains it will have a market value and that value will be related to the land's productivity, yet unquantified and uncosted SES have far-reaching economic implications. Even in a society without monetary currency it would be relatively straightforward to calculate the economic value of a given volume of soil, in conjunction with the land it occupies, purely in terms of its capacity for life-sustaining primary production; ask any farmer. Although we now have modern techniques to assess the productive capacity of land resources, a conceptual process of "following the money" has been at the heart of most farm or settlement emplacement decisions since the Neolithic.

For at least 80 years soil survey and land evaluation have been the traditional tools at our disposal for assessing the productive and economic potential of an area of land, alongside any limitations or hazards it may present. A range of soil properties is recorded alongside other local environmental data and, where appropriate, socioeconomic data. This data is combined to grade the capability or suitability of each parcel or zone of land according to specified use criteria. The modern terminology of multicriteria decision analysis (MCDA) is more a change of style than substance, to reflect the much greater reliance on information technology, especially GIS [174]. In the above approaches "criteria" is the operative word. Where these methodologies differ from analogous *ad hoc* historical techniques, is in their capacity to distil decades of practical and scientific observation to apply much greater predictive accuracy and precision to the land use decision-making process [175].

Land evaluation is not the same thing as land valuation, though they have always been interrelated and both are relevant to spatial planning. Land evaluation methodologies were never intended to calculate the true, i.e. total and holistic, economic value of an area of land or a volume of soil (which are themselves two very different things), but rather the capability or suitability of distinct parcels of land for specified forms of primary production, such as crops, pasture or forestry. The purpose is for land use decisions, albeit often with profound economic implications. The purpose of land valuation, however, is generally to set prices for landowners and taxes for governing authorities, and has always primarily been based on location, which admittedly is historically bound up with soil quality and land use, but with a myriad of other factors too.

In contrast, soil has always provided a variety of extrinsic benefits beyond primary production which were recognised and appreciated long before we referred to them as ecosystem services. To what extent our ancestors valued (or devalued) specific areas of soil or land for reasons other than primary production, or obvious physical location, is not always tangible, though it is clear that many indigenous communities have developed deep understanding of, and spiritual connection to, the land and soil on which they depend [83].

A wider all-embracing appreciation of soil and a few attempts to evaluate it in that context date from the 1960s [176]. This concept has gathered momentum and, more recently above all, with respect to C.

While it may seem too neat to earmark the last decade of the previous millennium as a critical turning point in time, it seems inescapable that this period represented an important paradigm shift in our collective understanding of the role of soils. The 1990s shines out as a lightbulb moment when soil science and climate science came together. Although the role of C in the soil has been studied for at least 150 years and its place in the C cycle appreciated for much of that time, it was only in the 1990s that a flurry of papers explicitly linked soil C fluxes to GHG concentrations in the atmosphere and climate change [177-179] which, after several decades of growing evidence, had started attracting worldwide media attention since 1988 [180].

At the onset of the new millennium soil scientists have more than ever before started to deconstruct and articulate the critical importance of SES [181, 182]. Experts on soil law have responded accordingly, emphasising that the value and status of soil far exceeds its role in agriculture [183]. Taking this a step further, several groups of researchers, building on earlier attempts to evaluate ecosystem services [184] and characterising soil as a form of natural capital, have developed tentative frameworks and quantitative methods to enumerate and evaluate SES in the fullest sense [185-189], sometimes using case studies to apply monetary values [190]. To what extent this is entirely feasible or even desirable, is debatable [176], especially with regard to qualitative or ethical issues, but such approaches may help policymakers and planners apply more meaningful criteria and priorities to the governance of soil and land protection.

Categories of soil governance

Juerges et al. [65] summarise the types of instruments applied to soil governance (i.e. regulatory, economic, and so on) and the levels at which these operate, from global to local. In this section we present a very different cross-cutting breakdown, based on issues and intended outcomes. One can identify three broad categories of governance approaches applied to preserve or protect soil or land, usually at a national or subnational level, with various implementation mechanisms, set out in Table 2 and elaborated further below:

Table 2: Categories of governance approaches and mechanisms to preserve or protect soil or land

CATEGORY	DESCRIPTION
1	Regulations to prohibit (or guidance to discourage) certain actions
(a)	- Enforced by penalties
(b)	- Facilitated by financial support
2	Restrictions on development involving change of use

(a)	- Enforced by zoning laws
(b)	- Enforced by laws (or encouraged by guidance) based on ad hoc site evaluation
(c)	- Enforced by public acquisition of land
(d)	- Facilitated by financial support
3	Generic incentives to preserve land or to enhance its ecosystem services value
(a)	- Taking land out of agricultural production
(b)	- Converting intensively farmed land via extensification
(c)	- Declining to develop land or intensify its use

The first category comprises regulations and laws to **prohibit** (or guidance to discourage) **certain actions** by landowners that may degrade land, e.g. pollution, sealing, construction, stubble burning, tree felling, or cultivation methods leading to soil erosion, acidification, salinization, compaction, and so on; whereby best practice may be either:

- a. Enforced by penalties for infringement or
- b. Facilitated by financial support, e.g. farming subsidies, PES or C credits

The second category comprises **restrictions** (or guidance) **on development involving conversion of use**, either to protect terrestrial natural resources (for the benefit of any or all of primary production, watershed management, biodiversity, landscape or cultural or historical value) or simply to retain a certain quota of agricultural or forested land within a state or designated region; whereby land conservation or preservation may be:

- a. Enforced by laws based on zoning, e.g. National Parks, nature reserves, green belt, urban growth areas, Sites of Special Scientific Interest (SSSI) and heritage landscapes such as the UK Areas of Outstanding Natural Beauty (AONB) or UNESCO World Heritage Sites;
- b. Enforced by more general laws (or encouraged by guidance) not specific to the protected areas described in 2(a), based on *ad hoc* site evaluation, such as an Environmental Impact Assessment (EIA), and

thereby affording protection to sites of ecological or landscape value, or prime farmland (e.g. based on ALC or LESA criteria);⁸

- c. Enforced by public acquisition of land, such as land trusts;
- d. Facilitated by financial support, e.g. using LESA to obtain Purchase of Development Rights (PDR) conservation easements (US).

The third category encompasses **generic incentives** to preserve land or to enhance its ecosystem services value (or disincentives to develop, such as the removal of subsidies), in the form of:

- a. Taking land out of agricultural production, purely for the purpose of conservation, e.g. EU set-aside, US land retirement, rewilding;
- b. Converting an area of intensive agricultural production (arable cropping) to more extensive agricultural production (such as pasture, silvopasture or agroforestry) or forestry;
- c. Declining to develop or intensify the use of uncultivated land that could otherwise be legally developed or intensified, in order to maintain ecological or ecosystem services value.

Overlapping categories

The above categories are presented as a useful framework for analysis of soil governance. There are of course many instances of overlap. For example, regimes that are not targeted specifically at soils can still fall within the categories above. There are also numerous cases where soil governance instruments span a number of the mechanisms listed above. The examples below illustrate these points.

Environmental Impact Assessment (EIA) of projects, Strategic Environmental Assessment (SEA) of plans and programmes, and related types of assessment, may apply to some of these categories, especially categories 2(a) or 2(b), and can be helpful in drawing attention to threats to natural resources, but there can be limitations in their application to soils. In the US, for example, EIA is applied primarily for federally-funded projects [191] which is partly why the NRCS developed the LESA system for smaller projects targeted at farmland. In the UK EIA is more widely applicable, but generally only where part of the area intended for development is uncultivated or only partially cultivated and hence might not be invoked where only arable land is at risk [192]. EIA regulations vary slightly in Scotland, however, where the site being assessed may consist exclusively of farmland where it exceeds 200 ha. The Scottish EPA also provides specific guidance on application of SEA to soils [193].

⁸ A legally binding example is the Indian Prohibition on Conversion of Agricultural Land for Non-Agricultural Use (No. 16 of 2010).

The broad North American planning term land preservation covers many of these categories with the main focus on zoning [165, 194]. Every state and city has its own subset or version of land preservation laws and regulations which are varied and complex. One example is a “conservation easement” whereby a landowner is bound by a covenant set by the government or some other organisation and in return receives an incentive, such as a tax rebate; this could fall under categories 1(b), 2(d), or 3(a-c).

Apple Valley City, Minnesota has adopted a Natural Resources Management Plan (NRPM) which essentially constitutes a 2 (b) type mechanism via a permitting system, but incorporates elements of 1(a) because it could apply to a single aspect of a development. It differs from most local planning laws and regulations by applying the broad principle of environmental impact to every development [195].

There is an interesting example of environmental scientists trying to set a legal precedent as expert witnesses in a 2(b) type scenario in New Zealand in 2011, using soil natural capital and ecosystem services arguments to prevent urban development on horticultural land. The lawyer representing the developer argued that the only measured ecosystem service of this soil was food production. The judge, while declining to engage in the natural capital debate, nevertheless upheld the local authority decision not to allow development in favour of the “holistic” argument to protect natural resources which, from the point of view of the scientists, came to the same thing [196].

The modern governance of soil resources worldwide

While the piecemeal governance of soil use is almost as old as human history, formal supranational global oversight of soil resources is only approximately 50 years old. In terms of the natural environment, the early 1970s stand out as a period when national policies, international initiatives and academic analyses of global problems coalesced in an unprecedented step change in attitudes. 1970 saw the conception of Earth Day, initially in the US but later worldwide, and consequently the creation of the US Environmental Protection Agency, an institution which also became replicated worldwide. In 1972 the Club of Rome think tank, published the highly influential *Limits to Growth*, with its prediction of societal collapse in the 21st century, and frequent references to soil degradation [197]. This period also saw the emergence of multidisciplinary stakeholder groups joining forces for sustainable development [198].

Emergence of soil in modern international law and policy

The 1968 *African Convention on the Conservation of Nature and Natural Resources* (the “Algiers Convention”) was a continent-wide treaty devoted to natural resource protection that included a reference to soil. Although it lacked the resources and institutional framework to be particularly effective, the Algiers Convention was nevertheless regarded as a milestone in international environmental law [199]. At around

the same time FAO was publishing several of its soils bulletins every year, including two landmark studies in 1971, on land degradation [200] and on legislative principles of soil conservation [201]. The latter was a relatively concise set of guidelines that any nation could use as a template for creating or improving its own legal framework for governing soil. The 1972 UN Conference on the Human Environment (UNCHE) in Stockholm was effectively the first major international conference on environmental protection and sustainable development, and has been referred to by Boer *et al.* as marking the: "...first phase of international soil protection law" [202, 203].

In 1972 the Council of Europe created and adopted the *European Soil Charter*, a concise but relatively holistic set of soil protection aspirations [204] which was non-binding, but regarded nevertheless as the first international legal instrument dedicated specifically to protecting soils [202, 205]. In 1977 the UN held its first Conference on Desertification (UNCOD), and produced its *Plan of Action to Combat Desertification* (PACD) [206]. In 1981 FAO adopted the *World Soil Charter* [207] and in 1982 UNEP developed a global soils policy [208] and the IUCN/UN *World Charter for Nature* explicitly referenced soil [209].

Soil as a feature of common concern

In 1987 the UN commissioned the Brundtland report, *Our Common Future*, which reinvigorated the debate, again stressing the severity of soil degradation, and annexed a summary of proposed legal principles for environmental protection and sustainable development [210]. In order to address these issues financially, in 1991 UN agencies and the World Bank created the Global Environment Facility (GEF), as a prerequisite for the 1992 UN Conference on Environment and Development (UNCED) or Rio Earth Summit [211], a milestone event. From this emerged the UN Commission on Sustainable Development (UNCSD) and three legally binding international treaties: the 1992 *UN Convention on Biological Diversity* (CBD) [212], the 1992 *UN Framework Convention on Climate Change* (UNFCCC) [213] and the 1994 *UN Convention to Combat Desertification* (UNCCD) [214], which came into force in 1993, 1994 and 1996, respectively (the Rio Conventions). There was appreciation of the interrelationships linking all three conventions to land degradation [215], but funding for the latter was mainly tied to the UNCCD which was always the weakest and poorest of the three due to donor scepticism [52]. The UNCCD has been called the first and only "legally binding global agreement directly dealing with the promotion of bio-productive land" [202], although it contains little in the way of substantive obligations, and formally applies only to "arid, semi-arid and dry sub-humid areas" [216]. Nevertheless, UNCCD is gaining momentum as a focal point for efforts to address the global land degradation neutrality (LDN) target [206, 217].

The interconnected nature of different environmental problems, as well as their linkages to socioeconomics and SLM, was taken a step further at the 2012 UN Conference on Sustainable Development (Rio+20). The UNCCD brought the concept of zero net land degradation (ZNLDD) for adoption at Rio+20, reflected in the

outcome document, *The Future We Want* [206], and this later became embedded into the 2015 SDGs as the global LDN target in SDG 15.3 [206, 218]. Land degradation was finally being acknowledged as a global problem that was intrinsically linked with climate change, biodiversity loss and poverty. The concept of ZNLD/LDN was that every effort should be made either to prevent further degradation or, where this is not possible, to rehabilitate equivalent areas of degraded land elsewhere [218]. The importance of land and soils is gaining more prominence in the context of the CBD [212], the UNFCCC [219] and related *Paris Agreement* [220]. The parties to the CBD have addressed soil biodiversity via an international initiative [221] and a global report [222]. It remains to be seen how soils will be reflected in the Global Biodiversity Framework to be adopted by the CBD COP 15, currently re-scheduled to take place in China (in phases spanning 2021 and 2022) especially given China's rapid advance in the field of soil science ⁹.

At its first Plenary Assembly, at the FAO Headquarters in Rome in June 2013, the Global Soil Partnership (GSP) created the Intergovernmental Technical Panel on Soils (ITPS), made up of 27 soil experts representing all the regions of the world. [73]. The GSP also proposed 2015 as the International Year of Soils (IYS) and the annual observance of a World Soil Day (on 5 December), both of which were adopted at the 68th UN General Assembly later that year [223]. Under the auspices of the GSP the *Revised World Soil Charter* [224] was adopted by FAO members in 2015, and the *Voluntary Guidelines for Sustainable Soil Management* (VGSSM) [225] were endorsed by the FAO Council in 2016.¹⁰ The work of the ITPS, UNCCD Science-Policy Interface (SPI), IPBES, the IPCC, and in particular the publication of the 2015 *Status of the World's Soil Resources* [73], the 2017 *Scientific Conceptual Framework for Land Degradation Neutrality* [226], the 2018 *Global Land Degradation and Restoration Assessment* [77], and the 2019 *Report on Climate Change and Land* [51], by each respectively, stand out as key influences in the soil governance narrative. The significance of the science-policy interface is increasingly important in determining the scope and extent of legal obligations [227].

In 2015, to coincide with UNFCCC COP21, the French government issued a bold entreaty to the global community to raise average soil organic C (SOC) levels by 0.4% (the "4 per mille Soils for Food Security and Climate" initiative), to offset annual global C emissions into the atmosphere, as well as improving food security [228, 229]. The choice of SOC as the target was fundamental and twofold, because it represents both a means of accumulating and sequestering atmospheric C, and the most widely accepted measure of soil health or productivity. The 4 per 1000 Initiative, to which many countries have signed up, has succeeded in highlighting the critical role of soil and agriculture in climate change mitigation and adaptation. There has

⁹ According to the SJR International Science Ranking website China ranks second only to the US in soil science (<https://www.scimagojr.com>; accessed 28th April, 2021).

¹⁰ <http://www.fao.org/3/bl813e/bl813e.pdf>

also been criticism [230], especially of the feasibility of the initiative and underpinning data. While the authors have defended the science, with caveats, they also stress that 4 per 1000 was never intended to be a precisely calculated solution, but a positive, politically-driven and symbolic aspirational target [231]. Other soil scientists agree with them, judging that the initiative would be a technically feasible, “no-regret” and indispensable climate action [232].

Regional and sectoral developments

At the regional level the Algiers Convention was revised in 2003 to become the *Revised African Convention on the Conservation of Nature and Natural Resources* (Maputo Convention), but only entered in force in 2016 [233]. In 1998 the Alpine Soil Protocol was the first legally binding treaty expressly devoted to soil, entering in force in 2006 [234]. For both the Maputo Convention and the Alpine Soil Protocol, implementation has not yet matched aspiration, but both serve as useful focal points for awareness raising regarding soil conservation, management and best practice. The ASEAN Agreement on the Conservation of Nature and Natural Resources, signed in 1985, included an article specifically on soil, but it has never come into force [202].

The Council of Europe revised the *European Soil Charter* in 2003 to become the *Revised European Charter for the Protection and Sustainable Management of Soil* [235]. In contrast, in the European Union, agreement on a soil instrument resembles a triumph of hope over experience. In 2006 the European Commission presented its *Thematic Strategy on Soil Protection* and presented a *Proposal for a Soil Framework Directive* to place healthy soil on a par with clean water and air [236]. The proposal was eventually withdrawn in 2014, because five member states were not willing to agree to strengthened EU-wide legislation addressing soil sealing and liability for contaminated land [237]. A new EU Soil Strategy is planned for adoption in 2021 as a key component of a European Green Deal (EGD), aimed at making Europe the first climate-neutral continent by 2050 [238]. The European Parliament has called for a binding legislative framework for soils. [239]

Sustainable Development Goals (SDGs)

All countries have signed up to the seventeen SDGs [240]. Land and soil have profound relevance for most, if not all, SDGs, though only a relatively small number of SDG targets make specific reference to them [240, 241]. In particular *SDG2: Zero hunger*, aims in target 2.4 to ensure sustainable food production systems and resilient agricultural practices that increase productivity and production, help maintain ecosystems, strengthen capacity for adaptation to climate change, extreme weather, drought, flooding and other disasters and progressively improve land and soil quality; and *SDG 15: Life on land*, aims to protect, restore and promote sustainable use of terrestrial ecosystems, including in target 15.3 combatting desertification, restoring degraded land and soil, and striving to achieve a land degradation neutral (LDN) world. SDG targets 3.9, 6.4, 6.5, 12.4 and 14.1 also have particularly obvious direct relevance to soils.

[241], and land and soil are fundamentally related to many other SDG targets and indicators, including those for SDG 13 on urgent climate action. While the SDGs themselves are not-legally binding, they do in some respects echo or amplify existing and emerging binding international commitments.

National soil legislation

Last year FAO and UNEP jointly published a key document, which is candid in its acknowledgement that implementation of international commitments can be difficult to enforce and monitor, especially when they are purely aspirational rather than legally binding [217]. The report provides many examples of innovative and progressive legislation in relation to land and soil, though questions still remain in relation to enforcement [68]. On a global scale the most urgent action needed may relate to category 2a in Table 2 - restrictions or guidance on development involving conversion of use, to protect terrestrial natural resources and enforced by laws based on zoning, e.g. National Parks, nature reserves, etc., to address issues such as deforestation in Amazonia or Southeast Asia. The instruments to regulate this type of anthropogenic threat are relatively straightforward and consistently defined across the world, but the obstacles are political and socioeconomic. This is also true of other existing laws that directly or indirectly protect soil, the success of which require that entrenched power imbalances are challenged [64].

Lack of enforcement and governance on the ground remain serious obstacles throughout much of the world. See, for example, the studies referenced at [66, 89, 92, 112, 114, 120, 146, 155, 242, 243]. Those ten studies cited, merely a subset of many more, draw on a range of data for their evidence, in addition to literature reviews, including land use and land evaluation records, census and demographic statistics, case studies, planning and legal decisions, stakeholder interviews and, perhaps most telling of all, spatial remote sensing of land use change, using GIS software tools such as CORINE. Nevertheless, the process of enacting soil and land legislation is ongoing and ubiquitous, and reflects each country's circumstances and priorities. China, for example, with possibly the greatest absolute area of contaminated soil on Earth [244-246], as well as a long history of soil erosion [8], has separate soil laws to address both of these problems.

Table 3 presents a condensed numerical summary of derived categories of soil-related legal instruments, presented as approximate percentages of countries that have created or adopted such instruments (as detailed in *Methods* above). It is evident that soil and land protection laws are conspicuous by their relative absence worldwide. Environmental legislation is not specifically recorded here but, for comparison, it has been almost universally adopted. It is also conspicuous from the data below that for many countries the word *soil* would be entirely absent from their legal portfolio were it not for such environmental legislation.

Table 3 Proportions of countries with soil-related legislation

(a) SOIL-SPECIFIC LEGAL INSTRUMENTS	COUNTRIES (%)
Explicit soil policy.	7
Soil conservation/erosion law(s) [explicit/explicit+implicit].	34/79
Soil contamination/pollution law(s) [explicit/explicit+implicit].	27/65
Soil sealing law(s) [explicit/explicit+implicit].	3/27
Generic or other soil protection law(s) [explicit/explicit+implicit].	28/71
Soil protection monitoring and/or targets [explicit/explicit+implicit].	21/47
Reference to soil embedded within environmental legislation.	88
Reference to soil embedded within agricultural or land rights legislation.	72
Reference to soil embedded within spatial/regional planning legislation.	50
(b) LEGAL INSTRUMENTS TO PRESERVE AGRICULTURAL LAND	
Policies designating zoning, including rural and/or agricultural land.	35
Policy advocating land take avoidance.	24
Land take targets.	9
Legal framework to facilitate farmland preservation schemes.	19
Prime farmland preservation guidance based on soil or land classification.	27
Law prohibiting or strongly restricting loss of all prime farmland.	21
Land Degradation Neutrality (LDN) policy (commitments).	66
(c) SOIL-RELATED AGRO-ENVIRONMENTAL POLICIES	
Policies promoting organic or regenerative agriculture, agroecology or PES.	40
Policies referring specifically to the critical ecosystem services of soils.	15

Prime farmland preservation (PFP)

The data presented in Table 3, along with our literature review, leads the authors to conclude that one of the most serious failings with respect to the ambiguity or absence of legal instruments relates to what we have termed category 2(b) in Table 2, that is land take outside protected areas. At most risk and of most concern is the conversion of high quality or prime farmland, which is usually afforded far less protection than natural ecosystems, or none at all, yet can sometimes provide equivalent, or even greater, ecosystem services. This

is an important and often misunderstood point that cannot be overestimated. It is easy to fall into the trap of assuming that agricultural land cannot provide environmental benefits comparable to those of natural ecosystems, especially when based on current land management practices, but this certainly does not reflect the intrinsic value of such land nor necessarily even its existential value [247].

Land which is densely covered in vegetation, with undisturbed topsoil, will tend to store more water and C, and foster greater biodiversity, but the principle reason land becomes prime farmland is its highly valued ecosystem attributes: low altitude and gentle gradients; deep soil with favourable texture and structure that retains water, nutrients and C; benign biochemistry; and a favourable moisture regime that is not susceptible to extremes of drought or waterlogging. These are the desirable attributes that facilitate not just food production, but terrestrial life in general, which ecologists assess in aggregate as NPP. As crude as it might be in some respects, especially in relation to cultural or scientific value, NPP is widely regarded as one of the best proxy measures of total ecosystem service contribution [248]. In contrast to this, many of the varied soil landscapes that currently support natural ecosystems can be biologically marginal, exhibiting low NPP. Even putting aside the utilitarian criterion of agricultural potential, these peripheral zones can also be limited with respect to a range of critical ecosystem services, including in terms of the biodiversity they support.

Prime farmland is not usually an obvious candidate for voluntary conversion to natural regeneration or rewilding, but high-grade arable land, even in its cultivated state, can provide abundant benefits in terms of flood control, water storage and filtration, wildlife habitat, landscape value and recreation. However, and critically, such benefits are greatly affected by land management. The negative impacts of modern intensive farming, on the soil in particular, such as compaction, erosion, pollution and reduced SOC, are precisely the factors that undermine the ability of the soil to provide ecosystem services [249, 250].

One very welcome trend in recent decades has been to roll back many of these harmful methods. A range of techniques that are loosely grouped under the term of conservation, or regenerative, agriculture, agroecology, or even “carbon farming” are increasingly being promoted and applied in many countries [78, 251-253]. Included within this approach are minimum or zero tillage, permanent soil cover, crop residue retention, cover cropping, intercropping and enhanced rotations including, for example, deep rooting “tillage crops”, leguminous leys and rotational livestock grazing, agroforestry and other forms of mixed cropping, perennial crops, integrated pest management (IPM) and biological control, and many other ways of conserving and enhancing soil quality and, by default, ameliorating environmental impacts. Hence, our advocacy of PFP is proposed in tandem with the adoption of such methods.

Emerging holistic conceptual frameworks

Amundson sums up soil governance in the US as a complex patchwork of transient interventions, as opposed to long-term solutions [254]. Other authors describe comparable situations elsewhere, for example,

in the EU [255, 256], in South America [257], in Australia [258], in Russia [259] and in other examples already cited in this article. Fromherz summarises many of the existing soil governance initiatives, as well as the failures, and makes an impassioned appeal for a dedicated, legally binding international soil governance instrument on the basis that, "...individual states lack both the power and the incentives to make these changes." [260]. This has been echoed by others [183, 203, 239, 261]. Fromherz also highlights a key point alluded to already in this article, the low profile (literal as well as metaphorical, one could say) of soil, which is often conspicuous in its absence from risk assessments of other natural resources [262, 263]. Gonzalez Lago *et al.* refer to a global soils policy vacuum and call for an urgent transdisciplinary framework approach to "re-politicise" soil [264], while a number of authors and institutions highlight the degree to which soil protection is inescapably enmeshed with ethics [265].

Participatory modelling and conceptual frameworks have been applied to complex cross-cutting problems such as addressing the SDGs, and these approaches continue to evolve [266]. The latest stages in the process of soil governance so far, in the last decade, are encouraging to some extent, at least with regard to what soil scientists are bringing to the table. An overarching conceptual framework that has been proposed by some of the world's leading soil scientists, broadly as a memorandum of understanding, is "soil security" [267]. This concept has five dimensions: capability, condition, capital, connectivity and codification, which encompasses the translation of soil knowledge into policy and legislation [268], e.g. aimed at achieving the SDGs [269], although an attempt to establish soil security as one of the SDGs was unsuccessful [270]. The soil security initiative is still at a high level, with the emphasis on policy rather than active governance, but progress is being made and reported on in some areas [271].

Also emerging are more formalised frameworks that place the concept of SES further than ever in the context of socioeconomic decisions, policy-making and governance. One such methodology developed and tested in Switzerland is SQUID (Soil Quality InDicator), an index for mapping soil quality with respect to SES, to guide spatial development [189]. Another example from New Zealand is the Land Resource Circle (LRC) framework, which goes further than any other approach known to the authors in extending land evaluation to incorporate SES. The framework identifies in-depth environmental and socioeconomic implications of land-use decisions via a scoring system which avoids reducing all outcomes to simplistic monetary values [272]. A recent LRC paper includes a very detailed hypothetical example which indicates how quantified soil variables can be combined and converted into societal costs and benefits. By contrast, the Resilience-Effectiveness-Efficiency-Legitimacy (REEL) framework from Germany provides a purely qualitative means of comparing different approaches to soil governance according to the four criteria (dimensions) embedded within its title [273]. The REEL framework also highlights interconnectedness and attempts to address situations where policy targets mismatch the causes of problems, either spatially or in

terms of scale or even time. This is almost a meta-framework, with the emphasis on socioeconomics and due diligence.

As with other forms of natural resource governance, politics and socioeconomics are often obstacles to effective soil governance, as are weak governance structures, but a critical distinction in relation to PFP is that there is much scope for creating more effective legal instruments to tackle this problem than currently exist in most countries, and also for improving clarity and consistency to bring PFP further in line with ecosystem and soil protection.

Conclusion

An appreciation of the tangible benefits of soil is woven into the fabric of history and reflected by the value and protections afforded to productive soil by society over millennia. Such protections include legislation, but *ad hoc*, inadequately enforced and rarely if ever proportionate to the total SES. Two challenges identified as hindering progress are terminological obstacles associated with soil and land, and the absence of a soil-centric policy framework. Moreover, soil exists as a component of land that is subject to both extra demands, e.g. of food production, and competing demands, such as urbanisation. Agriculture has responded to these demands by expansion and intensification, each approach posing threats to soil, the former often encroaching on natural ecosystems and the latter fostering soil degradation. These threats have spawned national and international legislation. However, the problem of land take, in particular the conversion of prime agricultural land to non-agricultural use, including largely irreversible soil sealing, is arguably at least as serious a threat as intensification because of the disproportionate loss of SES this represents, yet tends to be relatively inconspicuous, and subject to ambiguous legislation and governance.

Threats to natural ecosystems are explicitly addressed by environmental laws, and to soil degradation by agricultural or soil-specific laws. A few of these laws have been in place for centuries and more have appeared in recent decades as global environmental awareness has grown; but neither environmental nor agricultural legislation normally encompasses the loss of prime farmland, the fate of which then tends to fall within the remit of spatial planning. Planners' procedures must navigate many competing demands and interest groups, political as well as economic, and rarely prioritise soil or its intrinsic value to future generations. As a result, legal instruments that are frequently designed with little, if any, reference to soil, often have greatest power to transform soil and land use. This situation is further compounded by the fact that intense commercial pressure along with conventional economics, which has traditionally ignored the benefits of long-term ecosystem services, promotes land use options that are more profitable than farming.

In a time of so much environmental concern, a further paradoxical factor that has emerged in the last half century to weaken the case for farmland preservation (as opposed to soil protection *per se*), is the conflicted

nature of agriculture: on the one hand the producer of our sustenance and custodian of rural landscapes, while on the other, a significant cause of environmental harm and climate forcing. Lacking both the visual impact and the iconic status of natural landscapes, soil in an agricultural context does not constitute an obvious rallying point in the public consciousness and so the constant attrition of prime farmland occurs in something of a legal and ecological grey area, attracting far less attention than other environmental issues.

However, the case for protecting soil as a critical part of our natural capital is separately gaining ground. A key paradigm shift occurred in the 1990s when it became more widely appreciated that soil C is inextricably linked with atmospheric C. This development also approximately coincided with a broader gradual acceptance that land degradation was interrelated with climate change and biodiversity loss. This view of soil as a central component in limiting global warming, adapting to climate change and addressing the ecological crisis, is being framed unequivocally in the wider context of sustainability and the linking of science to policy. Emerging multidisciplinary frameworks, such as the concept of soil security and methodologies for holistically evaluating SES, contribute to this process.

Politicians and activists are calling for “green new deals” to emulate Roosevelt’s New Deal following the Great Depression, but with an emphasis on climate justice and ecosystem restoration. This encompasses sustainable land use, by default, and important components of that must be soil protection and land preservation, consistent with climate justice, including the attendant social and economic incentives necessary to achieve local and global climate goals, biodiversity objectives and food security. Profound reform can reach a tipping point when society perceives a binary moral choice that serves the common good. Two successful examples of global co-operation of this kind were the signing of the *Montreal Protocol* in 1987 to protect the ozone layer, and persuading the global community to reduce GHG emissions in the *Paris Agreement*, though hard-won and far from over. Ginzky cites other examples and, despite poor progress to date, suggests that a binding international treaty on soils is both necessary and achievable [274].

Clearly a lack of effective soil governance is often more significant a problem than an absence of legal instruments. The lesson here, for saving our soil, is to strive for a clear message, backed up by consistent policies and laws, and sound criteria to decide how to prioritise soil types or areas of land. This alone cannot solve the vast problem, but it will make the process more streamlined and more transparent, and facilitate governance for those who genuinely want to govern, and make infringements that bit more difficult to effect and pass unnoticed. The urgent need is to couple a compelling case with an achievable solution.

Data accessibility

The only original data presented is the categorisation percentages presented in Table 3.

Authors' contributions

L.R.P. conceived the idea, conducted much of the research and wrote the bulk of the text. C.R. made considerable contributions to the text, especially in relation to soil law, and performed much editing.

Competing interests

We declare we have no competing interests.

Funding

We received no funding for this study.

Acknowledgements

We wish to thank the following who helped to explain their local soil and land laws, or planning regulations, in theory and practice: Dr Thomas Daniels (Univ. Pennsylvania); Dr Mark Lapping (Univ. Southern Maine); Welsh Government Soil Policy Team (Natural Resources Wales); Planning and Local Government Specialist Team, Soils (Natural England); Prof. Fernando García Prechac (Univ. Uruguay, and former Director General of Natural Resources, Ministry of Livestock, Agriculture and Fisheries, Uruguay); Marcel Achkar (Univ. Uruguay); Gundula Prokop (EU Research and Territorial Co-operation Projects, Federal Environment Agency, Austria); Anna Shortly (Greenbelt Foundation, Ontario); Dr Elena Havlicek and Ruedi Stähli (Federal Office for the Environment (FOEN), Switzerland); Ljubiša Bezbradica (Institute of Architecture and Urban & Spatial Planning of Serbia); Tom Corser (New Zealand Ministry for the Environment); and Dr Suhail Ahmad (Cluster University Of Srinagar, Kashmir). In addition: Dr David Dent (former Director, ISRIC) gave advice on soil degradation assessments which he co-authored at ISRIC; CPRE posted a free copy of *Valuing the Land* which is unavailable online; and two anonymous reviewers helped us to improve the balance and style of the paper.

References

1. Brussaard, L. 2021 Biodiversity and ecosystem functioning in soil: The dark side of nature and the bright side of life. *Ambio*. <https://doi.org/10.1007/s13280-021-01507-z>
2. Smith, P, Keesstra, SD, Silver, WL, Adhya, TK, De Deyn, GB, Carvalheiro, LG, Giltrap, DL, Renforth, P, Cheng, K, Sarkar, B, et al. 2021 Soil-derived Nature's Contributions to People and their contribution to the UN Sustainable Development Goals. *Philosophical Transactions of the Royal Society B: Biological Sciences* **376**, 20200185. <https://doi.org/doi:10.1098/rstb.2020.0185>

3. Almond, REA, Grooten, M & Petersen, T. 2020 WWF (2020) Living Planet Report 2020 - Bending the curve of biodiversity loss. eds. REA Almond, M Grooten & T Petersen. Gland, Switzerland, WWF.
4. FAO. FAOLEX Database: <http://www.fao.org/faolex/country-profiles/en/>.
5. Government of Ukraine. 2020 Land Code (No. 2786-III of 2001; revised 16.10.2020).
6. Bennett, HH. 1939 *Soil conservation*. New York, London, McGraw-Hill.
7. Butzer, K. 2005 Environmental history in the Mediterranean world: cross-disciplinary investigation of cause-and-effect for degradation and soil erosion. *Journal of Archaeological Science* **32**, 1773-1800. (doi: 1710.1016/j.jas.2005.1706.1001).
8. Dotterweich, M. 2013 The history of human-induced soil erosion: Geomorphic legacies, early descriptions and research, and the development of soil conservation—A global synopsis. *Geomorphology* **201**, 1-34. <https://doi.org/10.1016/j.geomorph.2013.07.021>
9. Hyams, E. 1952 *Soil and Civilization*, Thames and Hudson.
10. Redman, CL. 1999 *Human impact on ancient environments*. Tucson, University of Arizona Press.
11. Brevik, EC & Hartemink, AE. 2010 Early soil knowledge and the birth and development of soil science. *CATENA* **83**, 23-33. <https://doi.org/10.1016/j.catena.2010.06.011>
12. Verheye, WH. 2009 *Land Use, Land Cover and Soil Sciences - Volume VII: Soils and Soil Sciences - 2*, EOLSS Publ.
13. Homer-Dixon, TF. 1994 Environmental Scarcities and Violent Conflict: Evidence from Cases. *International Security* **19**, 5-40. <https://doi.org/10.2307/2539147>
14. Berman, N, Couttenier, M & Soubeyran, R. 2019 Fertile Ground for Conflict. *Journal of the European Economic Association*. <https://doi.org/10.1093/ieea/jvz068>
15. Fearon, J & Laitin, D. 2014 Does Contemporary Armed Conflict Have 'Deep Historical Roots'? *SSRN Electronic Journal*. <https://doi.org/10.2139/ssrn.1922249>
16. Global Witness. 2020 Defending Tomorrow: The climate crisis and threats against land and environmental defenders.
17. Le Billon, P & Lujala, P. 2020 Environmental and land defenders: Global patterns and determinants of repression. *Global Environmental Change* **65**, 102163. <https://doi.org/10.1016/j.gloenvcha.2020.102163>
18. Gong, Z, Zhang, X, Chen, J & Zhang, G. 2003 Origin and development of soil science in ancient China. *Geoderma* **115**, 3-13. [https://doi.org/10.1016/S0016-7061\(03\)00071-5](https://doi.org/10.1016/S0016-7061(03)00071-5)
19. Barrera-Bassols, N, Alfred Zinck, J & Van Ranst, E. 2006 Symbolism, knowledge and management of soil and land resources in indigenous communities: Ethnopedology at global, regional and local scales. *CATENA* **65**, 118-137. <https://doi.org/10.1016/j.catena.2005.11.001>
20. Ellickson, RC & Thorland, CD. 1995 Ancient Land Law: Mesopotamia, Egypt, Israel. *Chicago-Kent Law Review* **71**, 321.

21. Barbieri-Low, AJ & Yates, RDS. 2015 *Law, State, and Society in Early Imperial China (2 vols)*, Brill.
22. Shi, S. 1959 *On "Fan Shêng-chih shu" : an agriculturist book of China written by Fan Shêng-chih in the first century B.C. / translated and commented upon by Shih Sheng-han = Fan Shengzhi shu jin shi / Shi Shenghan yi shi*. Peking, Science Press.
23. Milde, KF. 1950 Roman Contributions to the Law of Soil Conservation. *Fordham L. Rev.* **19**, 192-196.
24. Stocking, M. 1985 Soil Conservation Policy in Colonial Africa. *Agricultural History* **59**, 148-161.
25. Sandbach, F. 1980 *Environment, Ideology, and Policy*, Allandheld, Osmun.
26. Williams, GW & United States. Forest Service. 2005 *The USDA Forest Service : the first century*. Washington, DC, USDA Forest Service; iv, 156 p. p).
27. Wynn, G. 1979 Pioneers, politicians and the conservation of forests in early New Zealand. *Journal of Historical Geography* **5**, 171-188. [https://doi.org/10.1016/0305-7488\(79\)90132-4](https://doi.org/10.1016/0305-7488(79)90132-4)
28. Reddy, B, Hoag, D & Shobha, B. 2004 Economic incentives for soil conservation in India. In *Proceedings of the 13th International Soil Conservation Organization Conference, Brisbane*.
29. Crofts, R & Olgeirsson, FG. 2011 *Healing the Land: The Story of Land Reclamation and Soil Conservation in Iceland*, Soil conservation service.
30. McLeman, RA, Dupre, J, Berrang Ford, L, Ford, J, Gajewski, K & Marchildon, G. 2014 What we learned from the Dust Bowl: lessons in science, policy, and adaptation. *Population and Environment* **35**, 417-440. <https://doi.org/10.1007/s11111-013-0190-z>
31. Pretty, JN. 1995 *Regenerating agriculture : policies and practice for sustainability and self-reliance*. London, Earthscan.
32. Sylvester, KM & Rupley, ES. 2012 Revising the Dust Bowl: High Above the Kansas Grasslands. *Environ Hist Durh N C* **17**, 603-633. <https://doi.org/10.1093/envhis/ems047>
33. Peake, L. 1986 *Erosion, Crop Yields and Time: A Reassessment of Quantitative Relationships*, University of East Anglia School of Development Studies.
34. Konyushkov, D. 2014 Russian: Evolution and Examples. In *Reference Module in Earth Systems and Environmental Sciences* (Elsevier).
35. Carter-Whitney, M, Esakin, TC & Canadian Institute for Environmental Law and, P. 2010 Ontario's greenbelt in an international context.
36. Simonson, RW. 1989 *Historical Highlights of Soil Survey and Soil Classification with Emphasis on the United States, 1899-1970*. Wageningen, the Netherlands, International Soil Reference and Information Centre (ISRIC).
37. Young, A. 1980 *Tropical Soils and Soil Survey*, Cambridge University Press.
38. Robinson, A. 1943 The Scott and Uthwatt Reports on Land Utilisation. *The Economic Journal* **53**, 28-38. <https://doi.org/10.2307/2226286>

39. Sheail, J. 1997 Scott revisited: Post-war agriculture, planning and the British countryside. *Journal of Rural Studies* **13**, 387-398. [https://doi.org/10.1016/S0743-0167\(97\)00028-4](https://doi.org/10.1016/S0743-0167(97)00028-4)
40. CPRE. 2017 LANDLINES: why we need a strategic approach to land. London, CPRE.
41. Monk, S, Whitehead, C, Burgess, G & Tang, C. 2013 International review of land supply and planning systems. York, UK, Joseph Rowntree Foundation.
42. Natural England. 2012 Agricultural Land Classification: protecting the best and most versatile agricultural land. Technical Information Note TIN049.
43. Daniels, T. 1990 Using LESA in a purchase of development rights program. *Journal of Soil and Water Conservation* **45**, 617-621.
44. Lee, ATM. 1948 Soil Conservation—An International Study, Food and Agriculture Organization of the United Nations, Washington, D. C., 1948. *American Journal of Agricultural Economics* **30**, 784-786. <https://doi.org/10.2307/1232795>
45. United Nations. 1949 *UN Yearbook 1947-48*. Washington.
46. Bunce, AC. 1942 *The economics of soil conservation*. Ames, Ia., Iowa State college Press.
47. Ciriacy-Wantrup, SV. 1952 *Resource Conservation: Economics and Policies*, University of California, Division of Agricultural Sciences, Agricultural Experiment Station.
48. Sauer, EL. 1967 *Economics of Soil Conservation, Reclamation and Rehabilitation*.
49. Morgan, RJ & Future, Rft. 1965 *Governing Soil Conservation: Thirty Years of the New Decentralization*, Resources for the Future.
50. Aubreville, A. 1949 *Climats, Forêts, et Désertification de l'Afrique Tropicale*. Société d'Editions Géographiques, Maritimes et Tropicales. Paris.
51. IPCC. 2019 Climate Change and Land: an IPCC special report on climate change, desertification, land degradation, sustainable land management, food security, and greenhouse gas fluxes in terrestrial ecosystems.
52. Gisladdottir, G & Stocking, M. 2005 Land degradation control and its global environmental benefits. *Land Degradation & Development* **16**, 99-112. <https://doi.org/10.1002/ldr.687>
53. Thomas, DSG & Middleton, NJ. 1994 *Desertification: Exploding the Myth*. London, Wiley.
54. Davidson, J & Wibberley, GP. 1977 *Planning and the rural environment*.
55. Pingali, PL. 2012 Green Revolution: Impacts, limits, and the path ahead. *Proceedings of the National Academy of Sciences* **109**, 12302-12308. <https://doi.org/10.1073/pnas.0912953109>
56. Mannion, AM. 1991 *Global environmental change : a natural and cultural environmental history*. Harlow, Longman Scientific & Technical.
57. Agricultural Advisory Council. 1970 *Modern Farming and the Soil: Report of the Agricultural Advisory Council on Soil Structure and Soil Fertility*. London, HM Stationery Office.

58. Biniek, JP. 1979 *Agricultural and environmental relationships issues and priorities: report*. Washington, Library of Congress, U.S. Govt. Print. Off.; xvi, [1], 696 p. p).
59. USDA. 1973 *Monoculture in Agriculture: Extent, Causes, and Problems-report of the Task Force on Spatial Heterogeneity in Agricultural Landscapes and Enterprises*.
60. Carson, R. 1962 *Silent spring*. Boston, Cambridge, Mass., Houghton Mifflin; Riverside Press; x, 368 p. p).
61. Schneider, SH. 1989 The greenhouse effect: science and policy. *Science* **243**, 771-781.
62. Leahy, S, Clark, H & Reisinger, A. 2020 Challenges and Prospects for Agricultural Greenhouse Gas Mitigation Pathways Consistent With the Paris Agreement. *Frontiers in Sustainable Food Systems* **4**. <https://doi.org/10.3389/fsufs.2020.00069>
63. Shivanna, KR. 2020 The Sixth Mass Extinction Crisis and its Impact on Biodiversity and Human Welfare. *Resonance* **25**, 93-109. <https://doi.org/10.1007/s12045-019-0924-z>
64. Weigelt, J, Müller, A, Janetschek, H & Töpfer, K. 2015 Land and soil governance towards a transformational post-2015 Development Agenda: an overview. *Current Opinion in Environmental Sustainability* **15**, 57-65. <https://doi.org/10.1016/j.cosust.2015.08.005>
65. Juerges, N & Hansjürgens, B. 2016 Soil governance in the transition towards a sustainable bioeconomy – A review. *Journal of Cleaner Production* **170**. <https://doi.org/10.1016/j.jclepro.2016.10.143>
66. Guereña, A. 2016 *Unearthed: Land, power and inequality in Latin America*. OXFAM.
67. van Schaik, L & Dinnissen, R. 2014 *Terra Incognita: Land degradation as underestimated threat amplifier. Clingendael report for the Netherlands Environmental Assessment Agency (PBL)*. The Hague, Clingendael.
68. Baba, SH, Wani, MH, Zargar, BA & Bhat, IF. 2020 Determinants of land degradation in Jammu and Kashmir: implications for land governance. *Agricultural Economics Research Review* **32**, 303645.
69. Sloan, T, Payne, R, Anderson, R, Bain, C, Chapman, S, Cowie, N, Gilbert, PJ, Lindsay, R, Mauquoy, D, Newton, A, et al. 2018 Peatland afforestation in the UK and consequences for carbon storage. *Mires and Peat* **23**. <https://doi.org/10.19189/Map.2017.OMB.315>
70. Bai, ZG, Dent, DL, Olsson, L & Schaepman, ME. 2008 Proxy global assessment of land degradation. *Soil Use and Management* **24**, 223-234. <https://doi.org/10.1111/j.1475-2743.2008.00169.x>
71. Sonneveld, BGJS & Dent, DL. 2009 How good is GLASOD? *J. Environ. Manage.* **90**, 274-283. <https://doi.org/10.1016/j.jenvman.2007.09.008>
72. Bai, ZG, Dent, DL, Olsson, L, Tengberg, A, Tucker, C & Yengoh, G. 2015 A longer, closer, look at land degradation. *Agriculture for Development* **24**, 3-9.
73. FAO & ITPS. 2015 *The Status of the World's Soil Resources*.
74. Nkonya, E, Mirzabaev, A & von Braun, J. 2016 *Economics of Land Degradation and Improvement – A Global Assessment for Sustainable Development*, Springer Nature.

75. Esch, S, Brink, B, Stehfest, E, Bakkenes, M, Sewell, A, Bouwman, A, Meijer, J, Westhoek, H & van den Berg, M. 2017 *Exploring future changes in land use and land condition and the impacts on food, water, climate change and biodiversity: Scenarios for the UNCCD Global Land Outlook*.
76. United Nations Convention to Combat Desertification. 2017 *The Global Land Outlook*, first edition. Bonn, Germany.
77. IPBES. 2018 *The IPBES assessment report on land degradation and restoration. Secretariat of the Intergovernmental Science-Policy Platform on Biodiversity and Ecosystem Services*. Bonn, Germany.
78. Zimmerer, KS. 2011 " Conservation booms" with agricultural growth? Sustainability and Shifting Environmental Governance in Latin America, 1985-2008 (Mexico, Costa Rica, Brazil, Peru, Bolivia). *Latin American Research Review*, 82-114.
79. Stubenrauch, J, Garske, B & Ekardt, F. 2018 Sustainable Land Use, Soil Protection and Phosphorus Management from a Cross-National Perspective. *Sustainability* **10**, 1988. <https://doi.org/10.3390/su10061988>
80. 2017 The most neglected threat to public health in China is toxic soil; and fixing it will be hard and costly. In *The Economist*.
81. Liqiang, H. 2019 New law on soil pollution will pinpoint responsibility: <http://www.chinadaily.com.cn/a/201901/02/WS5c2c0b6fa310d91214051f8b.html>. In *China Daily*.
82. Li, T, Liu, Y, Lin, S, Liu, Y & Xie, Y. 2019 Soil Pollution Management in China: A Brief Introduction. *Sustainability* **11**, 556. <https://doi.org/10.3390/su11030556>
83. Hendlin, YH. 2014 From Terra Nullius to Terra Communis Reconsidering Wild Land in an Era of Conservation and Indigenous Rights. *Environmental Philosophy* **11**, 141-174. <https://doi.org/10.5281/zenodo.260245>
84. Pearce, F. 2018 *Conflicting Data: How Fast Is the World Losing its Forests?* Yale Environment 360, Yale School of the Environment.
85. Curtis, PG, Slay, CM, Harris, NL, Tyukavina, A & Hansen, MC. 2018 Classifying drivers of global forest loss. *Science* **361**, 1108-1111. <https://doi.org/10.1126/science.aau3445>
86. Escobar, H. 2020 Deforestation in the Brazilian Amazon is still rising sharply. *Science* **369**, 613-613. <https://doi.org/10.1126/science.369.6504.613>
87. Young, A. 1999 Is there Really Spare Land? A Critique of Estimates of Available Cultivable Land in Developing Countries. *Environment, Development and Sustainability* **1**, 3-18. <https://doi.org/10.1023/A:1010055012699>
88. Chamberlin, J, Jayne, TS & Headey, D. 2014 Scarcity amidst abundance? Reassessing the potential for cropland expansion in Africa. *Food Policy* **48**, 51-65. <https://doi.org/10.1016/j.foodpol.2014.05.002>
89. Güneralp, B, Lwasa, S, Masundire, H, Parnell, S & Seto, KC. 2017 Urbanization in Africa: challenges and opportunities for conservation. *Environmental Research Letters* **13**, 015002. <https://doi.org/10.1088/1748-9326/aa94fe>

90. Rockström, J, Steffen, W, Noone, K, Persson, Å, Chapin, FS, Lambin, EF, Lenton, TM, Scheffer, M, Folke, C, Schellnhuber, HJ, et al. 2009 A safe operating space for humanity. *Nature* **461**, 472-475. <https://doi.org/10.1038/461472a>
91. Byerlee, D, Stevenson, J & Villoria, N. 2014 Does intensification slow crop land expansion or encourage deforestation? *Global Food Security* **3**, 92-98. <https://doi.org/10.1016/j.gfs.2014.04.001>
92. Abu Hatab, A, Cavinato, MER, Lindemer, A & Lagerkvist, C-J. 2019 Urban sprawl, food security and agricultural systems in developing countries: A systematic review of the literature. *Cities* **94**, 129-142. <https://doi.org/10.1016/j.cities.2019.06.001>
93. Metternicht, G. 2018 *Land Use and Spatial Planning: Enabling Sustainable Management of Land Resources*, Springer International Publishing.
94. Seitzinger, SP, Svedin, U, Crumley, CL, Steffen, W, Abdullah, SA, Alfsen, C, Broadgate, WJ, Biermann, F, Bondre, NR, Dearing, JA, et al. 2012 Planetary Stewardship in an Urbanizing World: Beyond City Limits. *AMBIO* **41**, 787-794. <https://doi.org/10.1007/s13280-012-0353-7>
95. Smith, P, Gregory, PJ, Vuuren, Dv, Obersteiner, M, Havlík, P, Rounsevell, M, Woods, J, Stehfest, E & Bellarby, J. 2010 Competition for land. *Philosophical Transactions of the Royal Society B: Biological Sciences* **365**, 2941-2957. <https://doi.org/10.1098/rstb.2010.0127>
96. FAO. 2011 Payments for ecosystem services and food security.
97. van Vliet, J, Eitelberg, DA & Verburg, PH. 2017 A global analysis of land take in cropland areas and production displacement from urbanization. *Global Environmental Change* **43**, 107-115. <https://doi.org/10.1016/j.gloenvcha.2017.02.001>
98. Bren d'Amour, C, Reitsma, F, Baiocchi, G, Barthel, S, Güneralp, B, Erb, KH, Haberl, H, Creutzig, F & Seto, KC. 2017 Future urban land expansion and implications for global croplands. *Proc Natl Acad Sci U S A* **114**, 8939-8944. <https://doi.org/10.1073/pnas.1606036114>
99. Chen, G, Li, X, Liu, X, Chen, Y, Liang, X, Leng, J, Xu, X, Liao, W, Qiu, Y, Wu, Q, et al. 2020 Global projections of future urban land expansion under shared socioeconomic pathways. *Nature Communications* **11**. <https://doi.org/10.1038/s41467-020-14386-x>
100. Gardi, C. 2017 *Urban expansion, land cover and soil ecosystem services*, Taylor & Francis.
101. European Environment Agency. 2017 Landscapes in transition: An account of 25 years of land cover change in Europe. In *EEA Report*. Copenhagen.
102. CPRE. 2000 Valuing the land: planning for the best and most versatile agricultural land.
103. Thurston, N, Kenyon, D, Starkings, D & Taylor, K. 2011 Review of the weight that should be given to the protection of best and most versatile (BMV) land., Defra.
104. Ustaoglu, E & Williams, B. 2017 Determinants of Urban Expansion and Agricultural Land Conversion in 25 EU Countries. *Environmental Management* **60**, 717-746. <https://doi.org/10.1007/s00267-017-0908-2>

105. OECD. 2009 Farmland Conversion – The Spatial Implications of Agricultural and Land-use Policies. ed. D Diakosavvas. Paris, OECD.
106. Perrin, C, Clément, C, Melot, R & Nougarèdes, B. 2020 Preserving Farmland on the Urban Fringe: A Literature Review on Land Policies in Developed Countries. *Land* **9**, 223. <https://doi.org/10.3390/land9070223>
107. Silva, C. 2019 Auckland's Urban Sprawl, Policy Ambiguities and the Peri-Urbanisation to Pukekohe. *Urban Science* **3**, 1. <https://doi.org/10.3390/urbansci3010001>
108. Indian Government. 2009 Report of the Committee on State Agrarian Relations and the Unfinished Task in Land Reforms.
109. OECD. 2010 *Regional Development Policies in OECD Countries.*, OECD Publishing.
110. Nishi, M. 2019 Multi-Level Governance of Agricultural Land in Japan: Farmers' Perspectives and Responses to Farmland Banking. New York, Columbia.
111. Oberndorf, RB. 2012 Legal review of recently enacted farmland law and vacant, fallow and virgin lands management law: improving the legal & policy frameworks relating to land management in Myanmar. *Food Security Working Group and Land Core Group.*
112. Fernández-Maldonado, AM. 2019 Unboxing the Black Box of Peruvian Planning. *Planning Practice & Research* **34**, 368-386. <https://doi.org/10.1080/02697459.2019.1618596>
113. Slätmo, E. 2017 Preservation of Agricultural Land as an Issue of Societal Importance. *Rural Landscapes: Society, Environment, History* **4**. <https://doi.org/10.16993/rl.39>
114. Jain, M. 2018 Contemporary urbanization as unregulated growth in India: The story of census towns. *Cities* **73**, 117-127. <https://doi.org/10.1016/j.cities.2017.10.017>
115. Sili, M & Soumoulou, L. 2011 The issue of land in Argentina: Conflicts and dynamics of use, holdings and concentration. Rome, IFAD.
116. Cho, J. 2005 Urban Planning and Urban Sprawl in Korea. *Urban Policy and Research* **23**, 203-218. <https://doi.org/10.1080/08111470500143304>
117. Debolini, M, Valette, E, François, M & Chéry, J-P. 2015 Mapping land use competition in the rural–urban fringe and future perspectives on land policies: A case study of Meknès (Morocco). *Land Use Pol.* **47**, 373-381. <https://doi.org/10.1016/j.landusepol.2015.01.035>
118. Naab, FZ, Dinye, R & Kasanga, RK. 2013 Urbanisation and its impact on agricultural lands in growing cities in developing countries: a case study of Tamale in Ghana. *European scientific journal* **2**, 256-287.
119. Mori, H. 1998 Land Conversion at the Urban Fringe: A Comparative Study of Japan, Britain and the Netherlands. *Urban Studies* **35**, 1541-1558. <https://doi.org/10.1080/0042098984277>
120. Ladu, JLC, Athiba, AL & Ondogo, EC. 2019 An Assessment of the Impact of Urbanization on Agricultural Land Use in Juba City, Central Equatoria State, Republic of South Sudan. *Journal of Applied Agricultural Economics and Policy Analysis* **2**, 22-30. <https://doi.org/10.12691/jaaepa-2-1-4>

121. Ullah, S, Khan, MA, Rahman, A & Mahmood, S. 2019 Evaluation of urban encroachment on farmland: a threat to urban agriculture in Peshawar City District, Pakistan. *Erdkunde* **73**, 127-142. <https://doi.org/10.3112/erdkunde.2019.02.04>
122. Visser, O. 2017 Running out of farmland? Investment discourses, unstable land values and the sluggishness of asset making. *Agric Human Values* **34**, 185-198. <https://doi.org/10.1007/s10460-015-9679-7>
123. Satterthwaite, D, McGranahan, G & Tacoli, C. 2010 Urbanization and its implications for food and farming. *Philosophical Transactions of the Royal Society B: Biological Sciences* **365**, 2809-2820. <https://doi.org/10.1098/rstb.2010.0136>
124. Cottrell, RS, Nash, KL, Halpern, BS, Remenyi, TA, Corney, SP, Fleming, A, Fulton, EA, Hornborg, S, John, A, Watson, RA, et al. 2019 Food production shocks across land and sea. *Nature Sustainability* **2**, 130-137. <https://doi.org/10.1038/s41893-018-0210-1>
125. Prishchepov, AV, Müller, D, Dubinin, M, Baumann, M & Radeloff, VC. 2013 Determinants of agricultural land abandonment in post-Soviet European Russia. *Land Use Pol.* **30**, 873-884. <https://doi.org/10.1016/j.landusepol.2012.06.011>
126. Lambin, EF & Meyfroidt, P. 2011 Global land use change, economic globalization, and the looming land scarcity. *Proceedings of the National Academy of Sciences* **108**, 3465-3472. <https://doi.org/10.1073/pnas.1100480108>
127. Kaz'min, MA. 2016 Transformation of agricultural land use in Russian regions in the course of modern socioeconomic reforms. *Regional Research of Russia* **6**, 87-94. <https://doi.org/10.1134/S2079970516010056>
128. Liefert, WM, Liefert, O, Vocke, G & Allen, EW. 2010 Former Soviet Union Region To Play Larger Role in Meeting World Wheat Needs. In *Amber Waves*, pp. 12-19.
129. Meyfroidt, P, Schierhorn, F, Prishchepov, AV, Müller, D & Kuemmerle, T. 2016 Drivers, constraints and trade-offs associated with recultivating abandoned cropland in Russia, Ukraine and Kazakhstan. *Global Environmental Change* **37**, 1-15. <https://doi.org/10.1016/j.gloenvcha.2016.01.003>
130. Schierhorn, F, Müller, D, Beringer, T, Prishchepov, AV, Kuemmerle, T & Balman, A. 2013 Post-Soviet cropland abandonment and carbon sequestration in European Russia, Ukraine, and Belarus. *Global Biogeochemical Cycles* **27**, 1175-1185. <https://doi.org/10.1002/2013gb004654>
131. Amati, M & Taylor, L. 2010 From Green Belts to Green Infrastructure. *Planning Practice & Research* **25**, 143-155. <https://doi.org/10.1080/02697451003740122>
132. Papworth, T. 2015 The green noose. *Adam Smith Institute* **14**.
133. Cheshire, P. 2014 Turning houses into gold: the failure of British planning. Centre for Economic Performance, LSE.
134. Defra. 2020 Farming for the future: Policy and progress update. UK Government.
135. Shortly & A. (Greenbelt Foundation, Toronto), Personal Communication.

136. Carter-Whitney, M, Esakin, TC & Canadian Institute for Environmental Law and, P. 2010 Ontario's greenbelt in an international context. Toronto, Ont., Canadian Institute for Environmental Law and Policy.
137. Loghrin, H. Forthcoming Global Greenbelts Updates since 2009 (2019). Toronto, Greenbelt Foundation.
138. Suu, NV. 2009 Agricultural land conversion and its effects on farmers in contemporary Vietnam. *Focaal* **2009**, 106. <https://doi.org/10.3167/fcl.2009.540109>
139. Paudel, B, Pandit, J & Reed, B. 2013 Fragmentation and conversion of agriculture land in Nepal and Land Use Policy 2012.
140. Chen, A, He, H, Wang, J, Li, M, Guan, Q & Hao, J. 2019 A Study on the Arable Land Demand for Food Security in China. *Sustainability* **11**, 4769. <https://doi.org/10.3390/su11174769>
141. Govindaprasad, PK & Manikandan, K. 2016 Farm Land Conversion and Food Security: Empirical Evidences from Three Villages of Tamil Nadu. In *Indian Journal of Agricultural Economics*, pp. 493-503.
142. Islam, GMT, Islam, A, Shopan, AA, Rahman, MM, Lázár, AN & Mukhopadhyay, A. 2015 Implications of agricultural land use change to ecosystem services in the Ganges delta. *J Environ Manage* **161**, 443-452. <https://doi.org/10.1016/j.jenvman.2014.11.018>
143. Robson, JS, Ayad, HM, Wasfi, RA & El-Geneidy, AM. 2012 Spatial disintegration and arable land security in Egypt: A study of small- and moderate-sized urban areas. *Habitat International* **36**, 253-260. <https://doi.org/10.1016/j.habitatint.2011.10.001>
144. Dai, W & Dai, W. 2019 Effects of urban expansion on environment by morphological study. *IOP Conference Series: Earth and Environmental Science* **227**, 052004. <https://doi.org/10.1088/1755-1315/227/5/052004>
145. Agrimonde-Terra. 2018 *Land Use and Food Security in 2050: a Narrow Road*, éditions Quae.
146. Bezbradica, L, Pantić, M & Gajić, A. 2019 The land use and soil protection: Planning and legal regulations in Serbia. *Zemljiste i biljka* **68**, 51-71. <https://doi.org/10.5937/ZemBilj1902051B>
147. Owusu Ansah, B & Chigbu, UE. 2020 The Nexus between Peri-Urban Transformation and Customary Land Rights Disputes: Effects on Peri-Urban Development in Trede, Ghana. *Land* **9**, 187. <https://doi.org/10.3390/land9060187>
148. Jiang, L & Zhang, Y. 2016 Modeling Urban Expansion and Agricultural Land Conversion in Henan Province, China: An Integration of Land Use and Socioeconomic Data. *Sustainability* **8**, 920. <https://doi.org/10.3390/su8090920>
149. Li, S. 2018 Change detection: how has urban expansion in Buenos Aires metropolitan region affected croplands. *International Journal of Digital Earth* **11**, 195-211. <https://doi.org/10.1080/17538947.2017.1311954>
150. Carreño, L, Frank, FC & Viglizzo, EF. 2012 Tradeoffs between economic and ecosystem services in Argentina during 50 years of land-use change. *Agriculture, Ecosystems & Environment* **154**, 68-77. <https://doi.org/10.1016/j.agee.2011.05.019>

151. Meadows, D & Randers, J. 2004 *Limits to Growth: The 30-Year Update*, Chelsea Green Publishing.
152. Schumacher, EF. 1973 *Small is beautiful; economics as if people mattered*. New York, Harper & Row.
153. Kuper, S. 2019 Thoreau, Leopold, & Carson: Challenging Capitalist Conceptions of the Natural Environment. *Consilience* **0**. <https://doi.org/10.7916/consilience.v0i13.3927>
154. Kennel, CF. The gathering anthropocene crisis. *The Anthropocene Review* **2**, 81-98. <https://doi.org/10.1177/2053019620957355>
155. Żróbek-Różańska, A & Zielińska-Szczepkowska, J. 2019 National Land Use Policy against the Misuse of the Agricultural Land—Causes and Effects. Evidence from Poland. *Sustainability* **11**, 6403. <https://doi.org/10.3390/su11226403>
156. McPike, JL. 2015 Creating Space for the Formal Amongst the Informal: An Examination of Urban Housing Policies, State Power, and Multi-Scalar Politics in Indian Cities. In *RC21 International Conference on “The Ideal City: between myth and reality. Representations, policies, contradictions and challenges for tomorrow's urban life” Urbino (Italy) 27-29 August 2015*.
157. Bartz, D. 2015 *Soil atlas: Facts and figures about earth, land and fields*. Berlin.
158. Appiah, DO, Asante, F & Nketiah, B. 2019 Perspectives on Agricultural Land Use Conversion and Food Security in Rural Ghana. *Sci* **1**, 14. <https://doi.org/10.3390/sci1010014.v1>
159. Schueler, V, Kuemmerle, T & Schröder, H. 2011 Impacts of Surface Gold Mining on Land Use Systems in Western Ghana. *AMBIO* **40**, 528-539. <https://doi.org/10.1007/s13280-011-0141-9>
160. Franco, J & Borras Jr, SM. 2013 *Land concentration, land grabbing and people's struggles in Europe*. Transnational Institute, Amsterdam.
161. Khalid, H & Yusuf, MD. 2012 *Resource management: Fragmentation of land ownership and its impact on sustainability of agriculture*.
162. Prokop, G, Jobstmann, H & Schönbauer, A. 2011 *Report on Best Practices for Limiting Soil Sealing and Mitigating Its Effects*. European Communities.
163. Scalenghe, R & Marsan, FA. 2009 The anthropogenic sealing of soils in urban areas. *Landscape and Urban Planning* **90**, 1-10. <https://doi.org/10.1016/j.landurbplan.2008.10.011>
164. Tobias, S, Conen, F, Duss, A, Wenzel, LM, Buser, C & Alewell, C. 2018 Soil sealing and unsealing: State of the art and examples. *Land Degradation & Development* **29**, 2015-2024. <https://doi.org/https://doi.org/10.1002/ldr.2919>
165. Daniels, T & Lapping, M. 2005 Land Preservation: An Essential Ingredient in Smart Growth. *Journal of Planning Literature* **19**, 316-329. <https://doi.org/10.1177/0885412204271379>
166. Desrousseaux, M, Schmitt, B, Billet, P, Béchet, B, Le Bissonnais, Y & Ruas, A. 2019 Artificialised Land and Land Take: What Policies Will Limit Its Expansion and/or Reduce Its Impacts? In *International Yearbook of Soil Law and Policy 2018* (eds. H Ginzky, E Dooley, IL Heuser, E Kasimbazi, T Markus & T Qin), pp. 149-165. Cham, Springer International Publishing).

167. Hellerstein, DR. 2002 *Farmland protection : the role of public preferences for rural amenities*. Washington, D.C., U.S. Dept. of Agriculture, Economic Research Service.
168. Gosnell, H, Kline, JD, Chrostek, G & Duncan, J. 2011 Is Oregon's land use planning program conserving forest and farm land? A review of the evidence. *Land Use Pol.* **28**, 185-192. <https://doi.org/10.1016/j.landusepol.2010.05.012>
169. Ludlow, D, Falconi, M, Carmichael, L, Croft, N, Di Leginio, M, Fumanti, F, Sheppard, A & Smith, N. 2013 Land Planning and Soil Evaluation Instruments in EEA Member and Cooperating Countries (with inputs from Eionet NRC Land Use and Spatial Planning). Final Report for EEA from ETC/SIA (EEA project managers: G. Louwagie and G. Dige). Available at: <http://www.eea.europa.eu/themes/landuse/document-library>. EEA & ETC/SIA.
170. Grass, I, Loos, J, Baensch, S, Batáry, P, Librán-Embid, F, Ficiciyan, A, Klaus, F, Riechers, M, Rosa, J, Tiede, J, et al. 2019 Land-sharing/-sparing connectivity landscapes for ecosystem services and biodiversity conservation. *People and Nature* **1**, 262-272. <https://doi.org/10.1002/pan3.21>
171. Huang, Q, Lu, J, Li, M, Chen, Z & Li, F. 2015 Developing Planning Measures to Preserve Farmland: A Case Study from China. *Sustainability* **7**, 13011-13028. <https://doi.org/10.3390/su71013011>
172. Indian Government. 2003 Law for the preservation of the agricultural lands.
173. Sartori, D, Catalano, G, Genco, M, Pancotti, C, Sirtori, E, Vignetti, S & Bo, C. 2014 Guide to Cost-benefit Analysis of Investment Projects. Economic appraisal tool for Cohesion Policy 2014-2020.
174. Malczewski, J. 2004 GIS-based land-use suitability analysis: a critical overview. *Progress in Planning* **62**, 3-65. <https://doi.org/10.1016/j.progress.2003.09.002>
175. Dent, D & Young, A. 1981 *Soil Survey and Land Evaluation*, Allen & Unwin.
176. Baveye, PC, Baveye, J & Gowdy, J. 2016 Soil "Ecosystem" Services and Natural Capital: Critical Appraisal of Research on Uncertain Ground. *Frontiers in Environmental Science* **4**. <https://doi.org/10.3389/fenvs.2016.00041>
177. Bouwman, AF & Sombroek, WG. 1990 Inputs to climatic change by soils and agriculture related activities: Present status and possible future trends. In *Soils on a Warmer Earth* (pp. 15 - 30. Amsterdam, Elsevier).
178. Bouwman, AF. 1990 Land use related sources of greenhouse gases: Present emissions and possible future trends. *Land Use Pol.* **7**, 154-164. [https://doi.org/10.1016/0264-8377\(90\)90006-K](https://doi.org/10.1016/0264-8377(90)90006-K)
179. Batjes, NH & Bridges, EM. 1992 *A Review of Soil Factors and Processes that Control Fluxes of Heat, Moisture and Greenhouse Gases*. Wageningen, The Netherlands., ISRIC.
180. Weart, SR. 2008 *The Discovery of Global Warming: Revised and Expanded Edition*, Harvard University Press.
181. Wall, DH. 2004 *Sustaining Biodiversity and Ecosystem Services in Soils and Sediments*, Island Press.

182. Karlen, DL, Mausbach, MJ, Doran, JW, Cline, RG, Harris, RF & Schuman, GE. 1997 Soil Quality: A Concept, Definition, and Framework for Evaluation (A Guest Editorial). *Soil Science Society of America Journal* **61**, 4-10. <https://doi.org/10.2136/sssaj1997.03615995006100010001x>
183. Hannam, I & Boer, B. 2002 Legal and institutional frameworks for sustainable soils.
184. Costanza, R, d'Arge, R, de Groot, R, Farber, S, Grasso, M, Hannon, B, Limburg, K, Naeem, S, O'Neill, RV, Paruelo, J, et al. 1997 The value of the world's ecosystem services and natural capital. *Nature* **387**, 253-260. <https://doi.org/10.1038/387253a0>
185. Breure, AM, De Deyn, GB, Dominati, E, Eglin, T, Hedlund, K, Van Orshoven, J & Posthuma, L. 2012 Ecosystem services: a useful concept for soil policy making! *Current Opinion in Environmental Sustainability* **4**, 578-585. <https://doi.org/10.1016/j.cosust.2012.10.010>
186. Lescourret, F, Magda, D, Richard, G, Adam-Blondon, A-F, Bardy, M, Baudry, J, Doussan, I, Dumont, B, Lefèvre, F, Litrico, I, et al. 2015 A social–ecological approach to managing multiple agro-ecosystem services. *Current Opinion in Environmental Sustainability* **14**, 68-75. <https://doi.org/10.1016/j.cosust.2015.04.001>
187. Robinson, DA, Hockley, N, Cooper, DM, Emmett, BA, Keith, AM, Lebron, I, Reynolds, B, Tipping, E, Tye, AM, Watts, CW, et al. 2013 Natural capital and ecosystem services, developing an appropriate soils framework as a basis for valuation. *Soil Biology and Biochemistry* **57**, 1023-1033. <https://doi.org/10.1016/j.soilbio.2012.09.008>
188. Ellili-Bargaoui, Y, Walter, C, Lemercier, B & Michot, D. 2021 Assessment of six soil ecosystem services by coupling simulation modelling and field measurement of soil properties. *Ecological Indicators* **121**, 107211. <https://doi.org/10.1016/j.ecolind.2020.107211>
189. Drobnik, T, Greiner, L, Keller, A & Grêt-Regamey, A. 2018 Soil quality indicators – From soil functions to ecosystem services. *Ecological Indicators* **94**, 151-169. <https://doi.org/https://doi.org/10.1016/j.ecolind.2018.06.052>
190. Dominati, E, Mackay, A, Green, S & Patterson, M. 2014 A soil change-based methodology for the quantification and valuation of ecosystem services from agro-ecosystems: A case study of pastoral agriculture in New Zealand. *Ecological Economics* **100**, 119-129. <https://doi.org/10.1016/j.ecolecon.2014.02.008>
191. Daniels, T. 2020 (Professor of City and Regional Planning, Weitzman School of Design, University of Pennsylvania), Personal communication.
192. Defra. 2020 EIA (Agriculture) regulations: apply to make changes to rural land.
193. 2019 Scottish Government, Guidance on consideration of soil in Strategic Environmental Assessment , v5, 5.4.19.
194. Caldwell, WJ, Hilts, S & Wilton, B. 2017 *Farmland Preservation: Land for Future Generations*, University of Manitoba Press.
195. Nolon, J. 2003 Land Preservation. *Pace Law Faculty Publications*.

196. Robinson, DA, Jackson, BM, Clothier, BE, Dominati, EJ, Marchant, SC, Cooper, DM & Bristow, KL. 2013 Advances in Soil Ecosystem Services: Concepts, Models, and Applications for Earth System Life Support. *Vadose Zone Journal* **12**, vzj2013.2001.0027.
<https://doi.org/https://doi.org/10.2136/vzj2013.01.0027>
197. Meadows, DH. 1972 *The Limits to growth; a report for the Club of Rome's project on the predicament of mankind*, New York : Universe Books, [1972].
198. Norström, AV, Cvitanovic, C, Löf, MF, West, S, Wyborn, C, Balvanera, P, Bednarek, AT, Bennett, EM, Biggs, R, de Bremond, A, et al. 2020 Principles for knowledge co-production in sustainability research. *Nature Sustainability* **3**, 182-190. <https://doi.org/10.1038/s41893-019-0448-2>
199. Erinosh, BT. 2013 The Revised African Convention on the Conservation of Nature and Natural Resources: Prospects for a Comprehensive Treaty for the Management of Africa's Natural Resources. *African Journal of International and Comparative Law* **21**, 378-397.
<https://doi.org/10.3366/ajicl.2013.0069>
200. FAO. 1971 *Land degradation*. Rome, Food and Agriculture Organization of the United Nations.
201. FAO. 1971 *Legislative principles of soil conservation*.
202. Boer, B, Ginzky, H & Heuser, I. 2017 International Soil Protection Law: History, Concepts and Latest Developments. (pp. 49-72.
203. Bodle, R, Stockhaus, H, Wolff, F, Scherf, C-S & Oberthür, S. 2019 *Improving international soil governance - Analysis and recommendations*. Berlin, Ecologic Institute and Öko-Institute.
204. Europe, Co. 1972 European Soil Charter. ed. Co Europe. Brussels.
205. Tóth, Z. 2018 International dimensions of EU soil policy – The main binding and non-binding legal instruments. *Hungarian Journal of Legal Studies Acta Juridica Hungarica* **59**, 290.
<https://doi.org/10.1556/2052.2018.59.3.4>
206. Byron-Cox, R. 2020 From Desertification to Land Degradation Neutrality: The UNCCD and the Development of Legal Instruments for Protection of Soils. In *Legal Instruments for Sustainable Soil Management in Africa* (eds. H Yahyah, H Ginzky, E Kasimbazi, R Kibugi & OC Ruppel), pp. 1-13. Cham, Springer International Publishing).
207. FAO. 1981 World Soil Charter.
208. UNEP. 1982 World Soils Policy.
209. Wood & Harold, W. 1985 The United Nations World Charter for Nature: The Developing Nations' Initiative to Establish Protections for the Environment. *Ecology Law Quarterly* **12**, 977.
210. World Commission on Environment and Development. 1987 *Our common future*. Oxford; New York, Oxford University Press.
211. World Bank. 2002 *The First Decade of the GEF: second overall performance study*. Washington DC, World Bank.

212. Wolff, F & Kaphengst, T. 2017 The UN Convention on Biological Diversity and Soils: Status and Future Options. In *International Yearbook of Soil Law and Policy 2016* (eds. H Ginzky, IL Heuser, T Qin, OC Ruppel & P Wegerdt), pp. 129-148. Cham, Springer International Publishing).
213. CBD/SBSTTA/24/7/Rev.1. 4 December 2020 Review of the international initiative for the conservation and sustainable use of soil biodiversity and updated plan of action.
214. United Nations. 1994 Convention to Combat Desertification in those Countries Experiencing Serious Drought and/or Desertification, Particularly in Africa. In *International Legal Materials*, pp. 1328-1382, 2017/02/27 ed, Cambridge University Press.
215. Global Environment Facility (GEF). 1999 Clarifying Linkages Between Land Degradation And The GEF Focal Areas: An Action Plan For Enhancing GEF Support: An Action Plan for Enhancing GEF Support, GEF/C.14/4, November 17, 1999. Washington DC, World Bank Group.
216. Hannam, I & Boer, B. 2019 Land degradation and international environmental law. In *Response to Land Degradation* (ed. IDH E. Michael Bridges, L. Roel Oldeman, Frits W.T. Penning de Vries, Sara J. Scherr, Samran Sombatpanit, Robin N. Leslie, Tanadol Compo, Apuntree Prueksapong), pp. 429-440.
217. FAO and UNEP. 2020 *Legislative approaches to sustainable agriculture and natural resources governance*. FAO Legislative Study No. 114. Rome, Food and Agriculture Organization of the United Nations.
218. Chasek, P, Safriel, U, Shikongo, S & Fuhrman, VF. 2015 Operationalizing Zero Net Land Degradation: The next stage in international efforts to combat desertification? *Journal of Arid Environments* **112**, 5-13. <https://doi.org/10.1016/j.jaridenv.2014.05.020>
219. United Nations Framework Convention on Climate Change, 9 May 1992 (in force 21 March 1994), 31 ILM 849 ; 1771 UNTS 10 7. (“UNFCCC”).
220. Fee, E. 2019 Implementing the Paris Climate Agreement: Risks and Opportunities for Sustainable Land Use. In *International Yearbook of Soil Law and Policy 2018* (eds. H Ginzky, E Dooley, IL Heuser, E Kasimbazi, T Markus & T Qin), pp. 249-270. Cham, Springer International Publishing).
221. Convention on Biological Diversity. 2020 Convention on Biological Diversity: Review of the International Initiative for the Conservation and Sustainable Use of Soil Biodiversity and Updated Plan of Action.
222. FAO, ITPS, GSBI, SCBD & EC. 2020 State of knowledge of soil biodiversity - Status, challenges and potentialities: Report 2020. FAO.
223. Rojas, RV & Caon, L. 2016 The international year of soils revisited: promoting sustainable soil management beyond 2015. *Environmental Earth Sciences* **75**. <https://doi.org/10.1007/s12665-016-5891-z>
224. FAO. 2015 Revised World Soil Charter. Rome, Italy.
225. FAO. 2017 Voluntary Guidelines for Sustainable Soil Management.
226. Orr, B, Cowie, A, Castillo Sanchez, V, Chasek, P, Crossman, N, Erlewein, A, Louwagie, G, Maron, M, Metternicht, G & Minelli, S. 2017 Scientific conceptual framework for land degradation neutrality. In A

report of the science-policy interface. *United Nations Convention to Combat Desertification (UNCCD)*, Bonn, Germany, pp. 1-98.

227. International Law Association. 2020 ILA Guidelines on the Role of International Law in Sustainable Natural Resources Management for Development, Resolution 4/2020, 2020 Report of the Seventy Ninth Conference, Kyoto.
228. Minasny, B, Malone, BP, McBratney, AB, Angers, DA, Arrouays, D, Chambers, A, Chaplot, V, Chen, Z-S, Cheng, K, Das, BS, et al. 2017 Soil carbon 4 per mille. *Geoderma* **292**, 59-86. <https://doi.org/10.1016/j.geoderma.2017.01.002>
229. Chabbi, A, Lehmann, J, Ciais, P, Loescher, HW, Cotrufo, MF, Don, A, SanClements, M, Schipper, L, Six, J, Smith, P, et al. 2017 Aligning agriculture and climate policy. *Nature Climate Change* **7**, 307-309. <https://doi.org/10.1038/nclimate3286>
230. Poulton, P, Johnston, J, Macdonald, A, White, R & Powlson, D. 2018 Major limitations to achieving “4 per 1000” increases in soil organic carbon stock in temperate regions: Evidence from long-term experiments at Rothamsted Research, United Kingdom. *Glob. Change Biol.* **24**, 2563-2584. <https://doi.org/10.1111/gcb.14066>
231. Minasny, B, Arrouays, D, McBratney, AB, Angers, DA, Chambers, A, Chaplot, V, Chen, Z-S, Cheng, K, Das, BS, Field, DJ, et al. 2018 Rejoinder to Comments on Minasny et al., 2017 Soil carbon 4 per mille *Geoderma* **292**, 59–86. *Geoderma* **309**, 124-129. <https://doi.org/10.1016/j.geoderma.2017.05.026>
232. Soussana, J-F, Lutfalla, S, Ehrhardt, F, Rosenstock, T, Lamanna, C, Havlík, P, Richards, M, Lini, E, Wollenberg, E, Chotte, J-L, et al. 2019 Matching policy and science: Rationale for the '4 per 1000-soils for food security and climate' initiative. <https://doi.org/10.1016/j.still.2017.12.002>
233. Kibugi, R. 2018 Soil Health, Sustainable Land Management and Land Degradation in Africa: Legal Options on the Need for a Specific African Soil Convention or Protocol. In *International Yearbook of Soil Law and Policy 2017* (eds. H Ginzky, E Dooley, IL Heuser, E Kasimbazi, T Markus & T Qin), pp. 387-411. Cham, Springer International Publishing).
234. Markus, T. 2017 The Alpine Convention’s Soil Conservation Protocol: A Model Regime? In *International Yearbook of Soil Law and Policy 2016* (eds. H Ginzky, IL Heuser, T Qin, OC Ruppel & P Wegerdt), pp. 149-164. Cham, Springer International Publishing).
235. 2003 *Revised European Charter for the Protection and Sustainable Management of Soil*, Strasbourg, 17 July 2003 CO-DBP/documents/codbp2003/10e.
236. European Commission. 2006 Soil protection - The story behind the Strategy.
237. Stankovics, P, Tóth, G & Tóth, Z. 2018 Identifying Gaps between the Legislative Tools of Soil Protection in the EU Member States for a Common European Soil Protection Legislation. *Sustainability* **10**, 2886. <https://doi.org/10.3390/su10082886>
238. Montanarella, L & Panagos, P. 2021 The relevance of sustainable soil management within the European Green Deal. *Land Use Pol.* **100**, 104950. <https://doi.org/10.1016/j.landusepol.2020.104950>
239. 2021 European Parliament resolution of 28 April 2021 on soil protection.

240. Keesstra, SD, Bouma, J, Wallinga, J, Tiftonell, P, Smith, P, Cerdà, A, Montanarella, L, Quinton, JN, Pachepsky, Y, van der Putten, WH, et al. 2016 The significance of soils and soil science towards realization of the United Nations Sustainable Development Goals. *SOIL* **2**, 111-128. <https://doi.org/10.5194/soil-2-111-2016>
241. Tóth, G, Hermann, T, da Silva, MR & Montanarella, L. 2018 Monitoring soil for sustainable development and land degradation neutrality. *Environmental Monitoring and Assessment* **190**, 57. <https://doi.org/10.1007/s10661-017-6415-3>
242. Shen, X, Wang, X, Zhang, Z, Lu, Z & Lv, T. 2019 Evaluating the effectiveness of land use plans in containing urban expansion: An integrated view. *Land Use Pol.* **80**, 205-213. <https://doi.org/10.1016/j.landusepol.2018.10.001>
243. Roose, A, Kull, A, Gauk, M & Tali, T. 2013 Land use policy shocks in the post-communist urban fringe: A case study of Estonia. *Land Use Pol.* **30**, 76-83. <https://doi.org/10.1016/j.landusepol.2012.02.008>
244. Lu, Y, Song, S, Wang, R, Liu, Z, Meng, J, Sweetman, AJ, Jenkins, A, Ferrier, RC, Li, H, Luo, W, et al. 2015 Impacts of soil and water pollution on food safety and health risks in China. *Environment International* **77**, 5-15. <https://doi.org/10.1016/j.envint.2014.12.010>
245. Zhao, F-J, Ma, Y, Zhu, Y-G, Tang, Z & McGrath, SP. 2015 Soil Contamination in China: Current Status and Mitigation Strategies. *Environmental Science & Technology* **49**, 750-759. <https://doi.org/10.1021/es5047099>
246. Chen, R, de Sherbinin, A, Ye, C & Shi, G. 2014 China's Soil Pollution: Farms on the Frontline. *Science* **344**, 691-691. <https://doi.org/10.1126/science.344.6185.691-a>
247. Vogel, H-J, Eberhardt, E, Franko, U, Lang, B, Ließ, M, Weller, U, Wiesmeier, M & Wollschläger, U. 2019 Quantitative Evaluation of Soil Functions: Potential and State. *Frontiers in Environmental Science* **7**. <https://doi.org/10.3389/fenvs.2019.00164>
248. Vargas, L, Willemen, L & Hein, L. 2019 Assessing the Capacity of Ecosystems to Supply Ecosystem Services Using Remote Sensing and An Ecosystem Accounting Approach. *Environmental Management* **63**, 1-15. <https://doi.org/10.1007/s00267-018-1110-x>
249. Kibblewhite, MG, Ritz, K & Swift, MJ. 2008 Soil health in agricultural systems. *Philos Trans R Soc Lond B Biol Sci* **363**, 685-701. <https://doi.org/10.1098/rstb.2007.2178>
250. Powlson, DS, Gregory, PJ, Whalley, WR, Quinton, JN, Hopkins, DW, Whitmore, AP, Hirsch, PR & Goulding, KWT. 2011 Soil management in relation to sustainable agriculture and ecosystem services. *Food Policy* **36**, Supplement 1, S72-S87. <https://doi.org/10.1016/j.foodpol.2010.11.025>
251. Paustian, K, Lehmann, J, Ogle, S, Reay, D, Robertson, GP & Smith, P. 2016 Climate-smart soils. *Nature* **532**, 49. <https://doi.org/10.1038/nature17174>
252. LaCanne, CE & Lundgren, JG. 2018 Regenerative agriculture: merging farming and natural resource conservation profitably. *PeerJ* **6**, e4428-e4428. <https://doi.org/10.7717/peerj.4428>
253. Lal, R & Kosaki, T. 2018 *Soils and Sustainable Development Goals. GeoEcology Essay*, Schweizerbart Science Publishers.

254. Amundson, R. 2020 The policy challenges to managing global soil resources. *Geoderma* **379**, 114639. <https://doi.org/10.1016/j.geoderma.2020.114639>
255. Frelif-Larsen, A, Bowyer, C, Albrecht, S, Keenleyside, C, Kemper, M, Nanni, S, Naumann, S, Mottershead, RD, Langrebe, R, Andersen, E, et al. 2017 *Updated Inventory and Assessment of Soil Protection Policy Instruments in EU Member States*.
256. Ronchi, S, Salata, S, Arcidiacono, A, Piroli, E & Montanarella, L. 2019 Policy instruments for soil protection among the EU member states: A comparative analysis. *Land Use Pol.* **82**, 763-780. <https://doi.org/10.1016/j.landusepol.2019.01.017>
257. Wingeyer, AB, Amado, TJC, Pérez-Bidegain, M, Studdert, GA, Varela, CHP, Garcia, FO & Karlen, DL. 2015 Soil Quality Impacts of Current South American Agricultural Practices. *Sustainability* **7**, 2213-2242. <https://doi.org/10.3390/su7022213>
258. Webb, A, Kelly, G & Dougherty, W. 2015 Soil governance in the agricultural landscapes of New South Wales, Australia. *International Journal of Rural Law and Policy Special Edition* **1**, 1-16. <https://doi.org/10.5130/ijrlp.i1.2015.4169>
259. Chukov, SN & Yakovlev, AS. 2019 Soil and Land Categories in the Modern Legislation of Russia. *Eurasian Soil Science* **52**, 865-870. <https://doi.org/10.1134/S1064229319070020>
260. Fromherz, NA. 2012 The Case for a Global Treaty on Soil Conservation, Sustainable Farming, and the Preservation of Agrarian Culture. **39**. <https://doi.org/10.15779/Z38BC49>
261. IUCN. 2019 World Commission on Environmental Law, Soil Desertification & Sustainable Agriculture Specialist Group, 2019 Mid-Year Report. Online.
262. Hazelton, PA, Frossard, E, Blum, WEH & Warkentin, BP. 2006 Australian examples of the role of soils in environmental problems. In *Function of Soils for Human Societies and the Environment* (p. 0, Geological Society of London).
263. Sophia Antipolis. 2003 Threats to Soils in Mediterranean Countries: Document Review. In *Plan Bleu Papers*. Valbonne, France, Plan Bleu.
264. Gonzalez Lago, M, Plant, R & Jacobs, B. 2019 Re-politicising soils: What is the role of soil framings in setting the agenda? *Geoderma* **349**, 97-106. <https://doi.org/10.1016/j.geoderma.2019.04.021>
265. Heuser, IL. 2018 Development of Soil Awareness in Europe and Other Regions: Historical and Ethical Reflections About European (and International) Soil Protection Law. In *International Yearbook of Soil Law and Policy 2017* (eds. H Ginzky, E Dooley, IL Heuser, E Kasimbazi, T Markus & T Qin), pp. 451-474. Cham, Springer International Publishing).
266. Moallemi, EA, Haan, F, Hadjidakou, M, Khatami, S, Malekpour, S, Smajgl, A, Stafford Smith, M, Voinov, A, Bandari, R, Lamichhane, P, et al. 2021 Evaluating Participatory Modeling Methods for Co - creating Pathways to Sustainability. *Earth's Future* **9**. <https://doi.org/10.1029/2020EF001843>
267. Koch, A, McBratney, A, Adams, M, Field, D, Hill, R, Crawford, J, Minasny, B, Lal, R, Abbott, L, O'Donnell, A, et al. 2013 Soil Security: Solving the Global Soil Crisis. *Global Policy* **4**, 434-441. <https://doi.org/10.1111/1758-5899.12096>

268. McBratney, A, Field, DJ & Koch, A. 2014 The dimensions of soil security. *Geoderma* **213**, 203-213. <https://doi.org/10.1016/j.geoderma.2013.08.013>
269. Bouma, J. 2020 Soil security as a roadmap focusing soil contributions on sustainable development agendas. *Soil Security* **1**, 100001. <https://doi.org/10.1016/j.soisec.2020.100001>
270. Hill, R. 2017 The Place of Soil in International Government Policy. In *Global Soil Security* (eds. DJ Field, CLS Morgan & AB McBratney), pp. 443-449. Cham, Springer International Publishing).
271. Field, DJ, Morgan, CLS & McBratney, AB. 2017 *Global Soil Security*, Springer International Publishing, A. G.
272. Lilburne, L, Eger, A, Mudge, P, Ausseil, A-G, Stevenson, B, Herzig, A & Beare, M. 2020 The Land Resource Circle: Supporting land-use decision making with an ecosystem-service-based framework of soil functions. *Geoderma* **363**, 114134. <https://doi.org/10.1016/j.geoderma.2019.114134>
273. Juerges, N, Hagemann, N & Bartke, S. 2018 A tool to analyse instruments for soil governance: the REEL-framework. *Journal of Environmental Policy & Planning* **20**, 617-631. <https://doi.org/10.1080/1523908X.2018.1474731>
274. Ginzky, H. 2020 Good Governance for “Sustainable Management of Soil” on National and International Level: How to Do It? In *Legal Instruments for Sustainable Soil Management in Africa* (eds. H Yahyah, H Ginzky, E Kasimbazi, R Kibugi & OC Ruppel), pp. 35-54. Cham, Springer International Publishing).

Saving the ground beneath our feet

Establishing priorities and criteria for governing soil use and protection

Lewis Peake^{1,2} and Cairo Robb³

¹ School of Environmental Science, University of East Anglia, Norwich, UK;

² Tyndall Centre for Climate Change Research, University of East Anglia, Norwich, UK;

³ Legal Research Fellow, Centre for International Sustainable Development Law, <https://www.cisd.org>;

corresponding author: l.peake@uea.ac.uk

Abstract

The continual loss and impairment of soil ecosystem services across the globe calls for a fundamental reconsideration of soil governance mechanisms. This critical synthesis charts the history and evolution of national and international soil law and seeks to unravel certain challenges that have contributed to this failure in governance. It describes and categorises law and policy responses to different soil threats, and identifies a worrying widespread absence of legislation for oversight and protection of agricultural soils from urbanisation, as well as a lack of clear legal mechanisms to determine national priorities for soil protection. A reduction in the world's prime farmland threatens soil ecosystem services, including food security, carbon storage, and biodiversity. Falling between the stalls of agricultural and environmental law, the fate of farmland is often left to planners who do not see themselves as responsible for soils. Consequently, legal instruments with the greatest power to affect soil, sometimes irreversibly, are often framed and worded with little or no reference to soil. Nevertheless, emerging conceptual frameworks might offer positive outcomes. The authors advocate robust holistic policies of soil governance and land use planning that place soil ecosystem services and natural capital at the heart of decision making.

Keywords

agricultural land conversion, farmland preservation, land take, soil ecosystem services, soil governance, soil security

Introduction

Soil is a vital multifunctional resource that could be regarded as the metaphorical as well as the literal foundation of human civilization. Soil ecosystem services (SES) [1], and the broader concept of soil-derived Nature's Contributions to People (NCP) [2], is a relatively new academic subdiscipline, but human awareness of the life-supporting role of soil is as old as civilization, if not older. Clearly, the ground beneath our feet provides so much more than the ground beneath our feet, and transcends title deeds, boundary fences and even national borders. Arguably, we are more dependent on soil than ever, as our dominance as a species has resulted in greater demands and stressors on our environment. As the world's population has soared and almost every parcel of its terrestrial surface has been assigned to the various national states, and then further subdivided and transformed by governments, private corporations or individuals, the need to safeguard this increasingly degraded resource is more urgent than ever.

In addition to the familiar and very direct role that soil has in providing our food, fibre and timber, we often forget its many other interconnected functions, for example:

- Physical: providing building or landscaping material, and a stable substrate for infrastructure;
- Chemical: regulating the supply of plant nutrients;
- Biological: constituting biodiverse ecosystems and communities of useful organisms, converting toxic sewage and decomposing matter into plant nutrients, and providing a genetic reservoir;
- Hydrological: facilitating water absorption, storage and filtration, and attenuating flooding;
- Cultural: preserving cultural heritage and archaeological information, and providing recreation;
- Climatic: storing carbon (C), regulating greenhouse gas (GHG) fluxes and heat buffering.

The last of these would have been almost inconceivable to our forebears, only assuming critical importance approximately since the beginning of the new millennium.

However, in certain circumstances soil, like any other natural resource, can deliver negative impacts, such as emitting GHGs, harbouring pests and diseases, or contributing to the accumulation of sediment, dust storms and landslides. These events are reminders that good governance of soil is also about minimising ecosystem "disservices" as well as safeguarding essential ecosystem services. We live in a time of anthropogenic environmental crisis, unprecedented climate change and species extinction. A report recently released by WWF states that: "Since 1970, our Ecological Footprint has exceeded the Earth's rate of regeneration," and currently exceeds the world's capacity to support humanity by 56% [3]. Soil is an integral component within this system, both as an enabler and a victim of exponential human expansion.

This interdisciplinary critical synthesis reviews the role of soil governance in helping to achieve the objectives of the UN Sustainable Development Goals (SDGs), to mitigate and adapt to climate change and

contribute to a more secure future for humanity. We trace the historical path of the status and governance of soil; categorise anthropogenic threats to soil, and legal mechanisms that can be applied to address them; and discuss the semantics surrounding legal aspects of soil and land, aiming to disambiguate and illuminate the terminology. The current state of art of soil governance is explored in depth and a key finding is a failure to prevent or barely acknowledge the worldwide loss of prime farmland and the implications of this for society. The review concludes by looking to emerging approaches such as the soil security conceptual framework, proposed as a way of translating critical soil knowledge into sustainable development policy, and to holistic land evaluation methodologies that incorporate SES into land use planning.

Methods

An open-ended literature review was conducted, using combinations of the following search terms: “agriculture”, “biodiversity”, “conservation”, “contamination”, “conversion”, “degradation”, “ecosystem services”, “erosion”, “farmland”, “governance”, “land”, “land take”, “land use planning”, “law”, “legislation”, “non-agricultural”, “policy”, “preservation”, “prime”, “protection”, “SDG”, urbanis(z)ation”, “urban sprawl”, “soil”, “soil sealing”, etc.. By selective analysis we synthesised key points, such as our own categorisations, thus adding new content. By presenting in this way, the intention was twofold: to provide the reader with some clear signposts within a large and complex domain; and to provide ourselves with a framework within which the context of any given issue could be better understood and deconstructed.

To complement data and cases studies from the literature, a comprehensive search was conducted to abstract data from FAOLEX, the UN Food and Agriculture Organization (FAO) online database of policies and laws [4]. FAOLEX documents legal instruments in force in nearly 200 countries. This database is organised by country, into the following groups, of which those marked with an asterisk were reviewed by the lead author (where they existed for the country in question): **Policies***; **Legislation: Agricultural and rural development***, *Climate change*, *Cultivated plants*, *Disaster risk management*, *Environment**, *Fisheries*, *Food and nutrition*, *Forestry**, *Land and soil**, *Livestock*, *Sea*, *Water*, *Wild species and ecosystems**; **International Agreements***. *Land and soil* was the most relevant, but it was necessary to search further, because in many cases this group is primarily devoted to the administration of land ownership and land reform. The groups are not mutually exclusive, so some legislation appears more than once.

The name of the FAOLEX entry alone was sometimes broadly sufficient to explain its meaning, e.g. “Soil conservation law”, but in many cases it was necessary to select the embedded hyperlink to access its summary (usually in English). Ambiguous or simplified wording often made it necessary to select a further link to the full text, either within FAOLEX or on the websites of national governments. These documents were often in the local language, sometimes in local script, and where necessary Google Translate was used. In some cases, individuals professionally familiar with the legislation were contacted for clarification and

asked whether the laws in question were implemented and effectively governed. Approximately 1% of countries had either created no such instruments or lacked any accessible data in FAOLEX. This information was compiled from the summer of 2020 until the summer of 2021. A spreadsheet was created with one row per country and columns indicating categories of legal instrument, and from this it was possible to calculate percentages of countries with such measures in force (Table 3). The FAOLEX groups are powerful retrieval mechanisms, but unrelated to the categories we derived. While our research was being conducted FAO was in the process of creating and populating a similar but soil-specific repository, SoiLEX, including data from FAOLEX¹. In terms of structural organisation and ease of use, SoiLEX is superior to FAOLEX but limited to highly soil-specific categories of legislation, whereas FAOLEX is wider in scope and, crucially for our purposes, contains laws relating to the protection or preservation of categories of land such as prime farmland and natural ecosystems.

Some caveats are necessary. Such a synthesis can only ever be approximate and is undoubtedly incomplete. Several polities listed in FAOLEX as countries actually comprise a number of countries (e.g. the UK), or separate provinces or states (e.g. the US), with differing laws. Many instruments, especially policies, are generalised, e.g.: “The land is the main national wealth, which is under special protection of the state...ensuring rational use and protection of land,” [5] iterated many times, but with little or no further qualification. A law stating that land will be classified as urban or rural could imply protected status or simply a degree of spatial planning; the latter was assumed unless explicitly otherwise. Furthermore, this is primarily a review of officially documented policies and laws, not a comprehensive assessment of enforcement or effective governance. While an overview of international soil law and policy context is provided, the synthesis does not deal in detail with states’ external relations or human and peoples’ rights relating to land and soil, nor does it address global or national soil information systems or soil education and literacy measures, all of which are also important aspects of comprehensive soil governance.

Soil as a valued resource and legal entity

The evolution of soil law

The ancient Egyptians called their nation state *Kemet* which means black land or dark earth, in contrast to the barren *Deshret*, or red land, stretching out on either side. The black silty clay of the Nile valley was not simply a metaphor for their country; in their minds, it *was* their country. Human dependence on productive

¹ SoiLEX is described as a global database on national legislation on soil protection, conservation and restoration to facilitate access to information on the existing legal instruments in force and bridge the gap between the various soil stakeholders.:

www.fao.org/soils-portal/soilex/en/

soil and the impact of its degradation on civilisation throughout history is well attested [6-10]. Mesopotamia experienced at least two catastrophic regime changes due respectively to soil salinity and soil erosion [11, 12]. The Greek and Roman writers make multiple references both to the benefits of soil quality and the threat of soil degradation, and by the 1500s in Europe soil was regarded as the key factor of an economy [11]. History is full of examples of land-related wars and conquests, often correlated with soil quality [13-15], and repression and killing of those seeking to protect land and soil quality continues today [16, 17].

The earliest known attempts to put an economic value on productive land were Chinese soil classification and maps created 2000 years BCE for taxation purposes [18]. Other ancient peoples also independently derived similar systems, for example in Mesoamerica [19]. However, while there are many historical examples of land ownership rights and transactions [20], it is difficult to pin down when laws or policies were first introduced to protect soil or land as a vulnerable or scarce resource in its own right, that is, part of the natural capital for the common good. The *Statutes on Agriculture*, compiled during the Qin Dynasty of the late 4th century BCE is China's, and possibly the world's, earliest known set of soil-related laws [21]. The *Statutes* were followed a century later, during the Han Dynasty (206 BCE-8 CE), by the *Book of Fan Shengzhi*, China's oldest surviving agronomic treatise which provides extraordinarily detailed instructions on most aspects of soil management and may well have been regarded by farmers as de facto law [22].

Perhaps the oldest surviving documented legislation explicitly addressing soil is that within the *Digesta* of the Eastern Roman emperor Justinian, published in the 6th century, but including laws dating from centuries earlier. While there was no overt implication that soil *per se* was a scarce resource in the Roman Empire, these edicts leave the reader in no doubt that soil was of great importance to everyone involved in its use, from the question of who benefited from its bounty to who was responsible for its maintenance [23].

In the modern era, the first specific instances of soil legislation were usually initiated in response to the threat of severe soil erosion, typically the result of inappropriate land management combined with extreme climatic conditions. The most iconic example is the US Dust Bowl of the 1930s, leading to the first major piece of soil-related legislation in the US, the 1935 Soil Conservation Act, but even before then soil erosion was perceived as an ever-present threat to American farmers and food production. By the 1870s there was widespread awareness of the problem of erosion [24]. The emerging environmental movement in Europe and America also had a growing influence on policy [25, 26], and forest protection laws incorporating the principle of soil conservation appeared, for example in the US from 1873 [26] and New Zealand in 1874 [27]. In 1894 the US banned grazing in federal reserves to prevent land degradation [26] and in 1901 the Indian state of Punjab, under British administration, passed what might be the first soil and water conservation law, the *Punjab Land Preservation Act*, still in force today [28]. In 1907 Iceland passed the first national parliamentary soil conservation act [29].

A significant result of the Dust Bowl and President Roosevelt's generous response to it was the huge contribution made by the United States Department of Agriculture (USDA) and American scientists to the worldwide study and management of soils. Despite some criticism [30-33], the far-reaching impact of this intense period of activity, barely perceived outside the sphere of soil science, is perhaps unparalleled in human history. Building on a legacy of international scholarship, especially in Germany and Russia [34], in the space of a decade at least three trailblazing conceptual tools were created: the Soil Taxonomy, the Land Capability Classification (LCC) methodology and the Universal Soil Loss Equation (USLE). These would radically transform the international science, management and legislation of soils.

From the 1930s other nations rapidly passed their own soil and land legislation. Food security was a major driver, but so was the loss of rural landscapes and wilderness. Other kinds of environmental protection were also implemented throughout the 20th century, such as the creation of green belts around cities, even as early as 1901 [35]. In the 1940s, however, World War II impacted food production and distribution, especially in Europe, fostering more self-reliance. In the postwar world agricultural productivity took on a new significance and the new technique of land evaluation was enthusiastically applied [36, 37].

In the UK, even before the end of the war, the Scott Report on rural land use recommended the protection of: "good agricultural land," but also signalled a new approach to planning that incorporated landscape amenity [38], and has been cited as prescient in its ethos of sustainable development [39]. The report fed into the 1944 White Paper, *The Control of Land Use* [40] which paved the way for one of the earliest and most radical planning policies of its kind anywhere, enshrined in the UK 1947 Town and Country Planning Act which nationalised land use regulations and has been emulated worldwide [41]. While the driving force of the USDA LCC had been farm management and soil conservation, the British system was primarily a tool of the new planning system, being put to work by soil surveyors in Britain not to find new land to cultivate, but to identify and protect existing prime farmland. The latest incarnation of this system in England and Wales is the 1988 Agricultural Land Classification (ALC) system for identifying the "best and most versatile" (BMV) land [42]. In 1984 the USDA Natural Resources Conservation Service (NRCS) developed its own system oriented towards planning legislation, Land Evaluation and Site Assessment (LESA) [43].

The postwar era is notable for the creation of global and regional institutions such as the FAO, the International Union for Conservation of Nature (IUCN), and the European Union (EU), all of which have been instrumental in relation to land and soil governance. Shortly after its creation in 1945, the FAO embarked on the world's first formal international attempt to address soil conservation as a global issue, via a report [44] and a conference in 1948 [45]. Underpinning such initiatives was a growing awareness of the economic impact of soil degradation, to society at large as well as individual land users, with far-sighted hints at the valuation of ecosystem services and implications for government policy [46]. Similar analyses followed in the ensuing decades [47, 48], and soil law continued to evolve, with the US taking the lead [49].

In 1949 the botanist Aubreville coined the term desertification to describe the transition of agricultural land in arid or semiarid areas to an uncultivable state lacking ecological viability due to a combination of climatic and human factors [50]. By 1958 the Chinese government acknowledged the threat of desertification to the wellbeing of nearly 200 million people, and has initiated afforestation programmes since 1978 [51]. Despite controversy over its meaning, the concept of desertification became and continues to be a significant driver of sustainable land management (SLM) initiatives designed to tackle land degradation [52, 53].

The conflicted role of agriculture

In a sign of things to come, the UK postwar agricultural policy came in for subsequent criticism for virtually exempting farmers from planning regulations, the assumption being that the role of farmers in protecting the rural landscape made controls on agricultural land use unnecessary [39, 54]. After WWII, geopolitics stabilised and agricultural productivity soared, due to a range of technological innovations [31], which from the 1960s reached lower income economies as the Green Revolution [55]. This period represents a radical reframing of society's attitudes towards the environment and agriculture [56]. In some parts of the world food security was no longer regarded as a serious threat, whereas agriculture itself was increasingly perceived as the primary threat to the environment, and hence a mixed blessing. Both the expansion and intensification of agriculture led to unprecedented wildlife habitat loss, encompassing deforestation, wetland drainage, conversion of grassland to arable and hedgerow clearance. Added to this were many other environmental impacts associated with industrial agriculture, including pollution, soil degradation and fuel inefficiency [57-59].

The publication of *Silent Spring* by Carson [60] led to a major international turning point in changing attitudes and policies in response to the excessive use of pesticides and herbicides. In Europe in the 1970s and 1980s the Common Agricultural Policy (CAP) came to epitomise the ironic twin dilemma of taxpayers subsidising overproduction at the expense of environmental destruction [40, 54, 56]. With the public emergence of climate change science since the late 1980s [61], and growing evidence of the significant climate-forcing influence of GHG emissions from farming [62], alongside the ongoing loss of biodiversity [63], agriculture finds itself more implicated than ever as a cause of environmental crisis as well as its victim. The negative image of modern agriculture as an industry that degrades soil and threatens ecosystems risks devaluing the public perception of agricultural land, as if the sealing of such land would not result in a loss of environmental benefits. Meanwhile soil science, straddling geology and biology, became virtually a subdiscipline of agriculture at least halfway into the 20th century [11]. Hence soil, by association and also because largely hidden, rarely evokes the kind of intrinsic concern that attaches to wildlife or landscape.

The relationship between soil and land

Soil and land are inextricably linked in the context of governance and, indeed, many languages conflate the two in common parlance. Weigelt *et al.* [64] stress a distinction between soil and land, but also identify a gap in the literature to address the critical importance of governing them together to achieve sustainability. An absence of land rights also often encourages poor soil management and land degradation [65-68]. From a legal perspective the concept of land is usually operative in matters of ownership and boundaries, and also in terms of spatial and territorial planning. Soil tends to enter the legal fray when it is transformed, harmed or threatened by a specific activity, whether by the owner of the land containing the soil, or by another party. This distinction signals two very different aspects of soil law. The former perspective generally encompasses property rights, whereby the injured party is primarily the landowner. The latter perspective offers protection to soil as a legal entity in its own right, even from its “owner”, for the common good.

Where the blurring of land and soil becomes problematic is in instances of proposed changes that are framed only in relation to land despite having significant impact on the soil within, or adjacent to, that land. This opacity can apply to legislation specific to agriculture, the environment or planning, and as a result important soil implications of land use conversion may fall into the gaps between all three. The creators of agricultural laws may feel that their remit regarding soil protection extends only to the direct impact of farming on soil, while the creators of environmental laws may feel that their remit for soil protection and conservation extends only to an unspecified component within a larger ecosystem or protected landscape. This then leaves rural land use conversion in the hands of the creators and implementers of planning laws, who may feel that soil *per se* is outside their jurisdiction, despite the soil-related implications of the land use changes that their legislation may encompass. There is in effect a blind spot whereby the legal instruments with the greatest power to affect soil, sometimes irreversibly, are often framed and worded with little or no reference to soil. Raising this in a meeting with government soil scientists caused heads to nod in agreement and perusing comparative laws reveals that this is a recurring theme throughout the world [4].

Anthropogenic threats to soil and land

Humans have learnt to transform soil to their advantage, and the overall impact of human society on soil, both intended and unintended, has been profound. Short-term gains have often been at the expense of long-term harms. The picture is further complicated by the fact that human impact exacerbates or unbalances natural processes such as soil erosion, salinization or flooding. Soil or land is typically vulnerable to three types of anthropogenic threat, set out in Table 1, and described further below.

Table 1: Types of anthropogenic threat to soil or land

1. **Soil degradation** (i.e. harm to soil, other than soil sealing, which is covered in 2 and 3 below)
2. **Conversion of natural ecosystems** or other semi-natural and uncultivated land to another form of land use, such as agriculture, forestry, urbanisation, or industry, including mining or energy infrastructure (conversion is also known as ‘land take’)
3. **Conversion of farmland** to urban or industrial use² [69] (‘agricultural land take’)

1. *Soil degradation*

In this context, degradation refers to any harm done to soil which is potentially ameliorated by remedial measures and includes acute problems such as contamination, salinization, acidification, sodification and compaction or other forms of structural collapse³. The term soil degradation has also been widely used to describe gradual and chronic loss of productivity, typically the result of over-exploitation and inadequate management, invariably bound up with erosion, organic C depletion, reduction in soil biodiversity and low nutrient status. While acute soil degradation is often the result of breaches of regulations, chronic soil degradation tends to be associated with either intensive agriculture or rural poverty, and generally requires supportive rather than punitive legislation. These problems can be complex and linked to natural causes or intrinsic to certain soil types. In the worst cases large tracts of land have been abandoned or declared unfit for food production with severe economic impacts. Gisladdottir and Stocking articulate the interlinkages of land degradation with other environmental problems such as climate change and biodiversity loss [52].

In the first study of its kind, UNEP commissioned the International Soil Reference and Information Centre (ISRIC) to conduct the Global Assessment of Soil Degradation (GLASOD). GLASOD concluded that from World War II to 1990 15% of all land worldwide (almost 2 billion hectares) was degraded, equating to 23% of inhabited land (14% seriously). Soil erosion was by far the most widespread form of soil degradation, affecting 83% of degraded land. Approximately 20 years later ISRIC scientists reviewed GLASOD and used satellite imagery to measure an overlapping and subsequently ongoing period (1981-2003 [70, 71]). The authors detected a further 24% of the total land area that was degrading *mostly in addition to* GLASOD’s 15% already degraded, i.e. a cumulative process, implying that at least a third of the world’s land appeared

² Converting farmland to tree cover is not usually regarded as a “threat”, though there are scenarios where this may be the case, for example, in relation to inappropriate afforestation of peatlands, or replacement of biodiverse systems with plantation monoculture.

³ The term “soil degradation” can also, more generally, be understood to encompass the other two threats listed here, including soil sealing.

to be degraded to some extent by 2003⁴. Additional key findings were that land degradation was not primarily a problem of drylands, as had long been proposed, but was a worldwide phenomenon.

In a 2015 follow-up study the global estimate of degrading land was revised down slightly to 22%, with 14% of land showing some improvement [72]. Also in 2015 FAO/ITPS⁵ claimed that 33% of land was degraded, and could reach 90% by 2050 [73]. A 2016 study produced a global figure of 30% degraded land [74]. A UNCCD analysis in 2017 revised the total to 23% [75], but states that: "...over the last two decades, approximately 20 per cent of the Earth's vegetated surface shows persistent declining trends in productivity." [76]. In 2018 IPBES⁶ concluded that 75% of the Earth's land surface is either transformed or degraded to some extent by human activity, impairing 3.2 billion livelihoods, and costing 10% of the annual global gross product, whereas the benefits of restoration are typically 10 times higher than the costs [77].

Most countries now have some form of regulation addressing soil degradation, typically embedded within their agricultural legislation, and in a number of these countries the law is strictly enforced. In many cases, soil policies go beyond prohibitive and punitive laws dealing with acute soil degradation, and incorporate support and incentives to foster improved long-term soil health and protection [78]. Compared to land use conversion, soil degradation, even though it may be the result of deliberate law-breaking, is generally recognised as unintended and unwelcome, so there are fewer obvious social pressures to oppose such laws. The global trend has been greater governance, but the cost of remediation can be prohibitive, whether for those held responsible or those who bear the cost, e.g. taxpayers, so circumvention is a constant risk [79]. China stands out as the country probably most afflicted by soil degradation, of every kind, but by contamination in particular, on a scale that appears irreparable and unaffordable, requiring more than the world's entire wealth to remediate, according to *The Economist* [80]. Nevertheless, no country seems more acutely aware of this or determined to address it than China itself [81, 82].

2. *Conversion of natural ecosystems*

Natural ecosystems or semi-natural landscapes, that is land that is uncultivated and largely unmanaged [83], are ubiquitously valued for a variety of reasons: their importance for Indigenous peoples and other local communities; their aesthetic characteristics and recreational potential; their educational and scientific worth;

⁴ The authors made no attempt to merge the data because of the contrasting methods of the separate studies and the fact that the GLASOD data was unverifiable, but one of the authors opines that: "It is not unreasonable to judge that all land now under anything less than natural climax vegetation is degraded in terms of biodiversity and stored carbon and nitrogen, perhaps 66% of the world's land surface and rising." (David Dent, personal communication.)

⁵ The Intergovernmental Technical Panel on Soils.

⁶ The Intergovernmental Science-Policy Platform on Biodiversity and Ecosystem Services.

and, with increasing urgency, their vital ecosystem service benefits. Aside from largely uninhabitable areas, this category of land includes forest, wetland, native grassland (e.g. prairie or steppe) and heath-/moor-/peatland. Globally the primary threat to such land is from conversion to agriculture, responsible for 80% of deforestation according to WWF [3], but the full picture is confused by claims of both overestimation and underestimation [84].

The dynamics of forest change are also complicated by temporary deforestation and forest gain, e.g. via afforestation, reforestation or regrowth on abandoned farmland. A recent study [85] using satellite imagery estimated that between 2001 and 2015, only 27% of global tree cover loss was permanent land use change for commodity production, i.e. large-scale agriculture, mining and energy infrastructure. Urbanisation accounted for less than 1%. Impermanent changes included forestry (26%), shifting agriculture (24%), and wildfire (23%). Regional contrasts were stark with permanent deforestation accounting for most of the tree loss that occurred in Latin America and Southeast Asia, mainly driven respectively by ranching and oil palm plantations [85]. Indonesia and Malaysia stand out as areas of increased deforestation. The rate of loss had declined markedly in Brazil, but since the change of regime in Brazil in 2018 forest destruction is reported to have increased again dramatically [86]. Other regions have relatively low rates of permanent deforestation, with temporary disturbance associated with managed forestry and, in North America, Russia and Australia, wildfire. A few countries have achieved a net forest gain.

Natural ecosystems tend to be better protected in the higher income countries, but historically this was largely because they lack the natural resources that facilitate intensive land use and often present severe obstacles to development. Very little "productive" wilderness remains, and some parts of the tropics have reached or are close to this point [87]. In sub-Saharan Africa, even where natural forest exists near settlements, it is difficult and expensive for farmers to encroach on such land, which also often has poor soil [88]. The alarm has also been raised concerning the impact of rising and uncontrolled urbanisation on biodiversity in Africa [89]. In 2009 Rockström *et al.* [90] suggested we are approaching the limits of planetary land use conversion. This makes the preservation of natural ecosystems worldwide all the more urgent, especially given their far-reaching environmental benefits, such as C storage, biodiversity and soil and water conservation. A combination of agricultural support and incentives, and rational territorial planning can raise farm incomes, improve food security and reduce loss and expansion of farmland [91-95]. Nevertheless, conversion of natural lands to agriculture is rarely as drastic or permanent as urbanisation, and many countries are increasingly rewarding landowners, land managers or farmers, for example via payments for ecosystem services (PES), to implement more environmentally benign land use and practices [96]. Putting aside the question of how effectively each government responds to the loss of natural ecosystems, the issue is relatively unambiguous and virtually every country on Earth, theoretically at least, protects such land with environmental laws and policies. Furthermore, soil protection *per se* is rarely the driving force

behind such legislation, which tends to be framed in terms of ecology, biodiversity or watershed protection, or occasionally preservation of landscape or culture.

3. Conversion of farmland to urban or industrial use

The vast majority of land consumed by urban and industrial expansion throughout the world is agricultural land and this type of threat, which encompasses much soil sealing, is arguably the least reversible and the most profound of these three types of anthropogenic threats. Furthermore, for sound socioeconomic reasons, urban settlement has historically developed near, and often surrounded by, prime farmland. Hence it is frequently the best quality land that is most vulnerable to conversion [97-100]. Peri-urban farmland is particularly attractive to developers because unless it is subject to strict green belt or zoning laws, it lacks the environmental protections of natural ecosystems, is usually cheaper to develop than brownfield sites [101] and tends to be conveniently situated in terms of existing facilities and infrastructure. In the UK there has been a gradual weakening of the protection afforded to BMV land, despite an official policy to the contrary [102, 103]. Furthermore, a study of 25 EU countries found that the land most at risk was that slightly further out from city boundaries, which typically has more fertile soil than that closer to the city [104], while, interestingly, the influence of CAP subsidies reduced the rate of land take in general.

Although agricultural land take is rarely covered by either environmental or agricultural legislation, an exception to this is where farmland is protected for its above-ground biodiversity value, e.g. high natural value (HNV) [104], but this is not the same thing as protecting land for its intrinsic SES value. The last resort for farmland preservation, if it occurs at all, is usually some form of spatial or territorial planning, but the primary motive for such planning regulation is typically urban expansion driven by population and commercial pressure, rather than rural protection [105, 106]. There is a distinct lack of robust legal instruments to prohibit or restrict agricultural land take and the approach taken varies considerably between and within countries, producing policies that are frequently complex and sometimes ambiguous or conflicting [40, 106-113]. Added to this is the prevalence of “informal” governance of planning regulations in many low- and middle-income countries [111, 114, 115].

The mere potential for urban development, even without planning permission, typically increases the value of farmland by, for example, fivefold in South Korea [116], sixfold in Morocco [117], nearly tenfold in New Zealand [107], and even by orders of magnitude more than this: 100-fold in Ghana over ten years [118] and similar multipliers in Britain and Japan [119]. This creates enormous commercial pressure for landowners to sell, especially in lower income countries where agricultural livelihoods may be precarious and lack government support [120], but do not necessarily leave poorer farmers with any long-term benefits. With the exception of a few who may be able to exploit new urban markets, most farmers will use this temporary windfall to pay off debts and become landless farmers or seek other occupations [121]. The prospects of re-

investing in cheaper land to cultivate are greatly diminished by the fragmentation of holdings and the creeping outward shift of the agricultural zone onto lower quality land [121].

It is worth addressing arguments advocating, or not opposed to, farmland conversion. Visser takes this logic to a purely economic extreme of treating natural resources as tradeable commodities or assets and even suggests that land markets would benefit from greater scarcity of farmland [122]. Satterthwaite *et al.* [123] argue that the world has abundant arable land and a global economy from which urban populations can import food without depending on their own agricultural hinterland. This suggestion presupposes reliable food supplies and reliable trading partners, yet geopolitical instability and climate change increasingly threaten global food security [124], not to mention the additional risks associated with a global pandemic.

In defence of conversion one might argue that much farmland, often government-owned, has been abandoned or under-utilised for various reasons, such as de-population of rural areas or, as in the case of the former Soviet Union, incomplete land reform [125], and that this land could be recultivated. There may be limited capacity for rehabilitated farmland to compensate for conversion elsewhere [126], but this raises other issues. One reason for abandoning farmland is the difficulty in extracting an adequate livelihood because the land is degraded [127] or marginal [125]. Such land will be intrinsically less productive so, even if farmers have the means and incentive to restore it, a greater area might be needed to achieve requisite returns. Where productive land has been abandoned for socioeconomic reasons, such as a lack of infrastructure in remote areas which could be remedied, recultivation is feasible [128, 129]. However, abandoned farmland presents another global narrative, for it is swathes of such land, in Russia, US, China, Australia, Latin America and elsewhere, that have inadvertently facilitated a global pattern of reforestation and rewilding [126] that goes some way towards buffering the effects of deforestation elsewhere [126, 130].

Some advocates of farmland conversion in high-income countries like the UK employ primarily socioeconomic arguments, for example with respect to a scarcity of land for housing in the London Metropolitan Green Belt (MGB), arguing that the MGB policy is rigid and anachronistic in protecting all farmland regardless of its environmental value, while constraining suburban gardens which can harbour more biodiversity [54, 131-133]. However, residential land undergoes much sealing and topsoil removal, while the biodiversity of its gardens, though potentially of great value, is entirely arbitrary. Farmland, on the other hand, remains largely unsealed and, given appropriate policy incentives, such as PES, can become more biodiverse and provide greater ecosystem services, as has been a gradual trend in the UK since the late 1980s [102], and looks to continue with the proposed new Environmental Land Management (ELM) schemes [134]. Where such authors have a much stronger case is in criticising the binary distinction between green belt land, where all farmland is protected regardless of its agricultural quality, and non-green belt rural land where prime farmland should be protected but is often developed [132].

The MGB, one of the oldest green belt zones in the world which constrains one of Europe's largest cities, is continually under strain from pressure for housing and transit development. In spite of strict regulations, planning permission is devolved and inconsistent, and much development leapfrogs onto agricultural land beyond. Similar but younger zones have yet to experience such pressures, but in the case of the Ontario Greenbelt, for example, this is partly due to an approach which is arguably more integrated and enlightened, incorporating green infrastructure, principles of environmental, economic and social sustainability, greater public participation and underpinned by a legal framework, centrally owned and controlled by the provincial government, yet both robust and flexible [35, 131, 135]. Farmland preservation in Ontario is not simply a matter of locally *ad hoc* prohibition in the teeth of fierce opposition from planners and developers, but is integrated into a holistic policy that includes agricultural support, local food marketing, employment opportunities, recreation, tourism and ecosystem services in a regional context [131, 135-137].

The view that urbanisation represents economic progress [123] has led many administrators and politicians, especially in lower income countries, to embrace urban expansion policies with enthusiasm [138]. However, at the local level there are serious concerns, even within government [108], that the wider implications of farmland loss are increasing food prices and imports, escalating rural poverty, land degradation and conflict [108, 120, 121, 139-141], as well as the less appreciated impact of soil sealing [100, 108, 142], including poor sanitation, flooding and pollution, which disproportionately affect rural communities [143, 144]. Much of the land consumed by urban sprawl is common land on which many rural communities depend [108]. Developers and officials or agents may benefit from these transactions [145], but the cumulative negative impacts are felt by the whole community. Those highlighting these issues include local academics [120, 141, 146], journalists and blighted farmers taking their grievances to the courts, if they can [145, 147]. Social inequalities drive many to facilitate the very land use changes from which they gain the least, sometimes converting natural ecosystems to replace what they have lost. There is evidence that appropriate government support reduces farmland sale for non-agricultural development, and enhances food security and biodiversity [104, 105, 148] so any attempt to prohibit or restrict agricultural land conversion, must be accompanied by economic support and viable alternatives. No amount of legislation or governance will succeed without this. While all three types of anthropogenic threats to land and soil listed in Table 1 above endanger ecosystem services, from a law and policy point of view, compared to soil degradation and natural ecosystem protection, agricultural land take is often "out of sight and out of mind".

4. *Cross-cutting issues*

A few academics argue that land conversion or degradation may be justified in some cases, on the basis that the economic benefits may outweigh the environmental costs [149], but despite the best efforts of environmental economists, such costs remain intangible [150] and often profound. Furthermore, such

benefits are not necessarily sustainable and are ultimately dependent on ecosystem services that economists have traditionally treated as “free” and, more to the point, inexhaustible [90]. This market-driven worldview has been strongly criticised for at least 50 years [151, 152] and arguably much longer [153], underscoring what is becoming known as the Anthropocene crisis [154]. Economic benefits also accrue unevenly and not necessarily in the national interest. The escalating profits from development can foster an “unholy alliance” between the public and private sector that rewards elaborate circumvention of the law [105, 155] or outright infringements [66, 75-77, 112, 146]. However, it is also important to appreciate that what may appear to outside observers to be infringements may, in some cases, simply be the result of longstanding traditions of informal governance or customary tenure [145, 156], or simply a lack of institutional capacity [123].

Mining, energy infrastructure and other forms of industrial land use represent a very small proportion of land loss or degradation overall [157], although this is increasing [76] and cannot go unmentioned because of their extreme impact in some locations. In western Ghana, for example, gold and diamond mining has had devastating effects on rural communities and natural ecosystems [158, 159]. This activity includes both corporate mechanised extraction and illegal and artisanal hand-digging, a form of low-input mining which also provides construction material. These can involve all three types of anthropogenic threats to soil listed above, going far beyond the loss of land, and including local water contamination and depletion, illegal logging, extensive soil and subsoil removal, and sometimes violent land disputes. The highly lucrative and largely unregulated context in which these enterprises operate tolerates and even encourages infringements of the law, from which corporations operating legal concessions are not exempt [159].

The jurisprudence⁷ of soil and land

The implementation of soil legislation

There are essentially two components to legally protecting or preserving soil resources: (i) applying a method of evaluating and prioritising areas or bodies of soil and (ii) implementing the requisite (and effective) legal instruments and governance structures. Both of these components exist throughout the world, but only within a fragmented and, in some cases, theoretical patchwork of initiatives. Before dealing effectively with the issues of which forms of governance structures or legal mechanisms might be most applicable to soils, one must consider what criteria to apply. This is essential whether focusing primarily on specified spatial areas of land or on the functional aspects of soils in relation to their current status or use. However, preceding even all of this is a minefield of ambiguous terminology to navigate.

⁷ Jurisprudence here refers to both legal theory and legal systems.

Semantics and legal terminology

When legal documents and policies refer to the way in which soil is affected by those managing the land it occupies, and hence the possible harms it may encounter, the word “protection” is the term most widely used. Many if not most countries have some form of explicit or implicit soil protection policy in place, but this is more often than not subsumed within other legislative and policy domains, for example the environment, agriculture or spatial planning. Soil legislation is commonly sub-divided further into categories of protection, reflecting the severity of particular problems within a given country or the gravity attached to the problem by its authorities. For whatever reason, certain soil laws are almost always framed in negative harmful terms and others in positive or remedial terms. Hence there are soil pollution or contamination laws, but hardly ever soil erosion laws. Instead, there are soil conservation laws or occasionally, in a similar vein, soil improvement laws.

Harder to pin down is the terminology applied to the conversion (effectively the loss) of agricultural or other undeveloped rural land to non-agricultural use which typically incorporates substantial additions of infrastructure (e.g. housing). In this domain several terms are used, some in the negative or destructive sense, and others in the positive or protective sense. It can be argued that the term land conversion, though technically describing the act of change or loss, is neutral, because unlike the unintended consequences of the harms done to soil, this refers to a deliberate act with an intended outcome and various beneficiaries.

The many approximate synonyms for rural land conversion with a more negative connotation include urbanisation, urban spread/sprawl/expansion/encroachment, land take/consumption/loss, long-term land cover change and (in France) artificialisation [160]. Other phrases commonly used in connection with rural land conversion are land competition and fragmentation, of farmland in particular, because this is one of the ways in which urbanisation becomes self-perpetuating [161]. Soil sealing, the permanent covering of soil, for example by concrete or tarmac, is a phrase widely used in conjunction with land take, because they typically occur together, but the meanings are distinct. While land take refers to a change of use which, by definition, usually entails some loss of land resource from its former use, sealing constitutes a much more permanent and intrinsic alteration of the land surface, typically with more far-reaching consequences for soil functions, especially drainage, and potentially severe effects on biodiversity [100, 162, 163]. Nevertheless, land take almost always involves some degree of sealing, and where it does, that constitutes a double impact – the spatial loss of land resource and the additional degradation of the ecosystem services that that resource provides in the round. However, sealing also occurs independently of more general land take, even in highly protected areas, for instance in the form of roads [162]. There are a few examples of unsealing, as a form of compensation for sealing elsewhere, but it is notable that these did not equate to total restoration [164].

On the positive land-saving side of the same lexical coin, the term predominantly used in North America, and in many countries, is land preservation [165, 166], but the terms land conservation or land protection are used almost interchangeably with it in the US [167, 168] and throughout the world. In most cases these phrases are applied to specific bounded areas, at various scales, which could mean a protected zone or alternatively a cluster of holdings, a single estate or even one field. The expressions “no net land take” or “zero land take”, which have appeared in EU documents in recent years [101, 169], are similar in principle, but refer to overall quotas or targets, with the additional challenges relating to relevant spatial scale and overall quantification. In contrast ‘land sparing’ is used in juxtaposition to ‘land sharing’, referring to the concept of safeguarding biodiversity, either by sparing natural lands from agricultural use (land sparing) or by incentivising farmers to support more biodiversity (land sharing) [170].

Globally words or phrases used to mean approximately the same thing as land preservation can sometimes have other connotations. Land conservation has historically been most strongly associated with nature conservation, in the sense of land that is not recognised as developed or cultivated and has some form of protected status. Such land is often threatened by agricultural incursion at least as much as by urbanisation. Land conservation is not necessarily synonymous with ‘soil conservation’, a term used professionally and more widely to refer specifically to the prevention of soil erosion with respect to land management practices.

Land protection is often synonymous with land preservation, but in some examples of legislation it is intended to mean soil protection, that is *in-situ* safeguarding. Soil protection is sometimes intended to mean or encompass land preservation. These semantic issues may sometimes result from translation because the full texts of many laws appear only in their original language and script [4]. Occasionally countries have all-embracing “soil protection” policies which include the concept of spatial land preservation in context [171, 172] or conversely all-embracing “land preservation (or protection)” policies which include the concept of soil protection [5, 172]. In some cases this blurring seems to be deliberate, such as when land preservation is promoted as a means to protect SES [5, 162]. Pragmatically soil scientists may need to accommodate themselves to the language of policy and spatial land use planning in order to engage with the process.

Finally, it is important to note that while many legal instruments have been designed specifically to protect only prime (i.e. highly productive) farmland, this is not always the main driver of land preservation or protection, which may prioritise other factors such as landscape, heritage or SES. In this article this distinction is highlighted by using the term prime farmland preservation (PFP) where appropriate.

Putting a value on soil

At the root of our current environmental crisis is that humanity has traditionally treated abundant natural resources as free goods and services. Economists and accountants are no exception: cost-benefit analyses include tradeable assets and products, but routinely exclude the ever-present prerequisites of life.

Furthermore, traditional cost-benefit analysis (CBA) is short-termism by definition because it usually does not even consider impacts outside a 25-year window [173]. When a tonne of soil or water is traded, the allocated price is normally derived from the aggregate cost of acquisition, processing and delivery, with no attempt to assess the intrinsic value of the substance. This applies not only to soil as an intrinsic good, but also to some extent to its economic potential, i.e. the land that contains it will have a market value and that value will be related to the land's productivity, yet unquantified and uncoded SES have far-reaching economic implications. Even in a society without monetary currency it would be relatively straightforward to calculate the economic value of a given volume of soil, in conjunction with the land it occupies, purely in terms of its capacity for life-sustaining primary production; ask any farmer. Although we now have modern techniques to assess the productive capacity of land resources, a conceptual process of "following the money" has been at the heart of most farm or settlement emplacement decisions since the Neolithic.

For at least 80 years soil survey and land evaluation have been the traditional tools at our disposal for assessing the productive and economic potential of an area of land, alongside any limitations or hazards it may present. A range of soil properties is recorded alongside other local environmental data and, where appropriate, socioeconomic data. This data is combined to grade the capability or suitability of each parcel or zone of land according to specified use criteria. The modern terminology of multicriteria decision analysis (MCDA) is more a change of style than substance, to reflect the much greater reliance on information technology, especially GIS [174]. In the above approaches "criteria" is the operative word. Where these methodologies differ from analogous *ad hoc* historical techniques, is in their capacity to distil decades of practical and scientific observation to apply much greater predictive accuracy and precision to the land use decision-making process [175].

Land evaluation is not the same thing as land valuation, though they have always been interrelated and both are relevant to spatial planning. Land evaluation methodologies were never intended to calculate the true, i.e. total and holistic, economic value of an area of land or a volume of soil (which are themselves two very different things), but rather the capability or suitability of distinct parcels of land for specified forms of primary production, such as crops, pasture or forestry. The purpose is for land use decisions, albeit often with profound economic implications. The purpose of land valuation, however, is generally to set prices for landowners and taxes for governing authorities, and has always primarily been based on location, which admittedly is historically bound up with soil quality and land use, but with a myriad of other factors too.

In contrast, soil has always provided a variety of extrinsic benefits beyond primary production which were recognised and appreciated long before we referred to them as ecosystem services. To what extent our ancestors valued (or devalued) specific areas of soil or land for reasons other than primary production, or obvious physical location, is not always tangible, though it is clear that many indigenous communities have developed deep understanding of, and spiritual connection to, the land and soil on which they depend [83].

A wider all-embracing appreciation of soil and a few attempts to evaluate it in that context date from the 1960s [176]. This concept has gathered momentum and, more recently above all, with respect to C.

While it may seem too neat to earmark the last decade of the previous millennium as a critical turning point in time, it seems inescapable that this period represented an important paradigm shift in our collective understanding of the role of soils. The 1990s shines out as a lightbulb moment when soil science and climate science came together. Although the role of C in the soil has been studied for at least 150 years and its place in the C cycle appreciated for much of that time, it was only in the 1990s that a flurry of papers explicitly linked soil C fluxes to GHG concentrations in the atmosphere and climate change [177-179] which, after several decades of growing evidence, had started attracting worldwide media attention since 1988 [180].

At the onset of the new millennium soil scientists have more than ever before started to deconstruct and articulate the critical importance of SES [181, 182]. Experts on soil law have responded accordingly, emphasising that the value and status of soil far exceeds its role in agriculture [183]. Taking this a step further, several groups of researchers, building on earlier attempts to evaluate ecosystem services [184] and characterising soil as a form of natural capital, have developed tentative frameworks and quantitative methods to enumerate and evaluate SES in the fullest sense [185-189], sometimes using case studies to apply monetary values [190]. To what extent this is entirely feasible or even desirable, is debatable [176], especially with regard to qualitative or ethical issues, but such approaches may help policymakers and planners apply more meaningful criteria and priorities to the governance of soil and land protection.

Categories of soil governance

Juerges et al. [65] summarise the types of instruments applied to soil governance (i.e. regulatory, economic, and so on) and the levels at which these operate, from global to local. In this section we present a very different cross-cutting breakdown, based on issues and intended outcomes. One can identify three broad categories of governance approaches applied to preserve or protect soil or land, usually at a national or subnational level, with various implementation mechanisms, set out in Table 2 and elaborated further below:

Table 2: Categories of governance approaches and mechanisms to preserve or protect soil or land

CATEGORY	DESCRIPTION
1	Regulations to prohibit (or guidance to discourage) certain actions
(a)	- Enforced by penalties
(b)	- Facilitated by financial support
2	Restrictions on development involving change of use

(a)	- Enforced by zoning laws
(b)	- Enforced by laws (or encouraged by guidance) based on ad hoc site evaluation
(c)	- Enforced by public acquisition of land
(d)	- Facilitated by financial support
3	Generic incentives to preserve land or to enhance its ecosystem services value
(a)	- Taking land out of agricultural production
(b)	- Converting intensively farmed land via extensification
(c)	- Declining to develop land or intensify its use

The first category comprises regulations and laws to **prohibit** (or guidance to discourage) **certain actions** by landowners that may degrade land, e.g. pollution, sealing, construction, stubble burning, tree felling, or cultivation methods leading to soil erosion, acidification, salinization, compaction, and so on; whereby best practice may be either:

- a. Enforced by penalties for infringement or
- b. Facilitated by financial support, e.g. farming subsidies, PES or C credits

The second category comprises **restrictions** (or guidance) **on development involving conversion of use**, either to protect terrestrial natural resources (for the benefit of any or all of primary production, watershed management, biodiversity, landscape or cultural or historical value) or simply to retain a certain quota of agricultural or forested land within a state or designated region; whereby land conservation or preservation may be:

- a. Enforced by laws based on zoning, e.g. National Parks, nature reserves, green belt, urban growth areas, Sites of Special Scientific Interest (SSSI) and heritage landscapes such as the UK Areas of Outstanding Natural Beauty (AONB) or UNESCO World Heritage Sites;
- b. Enforced by more general laws (or encouraged by guidance) not specific to the protected areas described in 2(a), based on *ad hoc* site evaluation, such as an Environmental Impact Assessment (EIA), and

thereby affording protection to sites of ecological or landscape value, or prime farmland (e.g. based on ALC or LESA criteria);⁸

- c. Enforced by public acquisition of land, such as land trusts;
- d. Facilitated by financial support, e.g. using LESA to obtain Purchase of Development Rights (PDR) conservation easements (US).

The third category encompasses **generic incentives** to preserve land or to enhance its ecosystem services value (or disincentives to develop, such as the removal of subsidies), in the form of:

- a. Taking land out of agricultural production, purely for the purpose of conservation, e.g. EU set-aside, US land retirement, rewilding;
- b. Converting an area of intensive agricultural production (arable cropping) to more extensive agricultural production (such as pasture, silvopasture or agroforestry) or forestry;
- c. Declining to develop or intensify the use of uncultivated land that could otherwise be legally developed or intensified, in order to maintain ecological or ecosystem services value.

Overlapping categories

The above categories are presented as a useful framework for analysis of soil governance. There are of course many instances of overlap. For example, regimes that are not targeted specifically at soils can still fall within the categories above. There are also numerous cases where soil governance instruments span a number of the mechanisms listed above. The examples below illustrate these points.

Environmental Impact Assessment (EIA) of projects, Strategic Environmental Assessment (SEA) of plans and programmes, and related types of assessment, may apply to some of these categories, especially categories 2(a) or 2(b), and can be helpful in drawing attention to threats to natural resources, but there can be limitations in their application to soils. In the US, for example, EIA is applied primarily for federally-funded projects [191] which is partly why the NRCS developed the LESA system for smaller projects targeted at farmland. In the UK EIA is more widely applicable, but generally only where part of the area intended for development is uncultivated or only partially cultivated and hence might not be invoked where only arable land is at risk [192]. EIA regulations vary slightly in Scotland, however, where the site being assessed may consist exclusively of farmland where it exceeds 200 ha. The Scottish EPA also provides specific guidance on application of SEA to soils [193].

⁸ A legally binding example is the Indian Prohibition on Conversion of Agricultural Land for Non-Agricultural Use (No. 16 of 2010).

The broad North American planning term land preservation covers many of these categories with the main focus on zoning [165, 194]. Every state and city has its own subset or version of land preservation laws and regulations which are varied and complex. One example is a “conservation easement” whereby a landowner is bound by a covenant set by the government or some other organisation and in return receives an incentive, such as a tax rebate; this could fall under categories 1(b), 2(d), or 3(a-c).

Apple Valley City, Minnesota has adopted a Natural Resources Management Plan (NRPM) which essentially constitutes a 2 (b) type mechanism via a permitting system, but incorporates elements of 1(a) because it could apply to a single aspect of a development. It differs from most local planning laws and regulations by applying the broad principle of environmental impact to every development [195].

There is an interesting example of environmental scientists trying to set a legal precedent as expert witnesses in a 2(b) type scenario in New Zealand in 2011, using soil natural capital and ecosystem services arguments to prevent urban development on horticultural land. The lawyer representing the developer argued that the only measured ecosystem service of this soil was food production. The judge, while declining to engage in the natural capital debate, nevertheless upheld the local authority decision not to allow development in favour of the “holistic” argument to protect natural resources which, from the point of view of the scientists, came to the same thing [196].

The modern governance of soil resources worldwide

While the piecemeal governance of soil use is almost as old as human history, formal supranational global oversight of soil resources is only approximately 50 years old. In terms of the natural environment, the early 1970s stand out as a period when national policies, international initiatives and academic analyses of global problems coalesced in an unprecedented step change in attitudes. 1970 saw the conception of Earth Day, initially in the US but later worldwide, and consequently the creation of the US Environmental Protection Agency, an institution which also became replicated worldwide. In 1972 the Club of Rome think tank, published the highly influential *Limits to Growth*, with its prediction of societal collapse in the 21st century, and frequent references to soil degradation [197]. This period also saw the emergence of multidisciplinary stakeholder groups joining forces for sustainable development [198].

Emergence of soil in modern international law and policy

The 1968 *African Convention on the Conservation of Nature and Natural Resources* (the “Algiers Convention”) was a continent-wide treaty devoted to natural resource protection that included a reference to soil. Although it lacked the resources and institutional framework to be particularly effective, the Algiers Convention was nevertheless regarded as a milestone in international environmental law [199]. At around

the same time FAO was publishing several of its soils bulletins every year, including two landmark studies in 1971, on land degradation [200] and on legislative principles of soil conservation [201]. The latter was a relatively concise set of guidelines that any nation could use as a template for creating or improving its own legal framework for governing soil. The 1972 UN Conference on the Human Environment (UNCHE) in Stockholm was effectively the first major international conference on environmental protection and sustainable development, and has been referred to by Boer *et al.* as marking the: "...first phase of international soil protection law" [202, 203].

In 1972 the Council of Europe created and adopted the *European Soil Charter*, a concise but relatively holistic set of soil protection aspirations [204] which was non-binding, but regarded nevertheless as the first international legal instrument dedicated specifically to protecting soils [202, 205]. In 1977 the UN held its first Conference on Desertification (UNCOD), and produced its *Plan of Action to Combat Desertification* (PACD) [206]. In 1981 FAO adopted the *World Soil Charter* [207] and in 1982 UNEP developed a global soils policy [208] and the IUCN/UN *World Charter for Nature* explicitly referenced soil [209].

Soil as a feature of common concern

In 1987 the UN commissioned the Brundtland report, *Our Common Future*, which reinvigorated the debate, again stressing the severity of soil degradation, and annexed a summary of proposed legal principles for environmental protection and sustainable development [210]. In order to address these issues financially, in 1991 UN agencies and the World Bank created the Global Environment Facility (GEF), as a prerequisite for the 1992 UN Conference on Environment and Development (UNCED) or Rio Earth Summit [211], a milestone event. From this emerged the UN Commission on Sustainable Development (UNCSD) and three legally binding international treaties: the 1992 *UN Convention on Biological Diversity* (CBD) [212], the 1992 *UN Framework Convention on Climate Change* (UNFCCC) [213] and the 1994 *UN Convention to Combat Desertification* (UNCCD) [214], which came into force in 1993, 1994 and 1996, respectively (the Rio Conventions). There was appreciation of the interrelationships linking all three conventions to land degradation [215], but funding for the latter was mainly tied to the UNCCD which was always the weakest and poorest of the three due to donor scepticism [52]. The UNCCD has been called the first and only "legally binding global agreement directly dealing with the promotion of bio-productive land" [202], although it contains little in the way of substantive obligations, and formally applies only to "arid, semi-arid and dry sub-humid areas" [216]. Nevertheless, UNCCD is gaining momentum as a focal point for efforts to address the global land degradation neutrality (LDN) target [206, 217].

The interconnected nature of different environmental problems, as well as their linkages to socioeconomics and SLM, was taken a step further at the 2012 UN Conference on Sustainable Development (Rio+20). The UNCCD brought the concept of zero net land degradation (ZNLDD) for adoption at Rio+20, reflected in the

outcome document, *The Future We Want* [206], and this later became embedded into the 2015 SDGs as the global LDN target in SDG 15.3 [206, 218]. Land degradation was finally being acknowledged as a global problem that was intrinsically linked with climate change, biodiversity loss and poverty. The concept of ZNLD/LDN was that every effort should be made either to prevent further degradation or, where this is not possible, to rehabilitate equivalent areas of degraded land elsewhere [218]. The importance of land and soils is gaining more prominence in the context of the CBD [212], the UNFCCC [219] and related *Paris Agreement* [220]. The parties to the CBD have addressed soil biodiversity via an international initiative [221] and a global report [222]. It remains to be seen how soils will be reflected in the Global Biodiversity Framework to be adopted by the CBD COP 15, currently re-scheduled to take place in China (in phases spanning 2021 and 2022) especially given China's rapid advance in the field of soil science ⁹.

At its first Plenary Assembly, at the FAO Headquarters in Rome in June 2013, the Global Soil Partnership (GSP) created the Intergovernmental Technical Panel on Soils (ITPS), made up of 27 soil experts representing all the regions of the world. [73]. The GSP also proposed 2015 as the International Year of Soils (IYS) and the annual observance of a World Soil Day (on 5 December), both of which were adopted at the 68th UN General Assembly later that year [223]. Under the auspices of the GSP the *Revised World Soil Charter* [224] was adopted by FAO members in 2015, and the *Voluntary Guidelines for Sustainable Soil Management* (VGSSM) [225] were endorsed by the FAO Council in 2016.¹⁰ The work of the ITPS, UNCCD Science-Policy Interface (SPI), IPBES, the IPCC, and in particular the publication of the 2015 *Status of the World's Soil Resources* [73], the 2017 *Scientific Conceptual Framework for Land Degradation Neutrality* [226], the 2018 *Global Land Degradation and Restoration Assessment* [77], and the 2019 *Report on Climate Change and Land* [51], by each respectively, stand out as key influences in the soil governance narrative. The significance of the science-policy interface is increasingly important in determining the scope and extent of legal obligations [227].

In 2015, to coincide with UNFCCC COP21, the French government issued a bold entreaty to the global community to raise average soil organic C (SOC) levels by 0.4% (the "4 per mille Soils for Food Security and Climate" initiative), to offset annual global C emissions into the atmosphere, as well as improving food security [228, 229]. The choice of SOC as the target was fundamental and twofold, because it represents both a means of accumulating and sequestering atmospheric C, and the most widely accepted measure of soil health or productivity. The 4 per 1000 Initiative, to which many countries have signed up, has succeeded in highlighting the critical role of soil and agriculture in climate change mitigation and adaptation. There has

⁹ According to the SJR International Science Ranking website China ranks second only to the US in soil science (<https://www.scimagojr.com>; accessed 28th April, 2021).

¹⁰ <http://www.fao.org/3/bl813e/bl813e.pdf>

also been criticism [230], especially of the feasibility of the initiative and underpinning data. While the authors have defended the science, with caveats, they also stress that 4 per 1000 was never intended to be a precisely calculated solution, but a positive, politically-driven and symbolic aspirational target [231]. Other soil scientists agree with them, judging that the initiative would be a technically feasible, “no-regret” and indispensable climate action [232].

Regional and sectoral developments

At the regional level the Algiers Convention was revised in 2003 to become the *Revised African Convention on the Conservation of Nature and Natural Resources* (Maputo Convention), but only entered in force in 2016 [233]. In 1998 the Alpine Soil Protocol was the first legally binding treaty expressly devoted to soil, entering in force in 2006 [234]. For both the Maputo Convention and the Alpine Soil Protocol, implementation has not yet matched aspiration, but both serve as useful focal points for awareness raising regarding soil conservation, management and best practice. The ASEAN Agreement on the Conservation of Nature and Natural Resources, signed in 1985, included an article specifically on soil, but it has never come into force [202].

The Council of Europe revised the *European Soil Charter* in 2003 to become the *Revised European Charter for the Protection and Sustainable Management of Soil* [235]. In contrast, in the European Union, agreement on a soil instrument resembles a triumph of hope over experience. In 2006 the European Commission presented its *Thematic Strategy on Soil Protection* and presented a *Proposal for a Soil Framework Directive* to place healthy soil on a par with clean water and air [236]. The proposal was eventually withdrawn in 2014, because five member states were not willing to agree to strengthened EU-wide legislation addressing soil sealing and liability for contaminated land [237]. A new EU Soil Strategy is planned for adoption in 2021 as a key component of a European Green Deal (EGD), aimed at making Europe the first climate-neutral continent by 2050 [238]. The European Parliament has called for a binding legislative framework for soils. [239]

Sustainable Development Goals (SDGs)

All countries have signed up to the seventeen SDGs [240]. Land and soil have profound relevance for most, if not all, SDGs, though only a relatively small number of SDG targets make specific reference to them [240, 241]. In particular *SDG2: Zero hunger*, aims in target 2.4 to ensure sustainable food production systems and resilient agricultural practices that increase productivity and production, help maintain ecosystems, strengthen capacity for adaptation to climate change, extreme weather, drought, flooding and other disasters and progressively improve land and soil quality; and *SDG 15: Life on land*, aims to protect, restore and promote sustainable use of terrestrial ecosystems, including in target 15.3 combatting desertification, restoring degraded land and soil, and striving to achieve a land degradation neutral (LDN) world. SDG targets 3.9, 6.4, 6.5, 12.4 and 14.1 also have particularly obvious direct relevance to soils.

[241], and land and soil are fundamentally related to many other SDG targets and indicators, including those for SDG 13 on urgent climate action. While the SDGs themselves are not-legally binding, they do in some respects echo or amplify existing and emerging binding international commitments.

National soil legislation

Last year FAO and UNEP jointly published a key document, which is candid in its acknowledgement that implementation of international commitments can be difficult to enforce and monitor, especially when they are purely aspirational rather than legally binding [217]. The report provides many examples of innovative and progressive legislation in relation to land and soil, though questions still remain in relation to enforcement [68]. On a global scale the most urgent action needed may relate to category 2a in Table 2 - restrictions or guidance on development involving conversion of use, to protect terrestrial natural resources and enforced by laws based on zoning, e.g. National Parks, nature reserves, etc., to address issues such as deforestation in Amazonia or Southeast Asia. The instruments to regulate this type of anthropogenic threat are relatively straightforward and consistently defined across the world, but the obstacles are political and socioeconomic. This is also true of other existing laws that directly or indirectly protect soil, the success of which require that entrenched power imbalances are challenged [64].

Lack of enforcement and governance on the ground remain serious obstacles throughout much of the world. See, for example, the studies referenced at [66, 89, 92, 112, 114, 120, 146, 155, 242, 243]. Those ten studies cited, merely a subset of many more, draw on a range of data for their evidence, in addition to literature reviews, including land use and land evaluation records, census and demographic statistics, case studies, planning and legal decisions, stakeholder interviews and, perhaps most telling of all, spatial remote sensing of land use change, using GIS software tools such as CORINE. Nevertheless, the process of enacting soil and land legislation is ongoing and ubiquitous, and reflects each country's circumstances and priorities. China, for example, with possibly the greatest absolute area of contaminated soil on Earth [244-246], as well as a long history of soil erosion [8], has separate soil laws to address both of these problems.

Table 3 presents a condensed numerical summary of derived categories of soil-related legal instruments, presented as approximate percentages of countries that have created or adopted such instruments (as detailed in *Methods* above). It is evident that soil and land protection laws are conspicuous by their relative absence worldwide. Environmental legislation is not specifically recorded here but, for comparison, it has been almost universally adopted. It is also conspicuous from the data below that for many countries the word *soil* would be entirely absent from their legal portfolio were it not for such environmental legislation.

Table 3 Proportions of countries with soil-related legislation

(a) SOIL-SPECIFIC LEGAL INSTRUMENTS	COUNTRIES (%)
Explicit soil policy.	7
Soil conservation/erosion law(s) [explicit/explicit+implicit].	34/79
Soil contamination/pollution law(s) [explicit/explicit+implicit].	27/65
Soil sealing law(s) [explicit/explicit+implicit].	3/27
Generic or other soil protection law(s) [explicit/explicit+implicit].	28/71
Soil protection monitoring and/or targets [explicit/explicit+implicit].	21/47
Reference to soil embedded within environmental legislation.	88
Reference to soil embedded within agricultural or land rights legislation.	72
Reference to soil embedded within spatial/regional planning legislation.	50
(b) LEGAL INSTRUMENTS TO PRESERVE AGRICULTURAL LAND	
Policies designating zoning, including rural and/or agricultural land.	35
Policy advocating land take avoidance.	24
Land take targets.	9
Legal framework to facilitate farmland preservation schemes.	19
Prime farmland preservation guidance based on soil or land classification.	27
Law prohibiting or strongly restricting loss of all prime farmland.	21
Land Degradation Neutrality (LDN) policy (commitments).	66
(c) SOIL-RELATED AGRO-ENVIRONMENTAL POLICIES	
Policies promoting organic or regenerative agriculture, agroecology or PES.	40
Policies referring specifically to the critical ecosystem services of soils.	15

Prime farmland preservation (PFP)

The data presented in Table 3, along with our literature review, leads the authors to conclude that one of the most serious failings with respect to the ambiguity or absence of legal instruments relates to what we have termed category 2(b) in Table 2, that is land take outside protected areas. At most risk and of most concern is the conversion of high quality or prime farmland, which is usually afforded far less protection than natural ecosystems, or none at all, yet can sometimes provide equivalent, or even greater, ecosystem services. This

is an important and often misunderstood point that cannot be overestimated. It is easy to fall into the trap of assuming that agricultural land cannot provide environmental benefits comparable to those of natural ecosystems, especially when based on current land management practices, but this certainly does not reflect the intrinsic value of such land nor necessarily even its existential value [247].

Land which is densely covered in vegetation, with undisturbed topsoil, will tend to store more water and C, and foster greater biodiversity, but the principle reason land becomes prime farmland is its highly valued ecosystem attributes: low altitude and gentle gradients; deep soil with favourable texture and structure that retains water, nutrients and C; benign biochemistry; and a favourable moisture regime that is not susceptible to extremes of drought or waterlogging. These are the desirable attributes that facilitate not just food production, but terrestrial life in general, which ecologists assess in aggregate as NPP. As crude as it might be in some respects, especially in relation to cultural or scientific value, NPP is widely regarded as one of the best proxy measures of total ecosystem service contribution [248]. In contrast to this, many of the varied soil landscapes that currently support natural ecosystems can be biologically marginal, exhibiting low NPP. Even putting aside the utilitarian criterion of agricultural potential, these peripheral zones can also be limited with respect to a range of critical ecosystem services, including in terms of the biodiversity they support.

Prime farmland is not usually an obvious candidate for voluntary conversion to natural regeneration or rewilding, but high-grade arable land, even in its cultivated state, can provide abundant benefits in terms of flood control, water storage and filtration, wildlife habitat, landscape value and recreation. However, and critically, such benefits are greatly affected by land management. The negative impacts of modern intensive farming, on the soil in particular, such as compaction, erosion, pollution and reduced SOC, are precisely the factors that undermine the ability of the soil to provide ecosystem services [249, 250].

One very welcome trend in recent decades has been to roll back many of these harmful methods. A range of techniques that are loosely grouped under the term of conservation, or regenerative, agriculture, agroecology, or even “carbon farming” are increasingly being promoted and applied in many countries [78, 251-253]. Included within this approach are minimum or zero tillage, permanent soil cover, crop residue retention, cover cropping, intercropping and enhanced rotations including, for example, deep rooting “tillage crops”, leguminous leys and rotational livestock grazing, agroforestry and other forms of mixed cropping, perennial crops, integrated pest management (IPM) and biological control, and many other ways of conserving and enhancing soil quality and, by default, ameliorating environmental impacts. Hence, our advocacy of PFP is proposed in tandem with the adoption of such methods.

Emerging holistic conceptual frameworks

Amundson sums up soil governance in the US as a complex patchwork of transient interventions, as opposed to long-term solutions [254]. Other authors describe comparable situations elsewhere, for example,

in the EU [255, 256], in South America [257], in Australia [258], in Russia [259] and in other examples already cited in this article. Fromherz summarises many of the existing soil governance initiatives, as well as the failures, and makes an impassioned appeal for a dedicated, legally binding international soil governance instrument on the basis that, "...individual states lack both the power and the incentives to make these changes." [260]. This has been echoed by others [183, 203, 239, 261]. Fromherz also highlights a key point alluded to already in this article, the low profile (literal as well as metaphorical, one could say) of soil, which is often conspicuous in its absence from risk assessments of other natural resources [262, 263]. Gonzalez Lago *et al.* refer to a global soils policy vacuum and call for an urgent transdisciplinary framework approach to "re-politicise" soil [264], while a number of authors and institutions highlight the degree to which soil protection is inescapably enmeshed with ethics [265].

Participatory modelling and conceptual frameworks have been applied to complex cross-cutting problems such as addressing the SDGs, and these approaches continue to evolve [266]. The latest stages in the process of soil governance so far, in the last decade, are encouraging to some extent, at least with regard to what soil scientists are bringing to the table. An overarching conceptual framework that has been proposed by some of the world's leading soil scientists, broadly as a memorandum of understanding, is "soil security" [267]. This concept has five dimensions: capability, condition, capital, connectivity and codification, which encompasses the translation of soil knowledge into policy and legislation [268], e.g. aimed at achieving the SDGs [269], although an attempt to establish soil security as one of the SDGs was unsuccessful [270]. The soil security initiative is still at a high level, with the emphasis on policy rather than active governance, but progress is being made and reported on in some areas [271].

Also emerging are more formalised frameworks that place the concept of SES further than ever in the context of socioeconomic decisions, policy-making and governance. One such methodology developed and tested in Switzerland is SQUID (Soil Quality InDicator), an index for mapping soil quality with respect to SES, to guide spatial development [189]. Another example from New Zealand is the Land Resource Circle (LRC) framework, which goes further than any other approach known to the authors in extending land evaluation to incorporate SES. The framework identifies in-depth environmental and socioeconomic implications of land-use decisions via a scoring system which avoids reducing all outcomes to simplistic monetary values [272]. A recent LRC paper includes a very detailed hypothetical example which indicates how quantified soil variables can be combined and converted into societal costs and benefits. By contrast, the Resilience-Effectiveness-Efficiency-Legitimacy (REEL) framework from Germany provides a purely qualitative means of comparing different approaches to soil governance according to the four criteria (dimensions) embedded within its title [273]. The REEL framework also highlights interconnectedness and attempts to address situations where policy targets mismatch the causes of problems, either spatially or in

terms of scale or even time. This is almost a meta-framework, with the emphasis on socioeconomics and due diligence.

As with other forms of natural resource governance, politics and socioeconomics are often obstacles to effective soil governance, as are weak governance structures, but a critical distinction in relation to PFP is that there is much scope for creating more effective legal instruments to tackle this problem than currently exist in most countries, and also for improving clarity and consistency to bring PFP further in line with ecosystem and soil protection.

Conclusion

An appreciation of the tangible benefits of soil is woven into the fabric of history and reflected by the value and protections afforded to productive soil by society over millennia. Such protections include legislation, but *ad hoc*, inadequately enforced and rarely if ever proportionate to the total SES. Two challenges identified as hindering progress are terminological obstacles associated with soil and land, and the absence of a soil-centric policy framework. Moreover, soil exists as a component of land that is subject to both extra demands, e.g. of food production, and competing demands, such as urbanisation. Agriculture has responded to these demands by expansion and intensification, each approach posing threats to soil, the former often encroaching on natural ecosystems and the latter fostering soil degradation. These threats have spawned national and international legislation. However, the problem of land take, in particular the conversion of prime agricultural land to non-agricultural use, including largely irreversible soil sealing, is arguably at least as serious a threat as intensification because of the disproportionate loss of SES this represents, yet tends to be relatively inconspicuous, and subject to ambiguous legislation and governance.

Threats to natural ecosystems are explicitly addressed by environmental laws, and to soil degradation by agricultural or soil-specific laws. A few of these laws have been in place for centuries and more have appeared in recent decades as global environmental awareness has grown; but neither environmental nor agricultural legislation normally encompasses the loss of prime farmland, the fate of which then tends to fall within the remit of spatial planning. Planners' procedures must navigate many competing demands and interest groups, political as well as economic, and rarely prioritise soil or its intrinsic value to future generations. As a result, legal instruments that are frequently designed with little, if any, reference to soil, often have greatest power to transform soil and land use. This situation is further compounded by the fact that intense commercial pressure along with conventional economics, which has traditionally ignored the benefits of long-term ecosystem services, promotes land use options that are more profitable than farming.

In a time of so much environmental concern, a further paradoxical factor that has emerged in the last half century to weaken the case for farmland preservation (as opposed to soil protection *per se*), is the conflicted

nature of agriculture: on the one hand the producer of our sustenance and custodian of rural landscapes, while on the other, a significant cause of environmental harm and climate forcing. Lacking both the visual impact and the iconic status of natural landscapes, soil in an agricultural context does not constitute an obvious rallying point in the public consciousness and so the constant attrition of prime farmland occurs in something of a legal and ecological grey area, attracting far less attention than other environmental issues.

However, the case for protecting soil as a critical part of our natural capital is separately gaining ground. A key paradigm shift occurred in the 1990s when it became more widely appreciated that soil C is inextricably linked with atmospheric C. This development also approximately coincided with a broader gradual acceptance that land degradation was interrelated with climate change and biodiversity loss. This view of soil as a central component in limiting global warming, adapting to climate change and addressing the ecological crisis, is being framed unequivocally in the wider context of sustainability and the linking of science to policy. Emerging multidisciplinary frameworks, such as the concept of soil security and methodologies for holistically evaluating SES, contribute to this process.

Politicians and activists are calling for “green new deals” to emulate Roosevelt’s New Deal following the Great Depression, but with an emphasis on climate justice and ecosystem restoration. This encompasses sustainable land use, by default, and important components of that must be soil protection and land preservation, consistent with climate justice, including the attendant social and economic incentives necessary to achieve local and global climate goals, biodiversity objectives and food security. Profound reform can reach a tipping point when society perceives a binary moral choice that serves the common good. Two successful examples of global co-operation of this kind were the signing of the *Montreal Protocol* in 1987 to protect the ozone layer, and persuading the global community to reduce GHG emissions in the *Paris Agreement*, though hard-won and far from over. Ginzky cites other examples and, despite poor progress to date, suggests that a binding international treaty on soils is both necessary and achievable [274].

Clearly a lack of effective soil governance is often more significant a problem than an absence of legal instruments. The lesson here, for saving our soil, is to strive for a clear message, backed up by consistent policies and laws, and sound criteria to decide how to prioritise soil types or areas of land. This alone cannot solve the vast problem, but it will make the process more streamlined and more transparent, and facilitate governance for those who genuinely want to govern, and make infringements that bit more difficult to effect and pass unnoticed. The urgent need is to couple a compelling case with an achievable solution.

Data accessibility

The only original data presented is the categorisation percentages presented in Table 3.

Authors' contributions

L.R.P. conceived the idea, conducted much of the research and wrote the bulk of the text. C.R. made considerable contributions to the text, especially in relation to soil law, and performed much editing.

Competing interests

We declare we have no competing interests.

Funding

We received no funding for this study.

Acknowledgements

We wish to thank the following who helped to explain their local soil and land laws, or planning regulations, in theory and practice: Dr Thomas Daniels (Univ. Pennsylvania); Dr Mark Lapping (Univ. Southern Maine); Welsh Government Soil Policy Team (Natural Resources Wales); Planning and Local Government Specialist Team, Soils (Natural England); Prof. Fernando García Prechac (Univ. Uruguay, and former Director General of Natural Resources, Ministry of Livestock, Agriculture and Fisheries, Uruguay); Marcel Achkar (Univ. Uruguay); Gundula Prokop (EU Research and Territorial Co-operation Projects, Federal Environment Agency, Austria); Anna Shortly (Greenbelt Foundation, Ontario); Dr Elena Havlicek and Ruedi Stähli (Federal Office for the Environment (FOEN), Switzerland); Ljubiša Bezbradica (Institute of Architecture and Urban & Spatial Planning of Serbia); Tom Corser (New Zealand Ministry for the Environment); and Dr Suhail Ahmad (Cluster University Of Srinagar, Kashmir). In addition: Dr David Dent (former Director, ISRIC) gave advice on soil degradation assessments which he co-authored at ISRIC; CPRE posted a free copy of *Valuing the Land* which is unavailable online; and two anonymous reviewers helped us to improve the balance and style of the paper.

References

1. Brussaard, L. 2021 Biodiversity and ecosystem functioning in soil: The dark side of nature and the bright side of life. *Ambio*. <https://doi.org/10.1007/s13280-021-01507-z>
2. Smith, P, Keesstra, SD, Silver, WL, Adhya, TK, De Deyn, GB, Carvalheiro, LG, Giltrap, DL, Renforth, P, Cheng, K, Sarkar, B, et al. 2021 Soil-derived Nature's Contributions to People and their contribution to the UN Sustainable Development Goals. *Philosophical Transactions of the Royal Society B: Biological Sciences* **376**, 20200185. <https://doi.org/doi:10.1098/rstb.2020.0185>

3. Almond, REA, Grooten, M & Petersen, T. 2020 WWF (2020) Living Planet Report 2020 - Bending the curve of biodiversity loss. eds. REA Almond, M Grooten & T Petersen. Gland, Switzerland, WWF.
4. FAO. FAOLEX Database: <http://www.fao.org/faolex/country-profiles/en/>.
5. Government of Ukraine. 2020 Land Code (No. 2786-III of 2001; revised 16.10.2020).
6. Bennett, HH. 1939 *Soil conservation*. New York, London, McGraw-Hill.
7. Butzer, K. 2005 Environmental history in the Mediterranean world: cross-disciplinary investigation of cause-and-effect for degradation and soil erosion. *Journal of Archaeological Science* **32**, 1773-1800. (doi: 1710.1016/j.jas.2005.1706.1001).
8. Dotterweich, M. 2013 The history of human-induced soil erosion: Geomorphic legacies, early descriptions and research, and the development of soil conservation—A global synopsis. *Geomorphology* **201**, 1-34. <https://doi.org/10.1016/j.geomorph.2013.07.021>
9. Hyams, E. 1952 *Soil and Civilization*, Thames and Hudson.
10. Redman, CL. 1999 *Human impact on ancient environments*. Tucson, University of Arizona Press.
11. Brevik, EC & Hartemink, AE. 2010 Early soil knowledge and the birth and development of soil science. *CATENA* **83**, 23-33. <https://doi.org/10.1016/j.catena.2010.06.011>
12. Verheye, WH. 2009 *Land Use, Land Cover and Soil Sciences - Volume VII: Soils and Soil Sciences - 2*, EOLSS Publ.
13. Homer-Dixon, TF. 1994 Environmental Scarcities and Violent Conflict: Evidence from Cases. *International Security* **19**, 5-40. <https://doi.org/10.2307/2539147>
14. Berman, N, Couttenier, M & Soubeyran, R. 2019 Fertile Ground for Conflict. *Journal of the European Economic Association*. <https://doi.org/10.1093/ieea/jvz068>
15. Fearon, J & Laitin, D. 2014 Does Contemporary Armed Conflict Have 'Deep Historical Roots'? *SSRN Electronic Journal*. <https://doi.org/10.2139/ssrn.1922249>
16. Global Witness. 2020 Defending Tomorrow: The climate crisis and threats against land and environmental defenders.
17. Le Billon, P & Lujala, P. 2020 Environmental and land defenders: Global patterns and determinants of repression. *Global Environmental Change* **65**, 102163. <https://doi.org/10.1016/j.gloenvcha.2020.102163>
18. Gong, Z, Zhang, X, Chen, J & Zhang, G. 2003 Origin and development of soil science in ancient China. *Geoderma* **115**, 3-13. [https://doi.org/10.1016/S0016-7061\(03\)00071-5](https://doi.org/10.1016/S0016-7061(03)00071-5)
19. Barrera-Bassols, N, Alfred Zinck, J & Van Ranst, E. 2006 Symbolism, knowledge and management of soil and land resources in indigenous communities: Ethnopedology at global, regional and local scales. *CATENA* **65**, 118-137. <https://doi.org/10.1016/j.catena.2005.11.001>
20. Ellickson, RC & Thorland, CD. 1995 Ancient Land Law: Mesopotamia, Egypt, Israel. *Chicago-Kent Law Review* **71**, 321.

21. Barbieri-Low, AJ & Yates, RDS. 2015 *Law, State, and Society in Early Imperial China (2 vols)*, Brill.
22. Shi, S. 1959 *On "Fan Shêng-chih shu" : an agriculturist book of China written by Fan Shêng-chih in the first century B.C. / translated and commented upon by Shih Sheng-han = Fan Shengzhi shu jin shi / Shi Shenghan yi shi*. Peking, Science Press.
23. Milde, KF. 1950 Roman Contributions to the Law of Soil Conservation. *Fordham L. Rev.* **19**, 192-196.
24. Stocking, M. 1985 Soil Conservation Policy in Colonial Africa. *Agricultural History* **59**, 148-161.
25. Sandbach, F. 1980 *Environment, Ideology, and Policy*, Allandheld, Osmun.
26. Williams, GW & United States. Forest Service. 2005 *The USDA Forest Service : the first century*. Washington, DC, USDA Forest Service; iv, 156 p. p).
27. Wynn, G. 1979 Pioneers, politicians and the conservation of forests in early New Zealand. *Journal of Historical Geography* **5**, 171-188. [https://doi.org/10.1016/0305-7488\(79\)90132-4](https://doi.org/10.1016/0305-7488(79)90132-4)
28. Reddy, B, Hoag, D & Shobha, B. 2004 Economic incentives for soil conservation in India. In *Proceedings of the 13th International Soil Conservation Organization Conference, Brisbane*.
29. Crofts, R & Olgeirsson, FG. 2011 *Healing the Land: The Story of Land Reclamation and Soil Conservation in Iceland*, Soil conservation service.
30. McLeman, RA, Dupre, J, Berrang Ford, L, Ford, J, Gajewski, K & Marchildon, G. 2014 What we learned from the Dust Bowl: lessons in science, policy, and adaptation. *Population and Environment* **35**, 417-440. <https://doi.org/10.1007/s11111-013-0190-z>
31. Pretty, JN. 1995 *Regenerating agriculture : policies and practice for sustainability and self-reliance*. London, Earthscan.
32. Sylvester, KM & Rupley, ES. 2012 Revising the Dust Bowl: High Above the Kansas Grasslands. *Environ Hist Durh N C* **17**, 603-633. <https://doi.org/10.1093/envhis/ems047>
33. Peake, L. 1986 *Erosion, Crop Yields and Time: A Reassessment of Quantitative Relationships*, University of East Anglia School of Development Studies.
34. Konyushkov, D. 2014 Russian: Evolution and Examples. In *Reference Module in Earth Systems and Environmental Sciences* (Elsevier).
35. Carter-Whitney, M, Esakin, TC & Canadian Institute for Environmental Law and, P. 2010 Ontario's greenbelt in an international context.
36. Simonson, RW. 1989 *Historical Highlights of Soil Survey and Soil Classification with Emphasis on the United States, 1899-1970*. Wageningen, the Netherlands, International Soil Reference and Information Centre (ISRIC).
37. Young, A. 1980 *Tropical Soils and Soil Survey*, Cambridge University Press.
38. Robinson, A. 1943 The Scott and Uthwatt Reports on Land Utilisation. *The Economic Journal* **53**, 28-38. <https://doi.org/10.2307/2226286>

39. Sheail, J. 1997 Scott revisited: Post-war agriculture, planning and the British countryside. *Journal of Rural Studies* **13**, 387-398. [https://doi.org/10.1016/S0743-0167\(97\)00028-4](https://doi.org/10.1016/S0743-0167(97)00028-4)
40. CPRE. 2017 LANDLINES: why we need a strategic approach to land. London, CPRE.
41. Monk, S, Whitehead, C, Burgess, G & Tang, C. 2013 International review of land supply and planning systems. York, UK, Joseph Rowntree Foundation.
42. Natural England. 2012 Agricultural Land Classification: protecting the best and most versatile agricultural land. Technical Information Note TIN049.
43. Daniels, T. 1990 Using LESA in a purchase of development rights program. *Journal of Soil and Water Conservation* **45**, 617-621.
44. Lee, ATM. 1948 Soil Conservation—An International Study, Food and Agriculture Organization of the United Nations, Washington, D. C., 1948. *American Journal of Agricultural Economics* **30**, 784-786. <https://doi.org/10.2307/1232795>
45. United Nations. 1949 *UN Yearbook 1947-48*. Washington.
46. Bunce, AC. 1942 *The economics of soil conservation*. Ames, Ia., Iowa State college Press.
47. Ciriacy-Wantrup, SV. 1952 *Resource Conservation: Economics and Policies*, University of California, Division of Agricultural Sciences, Agricultural Experiment Station.
48. Sauer, EL. 1967 *Economics of Soil Conservation, Reclamation and Rehabilitation*.
49. Morgan, RJ & Future, Rft. 1965 *Governing Soil Conservation: Thirty Years of the New Decentralization*, Resources for the Future.
50. Aubreville, A. 1949 *Climats, Forêts, et Désertification de l'Afrique Tropicale*. Société d'Editions Géographiques, Maritimes et Tropicales. Paris.
51. IPCC. 2019 *Climate Change and Land: an IPCC special report on climate change, desertification, land degradation, sustainable land management, food security, and greenhouse gas fluxes in terrestrial ecosystems*.
52. Gisladdottir, G & Stocking, M. 2005 Land degradation control and its global environmental benefits. *Land Degradation & Development* **16**, 99-112. <https://doi.org/10.1002/ldr.687>
53. Thomas, DSG & Middleton, NJ. 1994 *Desertification: Exploding the Myth*. London, Wiley.
54. Davidson, J & Wibberley, GP. 1977 *Planning and the rural environment*.
55. Pingali, PL. 2012 Green Revolution: Impacts, limits, and the path ahead. *Proceedings of the National Academy of Sciences* **109**, 12302-12308. <https://doi.org/10.1073/pnas.0912953109>
56. Mannion, AM. 1991 *Global environmental change : a natural and cultural environmental history*. Harlow, Longman Scientific & Technical.
57. Agricultural Advisory Council. 1970 *Modern Farming and the Soil: Report of the Agricultural Advisory Council on Soil Structure and Soil Fertility*. London, HM Stationery Office.

58. Biniek, JP. 1979 *Agricultural and environmental relationships issues and priorities: report*. Washington, Library of Congress, U.S. Govt. Print. Off.; xvi, [1], 696 p. p).
59. USDA. 1973 *Monoculture in Agriculture: Extent, Causes, and Problems-report of the Task Force on Spatial Heterogeneity in Agricultural Landscapes and Enterprises*.
60. Carson, R. 1962 *Silent spring*. Boston, Cambridge, Mass., Houghton Mifflin; Riverside Press; x, 368 p. p).
61. Schneider, SH. 1989 The greenhouse effect: science and policy. *Science* **243**, 771-781.
62. Leahy, S, Clark, H & Reisinger, A. 2020 Challenges and Prospects for Agricultural Greenhouse Gas Mitigation Pathways Consistent With the Paris Agreement. *Frontiers in Sustainable Food Systems* **4**. <https://doi.org/10.3389/fsufs.2020.00069>
63. Shivanna, KR. 2020 The Sixth Mass Extinction Crisis and its Impact on Biodiversity and Human Welfare. *Resonance* **25**, 93-109. <https://doi.org/10.1007/s12045-019-0924-z>
64. Weigelt, J, Müller, A, Janetschek, H & Töpfer, K. 2015 Land and soil governance towards a transformational post-2015 Development Agenda: an overview. *Current Opinion in Environmental Sustainability* **15**, 57-65. <https://doi.org/10.1016/j.cosust.2015.08.005>
65. Juerges, N & Hansjürgens, B. 2016 Soil governance in the transition towards a sustainable bioeconomy – A review. *Journal of Cleaner Production* **170**. <https://doi.org/10.1016/j.jclepro.2016.10.143>
66. Guereña, A. 2016 *Unearthed: Land, power and inequality in Latin America*. OXFAM.
67. van Schaik, L & Dinnissen, R. 2014 *Terra Incognita: Land degradation as underestimated threat amplifier. Clingendael report for the Netherlands Environmental Assessment Agency (PBL)*. The Hague, Clingendael.
68. Baba, SH, Wani, MH, Zargar, BA & Bhat, IF. 2020 Determinants of land degradation in Jammu and Kashmir: implications for land governance. *Agricultural Economics Research Review* **32**, 303645.
69. Sloan, T, Payne, R, Anderson, R, Bain, C, Chapman, S, Cowie, N, Gilbert, PJ, Lindsay, R, Mauquoy, D, Newton, A, et al. 2018 Peatland afforestation in the UK and consequences for carbon storage. *Mires and Peat* **23**. <https://doi.org/10.19189/Map.2017.OMB.315>
70. Bai, ZG, Dent, DL, Olsson, L & Schaepman, ME. 2008 Proxy global assessment of land degradation. *Soil Use and Management* **24**, 223-234. <https://doi.org/10.1111/j.1475-2743.2008.00169.x>
71. Sonneveld, BGJS & Dent, DL. 2009 How good is GLASOD? *J. Environ. Manage.* **90**, 274-283. <https://doi.org/10.1016/j.jenvman.2007.09.008>
72. Bai, ZG, Dent, DL, Olsson, L, Tengberg, A, Tucker, C & Yengoh, G. 2015 A longer, closer, look at land degradation. *Agriculture for Development* **24**, 3-9.
73. FAO & ITPS. 2015 *The Status of the World's Soil Resources*.
74. Nkonya, E, Mirzabaev, A & von Braun, J. 2016 *Economics of Land Degradation and Improvement – A Global Assessment for Sustainable Development*, Springer Nature.

75. Esch, S, Brink, B, Stehfest, E, Bakkenes, M, Sewell, A, Bouwman, A, Meijer, J, Westhoek, H & van den Berg, M. 2017 *Exploring future changes in land use and land condition and the impacts on food, water, climate change and biodiversity: Scenarios for the UNCCD Global Land Outlook*.
76. United Nations Convention to Combat Desertification. 2017 *The Global Land Outlook*, first edition. Bonn, Germany.
77. IPBES. 2018 *The IPBES assessment report on land degradation and restoration. Secretariat of the Intergovernmental Science-Policy Platform on Biodiversity and Ecosystem Services*. Bonn, Germany.
78. Zimmerer, KS. 2011 " Conservation booms" with agricultural growth? Sustainability and Shifting Environmental Governance in Latin America, 1985-2008 (Mexico, Costa Rica, Brazil, Peru, Bolivia). *Latin American Research Review*, 82-114.
79. Stubenrauch, J, Garske, B & Ekardt, F. 2018 Sustainable Land Use, Soil Protection and Phosphorus Management from a Cross-National Perspective. *Sustainability* **10**, 1988. <https://doi.org/10.3390/su10061988>
80. 2017 The most neglected threat to public health in China is toxic soil; and fixing it will be hard and costly. In *The Economist*.
81. Liqiang, H. 2019 New law on soil pollution will pinpoint responsibility: <http://www.chinadaily.com.cn/a/201901/02/WS5c2c0b6fa310d91214051f8b.html>. In *China Daily*.
82. Li, T, Liu, Y, Lin, S, Liu, Y & Xie, Y. 2019 Soil Pollution Management in China: A Brief Introduction. *Sustainability* **11**, 556. <https://doi.org/10.3390/su11030556>
83. Hendlin, YH. 2014 From Terra Nullius to Terra Communis Reconsidering Wild Land in an Era of Conservation and Indigenous Rights. *Environmental Philosophy* **11**, 141-174. <https://doi.org/10.5281/zenodo.260245>
84. Pearce, F. 2018 *Conflicting Data: How Fast Is the World Losing its Forests?* Yale Environment 360, Yale School of the Environment.
85. Curtis, PG, Slay, CM, Harris, NL, Tyukavina, A & Hansen, MC. 2018 Classifying drivers of global forest loss. *Science* **361**, 1108-1111. <https://doi.org/10.1126/science.aau3445>
86. Escobar, H. 2020 Deforestation in the Brazilian Amazon is still rising sharply. *Science* **369**, 613-613. <https://doi.org/10.1126/science.369.6504.613>
87. Young, A. 1999 Is there Really Spare Land? A Critique of Estimates of Available Cultivable Land in Developing Countries. *Environment, Development and Sustainability* **1**, 3-18. <https://doi.org/10.1023/A:1010055012699>
88. Chamberlin, J, Jayne, TS & Headey, D. 2014 Scarcity amidst abundance? Reassessing the potential for cropland expansion in Africa. *Food Policy* **48**, 51-65. <https://doi.org/10.1016/j.foodpol.2014.05.002>
89. Güneralp, B, Lwasa, S, Masundire, H, Parnell, S & Seto, KC. 2017 Urbanization in Africa: challenges and opportunities for conservation. *Environmental Research Letters* **13**, 015002. <https://doi.org/10.1088/1748-9326/aa94fe>

90. Rockström, J, Steffen, W, Noone, K, Persson, Å, Chapin, FS, Lambin, EF, Lenton, TM, Scheffer, M, Folke, C, Schellnhuber, HJ, et al. 2009 A safe operating space for humanity. *Nature* **461**, 472-475. <https://doi.org/10.1038/461472a>
91. Byerlee, D, Stevenson, J & Villoria, N. 2014 Does intensification slow crop land expansion or encourage deforestation? *Global Food Security* **3**, 92-98. <https://doi.org/10.1016/j.gfs.2014.04.001>
92. Abu Hatab, A, Cavinato, MER, Lindemer, A & Lagerkvist, C-J. 2019 Urban sprawl, food security and agricultural systems in developing countries: A systematic review of the literature. *Cities* **94**, 129-142. <https://doi.org/10.1016/j.cities.2019.06.001>
93. Metternicht, G. 2018 *Land Use and Spatial Planning: Enabling Sustainable Management of Land Resources*, Springer International Publishing.
94. Seitzinger, SP, Svedin, U, Crumley, CL, Steffen, W, Abdullah, SA, Alfsen, C, Broadgate, WJ, Biermann, F, Bondre, NR, Dearing, JA, et al. 2012 Planetary Stewardship in an Urbanizing World: Beyond City Limits. *AMBIO* **41**, 787-794. <https://doi.org/10.1007/s13280-012-0353-7>
95. Smith, P, Gregory, PJ, Vuuren, Dv, Obersteiner, M, Havlík, P, Rounsevell, M, Woods, J, Stehfest, E & Bellarby, J. 2010 Competition for land. *Philosophical Transactions of the Royal Society B: Biological Sciences* **365**, 2941-2957. <https://doi.org/10.1098/rstb.2010.0127>
96. FAO. 2011 Payments for ecosystem services and food security.
97. van Vliet, J, Eitelberg, DA & Verburg, PH. 2017 A global analysis of land take in cropland areas and production displacement from urbanization. *Global Environmental Change* **43**, 107-115. <https://doi.org/10.1016/j.gloenvcha.2017.02.001>
98. Bren d'Amour, C, Reitsma, F, Baiocchi, G, Barthel, S, Güneralp, B, Erb, KH, Haberl, H, Creutzig, F & Seto, KC. 2017 Future urban land expansion and implications for global croplands. *Proc Natl Acad Sci U S A* **114**, 8939-8944. <https://doi.org/10.1073/pnas.1606036114>
99. Chen, G, Li, X, Liu, X, Chen, Y, Liang, X, Leng, J, Xu, X, Liao, W, Qiu, Y, Wu, Q, et al. 2020 Global projections of future urban land expansion under shared socioeconomic pathways. *Nature Communications* **11**. <https://doi.org/10.1038/s41467-020-14386-x>
100. Gardi, C. 2017 *Urban expansion, land cover and soil ecosystem services*, Taylor & Francis.
101. European Environment Agency. 2017 Landscapes in transition: An account of 25 years of land cover change in Europe. In *EEA Report*. Copenhagen.
102. CPRE. 2000 Valuing the land: planning for the best and most versatile agricultural land.
103. Thurston, N, Kenyon, D, Starkings, D & Taylor, K. 2011 Review of the weight that should be given to the protection of best and most versatile (BMV) land., Defra.
104. Ustaoglu, E & Williams, B. 2017 Determinants of Urban Expansion and Agricultural Land Conversion in 25 EU Countries. *Environmental Management* **60**, 717-746. <https://doi.org/10.1007/s00267-017-0908-2>

105. OECD. 2009 Farmland Conversion – The Spatial Implications of Agricultural and Land-use Policies. ed. D Diakosavvas. Paris, OECD.
106. Perrin, C, Clément, C, Melot, R & Nougarèdes, B. 2020 Preserving Farmland on the Urban Fringe: A Literature Review on Land Policies in Developed Countries. *Land* **9**, 223. <https://doi.org/10.3390/land9070223>
107. Silva, C. 2019 Auckland's Urban Sprawl, Policy Ambiguities and the Peri-Urbanisation to Pukekohe. *Urban Science* **3**, 1. <https://doi.org/10.3390/urbansci3010001>
108. Indian Government. 2009 Report of the Committee on State Agrarian Relations and the Unfinished Task in Land Reforms.
109. OECD. 2010 *Regional Development Policies in OECD Countries.*, OECD Publishing.
110. Nishi, M. 2019 Multi-Level Governance of Agricultural Land in Japan: Farmers' Perspectives and Responses to Farmland Banking. New York, Columbia.
111. Oberndorf, RB. 2012 Legal review of recently enacted farmland law and vacant, fallow and virgin lands management law: improving the legal & policy frameworks relating to land management in Myanmar. *Food Security Working Group and Land Core Group.*
112. Fernández-Maldonado, AM. 2019 Unboxing the Black Box of Peruvian Planning. *Planning Practice & Research* **34**, 368-386. <https://doi.org/10.1080/02697459.2019.1618596>
113. Slätmo, E. 2017 Preservation of Agricultural Land as an Issue of Societal Importance. *Rural Landscapes: Society, Environment, History* **4**. <https://doi.org/10.16993/rl.39>
114. Jain, M. 2018 Contemporary urbanization as unregulated growth in India: The story of census towns. *Cities* **73**, 117-127. <https://doi.org/10.1016/j.cities.2017.10.017>
115. Sili, M & Soumoulou, L. 2011 The issue of land in Argentina: Conflicts and dynamics of use, holdings and concentration. Rome, IFAD.
116. Cho, J. 2005 Urban Planning and Urban Sprawl in Korea. *Urban Policy and Research* **23**, 203-218. <https://doi.org/10.1080/08111470500143304>
117. Debolini, M, Valette, E, François, M & Chéry, J-P. 2015 Mapping land use competition in the rural–urban fringe and future perspectives on land policies: A case study of Meknès (Morocco). *Land Use Pol.* **47**, 373-381. <https://doi.org/10.1016/j.landusepol.2015.01.035>
118. Naab, FZ, Dinye, R & Kasanga, RK. 2013 Urbanisation and its impact on agricultural lands in growing cities in developing countries: a case study of Tamale in Ghana. *European scientific journal* **2**, 256-287.
119. Mori, H. 1998 Land Conversion at the Urban Fringe: A Comparative Study of Japan, Britain and the Netherlands. *Urban Studies* **35**, 1541-1558. <https://doi.org/10.1080/0042098984277>
120. Ladu, JLC, Athiba, AL & Ondogo, EC. 2019 An Assessment of the Impact of Urbanization on Agricultural Land Use in Juba City, Central Equatoria State, Republic of South Sudan. *Journal of Applied Agricultural Economics and Policy Analysis* **2**, 22-30. <https://doi.org/10.12691/jaaepa-2-1-4>

121. Ullah, S, Khan, MA, Rahman, A & Mahmood, S. 2019 Evaluation of urban encroachment on farmland: a threat to urban agriculture in Peshawar City District, Pakistan. *Erdkunde* **73**, 127-142. <https://doi.org/10.3112/erdkunde.2019.02.04>
122. Visser, O. 2017 Running out of farmland? Investment discourses, unstable land values and the sluggishness of asset making. *Agric Human Values* **34**, 185-198. <https://doi.org/10.1007/s10460-015-9679-7>
123. Satterthwaite, D, McGranahan, G & Tacoli, C. 2010 Urbanization and its implications for food and farming. *Philosophical Transactions of the Royal Society B: Biological Sciences* **365**, 2809-2820. <https://doi.org/10.1098/rstb.2010.0136>
124. Cottrell, RS, Nash, KL, Halpern, BS, Remenyi, TA, Corney, SP, Fleming, A, Fulton, EA, Hornborg, S, John, A, Watson, RA, et al. 2019 Food production shocks across land and sea. *Nature Sustainability* **2**, 130-137. <https://doi.org/10.1038/s41893-018-0210-1>
125. Prishchepov, AV, Müller, D, Dubinin, M, Baumann, M & Radeloff, VC. 2013 Determinants of agricultural land abandonment in post-Soviet European Russia. *Land Use Pol.* **30**, 873-884. <https://doi.org/10.1016/j.landusepol.2012.06.011>
126. Lambin, EF & Meyfroidt, P. 2011 Global land use change, economic globalization, and the looming land scarcity. *Proceedings of the National Academy of Sciences* **108**, 3465-3472. <https://doi.org/10.1073/pnas.1100480108>
127. Kaz'min, MA. 2016 Transformation of agricultural land use in Russian regions in the course of modern socioeconomic reforms. *Regional Research of Russia* **6**, 87-94. <https://doi.org/10.1134/S2079970516010056>
128. Liefert, WM, Liefert, O, Vocke, G & Allen, EW. 2010 Former Soviet Union Region To Play Larger Role in Meeting World Wheat Needs. In *Amber Waves*, pp. 12-19.
129. Meyfroidt, P, Schierhorn, F, Prishchepov, AV, Müller, D & Kuemmerle, T. 2016 Drivers, constraints and trade-offs associated with recultivating abandoned cropland in Russia, Ukraine and Kazakhstan. *Global Environmental Change* **37**, 1-15. <https://doi.org/10.1016/j.gloenvcha.2016.01.003>
130. Schierhorn, F, Müller, D, Beringer, T, Prishchepov, AV, Kuemmerle, T & Balman, A. 2013 Post-Soviet cropland abandonment and carbon sequestration in European Russia, Ukraine, and Belarus. *Global Biogeochemical Cycles* **27**, 1175-1185. <https://doi.org/10.1002/2013gb004654>
131. Amati, M & Taylor, L. 2010 From Green Belts to Green Infrastructure. *Planning Practice & Research* **25**, 143-155. <https://doi.org/10.1080/02697451003740122>
132. Papworth, T. 2015 The green noose. *Adam Smith Institute* **14**.
133. Cheshire, P. 2014 Turning houses into gold: the failure of British planning. Centre for Economic Performance, LSE.
134. Defra. 2020 Farming for the future: Policy and progress update. UK Government.
135. Shortly & A. (Greenbelt Foundation, Toronto), Personal Communication.

136. Carter-Whitney, M, Esakin, TC & Canadian Institute for Environmental Law and, P. 2010 Ontario's greenbelt in an international context. Toronto, Ont., Canadian Institute for Environmental Law and Policy.
137. Loghrin, H. Forthcoming Global Greenbelts Updates since 2009 (2019). Toronto, Greenbelt Foundation.
138. Suu, NV. 2009 Agricultural land conversion and its effects on farmers in contemporary Vietnam. *Focaal* **2009**, 106. <https://doi.org/10.3167/fcl.2009.540109>
139. Paudel, B, Pandit, J & Reed, B. 2013 Fragmentation and conversion of agriculture land in Nepal and Land Use Policy 2012.
140. Chen, A, He, H, Wang, J, Li, M, Guan, Q & Hao, J. 2019 A Study on the Arable Land Demand for Food Security in China. *Sustainability* **11**, 4769. <https://doi.org/10.3390/su11174769>
141. Govindaprasad, PK & Manikandan, K. 2016 Farm Land Conversion and Food Security: Empirical Evidences from Three Villages of Tamil Nadu. In *Indian Journal of Agricultural Economics*, pp. 493-503.
142. Islam, GMT, Islam, A, Shopan, AA, Rahman, MM, Lázár, AN & Mukhopadhyay, A. 2015 Implications of agricultural land use change to ecosystem services in the Ganges delta. *J Environ Manage* **161**, 443-452. <https://doi.org/10.1016/j.jenvman.2014.11.018>
143. Robson, JS, Ayad, HM, Wasfi, RA & El-Geneidy, AM. 2012 Spatial disintegration and arable land security in Egypt: A study of small- and moderate-sized urban areas. *Habitat International* **36**, 253-260. <https://doi.org/10.1016/j.habitatint.2011.10.001>
144. Dai, W & Dai, W. 2019 Effects of urban expansion on environment by morphological study. *IOP Conference Series: Earth and Environmental Science* **227**, 052004. <https://doi.org/10.1088/1755-1315/227/5/052004>
145. Agrimonde-Terra. 2018 *Land Use and Food Security in 2050: a Narrow Road*, éditions Quae.
146. Bezbradica, L, Pantić, M & Gajić, A. 2019 The land use and soil protection: Planning and legal regulations in Serbia. *Zemljiste i biljka* **68**, 51-71. <https://doi.org/10.5937/ZemBilj1902051B>
147. Owusu Ansah, B & Chigbu, UE. 2020 The Nexus between Peri-Urban Transformation and Customary Land Rights Disputes: Effects on Peri-Urban Development in Trede, Ghana. *Land* **9**, 187. <https://doi.org/10.3390/land9060187>
148. Jiang, L & Zhang, Y. 2016 Modeling Urban Expansion and Agricultural Land Conversion in Henan Province, China: An Integration of Land Use and Socioeconomic Data. *Sustainability* **8**, 920. <https://doi.org/10.3390/su8090920>
149. Li, S. 2018 Change detection: how has urban expansion in Buenos Aires metropolitan region affected croplands. *International Journal of Digital Earth* **11**, 195-211. <https://doi.org/10.1080/17538947.2017.1311954>
150. Carreño, L, Frank, FC & Viglizzo, EF. 2012 Tradeoffs between economic and ecosystem services in Argentina during 50 years of land-use change. *Agriculture, Ecosystems & Environment* **154**, 68-77. <https://doi.org/10.1016/j.agee.2011.05.019>

151. Meadows, D & Randers, J. 2004 *Limits to Growth: The 30-Year Update*, Chelsea Green Publishing.
152. Schumacher, EF. 1973 *Small is beautiful; economics as if people mattered*. New York, Harper & Row.
153. Kuper, S. 2019 Thoreau, Leopold, & Carson: Challenging Capitalist Conceptions of the Natural Environment. *Consilience* **0**. <https://doi.org/10.7916/consilience.v0i13.3927>
154. Kennel, CF. The gathering anthropocene crisis. *The Anthropocene Review* **2**, 81-98. <https://doi.org/10.1177/2053019620957355>
155. Żróbek-Róžańska, A & Zielińska-Szczepkowska, J. 2019 National Land Use Policy against the Misuse of the Agricultural Land—Causes and Effects. Evidence from Poland. *Sustainability* **11**, 6403. <https://doi.org/10.3390/su11226403>
156. McPike, JL. 2015 Creating Space for the Formal Amongst the Informal: An Examination of Urban Housing Policies, State Power, and Multi-Scalar Politics in Indian Cities. In *RC21 International Conference on “The Ideal City: between myth and reality. Representations, policies, contradictions and challenges for tomorrow's urban life” Urbino (Italy) 27-29 August 2015*.
157. Bartz, D. 2015 *Soil atlas: Facts and figures about earth, land and fields*. Berlin.
158. Appiah, DO, Asante, F & Nketiah, B. 2019 Perspectives on Agricultural Land Use Conversion and Food Security in Rural Ghana. *Sci* **1**, 14. <https://doi.org/10.3390/sci1010014.v1>
159. Schueler, V, Kuemmerle, T & Schröder, H. 2011 Impacts of Surface Gold Mining on Land Use Systems in Western Ghana. *AMBIO* **40**, 528-539. <https://doi.org/10.1007/s13280-011-0141-9>
160. Franco, J & Borrás Jr, SM. 2013 *Land concentration, land grabbing and people's struggles in Europe*. Transnational Institute, Amsterdam.
161. Khalid, H & Yusuf, MD. 2012 Resource management: Fragmentation of land ownership and its impact on sustainability of agriculture.
162. Prokop, G, Jobstmann, H & Schönbauer, A. 2011 *Report on Best Practices for Limiting Soil Sealing and Mitigating Its Effects*. European Communities.
163. Scalenghe, R & Marsan, FA. 2009 The anthropogenic sealing of soils in urban areas. *Landscape and Urban Planning* **90**, 1-10. <https://doi.org/10.1016/j.landurbplan.2008.10.011>
164. Tobias, S, Conen, F, Duss, A, Wenzel, LM, Buser, C & Alewell, C. 2018 Soil sealing and unsealing: State of the art and examples. *Land Degradation & Development* **29**, 2015-2024. <https://doi.org/https://doi.org/10.1002/ldr.2919>
165. Daniels, T & Lapping, M. 2005 Land Preservation: An Essential Ingredient in Smart Growth. *Journal of Planning Literature* **19**, 316-329. <https://doi.org/10.1177/0885412204271379>
166. Desrousseaux, M, Schmitt, B, Billet, P, Béchet, B, Le Bissonnais, Y & Ruas, A. 2019 Artificialised Land and Land Take: What Policies Will Limit Its Expansion and/or Reduce Its Impacts? In *International Yearbook of Soil Law and Policy 2018* (eds. H Ginzky, E Dooley, IL Heuser, E Kasimbazi, T Markus & T Qin), pp. 149-165. Cham, Springer International Publishing).

167. Hellerstein, DR. 2002 *Farmland protection : the role of public preferences for rural amenities*. Washington, D.C., U.S. Dept. of Agriculture, Economic Research Service.
168. Gosnell, H, Kline, JD, Chrostek, G & Duncan, J. 2011 Is Oregon's land use planning program conserving forest and farm land? A review of the evidence. *Land Use Pol.* **28**, 185-192. <https://doi.org/10.1016/j.landusepol.2010.05.012>
169. Ludlow, D, Falconi, M, Carmichael, L, Croft, N, Di Leginio, M, Fumanti, F, Sheppard, A & Smith, N. 2013 Land Planning and Soil Evaluation Instruments in EEA Member and Cooperating Countries (with inputs from Eionet NRC Land Use and Spatial Planning). Final Report for EEA from ETC/SIA (EEA project managers: G. Louwagie and G. Dige). Available at: <http://www.eea.europa.eu/themes/landuse/document-library>. EEA & ETC/SIA.
170. Grass, I, Loos, J, Baensch, S, Batáry, P, Librán-Embid, F, Ficiciyan, A, Klaus, F, Riechers, M, Rosa, J, Tiede, J, et al. 2019 Land-sharing/-sparing connectivity landscapes for ecosystem services and biodiversity conservation. *People and Nature* **1**, 262-272. <https://doi.org/10.1002/pan3.21>
171. Huang, Q, Lu, J, Li, M, Chen, Z & Li, F. 2015 Developing Planning Measures to Preserve Farmland: A Case Study from China. *Sustainability* **7**, 13011-13028. <https://doi.org/10.3390/su71013011>
172. Indian Government. 2003 Law for the preservation of the agricultural lands.
173. Sartori, D, Catalano, G, Genco, M, Pancotti, C, Sirtori, E, Vignetti, S & Bo, C. 2014 Guide to Cost-benefit Analysis of Investment Projects. Economic appraisal tool for Cohesion Policy 2014-2020.
174. Malczewski, J. 2004 GIS-based land-use suitability analysis: a critical overview. *Progress in Planning* **62**, 3-65. <https://doi.org/10.1016/j.progress.2003.09.002>
175. Dent, D & Young, A. 1981 *Soil Survey and Land Evaluation*, Allen & Unwin.
176. Baveye, PC, Baveye, J & Gowdy, J. 2016 Soil "Ecosystem" Services and Natural Capital: Critical Appraisal of Research on Uncertain Ground. *Frontiers in Environmental Science* **4**. <https://doi.org/10.3389/fenvs.2016.00041>
177. Bouwman, AF & Sombroek, WG. 1990 Inputs to climatic change by soils and agriculture related activities: Present status and possible future trends. In *Soils on a Warmer Earth* (pp. 15 - 30. Amsterdam, Elsevier).
178. Bouwman, AF. 1990 Land use related sources of greenhouse gases: Present emissions and possible future trends. *Land Use Pol.* **7**, 154-164. [https://doi.org/10.1016/0264-8377\(90\)90006-K](https://doi.org/10.1016/0264-8377(90)90006-K)
179. Batjes, NH & Bridges, EM. 1992 *A Review of Soil Factors and Processes that Control Fluxes of Heat, Moisture and Greenhouse Gases*. Wageningen, The Netherlands., ISRIC.
180. Weart, SR. 2008 *The Discovery of Global Warming: Revised and Expanded Edition*, Harvard University Press.
181. Wall, DH. 2004 *Sustaining Biodiversity and Ecosystem Services in Soils and Sediments*, Island Press.

182. Karlen, DL, Mausbach, MJ, Doran, JW, Cline, RG, Harris, RF & Schuman, GE. 1997 Soil Quality: A Concept, Definition, and Framework for Evaluation (A Guest Editorial). *Soil Science Society of America Journal* **61**, 4-10. <https://doi.org/10.2136/sssaj1997.03615995006100010001x>
183. Hannam, I & Boer, B. 2002 Legal and institutional frameworks for sustainable soils.
184. Costanza, R, d'Arge, R, de Groot, R, Farber, S, Grasso, M, Hannon, B, Limburg, K, Naeem, S, O'Neill, RV, Paruelo, J, et al. 1997 The value of the world's ecosystem services and natural capital. *Nature* **387**, 253-260. <https://doi.org/10.1038/387253a0>
185. Breure, AM, De Deyn, GB, Dominati, E, Eglin, T, Hedlund, K, Van Orshoven, J & Posthuma, L. 2012 Ecosystem services: a useful concept for soil policy making! *Current Opinion in Environmental Sustainability* **4**, 578-585. <https://doi.org/10.1016/j.cosust.2012.10.010>
186. Lescourret, F, Magda, D, Richard, G, Adam-Blondon, A-F, Bardy, M, Baudry, J, Doussan, I, Dumont, B, Lefèvre, F, Litrico, I, et al. 2015 A social–ecological approach to managing multiple agro-ecosystem services. *Current Opinion in Environmental Sustainability* **14**, 68-75. <https://doi.org/10.1016/j.cosust.2015.04.001>
187. Robinson, DA, Hockley, N, Cooper, DM, Emmett, BA, Keith, AM, Lebron, I, Reynolds, B, Tipping, E, Tye, AM, Watts, CW, et al. 2013 Natural capital and ecosystem services, developing an appropriate soils framework as a basis for valuation. *Soil Biology and Biochemistry* **57**, 1023-1033. <https://doi.org/10.1016/j.soilbio.2012.09.008>
188. Ellili-Bargaoui, Y, Walter, C, Lemercier, B & Michot, D. 2021 Assessment of six soil ecosystem services by coupling simulation modelling and field measurement of soil properties. *Ecological Indicators* **121**, 107211. <https://doi.org/10.1016/j.ecolind.2020.107211>
189. Drobnik, T, Greiner, L, Keller, A & Grêt-Regamey, A. 2018 Soil quality indicators – From soil functions to ecosystem services. *Ecological Indicators* **94**, 151-169. <https://doi.org/https://doi.org/10.1016/j.ecolind.2018.06.052>
190. Dominati, E, Mackay, A, Green, S & Patterson, M. 2014 A soil change-based methodology for the quantification and valuation of ecosystem services from agro-ecosystems: A case study of pastoral agriculture in New Zealand. *Ecological Economics* **100**, 119-129. <https://doi.org/10.1016/j.ecolecon.2014.02.008>
191. Daniels, T. 2020 (Professor of City and Regional Planning, Weitzman School of Design, University of Pennsylvania), Personal communication.
192. Defra. 2020 EIA (Agriculture) regulations: apply to make changes to rural land.
193. 2019 Scottish Government, Guidance on consideration of soil in Strategic Environmental Assessment , v5, 5.4.19.
194. Caldwell, WJ, Hilts, S & Wilton, B. 2017 *Farmland Preservation: Land for Future Generations*, University of Manitoba Press.
195. Nolon, J. 2003 Land Preservation. *Pace Law Faculty Publications*.

196. Robinson, DA, Jackson, BM, Clothier, BE, Dominati, EJ, Marchant, SC, Cooper, DM & Bristow, KL. 2013 Advances in Soil Ecosystem Services: Concepts, Models, and Applications for Earth System Life Support. *Vadose Zone Journal* **12**, vzj2013.2001.0027.
<https://doi.org/https://doi.org/10.2136/vzj2013.01.0027>
197. Meadows, DH. 1972 *The Limits to growth; a report for the Club of Rome's project on the predicament of mankind*, New York : Universe Books, [1972].
198. Norström, AV, Cvitanovic, C, Löf, MF, West, S, Wyborn, C, Balvanera, P, Bednarek, AT, Bennett, EM, Biggs, R, de Bremond, A, et al. 2020 Principles for knowledge co-production in sustainability research. *Nature Sustainability* **3**, 182-190. <https://doi.org/10.1038/s41893-019-0448-2>
199. Erinosh, BT. 2013 The Revised African Convention on the Conservation of Nature and Natural Resources: Prospects for a Comprehensive Treaty for the Management of Africa's Natural Resources. *African Journal of International and Comparative Law* **21**, 378-397.
<https://doi.org/10.3366/ajicl.2013.0069>
200. FAO. 1971 *Land degradation*. Rome, Food and Agriculture Organization of the United Nations.
201. FAO. 1971 *Legislative principles of soil conservation*.
202. Boer, B, Ginzky, H & Heuser, I. 2017 International Soil Protection Law: History, Concepts and Latest Developments. (pp. 49-72.
203. Bodle, R, Stockhaus, H, Wolff, F, Scherf, C-S & Oberthür, S. 2019 *Improving international soil governance - Analysis and recommendations*. Berlin, Ecologic Institute and Öko-Institute.
204. Europe, Co. 1972 European Soil Charter. ed. Co Europe. Brussels.
205. Tóth, Z. 2018 International dimensions of EU soil policy – The main binding and non-binding legal instruments. *Hungarian Journal of Legal Studies Acta Juridica Hungarica* **59**, 290.
<https://doi.org/10.1556/2052.2018.59.3.4>
206. Byron-Cox, R. 2020 From Desertification to Land Degradation Neutrality: The UNCCD and the Development of Legal Instruments for Protection of Soils. In *Legal Instruments for Sustainable Soil Management in Africa* (eds. H Yahyah, H Ginzky, E Kasimbazi, R Kibugi & OC Ruppel), pp. 1-13. Cham, Springer International Publishing).
207. FAO. 1981 World Soil Charter.
208. UNEP. 1982 World Soils Policy.
209. Wood & Harold, W. 1985 The United Nations World Charter for Nature: The Developing Nations' Initiative to Establish Protections for the Environment. *Ecology Law Quarterly* **12**, 977.
210. World Commission on Environment and Development. 1987 *Our common future*. Oxford; New York, Oxford University Press.
211. World Bank. 2002 *The First Decade of the GEF: second overall performance study*. Washington DC, World Bank.

212. Wolff, F & Kaphengst, T. 2017 The UN Convention on Biological Diversity and Soils: Status and Future Options. In *International Yearbook of Soil Law and Policy 2016* (eds. H Ginzky, IL Heuser, T Qin, OC Ruppel & P Wegerdt), pp. 129-148. Cham, Springer International Publishing).
213. CBD/SBSTTA/24/7/Rev.1. 4 December 2020 Review of the international initiative for the conservation and sustainable use of soil biodiversity and updated plan of action.
214. United Nations. 1994 Convention to Combat Desertification in those Countries Experiencing Serious Drought and/or Desertification, Particularly in Africa. In *International Legal Materials*, pp. 1328-1382, 2017/02/27 ed, Cambridge University Press.
215. Global Environment Facility (GEF). 1999 Clarifying Linkages Between Land Degradation And The GEF Focal Areas: An Action Plan For Enhancing GEF Support: An Action Plan for Enhancing GEF Support, GEF/C.14/4, November 17, 1999. Washington DC, World Bank Group.
216. Hannam, I & Boer, B. 2019 Land degradation and international environmental law. In *Response to Land Degradation* (ed. IDH E. Michael Bridges, L. Roel Oldeman, Frits W.T. Penning de Vries, Sara J. Scherr, Samran Sombatpanit, Robin N. Leslie, Tanadol Compo, Apuntree Prueksapong), pp. 429-440.
217. FAO and UNEP. 2020 *Legislative approaches to sustainable agriculture and natural resources governance*. FAO Legislative Study No. 114. Rome, Food and Agriculture Organization of the United Nations.
218. Chasek, P, Safriel, U, Shikongo, S & Fuhrman, VF. 2015 Operationalizing Zero Net Land Degradation: The next stage in international efforts to combat desertification? *Journal of Arid Environments* **112**, 5-13. <https://doi.org/10.1016/j.jaridenv.2014.05.020>
219. United Nations Framework Convention on Climate Change, 9 May 1992 (in force 21 March 1994), 31 ILM 849 ; 1771 UNTS 10 7. (“UNFCCC”).
220. Fee, E. 2019 Implementing the Paris Climate Agreement: Risks and Opportunities for Sustainable Land Use. In *International Yearbook of Soil Law and Policy 2018* (eds. H Ginzky, E Dooley, IL Heuser, E Kasimbazi, T Markus & T Qin), pp. 249-270. Cham, Springer International Publishing).
221. Convention on Biological Diversity. 2020 Convention on Biological Diversity: Review of the International Initiative for the Conservation and Sustainable Use of Soil Biodiversity and Updated Plan of Action.
222. FAO, ITPS, GSBI, SCBD & EC. 2020 State of knowledge of soil biodiversity - Status, challenges and potentialities: Report 2020. FAO.
223. Rojas, RV & Caon, L. 2016 The international year of soils revisited: promoting sustainable soil management beyond 2015. *Environmental Earth Sciences* **75**. <https://doi.org/10.1007/s12665-016-5891-z>
224. FAO. 2015 Revised World Soil Charter. Rome, Italy.
225. FAO. 2017 Voluntary Guidelines for Sustainable Soil Management.
226. Orr, B, Cowie, A, Castillo Sanchez, V, Chasek, P, Crossman, N, Erlewein, A, Louwagie, G, Maron, M, Metternicht, G & Minelli, S. 2017 Scientific conceptual framework for land degradation neutrality. In A

report of the science-policy interface. *United Nations Convention to Combat Desertification (UNCCD)*, Bonn, Germany, pp. 1-98.

227. International Law Association. 2020 ILA Guidelines on the Role of International Law in Sustainable Natural Resources Management for Development, Resolution 4/2020, 2020 Report of the Seventy Ninth Conference, Kyoto.
228. Minasny, B, Malone, BP, McBratney, AB, Angers, DA, Arrouays, D, Chambers, A, Chaplot, V, Chen, Z-S, Cheng, K, Das, BS, et al. 2017 Soil carbon 4 per mille. *Geoderma* **292**, 59-86. <https://doi.org/10.1016/j.geoderma.2017.01.002>
229. Chabbi, A, Lehmann, J, Ciais, P, Loescher, HW, Cotrufo, MF, Don, A, SanClements, M, Schipper, L, Six, J, Smith, P, et al. 2017 Aligning agriculture and climate policy. *Nature Climate Change* **7**, 307-309. <https://doi.org/10.1038/nclimate3286>
230. Poulton, P, Johnston, J, Macdonald, A, White, R & Powlson, D. 2018 Major limitations to achieving “4 per 1000” increases in soil organic carbon stock in temperate regions: Evidence from long-term experiments at Rothamsted Research, United Kingdom. *Glob. Change Biol.* **24**, 2563-2584. <https://doi.org/10.1111/gcb.14066>
231. Minasny, B, Arrouays, D, McBratney, AB, Angers, DA, Chambers, A, Chaplot, V, Chen, Z-S, Cheng, K, Das, BS, Field, DJ, et al. 2018 Rejoinder to Comments on Minasny et al., 2017 Soil carbon 4 per mille *Geoderma* **292**, 59–86. *Geoderma* **309**, 124-129. <https://doi.org/10.1016/j.geoderma.2017.05.026>
232. Soussana, J-F, Lutfalla, S, Ehrhardt, F, Rosenstock, T, Lamanna, C, Havlík, P, Richards, M, Lini, E, Wollenberg, E, Chotte, J-L, et al. 2019 Matching policy and science: Rationale for the '4 per 1000-soils for food security and climate' initiative. <https://doi.org/10.1016/j.still.2017.12.002>
233. Kibugi, R. 2018 Soil Health, Sustainable Land Management and Land Degradation in Africa: Legal Options on the Need for a Specific African Soil Convention or Protocol. In *International Yearbook of Soil Law and Policy 2017* (eds. H Ginzky, E Dooley, IL Heuser, E Kasimbazi, T Markus & T Qin), pp. 387-411. Cham, Springer International Publishing).
234. Markus, T. 2017 The Alpine Convention’s Soil Conservation Protocol: A Model Regime? In *International Yearbook of Soil Law and Policy 2016* (eds. H Ginzky, IL Heuser, T Qin, OC Ruppel & P Wegerdt), pp. 149-164. Cham, Springer International Publishing).
235. 2003 *Revised European Charter for the Protection and Sustainable Management of Soil*, Strasbourg, 17 July 2003 CO-DBP/documents/codbp2003/10e.
236. European Commission. 2006 Soil protection - The story behind the Strategy.
237. Stankovics, P, Tóth, G & Tóth, Z. 2018 Identifying Gaps between the Legislative Tools of Soil Protection in the EU Member States for a Common European Soil Protection Legislation. *Sustainability* **10**, 2886. <https://doi.org/10.3390/su10082886>
238. Montanarella, L & Panagos, P. 2021 The relevance of sustainable soil management within the European Green Deal. *Land Use Pol.* **100**, 104950. <https://doi.org/10.1016/j.landusepol.2020.104950>
239. 2021 European Parliament resolution of 28 April 2021 on soil protection.

240. Keesstra, SD, Bouma, J, Wallinga, J, Tiftonell, P, Smith, P, Cerdà, A, Montanarella, L, Quinton, JN, Pachepsky, Y, van der Putten, WH, et al. 2016 The significance of soils and soil science towards realization of the United Nations Sustainable Development Goals. *SOIL* **2**, 111-128. <https://doi.org/10.5194/soil-2-111-2016>
241. Tóth, G, Hermann, T, da Silva, MR & Montanarella, L. 2018 Monitoring soil for sustainable development and land degradation neutrality. *Environmental Monitoring and Assessment* **190**, 57. <https://doi.org/10.1007/s10661-017-6415-3>
242. Shen, X, Wang, X, Zhang, Z, Lu, Z & Lv, T. 2019 Evaluating the effectiveness of land use plans in containing urban expansion: An integrated view. *Land Use Pol.* **80**, 205-213. <https://doi.org/10.1016/j.landusepol.2018.10.001>
243. Roose, A, Kull, A, Gauk, M & Tali, T. 2013 Land use policy shocks in the post-communist urban fringe: A case study of Estonia. *Land Use Pol.* **30**, 76-83. <https://doi.org/10.1016/j.landusepol.2012.02.008>
244. Lu, Y, Song, S, Wang, R, Liu, Z, Meng, J, Sweetman, AJ, Jenkins, A, Ferrier, RC, Li, H, Luo, W, et al. 2015 Impacts of soil and water pollution on food safety and health risks in China. *Environment International* **77**, 5-15. <https://doi.org/10.1016/j.envint.2014.12.010>
245. Zhao, F-J, Ma, Y, Zhu, Y-G, Tang, Z & McGrath, SP. 2015 Soil Contamination in China: Current Status and Mitigation Strategies. *Environmental Science & Technology* **49**, 750-759. <https://doi.org/10.1021/es5047099>
246. Chen, R, de Sherbinin, A, Ye, C & Shi, G. 2014 China's Soil Pollution: Farms on the Frontline. *Science* **344**, 691-691. <https://doi.org/10.1126/science.344.6185.691-a>
247. Vogel, H-J, Eberhardt, E, Franko, U, Lang, B, Ließ, M, Weller, U, Wiesmeier, M & Wollschläger, U. 2019 Quantitative Evaluation of Soil Functions: Potential and State. *Frontiers in Environmental Science* **7**. <https://doi.org/10.3389/fenvs.2019.00164>
248. Vargas, L, Willemsen, L & Hein, L. 2019 Assessing the Capacity of Ecosystems to Supply Ecosystem Services Using Remote Sensing and An Ecosystem Accounting Approach. *Environmental Management* **63**, 1-15. <https://doi.org/10.1007/s00267-018-1110-x>
249. Kibblewhite, MG, Ritz, K & Swift, MJ. 2008 Soil health in agricultural systems. *Philos Trans R Soc Lond B Biol Sci* **363**, 685-701. <https://doi.org/10.1098/rstb.2007.2178>
250. Powlson, DS, Gregory, PJ, Whalley, WR, Quinton, JN, Hopkins, DW, Whitmore, AP, Hirsch, PR & Goulding, KWT. 2011 Soil management in relation to sustainable agriculture and ecosystem services. *Food Policy* **36**, Supplement 1, S72-S87. <https://doi.org/10.1016/j.foodpol.2010.11.025>
251. Paustian, K, Lehmann, J, Ogle, S, Reay, D, Robertson, GP & Smith, P. 2016 Climate-smart soils. *Nature* **532**, 49. <https://doi.org/10.1038/nature17174>
252. LaCanne, CE & Lundgren, JG. 2018 Regenerative agriculture: merging farming and natural resource conservation profitably. *PeerJ* **6**, e4428-e4428. <https://doi.org/10.7717/peerj.4428>
253. Lal, R & Kosaki, T. 2018 *Soils and Sustainable Development Goals. GeoEcology Essay*, Schweizerbart Science Publishers.

254. Amundson, R. 2020 The policy challenges to managing global soil resources. *Geoderma* **379**, 114639. <https://doi.org/10.1016/j.geoderma.2020.114639>
255. Frelh-Larsen, A, Bowyer, C, Albrecht, S, Keenleyside, C, Kemper, M, Nanni, S, Naumann, S, Mottershead, RD, Langrebe, R, Andersen, E, et al. 2017 *Updated Inventory and Assessment of Soil Protection Policy Instruments in EU Member States*.
256. Ronchi, S, Salata, S, Arcidiacono, A, Piroli, E & Montanarella, L. 2019 Policy instruments for soil protection among the EU member states: A comparative analysis. *Land Use Pol.* **82**, 763-780. <https://doi.org/10.1016/j.landusepol.2019.01.017>
257. Wingeyer, AB, Amado, TJC, Pérez-Bidegain, M, Studdert, GA, Varela, CHP, Garcia, FO & Karlen, DL. 2015 Soil Quality Impacts of Current South American Agricultural Practices. *Sustainability* **7**, 2213-2242. <https://doi.org/10.3390/su7022213>
258. Webb, A, Kelly, G & Dougherty, W. 2015 Soil governance in the agricultural landscapes of New South Wales, Australia. *International Journal of Rural Law and Policy Special Edition* **1**, 1-16. <https://doi.org/10.5130/ijrlp.i1.2015.4169>
259. Chukov, SN & Yakovlev, AS. 2019 Soil and Land Categories in the Modern Legislation of Russia. *Eurasian Soil Science* **52**, 865-870. <https://doi.org/10.1134/S1064229319070020>
260. Fromherz, NA. 2012 The Case for a Global Treaty on Soil Conservation, Sustainable Farming, and the Preservation of Agrarian Culture. **39**. <https://doi.org/10.15779/Z38BC49>
261. IUCN. 2019 World Commission on Environmental Law, Soil Desertification & Sustainable Agriculture Specialist Group, 2019 Mid-Year Report. Online.
262. Hazelton, PA, Frossard, E, Blum, WEH & Warkentin, BP. 2006 Australian examples of the role of soils in environmental problems. In *Function of Soils for Human Societies and the Environment* (p. 0, Geological Society of London).
263. Sophia Antipolis. 2003 Threats to Soils in Mediterranean Countries: Document Review. In *Plan Bleu Papers*. Valbonne, France, Plan Bleu.
264. Gonzalez Lago, M, Plant, R & Jacobs, B. 2019 Re-politicising soils: What is the role of soil framings in setting the agenda? *Geoderma* **349**, 97-106. <https://doi.org/10.1016/j.geoderma.2019.04.021>
265. Heuser, IL. 2018 Development of Soil Awareness in Europe and Other Regions: Historical and Ethical Reflections About European (and International) Soil Protection Law. In *International Yearbook of Soil Law and Policy 2017* (eds. H Ginzky, E Dooley, IL Heuser, E Kasimbazi, T Markus & T Qin), pp. 451-474. Cham, Springer International Publishing).
266. Moallemi, EA, Haan, F, Hadjidakou, M, Khatami, S, Malekpour, S, Smajgl, A, Stafford Smith, M, Voinov, A, Bandari, R, Lamichhane, P, et al. 2021 Evaluating Participatory Modeling Methods for Co - creating Pathways to Sustainability. *Earth's Future* **9**. <https://doi.org/10.1029/2020EF001843>
267. Koch, A, McBratney, A, Adams, M, Field, D, Hill, R, Crawford, J, Minasny, B, Lal, R, Abbott, L, O'Donnell, A, et al. 2013 Soil Security: Solving the Global Soil Crisis. *Global Policy* **4**, 434-441. <https://doi.org/10.1111/1758-5899.12096>

268. McBratney, A, Field, DJ & Koch, A. 2014 The dimensions of soil security. *Geoderma* **213**, 203-213. <https://doi.org/10.1016/j.geoderma.2013.08.013>
269. Bouma, J. 2020 Soil security as a roadmap focusing soil contributions on sustainable development agendas. *Soil Security* **1**, 100001. <https://doi.org/10.1016/j.soisec.2020.100001>
270. Hill, R. 2017 The Place of Soil in International Government Policy. In *Global Soil Security* (eds. DJ Field, CLS Morgan & AB McBratney), pp. 443-449. Cham, Springer International Publishing).
271. Field, DJ, Morgan, CLS & McBratney, AB. 2017 *Global Soil Security*, Springer International Publishing, A. G.
272. Lilburne, L, Eger, A, Mudge, P, Ausseil, A-G, Stevenson, B, Herzig, A & Beare, M. 2020 The Land Resource Circle: Supporting land-use decision making with an ecosystem-service-based framework of soil functions. *Geoderma* **363**, 114134. <https://doi.org/10.1016/j.geoderma.2019.114134>
273. Juerges, N, Hagemann, N & Bartke, S. 2018 A tool to analyse instruments for soil governance: the REEL-framework. *Journal of Environmental Policy & Planning* **20**, 617-631. <https://doi.org/10.1080/1523908X.2018.1474731>
274. Ginzky, H. 2020 Good Governance for “Sustainable Management of Soil” on National and International Level: How to Do It? In *Legal Instruments for Sustainable Soil Management in Africa* (eds. H Yahyah, H Ginzky, E Kasimbazi, R Kibugi & OC Ruppel), pp. 35-54. Cham, Springer International Publishing).